# A Wigner-Eckart Theorem for Group Equivariant Convolution Kernels

**Leon Lang**[*]
AMLab, CSL
University of Amsterdam
l.lang@uva.nl

**Maurice Weiler**
AMLab, QUVA Lab
University of Amsterdam
m.weiler.ml@gmail.com

## Abstract

Group equivariant convolutional networks (GCNNs) endow classical convolutional networks with additional symmetry priors, which can lead to a considerably improved performance. Recent advances in the theoretical description of GCNNs revealed that such models can generally be understood as performing convolutions with *$G$-steerable* kernels, that is, kernels that satisfy an equivariance constraint themselves. While the $G$-steerability constraint has been *derived*, it has to date only been *solved* for specific use cases – a general characterization of $G$-steerable kernel spaces is still missing. This work provides such a characterization for the practically relevant case of $G$ being any compact group. Our investigation is motivated by a striking analogy between the constraints underlying steerable kernels on the one hand and spherical tensor operators from quantum mechanics on the other hand. By generalizing the famous Wigner-Eckart theorem for spherical tensor operators, we prove that steerable kernel spaces are fully understood and parameterized in terms of 1) generalized reduced matrix elements, 2) Clebsch-Gordan coefficients, and 3) harmonic basis functions on homogeneous spaces.

## 1 Introduction

Undoubtedly, symmetries play a central role in the formulation of physical theories. Any imposed symmetry greatly reduces the set of admissible physical laws and dynamics. Specifically in quantum mechanics, the Hilbert space of a system is equipped with a group representation which specifies the transformation law of system states. Quantum mechanical operators, which map between different states, are required to respect these transformation laws. That is, any symmetry transformation of a state on which they act should lead to a corresponding transformation of the resulting state after their action. This requirement imposes a *symmetry constraint on the operators* themselves – only specific operators can map between a given pair of states.

The situation in equivariant deep learning is remarkably similar to that in physics. Instead of a physical system, one considers in this case some learning task subject to symmetries. For instance, image segmentation is usually assumed to be translationally symmetric: a shift of the input image should lead to a corresponding shift of the predicted segmentation mask. Convolutional networks guarantee this property via their inherent translation equivariance. The role of the quantum states is in equivariant deep learning taken by the features in each layer, which are due to the enforced equivariance endowed with some transformation law. The analog of quantum mechanical operators, mapping between states, is the neural connectivity, mapping between features of consecutive layers. As in the case of operators, there is a *symmetry (equivariance) constraint on the neural connectivity* – only specific connectivity patterns guarantee a correct transformation law of the resulting features.

In this work we are considering group equivariant convolutional networks (GCNNs), which are convolutional networks that are equivariant w.r.t. symmetries of the space on which the convolution is performed. Typical examples are isometry equivariant CNNs on Euclidean spaces (Weiler & Cesa, 2019) or spherical CNNs (Cohen et al., 2018). Many different formulations of GCNNs have been proposed, however, it has recently been shown that $H$-equivariant GCNNs on homogeneous spaces

---

[*]This research has been conducted during an internship at QUVA lab, University of Amsterdam.

$H/G$ can in a fairly general setting be understood as performing convolutions with *G-steerable kernels* (Cohen et al., 2019b). Convolutional weight sharing hereby guarantees the equivariance under "translations" of the space while $G$-steerability is a constraint on the convolution kernel that ensures its equivariance under the action of the stabilizer subgroup $G < H$. Although the space of $G$-steerable kernels has been characterized for specific choices of groups $G$ and feature transformation laws, i.e., group representations $\rho$, see Section 5, no general solution was known so far. This work characterizes the solution space for arbitrary compact groups $G$.

Our solution is motivated by the close resemblance of the $G$-steerability kernel constraint to the defining constraint of *spherical tensor operators* (or more general representation operators (Jeevanjee, 2011)) in quantum mechanics. The famous *Wigner-Eckart theorem* describes the general structure of these operators by Clebsch-Gordan coefficients, with the degrees of freedom given by reduced matrix elements. By generalizing this theorem, we find a general characterization and parameterization of $G$-steerable kernel spaces. For specific examples, like $G = \mathrm{SO}(3)$ or compact subgroups of $G = \mathrm{O}(2)$, our kernel space solution specializes to earlier work, e.g., Worrall et al. (2016); Thomas et al. (2018); Weiler & Cesa (2019). Our main contributions are the following:

- We present a *generalized Wigner-Eckart theorem* 4.1 *for G-steerable kernels*. It describes the general structure of equivariant kernels in terms of 1) endomorphism bases, which generalize reduced matrix elements, 2) Clebsch-Gordan coefficients, and 3) harmonic basis functions on a suitable homogeneous space. In contrast to the usual formulation, we cover any compact group $G$ and both real and complex representations.
- Corollary 4.2 explains how to *parameterize G-steerable kernels* and thus GCNNs.
- We apply the theorem exemplarily to solve for the kernel spaces for the symmetry groups $\mathrm{SO}(2)$, $\mathbb{Z}/2$, $\mathrm{SO}(3)$ and $\mathrm{O}(3)$, considering both real and complex representations. Thereby, we demonstrate that the endomorphism bases, Clebsch-Gordan coefficients, and harmonic basis functions can usually be determined for practically relevant symmetry groups.

## 2   Symmetry-constrained Operators and their Matrix Elements

To motivate our generalized Wigner-Eckart theorem, we review quantum mechanical representation operators and $G$-steerable kernels with an emphasis on the similarity of their underlying symmetry constraints. Due to their symmetries, the matrix elements of such operators and kernels are fully specified by a comparatively small number of reduced matrix elements or learnable parameters, respectively. This reduction is for representation operators described by the Wigner-Eckart theorem. For clarity, we discuss this theorem in its most popular form, i.e., for spherical tensor operators (SO(3)-representation operators transforming under irreducible representations).

**The Representation Operator Constraint**    Consider a quantum mechanical system with symmetry under the action of some group $G$, for instance rotations. The action of this symmetry group on quantum states is modeled by some unitary $G$-representation[1] $U : G \to \mathrm{U}(\mathcal{H})$ on the Hilbert space $\mathcal{H}$. More specifically, $G$ acts on kets according to $|\psi\rangle \mapsto |\psi'\rangle := U(g)|\psi\rangle$ and on bras according to $\langle\psi| \mapsto \langle\psi'| := \langle\psi|U(g)^\dagger$, where $U(g)^\dagger$ is the adjoint of $U(g)$. Observables of the system correspond to self-adjoint operators $A = A^\dagger$. The expectation value of such an observable in some quantum state $|\psi\rangle$ is given by $\langle\psi|A|\psi\rangle \in \mathbb{R}$.

The transformation behaviors of states and observables need to be consistent with each other. As an example, consider a system consisting of a single, free particle in $\mathbb{R}^3$, which is (among other symmetries) symmetric under rotations $G = \mathrm{SO}(3)$. The momentum of the particle in the direction of the three frame axes is measured by the three momentum operators $(P_1, P_2, P_3)$. Since the momentum of a classical particle transforms geometrically like a vector, one needs to demand the same for the momentum observable expectation values. If we denote by $p_i := \langle\psi|P_i|\psi\rangle$ the expected momentum in $i$-direction, this means that the expected momentum of a rotated system is given by $p_i' = \sum_j R_{ij}\, p_j = \sum_j R_{ij}\langle\psi|P_j|\psi\rangle$, where $R \in \mathrm{SO}(3)$ is an element of the rotation group. This result should agree with the expectation values for rotated system states, that is, $p_i' = \langle\psi'|P_i|\psi'\rangle = \langle\psi|U(R)^\dagger P_i U(R)|\psi\rangle$. As this argument is independent from the particular choice of state $|\psi\rangle$, and making use of the linearity of the operations, this implies a consistency

---

[1]Unitary representations are explained in Section 3. The notation $U$ for the operator is distinct from the notation U of the unitary group U($H$).

constraint $\sum_j R_{ij} P_j = U(R)^\dagger P_i U(R)$, which identifies the collection $(P_1, P_2, P_3)$ as a *vector operator*. Other geometric quantities are required to satisfy similar constraints: For instance, energy is a scalar (i.e., invariant) quantity and the Hamilton operator $H$ is a scalar operator, satisfying $H = U(R)^\dagger H U(R)$. Similarly, any matrix valued classical quantity corresponds to a rank $(1,1)$ Cartesian tensor operator $(M_{ij})_{i,j=1,2,3}$ subject to $\sum_{kl} R_{ik} M_{kl} (R^{-1})_{lj} = U(R)^\dagger M_{ij} U(R)$. The overarching framework to study such situations is the notion of a *representation operator*, which we define as a family of operators $(A_1, \ldots, A_N)$ which are required to satisfy the constraint

$$\sum_{j=1}^{N} \pi(g)_{ij} A_j = U(g)^\dagger A_i U(g) \qquad \forall\, g \in G, \tag{1}$$

where $\pi : G \to \mathrm{U}(\mathbb{C}^N)$ is some unitary representation of the symmetry group under consideration. The examples above correspond to specific choices of representations, namely the trivial representation $\pi(R) = 1$ for scalars, the "standard" representation $\pi(R) = R$ for vectors and the tensor product representation $\pi(R) = R \otimes (R^{-1})^\top$ for matrices. Spherical tensor operators, discussed below, correspond to the irreps (irreducible representations) of $SO(3)$.

**The Steerable Kernel Constraint**   Convolution kernels of group equivariant CNNs are required to satisfy a very similar constraint to that in Eq. (1). Before coming to such GCNNs, consider the case of conventional CNNs, processing image-like signals on a Euclidean space $\mathbb{R}^d$. Such signals are formalized as $c$-channel feature maps $f : \mathbb{R}^d \to \mathbb{K}^c$ that assign a $c$-dimensional feature vector $f(x) \in \mathbb{K}^c$ to each point $x \in \mathbb{R}^d$, where we allow for $\mathbb{K}$ being either of the real or complex numbers $\mathbb{R}$ or $\mathbb{C}$. Each CNN layer maps its input feature map $f_{\mathrm{in}} : \mathbb{R}^d \to \mathbb{K}^{c_{\mathrm{in}}}$ via a convolution to an output feature map $f_{\mathrm{out}} := K \star f_{\mathrm{in}} : \mathbb{R}^d \to \mathbb{K}^{c_{\mathrm{out}}}$. Since the convolution maps $c_{\mathrm{in}}$ input channels to $c_{\mathrm{out}}$ output channels, the kernel $K : \mathbb{R}^d \to \mathbb{K}^{c_{\mathrm{out}} \times c_{\mathrm{in}}}$ is matrix-valued.

Conventional CNNs are translation equivariant, however it is often desirable that the convolution is equivariant w.r.t. a larger symmetry group, for instance the isometries $\mathrm{E}(d)$ of $\mathbb{R}^d$ (Weiler & Cesa, 2019). For simplicity, we consider semidirect product groups of the form $(\mathbb{R}^d, +) \rtimes G$, where $G \leq \mathrm{GL}(d)$ is any compact group. Group elements $tg \in (\mathbb{R}^d, +) \rtimes G$ are uniquely split into a translation $t \in (\mathbb{R}^d, +)$ and an element $g \in G$, stabilizing the origin. They act on $\mathbb{R}^d$ according to $x \mapsto (tg) \cdot x := gx + t$. The equivariance of a GCNN – which is the analog to the symmetry of a quantum mechanical system – requires the feature spaces to be endowed with a group action of the symmetry group. A natural choice is to model the feature spaces as spaces of *feature fields*, for instance scalar, vector or tensor fields (Cohen & Welling, 2016b).

Such feature fields are defined as functions $f : \mathbb{R}^d \to V$, where the difference to conventional feature maps is that the space $V \cong \mathbb{K}^c$ of feature vectors is equipped with a group representation $\rho : G \to \mathrm{GL}(V)$ of the stabilizer $G$. The full symmetry group acts on feature fields according to $f \mapsto (tg) \cdot f := \rho(g) \circ f \circ (tg)^{-1}$, which is known as the induced representation of $\rho$. As proven in (Weiler et al., 2018a), the most general linear and equivariant map from an input field $f_{\mathrm{in}} : \mathbb{R}^d \to V_{\mathrm{in}}$ to an output field $f_{\mathrm{out}} : \mathbb{R}^d \to V_{\mathrm{out}}$ is a convolution with a $G$-steerable kernel $K : \mathbb{R}^d \to \mathrm{Hom}_{\mathbb{K}}(V_{\mathrm{in}}, V_{\mathrm{out}}) \cong \mathbb{K}^{c_{\mathrm{out}} \times c_{\mathrm{in}}}$. Such kernels take values in the space of *linear operators* from $V_{\mathrm{in}}$ to $V_{\mathrm{out}}$ and are required to satisfy the *$G$-steerability* (equivariance) constraint

$$K(gx) = \rho_{\mathrm{out}}(g) \circ K(x) \circ \rho_{\mathrm{in}}(g)^{-1} \qquad \forall\, g \in G, \ x \in \mathbb{R}^d. \tag{2}$$

One can easily check that a convolution with a $G$-steerable kernel $K$ is indeed equivariant, i.e., satisfies $K \star ((tg) \cdot f) = (tg) \cdot (K \star f)$ for any $tg \in (\mathbb{R}^d, +) \rtimes G$. This result was later generalized to feature fields on homogeneous spaces $H/G$ of unimodular locally compact groups $H$ (Cohen et al., 2019b) and on Riemannian manifolds with structure group $G$ (Cohen et al., 2019a). That the equivariance of the convolutional network requires $G$-steerable kernels in any of these settings underlines the great practical relevance of our results.

The two constraints, Eq. (1) and Eq. (2), are remarkably similar: the left-hand-sides are in both cases given by a $G$-transformation of the operator or kernel itself while the right-hand-sides are given by pre- and postcomposition of the operator or kernel with unitary representations. More details on this comparison can be found in Appendix C.1.3.

**The Wigner-Eckart Theorem for Spherical Tensor Operators**   All information about a linear operator $A : \mathcal{H} \to \mathcal{H}$ is encoded by its *matrix elements* $A_{\mu\nu} := \langle \mu | A | \nu \rangle \in \mathbb{C}$ relative to a given basis, where $|\nu\rangle \in \mathcal{H}$ and $\langle \mu | \in \mathcal{H}^*$ denote basis elements of the Hilbert space and its

dual. Similarly, all information about a convolution kernel $K$ is encoded by its matrix elements $K_{\mu\nu}(x) := \langle\mu|K(x)|\nu\rangle \in \mathbb{K}$, where $|\nu\rangle \in V_{\text{in}}$ and $\langle\mu| \in V_{\text{out}}^*$ are elements of chosen bases for the input representation and dual output representation. Considering general operators and kernels, i.e., ignoring the symmetry constraints in Eqs. (1) and (2), all matrix elements are independent degrees of freedom. In the case of convolution kernels, they correspond directly to the $c_{\text{out}} \cdot c_{\text{in}}$ learnable parameters for every point of the kernel. However, if $A$ is a representation operator – or if $K$ is a $G$-steerable kernel – the symmetry constraints couple the matrix elements to each other such that they can not be chosen freely anymore. For representation operators, this statement is made precise by the Wigner-Eckart theorem.

The Wigner-Eckart theorem is best known in its classical form, which applies specifically to *spherical tensor operators*. These operators are the representation operators for the irreps of $\mathrm{SO}(3)$, i.e., the Wigner D-matrices $D_j : \mathrm{SO}(3) \to \mathrm{U}(\mathbb{C}^{2j+1})$. As such, spherical tensor operators of rank $j$ are defined as families $\boldsymbol{T}_j = (T_j^{-j}, \ldots, T_j^j)^\top$ of $2j+1$ operators $T_j^m$ that satisfy the constraint $\sum_{n=-j}^j D_j^{mn}(g) T_j^n = U(g)^\dagger T_j^m U(g)$ for any $g \in \mathrm{SO}(3)$.

In order to express the operators $T_j^m$ in terms of matrix elements, we need to fix a basis of $\mathcal{H}$. Due to the $\mathrm{SO}(3)$-symmetry of $\boldsymbol{T}_j$, a natural choice are the angular momentum eigenstates[2] $|ln\rangle$, where $l \in \mathbb{N}_{\geq 0}$ and $n = -l, \ldots, l$. For fixed quantum numbers $j$, $l$, and $J$, there are $2j+1$ components $T_j^m$ of $\boldsymbol{T}_j$, $2l+1$ basis kets $|ln\rangle$, and $2J+1$ basis bras $\langle JM|$. This implies that there are $(2J+1)(2j+1)(2l+1)$ different matrix elements $\langle JM|T_j^m|ln\rangle \in \mathbb{C}$ for these quantum numbers. According to the Wigner-Eckart theorem, all of these matrix elements are fully specified by one single number (Jeevanjee, 2011):

**Theorem 2.1** (Wigner-Eckart theorem for Spherical Tensor Operators). *Let* $j, l, J \in \mathbb{N}_{\geq 0}$ *and let* $\boldsymbol{T}_j$ *be a spherical tensor operator of rank* $j$. *Then there is a unique complex number, the* reduced matrix element $\lambda \in \mathbb{C}$ *(often written* $\langle J\|\boldsymbol{T}_j\|l\rangle \in \mathbb{C}$*), that completely determines any of the* $(2J+1)(2j+1)(2l+1)$ *matrix elements* $\langle JM|T_j^m|ln\rangle$ *by the relation*

$$\langle JM|T_j^m|ln\rangle = \lambda \cdot \langle JM|jm; ln\rangle.$$

*The coupling coefficients* $\langle JM|jm; ln\rangle$, *known as* Clebsch-Gordan coefficients, *are given by the projection of the tensor product basis* $|jm; ln\rangle := |jm\rangle \otimes |ln\rangle$ *on* $|JM\rangle$. *They are purely algebraic and therefore independent of the spherical tensor operator* $\boldsymbol{T}_j$.

This result generalizes to arbitrary representation operators of the form in Eq. (1) (Agrawala, 1980). The similarities between representation operators and $G$-steerable kernels suggests that a similar statement might hold for the matrix elements of $G$-steerable kernels as well. As proven below, this is indeed the case: our generalized Wigner-Eckart theorem separates their independent degrees of freedom from purely algebraic relations between mutually dependent matrix elements. It does therefore give an explicit parametrization of the space of $G$-steerable kernels.

## 3 BUILDING BLOCKS OF STEERABLE KERNELS

This chapter gives a brief introduction to the mathematical concepts that are required to formulate our Wigner-Eckart theorem for $G$-steerable kernels. The first two of the following paragraphs explain why it is w.l.o.g. possible to restrict attention to steerable kernels on homogeneous spaces and to irreducible representations. The following three paragraphs discuss the building blocks of steerable kernels, which are *endomorphisms*, *harmonic basis functions* described by the Peter-Weyl theorem, and *tensor product representations* and their *Clebsch-Gordan decomposition*. An illustration of the concepts introduced in this chapter is given in Appendix A.

**The Restriction to Homogeneous Spaces**     Convolution kernels are usually defined on a Euclidean space $\mathbb{R}^d$, i.e., they are functions $K : \mathbb{R}^d \to \mathrm{Hom}_{\mathbb{K}}(V_{\text{in}}, V_{\text{out}})$. The $G$-steerability constraint in Eq. (2) relates kernel values $K(x)$ at $x$ to kernel values $K(gx)$ at all other points $gx$ on the *orbit* $Gx := \{gx \mid g \in G\}$ of $x$. To solve the constraint, it is therefore w.l.o.g. sufficient to consider restrictions of kernels to the individual orbits, from which the full solution on $\mathbb{R}^d$ can be assembled (Weiler et al., 2018a). By construction, the orbits have the structure of a *homogeneous space*:

---

[2]The system could in general have further quantum numbers, which we suppress here for simplicity.

**Definition 3.1** (Homogeneous Space, Transitive Action). Let $\cdot : G \times X \to X$ be a continuous action of a compact group $G$ on a topological space $X$. Then $X$ is called a *homogeneous space* w.r.t. $G$ if $\emptyset \neq X$ and if for all $x, y \in X$ there is a $g \in G$ such that $gx = y$. The action is then called *transitive*.

We will in the following w.l.o.g. consider steerable kernels $K : X \to \mathrm{Hom}_{\mathbb{K}}(V_{\mathrm{in}}, V_{\mathrm{out}})$ on such homogeneous spaces $X$.

**Restriction to Irreducible Unitary Representations**   The theorems below apply specifically to unitary representations, that is, representations for which the automorphisms $\rho(g)$ preserve distances (Knapp, 2002). As asserted by Theorem B.20, this is not really a restriction as every finite-dimensional linear representation can be considered as being unitary. Thus, we assume $\rho : G \to \mathrm{U}(V)$, where $\mathrm{U}(V)$ is the *unitary group*, i.e., the group of distance-preserving linear functions on $V$. In the case of $\mathbb{K} = \mathbb{R}$ we say *orthogonal* instead of unitary and write $\mathrm{O}(V)$.

Additionally, prior research has shown that it is sufficient to solve the kernel constraint in Eq. (2) for *irreducible* (unitary) input- and output representations instead of arbitrary finite-dimensional representations (Weiler & Cesa, 2019). This is possible due to the linearity of the constraint and the fact that any finite-dimensional unitary representation decomposes by Proposition B.38 into an orthogonal direct sum of irreps. The solution for general representations can thus be recovered from the solutions for irreps. More details on these considerations can be found in Section D.1.3.

If two unitary irreps are related by an isometric intertwiner, they are *isomorphic*; see Definition B.18. The set of isomorphism classes of unitary irreps of $G$ is denoted by $\widehat{G}$. We assume that for each isomorphism class $j \in \widehat{G}$ we have picked a representative irrep $\rho_j : G \to \mathrm{U}(V_j)$. We denote by $d_j$ the dimension of $V_j$, so that we have $V_j \cong \mathbb{K}^{d_j}$.

Overall, we can w.l.o.g. replace $\mathbb{R}^d$ with $X$ and $\rho_{\mathrm{in}}$ and $\rho_{\mathrm{out}}$ by $\rho_l : G \to \mathrm{U}(V_l)$ and $\rho_J : G \to \mathrm{U}(V_J)$, where $X$ is a homogeneous space and $\rho_l$ and $\rho_J$ are (representatives of isomorphism classes of) irreducible unitary representations of $G$. This leads to our working definition of steerable kernels, to which we restrict from now on:

**Definition 3.2** (Steerable Kernel on a Homogeneous Space w.r.t. Unitary Irreps). Let $X$ be a homogeneous space of $G$ and $\rho_l : G \to \mathrm{U}(V_l)$ and $\rho_J : G \to \mathrm{U}(V_J)$ be representatives of isomorphism classes of irreducible unitary representations of $G$. A *G-steerable kernel* (on a homogeneous space and w.r.t. unitary irreps) is any function $K : X \to \mathrm{Hom}_{\mathbb{K}}(V_l, V_J)$ such that the following $G$-steerability constraint holds:

$$K(gx) = \rho_J(g) \circ K(x) \circ \rho_l(g)^{-1} \qquad \forall\, g \in G,\ x \in X. \tag{3}$$

**Endomorphisms**   An important concept, underlying the reduced matrix elements in the Wigner-Eckart theorem for spherical tensor operators, is that of endomorphisms of linear representations.

**Definition 3.3** (Endomorphism of a of Linear Representation). Let $\rho : G \to \mathrm{GL}(V)$ be a linear representation. An *endomorphism* of $\rho$ is a linear map $c : V \to V$ which satisfies $c \circ \rho(g) = \rho(g) \circ c$ for all $g \in G$. The space of all endomorphisms of $\rho$ is written $\mathrm{End}_{G,\mathbb{K}}(V)$.

Endomorphisms play a central role in our generalized Wigner-Eckart theorem for steerable kernels. To get an insight why this is the case, consider a given steerable kernel $K : X \to \mathrm{Hom}_{\mathbb{K}}(V_l, V_J)$. The post-composition $(c \circ K)(x) := c \circ (K(x))$ of this kernel with *any* endomorphism $c \in \mathrm{End}_{G,\mathbb{K}}(V_J)$ is obviously still steerable, i.e., satisfies Eq. (3). A basis of the space of steerable kernels is therefore partly explained by *bases of the endomorphism spaces*, and thus occurs in our general solution. In the following, we write $\{c_r \mid r = 1, \ldots, E_J\}$ for the basis of $\mathrm{End}_{G,\mathbb{K}}(V_J)$, where $E_J := \dim(\mathrm{End}_{G,\mathbb{K}}(V_J))$ is the dimension of the endomorphism space.[3]

**The Peter-Weyl Theorem and Harmonic Basis Functions**   A cornerstone in our proof of the Wigner-Eckart theorem for steerable kernels is Theorem C.7. It states that the space of steerable kernels, which are $G$-equivariant maps $K : X \to \mathrm{Hom}_{\mathbb{K}}(V_l, V_J)$, is isomorphic to the space of

---

[3]For $\mathbb{K} = \mathbb{C}$, Schur's Lemma D.8 tells us that the endomorphism spaces of irreducible representations are always 1-dimensional, generated by the identity. For $\mathbb{K} = \mathbb{R}$, however, one can show that they have either 1, 2, or 4 dimensions, see Bröcker & Dieck (2003), Theorem II.6.3.

*linear $G$-equivariant maps of the form* $\widehat{K} : L^2_{\mathbb{K}}(X) \to \mathrm{Hom}_{\mathbb{K}}(V_l, V_J)$. We are therefore interested in the representation theory of $L^2_{\mathbb{K}}(X)$, which is described by the *Peter-Weyl theorem*.[4]

**Theorem 3.4** (Peter-Weyl Theorem, Existence of Harmonic Basis Functions). *Let $G$ be a compact group and $X$ a homogeneous space. Let $\widehat{G}$ be the set of isomorphism classes of irreducible representations. For $j \in \widehat{G}$, let $\rho_j : G \to \mathrm{U}(V_j)$ be a representative with dimension $d_j = \dim(V_j)$. Then there are multiplicities $m_j \in \mathbb{N}_{\geq 0}$ with $m_j \leq d_j$, and for each $i = 1, \ldots, m_j$ there are* harmonic basis functions $Y^m_{ji} : X \to \mathbb{K}$, $m = 1, \ldots, d_j$, *such that the following three properties hold:*

1. *The $Y^m_{ji}$, for fixed $j$ and $i$, are* steerable *(Freeman & Adelson, 1991; Hel-Or & Teo, 1998), i.e., transformation via $g \in G$ can be expressed by shifting basis coefficients with $\rho_j$:*
   $$Y^m_{ji}(g^{-1}x) = \left( \sum_{m'=1}^{d_j} \rho^{m'm}_j(g) \, Y^{m'}_{ji} \right)(x).$$

2. *Any square-integrable function $f : X \to \mathbb{K}$ can be uniquely expanded in terms of harmonic basis functions, i.e., $f = \sum_{j \in \widehat{G}} \sum_{i=1}^{m_j} \sum_{m=1}^{d_j} \lambda_{jim} \, Y^m_{ji}$ with coefficients $\lambda_{jim} \in \mathbb{K}$.*

3. *The $Y^m_{ji}$ are an orthonormal system with respect to the scalar product given by integration:*
   $\int_X \overline{Y^m_{ji}(x)} \, Y^{m'}_{j'i'}(x) \, dx = \delta_{jj'} \, \delta_{ii'} \, \delta_{mm'}$.[5]

Note the similarity of these properties to those encountered in usual Fourier analysis. Indeed, the Peter-Weyl theorem can be viewed as describing the *harmonic analysis* on arbitrary compact groups and their homogeneous spaces.

**Tensor Products and Clebsch-Gordan Coefficients**   The last ingredients that we need to discuss are tensor product representations and Clebsch-Gordan coefficients. They appear, roughly speaking, in the following way: the kernel $K$ can be thought of as being built from harmonic basis functions $Y^m_{ji}$ which transform according to the corresponding irrep $\rho_j$. When a harmonic kernel component of type $\rho_j$ acts on an input feature field of type $\rho_l$, the combination will transform according to their tensor product $\rho_j \otimes \rho_l$. If the convolution should map to an output field of type $\rho_J$, not any harmonic component $Y^m_{ji}$ is admissible, but only those for which $\rho_J$ appears as a subrepresentation in the tensor product $\rho_j \otimes \rho_l$. The Clebsch-Gordan coefficients encode whether $\rho_j \otimes \rho_l$ contains $\rho_J$, and, if it does, in which way and how often $\rho_J$ is embedded in the tensor product. For more details on the definitions in this section see Appendix D.1.

**Definition 3.5** (Tensor product representation). Let $\rho : G \to \mathrm{U}(V)$ and $\tilde{\rho} : G \to \mathrm{U}(\tilde{V})$ be unitary representations. Then their tensor product $\rho \otimes \tilde{\rho} : G \to \mathrm{U}(V \otimes \tilde{V})$ is defined by: $\left[ (\rho \otimes \tilde{\rho})(g) \right](v \otimes \tilde{v}) = \left[ \rho(g) \right](v) \otimes \left[ \tilde{\rho}(g) \right](\tilde{v})$.

The tensor product $\rho_j \otimes \rho_l$ of two irreps is itself in general *not* irreducible anymore. However, as it is again a unitary representation, it splits by Proposition B.38 into a direct sum of irreducible unitary subrepresentations. Thus, there is an equivariant isomorphism

$$\mathrm{CG}_{jl} : V_j \otimes V_l \to \bigoplus_{J \in \widehat{G}} \bigoplus_{s=1}^{[J(jl)]} V_J. \tag{4}$$

The integer $[J(jl)]$ is the multiplicity of $V_J$ in $V_j \otimes V_l$, which is zero for all but finitely many $J$.

The matrix elements of $\mathrm{CG}_{jl}$ are denoted as Clebsch-Gordan coefficients:

**Definition 3.6** (Clebsch-Gordan Coefficients). Let $Y^m_j \otimes Y^n_l$ be the basis tensors in $V_j \otimes V_l$ and let the basis element $Y^M_{Js}$ be the copy of $Y^M_J$ with index $s$ in $\bigoplus_{J \in \widehat{G}} \bigoplus_{s=1}^{[J(jl)]} V_J$. Then the *Clebsch-Gordan coefficients* are the matrix elements of $\mathrm{CG}_{jl}$ relative to these bases,

$$\langle s, JM | jm; ln \rangle := \langle Y^M_{Js} | \mathrm{CG}_{jl} | Y^m_j \otimes Y^n_l \rangle,$$

i.e., the scalar product of $\mathrm{CG}_{jl}(Y^m_j \otimes Y^n_l)$ and $Y^M_{Js}$.

---

[4]Usually, the Peter-Weyl theorem uses $G$ itself as the homogeneous space and is formulated for complex representations (Knapp, 2002). However, generalizations to arbitrary homogeneous spaces and real representations are possible, as we explain in Appendix B.2

[5]From a representation theoretic viewpoint, the functions $Y^m_{ji}$ for fixed $j$ and $i$ span an irreducible subrepresentation $V_{ji}$ of the unitary representation $\lambda : G \to \mathrm{U}(L^2_{\mathbb{K}}(X))$ given by $\left[ \lambda(g) f \right](x) := f(g^{-1}x)$. $L^2_{\mathbb{K}}(X)$ then splits into an orthogonal direct sum $L^2_{\mathbb{K}}(X) = \widehat{\bigoplus}_{j \in \widehat{G}} \bigoplus_{i=1}^{m_j} V_{ji}$. This viewpoint is explained in the equivalent, more representation theoretic formulation of the Peter-Weyl theorem in Theorem B.22.

## 4 A WIGNER-ECKART THEOREM FOR $G$-STEERABLE KERNELS

Now that we have discussed all of the required ingredients, we are ready for stating our main theorem. Intuitively, our Wigner-Eckart theorem identifies exactly those combinations of harmonics, Clebsch-Gordan coefficients and endomorphisms that, when being assembled together, yield a $G$-steerable kernel $K : X \to \operatorname{Hom}_{\mathbb{K}}(V_l, V_J)$. The kernel will thereby comprise all those harmonics $Y_{ji}^m$ for which the tensor product $V_j \otimes V_l$ contains $V_J$ as a factor. The number of possible combinations depends therefore on the number of different isomorphism classes $j \in \widehat{G}$ for which $V_J$ appears as a factor in the tensor product, the multiplicity $[J(jl)]$ with which it occurs, and the multiplicities $m_j$ of harmonics $Y_{ji}^m$ in the Peter-Weyl decomposition that transform according to $\rho_j$. In addition, each individual combination can subsequently be composed with an endomorphism in $\operatorname{End}_{G,\mathbb{K}}(V_J)$, which increases the number of combinations by a factor of $E_J = \dim(\operatorname{End}_{G,\mathbb{K}}(V_J))$ to a total of $\Lambda_{Jl} := E_J \cdot \sum_{j \in \widehat{G}} [J(jl)] \cdot m_j$. This number is finite, as we explain in Remark D.18.

How are such assembled steerable kernels parameterized? The learnable parameters correspond to the degrees of freedom in the individual components from which the kernel is built. While the Clebsch-Gordan coefficients and harmonic basis functions are fixed, the endomorphisms are elements of the $E_J$-dimensional vector spaces $\operatorname{End}_{G,\mathbb{K}}(V_J)$. The degrees of freedom of a $G$-steerable kernel are therefore identified with the choice of endomorphisms.[6] This gives a total of $\Lambda_{Jl}$ parameters which take values in $\mathbb{K}$. Note that the choice of endomorphisms corresponds directly to the choice of reduced matrix elements of spherical tensor operators.

For a kernel $K : X \to \operatorname{Hom}_{\mathbb{K}}(V_l, V_J)$, we write $\langle JM|K(x)|ln\rangle$ for the matrix elements of $K(x) \in \operatorname{Hom}_{\mathbb{K}}(V_l, V_J)$ with indices $n \le d_l$ and $M \le d_J$, see also Definition D.9. Similarly, endomorphisms $c \in \operatorname{End}_{G,\mathbb{K}}(V_J)$ have matrix elements $\langle JM|c|JM'\rangle$ with $M, M' \le d_J$. We furthermore write $\langle i, jm|x\rangle := \overline{Y_{ji}^m(x)}$. Finally, we denote the space of $G$-steerable kernels by $\operatorname{Hom}_G(X, \operatorname{Hom}_{\mathbb{K}}(V_l, V_J))$.

Our main result is the following Wigner-Eckart theorem for $G$-steerable kernels. Other versions at different levels of abstraction can be found in Theorems D.13 and D.16.

**Theorem 4.1** (Wigner-Eckart Theorem for Steerable Kernels). *There is a vector space isomorphism*

$$\operatorname{GKer} : \bigoplus_{j \in \widehat{G}} \bigoplus_{i=1}^{m_j} \bigoplus_{s=1}^{[J(jl)]} \operatorname{End}_{G,\mathbb{K}}(V_J) \to \operatorname{Hom}_G(X, \operatorname{Hom}_{\mathbb{K}}(V_l, V_J)). \quad (5)$$

*A general steerable kernel $K = \operatorname{GKer}((c_{jis})_{jis})$ with $c_{jis} \in \operatorname{End}_{G,\mathbb{K}}(V_J)$ has matrix elements*

$$\underbrace{\langle JM|K(x)|ln\rangle}_{\text{kernel matrix elements}} = \sum_{j \in \widehat{G}} \sum_{i=1}^{m_j} \sum_{s=1}^{[J(jl)]} \sum_{m=1}^{d_j} \sum_{M'=1}^{d_J} \underbrace{\langle JM|c_{jis}|JM'\rangle}_{\text{endomorphisms}} \cdot \underbrace{\langle s, JM'|jm; ln\rangle}_{\text{Clebsch-Gordan}} \cdot \underbrace{\langle i, jm|x\rangle}_{\text{harmonics}}. \quad (6)$$

*Proof.* We shortly sketch a proof of this theorem. We use the notation $\operatorname{Hom}_{G,\mathbb{K}}$ to denote *linear* equivariant maps. The space of steerable kernels can be progressively transformed as follows:

$$\operatorname{Hom}_G(X, \operatorname{Hom}_{\mathbb{K}}(V_l, V_J)) \overset{(1)}{\cong} \operatorname{Hom}_{G,\mathbb{K}}(L_{\mathbb{K}}^2(X), \operatorname{Hom}_{\mathbb{K}}(V_l, V_J))$$

$$\overset{(2)}{\cong} \operatorname{Hom}_{G,\mathbb{K}}\left(\widehat{\bigoplus}_{j \in \widehat{G}} \bigoplus_{i=1}^{m_j} V_{ji}, \operatorname{Hom}_{\mathbb{K}}(V_l, V_J)\right) \overset{(3)}{\cong} \bigoplus_{j \in \widehat{G}} \bigoplus_{i=1}^{m_j} \operatorname{Hom}_{G,\mathbb{K}}(V_j, \operatorname{Hom}_{\mathbb{K}}(V_l, V_J))$$

$$\overset{(4)}{\cong} \bigoplus_{j \in \widehat{G}} \bigoplus_{i=1}^{m_j} \operatorname{Hom}_{G,\mathbb{K}}(V_j \otimes V_l, V_J) \overset{(5)}{\cong} \bigoplus_{j \in \widehat{G}} \bigoplus_{i=1}^{m_j} \operatorname{Hom}_{G,\mathbb{K}}\left(\bigoplus_{J' \in \widehat{G}} \bigoplus_{s=1}^{[J'(jl)]} V_{J'}, V_J\right)$$

$$\overset{(6)}{\cong} \bigoplus_{j \in \widehat{G}} \bigoplus_{i=1}^{m_j} \bigoplus_{s=1}^{[J(jl)]} \operatorname{Hom}_{G,\mathbb{K}}(V_J, V_J) \overset{(7)}{\cong} \bigoplus_{j \in \widehat{G}} \bigoplus_{i=1}^{m_j} \bigoplus_{s=1}^{[J(jl)]} \operatorname{End}_{G,\mathbb{K}}(V_J)$$

In (1), we *linearize* the kernels such that they become representation operators, as detailed in Theorem C.7. Step (2) applies the representation-theoretic version of the Peter-Weyl Theorem B.22 to decompose $L_{\mathbb{K}}^2(X)$ in harmonic basis functions. Step (3) makes use of the well-known fact that linear maps can be described on each direct summand individually. Topological details are explained

---

[6]This statement is made precise by the isomorphism $\operatorname{GKer}$, defined in Eq. (5) in Theorem 4.1.

in Lemma D.20. In (4), we use the hom-tensor adjunction Proposition D.23. In (5), we use the Clebsch-Gordan decomposition Eq. (4), which provides us with Clebsch-Gordan coefficients. In (6), we use that nontrivial linear equivariant maps from $V_{J'}$ to $V_J$ exist by Schur's Lemma B.29 only for $J' = J$ and, once again, that we can describe linear maps on each direct summand individually. Finally, in (7) we note that $\operatorname{Hom}_{G,\mathbb{K}}(V_J, V_J) = \operatorname{End}_{G,\mathbb{K}}(V_J)$ is the space of endomorphisms. The formula of the matrix elements Eq. (6) is fully proven in Theorem D.13 by carefully tracing back all the isomorphisms above.[7]

Technically, step (1) is the main gap that we had to bridge: it establishes that non-linear kernels on $X$ can be seen as linear representation operators on $L^2_{\mathbb{K}}(X)$. Steps (2) to (7) orient at the proof of the Wigner-Eckart theorem for representation operators by Agrawala (1980). However, it differs non-trivially from the reference by a) allowing the operator to be non-injective, b) topological considerations, since $L^2_{\mathbb{K}}(X)$ is not simply a direct sum of irreps but its *topological closure*, and c) the possibility to allow for real representations, which is why we end up with endomorphisms. $\qquad\square$

We obtain the following corollary, which clarifies how steerable kernels can be parameterized:

**Corollary 4.2.** *The space* $\operatorname{Hom}_G(X, \operatorname{Hom}_{\mathbb{K}}(V_l, V_J))$ *of steerable kernels is spanned by basis kernels* $\{K_{jisr} : X \to \operatorname{Hom}_{\mathbb{K}}(V_l, V_J) \mid j \in \widehat{G}, \ i \leq m_j, \ s \leq [J(jl)], \ r \leq E_J\}$ *with matrix elements*

$$\langle JM|K_{jisr}(x)|ln\rangle \ = \ \sum_{m=1}^{d_j} \sum_{M'=1}^{d_J} \langle JM|c_r|JM'\rangle \cdot \langle s, JM'|jm; ln\rangle \cdot \langle i, jm|x\rangle. \quad (7)$$

*A matrix-expression of the basis kernels from Eq. (7) is given in Eq. (22). Here, $c_r$ is one of the $E_J$ basis endomorphisms of* $\operatorname{End}_{G,\mathbb{K}}(V_J)$. *This means that a general steerable kernel $K : X \to \operatorname{Hom}_{\mathbb{K}}(V_l, V_J)$ is of the form $K = \sum_{j \in \widehat{G}} \sum_{i=1}^{m_j} \sum_{s=1}^{[J(jl)]} \sum_{r=1}^{E_J} \lambda_{jisr} \cdot K_{jisr}$ with a total of $\Lambda_{Jl} = E_J \cdot \sum_{j \in \widehat{G}} [J(jl)] \cdot m_j$ learnable parameters $\lambda_{jisr} \in \mathbb{K}$. Overall, the kernel space can therefore be parameterized with an isomorphism* $\overline{\operatorname{GKer}} : \mathbb{K}^{\Lambda_{Jl}} \to \operatorname{Hom}_G(X, \operatorname{Hom}_{\mathbb{K}}(V_l, V_J))$.

*Proof.* We simply choose $K_{jisr} := \operatorname{GKer}((c^{jisr}_{j'i's'})_{j'i's'})$ with $c^{jisr}_{j'i's'} = \delta_{j'j} \cdot \delta_{i'i} \cdot \delta_{s's} \cdot c_r$. Clearly, the $\boldsymbol{c}^{jisr}$ are a basis of $\bigoplus_{j \in \widehat{G}} \bigoplus_{i=1}^{m_j} \bigoplus_{s=1}^{[J(jl)]} \operatorname{End}_{G,\mathbb{K}}(V_J)$, and since $\operatorname{GKer}$ is an isomorphism, the $K_{jisr}$ form a basis of steerable kernels. $\qquad\square$

*Remark* 4.3. The matrix elements $\langle JM|c_{jis}|JM'\rangle$ relate to the reduced matrix elements $\lambda \in \mathbb{C}$ of spherical tensor operators as follows: in the case of spherical tensor operators one deals with complex irreps, whose endomorphism spaces are according to Schur's Lemma D.8 generated by the identity. Consequently, such endomorphisms $c$ have matrix-elements $\langle JM|c|JM'\rangle = \lambda\,\delta_{MM'}$ for some scaling factor $\lambda \in \mathbb{C}$. $\lambda$ is denoted as the *reduced matrix element* of the spherical tensor operator. The analog to $\lambda$ in our Wigner-Eckart theorem are the learnable parameters $\lambda_{jisr} \in \mathbb{K}$.

## 5 RELATED WORK

Harmonic convolution kernels date back to at least the early '80s (Hsu & Arsenault, 1982; Rosen & Shamir, 1988). The term *steerable filter* was coined in Freeman & Adelson (1991). Hel-Or & Teo (1998) generalized steerable filters to Lie groups. Reisert & Burkhardt (2007) proposed matrix valued steerable kernels between representation spaces, which are similar to our $G$-steerable kernels.

Steerable CNNs formulate GCNNs in the language of representation theory and feature fields. This design was proposed by Cohen & Welling (2016b), who specifically considered finite groups, for which the kernel constraint can be solved numerically. Weiler et al. (2018a) introduced the $G$-steerability constraint in the form in Eq. (2) for $G = \mathrm{SO}(3)$. The authors choose a slightly different approach to solve the constraint in which they decompose the space $\operatorname{Hom}_{\mathbb{R}}(V_l, V_J) \cong V_l^* \otimes V_J$ instead of $V_j \otimes V_l$ via Clebsch-Gordan coefficients. An essentially equivalent design was simultaneously proposed by Thomas et al. (2018), who decomposed $V_j \otimes V_l$ as in the present work, see Appendix E.5. The case of complex valued irreps of $\mathrm{SO}(2)$ was investigated by Worrall et al. (2016) and Wiersma et al. (2020); see Appendix E.1. Weiler & Cesa (2019) solve the constraint for any,

---

[7]In Eq. (6) we see terms $\langle i, jm|x\rangle$ which are not present in the original Wigner-Eckart Theorem 2.1. They appear through the linearization of steerable kernels by Theorem C.7.

not necessarily irreducible, representation of the groups $O(2)$, $SO(2)$, $D_N$ and $C_N$. Their solution strategy is based on an expansion of the kernel in the Fourier basis of $L^2_{\mathbb{R}}(S^1)$ and solving for the Fourier coefficients satisfying the constraint. This is a special case of the strategy that we employ in the proof of our Wigner-Eckart theorem. de Haan et al. (2020) solve for $SO(2)$-steerable kernels by viewing them as invariants of the tensor product representation $L^2_{\mathbb{R}}(S^1)^* \otimes V_l^* \otimes V_J$. As they use real valued irreps, they can use that the duals are isomorphic to their original counterparts. Our Wigner-Eckart theorem unifies all of these results in one general framework.

To which use cases does the proposed kernel space solution apply? As argued by Cohen et al. (2019b), *any $H$-equivariant convolutional network* on a homogeneous space $H/G$ needs to satisfy a $G$-steerability constraint — if $H$ is locally compact and unimodular. While these works proved the necessity of steerable kernels, they *did not solve the constraint* – a gap which is filled by our Wigner-Eckart theorem for compact groups $G$, see also Remark D.15. This framework includes in particular the popular *group convolutions* on flat spaces (Cohen & Welling, 2016a) and homogeneous spaces of compact groups (Kondor & Trivedi, 2018) and Lie groups (Bekkers, 2020), including for instance the sphere (Cohen et al., 2018). Specifically, if $\rho_{\text{in}}$ and $\rho_{\text{out}}$ are chosen to be regular representations $L^2_{\mathbb{K}}(G)$, steerable convolutions are *equivalent* to group convolutions (Weiler & Cesa, 2019).

A related line of work are *Clebsch-Gordan Networks* (Kondor et al., 2018; Kondor, 2018; Anderson et al., 2019; Bogatskiy et al., 2020). They apply bilinear equivariant nonlinearities which compute the tensor products of global irrep features. A subsequent Clebsch-Gordan decomposition disentangles the product features back into irrep features. Note that in this network design, the Clebsch-Gordan coefficients are used in the *nonlinear* part, which differs from our use of these coefficients in the construction of steerable basis kernels, i.e. in the *linear* part of the network.

## 6 EXAMPLE APPLICATIONS

Cohen et al. (2019b) showed in a fairly general setting that every GCNN is based on $G$-steerable kernels. In practice, a basis for the space of $G$-steerable kernels needs to be determined for parameterizing GCNNs. This work determines the general structure of these basis kernels for compact (point-)symmetry groups $G$ and their homogeneous spaces $X$: Corollary 4.2 explains that one needs to determine 1) the *irreps* $\rho_l$ of $G$, 2) harmonic basis functions $Y_{ji}^m$ in $L^2_{\mathbb{K}}(X)$ according to the Peter-Weyl Theorem 3.4, 3) the *Clebsch-Gordan* decomposition of $V_j \otimes V_l$, given by the Clebsch-Gordan coefficients $\langle s, JM|jm; ln\rangle$, and 4) a basis of *endomorphisms* $c_r \in \text{End}_{G,\mathbb{K}}(V_J)$ for any $J \in \widehat{G}$. Given these ingredients, they can in a fifth step be put together according to Eq. (7) to obtain a complete, $\Lambda_{Jl}$-dimensional basis of $G$-steerable kernels $K_{jisr} : X \to \text{Hom}_{\mathbb{K}}(V_l, V_J)$.

Appendix E demonstrates this for the examples of $G$ being $SO(2)$, $SO(3)$, $O(3)$, and $\mathbb{Z}/2$, considering both real and complex irreps. In any of these cases, we derive the kernel bases following *exactly* the five steps outlined above. This procedure can easily be applied to further compact groups, for instance $SU(2)$ or $SU(3)$, which play an important role in physics applications of deep learning.

## 7 CONCLUSIONS AND FUTURE WORK

Prior work revealed that group equivariant convolutions generally rely on $G$-steerable kernels. Our Wigner-Eckart theorem for $G$-steerable kernels characterizes them for the practically relevant case of $G$ being any compact group. The degrees of freedom – or learnable parameters – correspond thereby precisely to the choice of endomorphisms. This mirrors the situation in quantum mechanics, where the degrees of freedom of spherical tensor operators are given by reduced matrix elements.

It would be desirable to extend this result to non-compact groups, where the Peter-Weyl Theorem does not hold anymore. One alternative might be Pontryagin duality (Reiter, 1968), which describes the Fourier transform on locally compact *abelian* groups. Furthermore, for many non-compact, *nonabelian* groups, one can often find a *direct integral decomposition* of $L^2_{\mathbb{C}}(G)$. This generalization of the Peter-Weyl theorem can be found in Segal (1950) and Mautner (1955). Such generalizations of our Wigner-Eckart Theorem might lead to a better theoretical understanding of several recent work (Worrall & Welling, 2019; Bekkers, 2020; Sosnovik et al., 2020; Shutty & Wierzynski, 2020).

Finally, we hope that the analogies between steerable kernels and representation operators appearing in physics inspire further research in this fascinating crossdisciplinary domain. This could lead to applications of GCNNs for learning tasks with physical symmetries.

ACKNOWLEDGMENTS

We thank Lucas Lang for discussions on the Wigner-Eckart Theorem and observables in physics and Patrick Forré for discussions on the link between steerable kernels and representation operators. Additionally, we are greatful for discussions with Gabriele Cesa on the connection between real and complex representations of compact groups. Furthermore, we thank Stefan Dawydiak and Terrence Tao for online discussions on aspects surrounding a real version of the Peter-Weyl theorem. Finally, we thank Roberto Bondesan, Miranda Cheng, Tom Lieberum, and Rupert McCallum for feedback on different aspects of our work.

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

# APPENDIX

This appendix contains a detailed and rigorous treatment of the Wigner-Eckart theorem for steerable kernels, including background knowledge, proofs, and many example applications.

In Chapter A, we shortly look at the simple example $SO(2)$ for motivating the concepts and results in Section 3.

Everything afterwards, starting with Chapter B, can be read independently of the main paper and is a self-contained treatment of our investigations. In Chapter B, we start with the foundations of the representation theory of compact groups. We formulate the Peter-Weyl Theorem B.22, which tells us how to decompose the space of square-integrable functions on a homogeneous space into irreducible representations, leading to harmonic basis functions. In the second half, we include a proof of the more algebraic parts of this theorem. We do this since the theorem is usually only proven for complex representations in the literature, but we need it for real representation as well.

In Chapter C we investigate steerable kernels and show their similarities to representation operators from physics and representation theory. In Theorem C.7 we will then proof a precise isomorphism between steerable kernels and representation operators on the space of square-integrable functions on a homogeneous space. We call these *kernel operators*.

In Chapter D we will then formulate and prove the Wigner-Eckart theorem for steerable kernels of general compact groups D.13. The proof makes in essential parts use of the Peter-Weyl Theorem and Theorem C.7, and additionally of Schur's Lemma B.29, the hom-tensor adjunction Proposition D.23, and the Clebsch-Gordan decomposition of tensor products.

In Chapter E, we then look at specific example applications of our theory. In these examples, we look at specific compact transformation groups $G$, specific, relevant homogeneous spaces $X$ of the group and one of the fields $\mathbb{R}$ or $\mathbb{C}$. For this combination we derive a basis for the space of steerable kernels between arbitrary irreducible input- and output representations of the group. Specifically, we look at harmonic networks (Worrall et al., 2016), $SO(2)$-equivariant networks for real representations (Weiler & Cesa, 2019), $\mathbb{Z}_2$-equivariant networks for real representations, $SO(3)$-equivariant networks for both real and complex representations (Weiler et al., 2018a; Thomas et al., 2018), and $O(3)$-equivariant networks for both real and complex representations. The investigation of $\mathbb{Z}_2$-equivariant CNNs will additionally show that our result is consistent with group convolutional CNNs for the regular representation (Cohen & Welling, 2016a).

In Chapter F, we summarize some important notions and results from the theory of topological spaces, metric spaces, normed vector spaces, and (pre-)Hilbert spaces that we use throughout this appendix.

Chapters B, C, and D contain the bulk of the theoretical work. We recommend the reader to first only read the first halves of these chapters, Sections B.1, C.1 and D.1, since they contain the formulation of the most important results and the main intuitions, whereas the second halves of these chapters, i.e., Sections B.2, C.2 and D.2, mainly contain detailed proofs that can be skipped when going over the material for the first time.

## CONTENTS OF THE APPENDIX

## LIST OF SYMBOLS

### GENERAL SET THEORY AND FUNCTIONS

| | |
|---|---|
| $A \cap B$ | intersection of sets $A$ and $B$ |
| $A \cup B$ | union of sets $A$ and $B$ |
| $\bigcap_{i \in I} A_i$ | intersection of sets $A_i$ |
| $\bigcup_{i \in I} A_i$ | union of sets $A_i$ |
| $\bigsqcup_{i \in I} A_i$ | union of sets $A_i$ which are disjoint from each other |
| $A \subseteq B$ | $A$ is a subset of $B$ |
| $A \subsetneq B$ | $A$ is a strict subset of $B$ |
| $A \setminus B$ | set of all elements in $A$ which are not in $B$ |
| $A \times B$ | Cartesian product of sets or structures (e.g., groups) $A, B$ |
| $\emptyset$ | empty set |
| $X := Y$ | $X$ is defined as $Y$ |
| $\sim$ | often an equivalence relation |
| $[x]$ | equivalence class with respect to an equivalence relation |
| $\mathbf{1}_A$ | indicator function of set $A$ |
| $f \circ g$ | composition of two composable functions $f$ and $g$ |
| $f^{-1}$ | either the inverse of function $f$ or the preimage function |
| $f|_A$ | restriction of a function $f$ to a subset $A$ |

### NUMBERS AND COLLECTIONS OF NUMBERS

| | |
|---|---|
| $\mathbb{N}$ | natural numbers including $0$ |
| $\mathbb{Z}$ | integers |
| $\mathbb{R}$ | field of real numbers |
| $\mathbb{C}$ | field of complex numbers |
| $\mathbb{H}$ | skew-field of quaternions |
| $\mathbb{K}$ | one of the two fields $\mathbb{R}$ and $\mathbb{C}$ |
| $\mathbb{K}^n$ | $n$-dimensional canonical vector space over $\mathbb{K}$ |
| $\overline{x}$ | complex conjugate of $x$ |

GROUPS

| | |
|---|---|
| $G$ | a compact topological group |
| $1, e$ | neutral element of a group with multiplication as operation |
| $0$ | neutral element of an additive group |
| $G \rtimes H$ | semidirect product of two groups $G$ and $H$ |
| $C_N$ | group of planar rotations of a regular $N$-gon |
| $D_N$ | group of planar rotations and reflections of a regular $N$-gon |
| $SO(d)$ | special orthogonal group in $d$ real dimensions |
| $O(d)$ | orthogonal group in $d$ real dimensions |
| $O(V)$ | orthogonal group of a *real* Hilbert space $V$ |
| $SU(d)$ | special unitary group in $d$ complex dimensions |
| $U(d)$ | unitary group in $d$ complex dimensions |
| $U(V)$ | unitary group of a *complex* Hilbert space $V$ |
| $E(d)$ | Euclidean motion group in $d$ dimensions |

BASIC REPRESENTATION THEORY

| | |
|---|---|
| $\rho$ | a linear representation of a group |
| $\rho^v$ | The function $G \to V$, $g \mapsto \rho(g)(v)$ |
| $\rho^{uv}$ | matrix coefficient of the unitary representation $\rho$ |
| $\rho^{\text{in}}, \rho^{\text{out}}$ | representations of the in-field and out-field, respectively |
| $\rho_{\text{Hom}}$ | Hom-representation on $\text{Hom}_{\mathbb{K}}(V, V')$ of representations $\rho$ and $\rho'$ |
| $\rho \otimes \rho'$ | tensor product representation on $V \otimes V'$ of representations $\rho$ and $\rho'$ |
| $\text{Ind}_G^H \rho$ | induced representation on $H$ or a representation $\rho$ on $G$ |
| $\widehat{G}$ | set of isomorphism classes of unitary representations on $G$ |
| $l$ | an isomorphism class of unitary representations |
| $\rho_l$ | a representative of isomorphism class $l$ |
| $V_l$ | vector space on which $\rho_l$ acts |
| $v_l^i$ or $Y_l^n$ | fixed chosen orthonormal basis vector of $V_l$ |

VECTOR SPACES AND HILBERT SPACES

| | |
|---|---|
| $\dim(V)$ | dimension of $\mathbb{K}$-vectorspace $V$ |
| $V \perp W$ | $V$ and $W$ are perpendicular |
| $V \cong W$ | $V$ and $W$ are isomorphic with respect to their structures |
| $V \ncong W$ | $V$ and $W$ are not isomorphic with respect to their structures |
| $\langle f|g \rangle$ | bra-ket notation of a scalar product on a Hilbert space |
| $\langle y|f|x \rangle$ | equivalent to $\langle y|f(x) \rangle$ for a function $f$ |
| $\text{null}(f)$ | null space of $f$ |
| $\text{im}(f)$ | image of $f$ |
| $f^*$ | adjoint of the operator $f$ |
| $\text{id}_V$ | identity function on $V$ |

(HILBERT) SPACE CONSTRUCTIONS FROM OTHER SPACES

| | |
|---|---|
| $\text{Hom}_{\mathbb{K}}(V, W)$ | space of $\mathbb{K}$-linear functions from $V$ to $W$ |
| $GL(V)$ | space of invertible $\mathbb{K}$-linear functions from $V$ to itself, sometimes written $GL(V, \mathbb{K})$ in the literature |
| $\text{Hom}_{G,\mathbb{K}}(V, W)$ | space of intertwiners from $V$ to $W$ |
| $\text{Hom}_G(X, W)$ | space of $G$-equivariant continuous maps from $X$ to $W$, for a homogeneous space $X$ |
| $\text{End}_{G,\mathbb{K}}(V)$ | space of endomorphisms of $V$, i.e., intertwiners from $V$ to $V$ |

| | |
|---|---|
| $V \otimes W$ | tensor product of two vector spaces over their common field. Also denotes the tensor product of pre-Hilbert spaces |
| $\bigoplus_{i \in I} V_i$ | (orthogonal) direct sum of all $V_i$ |
| $\widehat{\bigoplus}_{i \in I} V_i$ | topological closure of the (orthogonal) direct sum of all $V_i$ |
| $\operatorname{span}_{\mathbb{K}}(M)$ | vector subspace of a $\mathbb{K}$-vector space spanned by $M$ |
| $V^{\perp}$ | orthogonal complement of $V$ |
| $E_{\lambda}(\varphi)$ | eigenspace of $\varphi$ for eigenvalue $\lambda$ |

TOPOLOGICAL SPACES, METRIC SPACES, NORMED SPACES

| | |
|---|---|
| $\mathcal{T}$ | topology |
| $U_x$ | open neighborhood of $x \in X$ |
| $\mathcal{U}_x$ | set of all open neighborhoods of $x \in X$ |
| $\lim_{U \in \mathcal{U}_x}$ | limit over the directed set of open neighborhoods of $x$ |
| $\lim_{k \to \infty} x_k$ | limit of the sequence $(x_k)_k$ |
| $\overline{A}$ | topological closure of $A \subseteq X$ |
| $\|x\|$ | norm of $x$ |
| $|x|$ | absolute value of $x$ |
| $d(x, x')$ | distance of $x, x'$ according to metric $d$ |
| $\mathrm{B}_{\epsilon}(x)$ | $\epsilon$-ball around $x$ according to some metric $d$ |

HOMOGENEOUS SPACES AND THE PETER-WEYL THEOREM

| | |
|---|---|
| $X$ | a homogeneous space of $G$ |
| $x^* \in X$ | arbitrary point |
| $S^n$ | $n$-dimensional sphere in $(n+1)$-dimensional space |
| $\mu$ | a measure on a compact group $G$ or its Homogeneous Space $X$ |
| $\int_X$ | integral on a space $X$ with respect to its measure |
| $L^2_{\mathbb{K}}(X), L^2_{\mathbb{K}}(G)$ | Hilbert space of square-integrable functions on $X$ and $G$ with values in $\mathbb{K}$ |
| $\lambda$ | unitary representation on $L^2_{\mathbb{K}}(X)$ or $L^2_{\mathbb{K}}(G)$ |
| $g(x)$ | arbitrary lift of $x$ with respect to projection $\pi : G \to X$, $g \mapsto gx^*$ |
| $\operatorname{av}(f)$ | average of $f : G \to \mathbb{K}$ along cosets |
| $\pi^*$ | lift of functions $L^2_{\mathbb{K}}(X) \to L^2_{\mathbb{K}}(G)$ |
| $\delta_x$ | Dirac delta function at point $x$ |
| $\delta_U$ | approximated Dirac delta function for nonempty open set $U$ |
| $\rho_l^{ij}$ | abbreviation for $\rho_l^{v^i v^j}$ for orthonormal basis vectors $v^i, v^j \in V_l$ |
| $\mathcal{E}$ | linear span of all matrix coefficients of irreducible unitary representations |
| $\mathcal{E}_l$ | linear span of all matrix coefficients of $\rho_l$ |
| $\mathcal{E}_l^j$ | linear span of all matrix coefficients $\rho_l^{ij}$ with varying $i$ but fixed $j$ |
| $n_l, m_l$ | multiplicity of $l$ in orthogonal decomposition of $L^2_{\mathbb{K}}(G)$ and $L^2_{\mathbb{K}}(X)$, respectively |
| $V_{li}$ | copy of $V_l$ appearing in the Peter-Weyl decomposition of $L^2_{\mathbb{K}}(X)$ |
| $p_{li}$ | canonical projection $p_{li} : L^2_{\mathbb{K}}(X) \to V_{li}$ and $p_{li} : \bigoplus_{l'i'} V_{l'i'} \to V_{li}$ |
| $\sin_m, \cos_m$ | the functions $x \mapsto \sin(mx)$ and $x \mapsto \cos(mx)$ |
| $Y_l^n, {}^r Y_l^n$ | complex- and real-valued version of a spherical harmonic |
| $D_l, {}^r D_l$ | complex- and real-valued version of Wigner D-matrix |

## KERNELS AND REPRESENTATION OPERATORS

| | |
|---|---|
| $K$ | kernel $K : X \to \mathrm{Hom}_{\mathbb{K}}(V_{\mathrm{in}}, V_{\mathrm{out}})$ |
| $K \star f$ | convolution of kernel $K$ with input $f$ |
| $\mathcal{K}$ | kernel operator or (more generally) representation operator $\mathcal{K} : T \to \mathrm{Hom}_{\mathbb{K}}(U, V)$ |
| $\widehat{K}$ | kernel operator $\widehat{K} : L^2_{\mathbb{K}}(X) \to \mathrm{Hom}_{\mathbb{K}}(V_{\mathrm{in}}, V_{\mathrm{out}})$ corresponding to a kernel $K$ |
| $\mathcal{K}\vert_X$ | kernel $\mathcal{K}\vert_X : X \to \mathrm{Hom}_{\mathbb{K}}(V_{\mathrm{in}}, V_{\mathrm{out}})$ corresponding to a kernel operator $\mathcal{K}$ |
| $\tilde{\mathcal{K}}$ | for a representation operator $\mathcal{K} : T \to \mathrm{Hom}_{\mathbb{K}}(U, V)$, this denotes the corresponding map $\tilde{\mathcal{K}} : T \otimes U \to V$ under the hom-tensor adjunction |

## THE WIGNER-ECKART THEOREM

| | |
|---|---|
| $\rho_l, \rho_J$ | input- and output representations on the spaces $V_l$ and $V_J$ |
| $Y_j^m, Y_l^n, Y_J^M$ | fixed chosen orthonormal basis vectors of the abstract irreducible representations $V_j$, $V_l$, $V_J$ |
| $\langle JM\vert K(x)\vert ln\rangle$ | matrix element of $K(x)$ for a kernel $K$ and $x \in X$ |
| $d_l$ | dimension of $l$'th irrep $V_l$ as $\mathbb{K}$-vector space |
| $m_j$ | number of times $V_j$ is in the Peter-Weyl decomposition of $L^2_{\mathbb{K}}(X)$ |
| $[J(jl)]$ | number of times $V_J$ is in the direct sum decomposition of $V_j \otimes V_l$ |
| $c, c_{jis}$ | endomorphisms, mostly on $V_J$. $c_{jis}$ are endomorphisms appearing in the Wigner-Eckart theorem for steerable kernels |
| $c_r$ | basis endomorphism of $\rho_J$, indexed with index set $r = 1, \ldots, E_J$ |
| $\langle JM\vert c\vert JM'\rangle$ | matrix element at indices $M, M'$ for endomorphism $c$ |
| $l_s, l_{jis}$ | linear equivariant isometric embeddings $l_s : V_J \to V_j \otimes V_l$ and $l_{jis} : V_J \to V_{ji} \otimes V_l$ |
| $p_{jis}$ | projection $p_{jis} : V_{ji} \otimes V_l \to V_J$ corresponding to (i.e.: adjoint to) the embedding $l_{jis}$ |
| $\langle s, JM\vert jm; ln\rangle$ | Clebsch-Gordan coefficient corresponding to $l_s$ |
| $\mathrm{CG}_{J(jl)s}$ | 3-dimensional matrix of Clebsch-Gordan coefficients |
| $Y_{ji}^m$ | harmonic basis function, for example, spherical harmonic. Element of $V_{ji} \subseteq L^2_{\mathbb{K}}(X)$ |
| $\langle i, jm\vert x\rangle$ | shorthand notation for $\lim_{U \in \mathcal{U}_x} \langle Y_{ji}^m\vert \delta_U\rangle$. Equal to $\overline{Y_{ji}^m(x)}$ |
| $\langle i, j\vert x\rangle$ | row vector with entries $\langle i, jm\vert x\rangle$ |
| $\mathrm{Rep}$ | isomorphism between tuples of endomorphisms and kernel operators |
| $\mathrm{GKer}$ | isomorphism between tuples of endomorphisms and steerable kernels |
| $K_{jisr}$ | basis kernel |
| $w_{jisr}$ | learnable parameter |

LIST OF THEOREMS

LIST OF DEFINITIONS

# A   BUILDING BLOCKS OF SO(2)-STEERABLE KERNELS – RUNNING EXAMPLE FOR SECTION 3

In this short chapter, we briefly explain the components of steerable kernels at the specific example of real valued irreps of the circle group $SO(2)$. While this example is quite simple, it shows

some non-trivial properties like 2-dimensional endomorphism spaces $\mathrm{End}_{\mathrm{SO}(2),\mathbb{R}}(V_J)$ for $J > 0$ and a Clebsch-Gordan decomposition in which the multiplicity $[J(jl)]$ can differ from 0 or 1. To give a quick overview: Example A.1 considers the circle as an orbit and homogeneous space of $\mathrm{SO}(2)$ while Example A.2 introduces the real valued irreps. Their endomorphisms are stated in Example A.3. As discussed in Example A.4, the Peter-Weyl theorem corresponds here to the usual Fourier series on $S^1$. The Clebsch-Gordan decomposition of tensor products of the irreps are discussed in Example A.5. With these ingredients, we are ready to instantiate the kernel spaces as described by our Wigner-Eckart Theorem 4.1 for steerable kernels, for which we refer, including proofs, to Section E.2.

$\mathrm{SO}(2)$-steerable kernels $K : \mathbb{R}^2 \to \mathrm{Hom}_{\mathbb{R}}(V_{\mathrm{in}}, V_{\mathrm{out}}) \cong \mathbb{R}^{c_{\mathrm{out}} \times c_{\mathrm{in}}}$ allow for rotation equivariant convolutions. For instance, a convolution with an $\mathrm{SO}(2)$-steerable kernel on $\mathbb{R}^2$ is guaranteed to be $\mathrm{SE}(2) = (\mathbb{R}^2, +) \rtimes \mathrm{SO}(2)$ equivariant while a convolution with an $\mathrm{SO}(2)$-steerable kernel on $S^2$ will be $\mathrm{SO}(3)$-equivariant.

**Homogeneous Spaces**   $\mathrm{SO}(2)$ acts on the kernel's domain $\mathbb{R}^2$ by rotating it. The orbits of the action are therefore given by 1) the origin $\{0\}$ and 2) circles of arbitrary radius. We know that the kernel constraint can be solved on each orbit individually, and so we can restrict to looking at those. Since $\{0\}$ is rather trivial, we specifically consider the circle $S^1$ as a more interesting homogeneous space.

**Example A.1.** Consider the circle $S^1$ and the rotation group $\mathrm{SO}(2)$. For convenience, we reparameterize both: we view $\mathrm{SO}(2)$ as the group of angles $\phi \in \mathbb{R}/2\pi\mathbb{Z} \cong [0, 2\pi]/_{0 \sim 2\pi}$ and $S^1$ as the space $\mathbb{R}/2\pi\mathbb{Z}$ as well. Then the action of $\mathrm{SO}(2)$ on $S^1$ is given by $\phi \cdot x := (\phi + x) \mod 2\pi$. It is easy to see that this action is transitive, which makes the circle a homogeneous space of $\mathrm{SO}(2)$.

**Irreducible Representations**   As it is sufficient to solve the kernel constraint for irreducible orthogonal input- and output representations, we now state a classification of those up to isomorphism.

**Example A.2.** The irreducible orthogonal representations $\rho_l : \mathrm{SO}(2) \to \mathrm{O}(V_l)$ of $\mathrm{SO}(2)$ are labeled by indices ("quantum numbers") $l \in \mathbb{N}_{\geq 0}$. For $l = 0$, one has the trivial representation with $V_0 = \mathbb{R}$ and $\rho_0(\phi) = \mathrm{id}_{\mathbb{R}}$. For $l \geq 1$, one has $V_l = \mathbb{R}^2$ and

$$\rho_l(\phi) = \begin{pmatrix} \cos(l\phi) & -\sin(l\phi) \\ \sin(l\phi) & \cos(l\phi) \end{pmatrix},$$

i.e., rotation matrices of "frequency $l$". The isomorphism classes of irreducible orthogonal representations are then given by $\widehat{\mathrm{SO}(2)} \cong \mathbb{N}_{\geq 0}$.

We are thus in the following considering $\mathrm{SO}(2)$-steerable kernels of the form

$$K : S^1 \to \mathrm{Hom}_{\mathbb{R}}(V_l, V_J),$$

where $l, J \geq 0$.

**Endomorphisms**   Remember that if $c : V_J \to V_J$ is an endomorphism, i.e., commutes with $\rho_J$, that $c \circ K$ is then steerable as well. Thus, we now look at a classification of the endomorphisms of the irreducible orthogonal representations:

**Example A.3.** Let $G = \mathrm{SO}(2)$ with the irreducible representations $\rho_J$ as in Example A.2. Clearly, the endomorphism space $\mathrm{End}_{\mathrm{SO}(2),\mathbb{R}}(V_0)$ is 1-dimensional, i.e., $E_0 = 1$. For all $J \geq 1$, the endomorphism space is two-dimensional ($E_J = 2$) and given by combinations of scalings and rotations[8] on $V_J = \mathbb{R}^2$. A basis of this space is given by the following two matrices:

$$c_1 = \mathrm{id}_{\mathbb{R}^2} = \begin{pmatrix} 1 & 0 \\ 0 & 1 \end{pmatrix}, \quad c_2 = \begin{pmatrix} 0 & -1 \\ 1 & 0 \end{pmatrix}, \quad \mathrm{span}_{\mathbb{R}}\{c_1, c_2\} = \mathrm{End}_{\mathrm{SO}(2),\mathbb{K}}(V_J).$$

---

[8]Another way to imagine this is to identify $V_J$ with the complex plane $\mathbb{C}$. Then an endomorphism is given by multiplication with an arbitrary complex number.

That $c_1$ is an endomorphism of $\rho_J$ for $J \geq 1$ is immediately clear. That the same holds for $c_2$ is checked by the following simple calculation:

$$
\begin{aligned}
c_2 \, \rho_J(\phi) &= \begin{pmatrix} 0 & -1 \\ 1 & 0 \end{pmatrix} \begin{pmatrix} \cos(J\phi) & -\sin(J\phi) \\ \sin(J\phi) & \cos(J\phi) \end{pmatrix} = \begin{pmatrix} -\sin(J\phi) & -\cos(J\phi) \\ \cos(J\phi) & -\sin(J\phi) \end{pmatrix} \\
&= \begin{pmatrix} \cos(J\phi) & -\sin(J\phi) \\ \sin(J\phi) & \cos(J\phi) \end{pmatrix} \begin{pmatrix} 0 & -1 \\ 1 & 0 \end{pmatrix} = \rho_J(\phi) \, c_2
\end{aligned}
$$

The proof that there are no other endomorphisms is sketched in Proposition E.5.

**Peter-Weyl and Harmonic Basis Functions**  Another ingredient that we need to construct $\mathrm{SO}(2)$-steerable kernels on $S^1$ is the decomposition of $L^2_{\mathbb{R}}(S^1)$ into its irreducible subrepresentations $V_{ji}$, which the Peter-Weyl theorems guarantees to exist. Less abstractly, we are interested in an orthonormal set of harmonic (steerable) basis functions on $S^1$ that span $L^2_{\mathbb{R}}(S^1)$ – which corresponds to the usual Fourier series on $S^1$.

**Example A.4.**  As in Example A.1, we assume $G = \mathrm{SO}(2)$ and $X = S^1$. A standard result in harmonic analysis says that square-integrable functions $f : S^1 \to \mathbb{R}$, i.e., $f \in L^2_{\mathbb{R}}(S^1)$, can be uniquely written as an infinite sum of sine and cosine terms,

$$
f(x) = a_0 + \sum_{j=1}^{\infty} a_j \cos(jx) + b_j \sin(jx) \,, \tag{8}
$$

where $a_0$ and $a_j, b_j$, $j \geq 1$ are real-valued expansion coefficients.

How does this result relate to the harmonic basis functions in the Peter-Weyl theorem 3.4? As stated above, we have isomorphism classes $\widehat{G} = \widehat{\mathrm{SO}(2)} \cong \mathbb{N}_{\geq 0}$ of irreps with representatives $\rho_j$. A comparison of the Fourier series in Eq. (8) with property 2 in the Peter-Weyl theorem 3.4 suggests the following identification of harmonic basis functions $Y_j^m$ and coefficients $\lambda_{jm}$,

$$
\begin{aligned}
Y_0^1 &= \cos_0 = 1 \,, & \lambda_{01} &= a_0 & \text{for} \quad j = 0 \\
Y_j^1 &= \cos_j \,, & \lambda_{j1} &= a_j & \text{for} \quad j \geq 1 \\
Y_j^2 &= \sin_j \,, & \lambda_{j2} &= b_j & \text{for} \quad j \geq 1 \,,
\end{aligned}
$$

where we introduced the shorthand notations $\cos_j(x) := \cos(jx)$ and $\sin_j(x) := \sin(jx)$. Note that we dropped the index $i = 1, \ldots, m_j$ since $m_j = 1$ for any $j \in \widehat{\mathrm{SO}(2)}$. As expected, we have indices $m = 1$ for $j = 0$ with $d_j = \dim(V_0) = 1$ and indices $m = 1, 2$ for $j \geq 1$ with $d_j = \dim(V_j) = 2$. The orthogonality relations in property 3 of the Peter-Weyl theorem hold up to a simple normalization of these basis functions and are easily checked by explicitly computing the scalar products. Property 1, i.e., the $\mathrm{SO}(2)$-steerability of the harmonic bases, is trivial for $j = 0$. For $j \geq 1$, the standard angle summation formulas for cosines and sines lead to the following expressions for harmonics that are translated by $\phi \in \mathrm{SO}(2)$:

$$
\begin{aligned}
\cos_j(x - \phi) &= \cos_j(x)\cos_j(-\phi) - \sin_j(x)\sin_j(-\phi) = \left( \rho_j^{11}(\phi)\cos_j + \rho_j^{21}(\phi)\sin_j \right)(x) \\
\sin_j(x - \phi) &= \cos_j(x)\sin_j(-\phi) + \sin_j(x)\cos_j(-\phi) = \left( \rho_j^{12}(\phi)\cos_j + \rho_j^{22}(\phi)\sin_j \right)(x) \,,
\end{aligned}
$$

which is just property 1 in the Peter-Weyl theorem. This is concisely summarized by

$$
\begin{pmatrix} \cos_j \\ \sin_j \end{pmatrix} (\phi^{-1} \cdot x) = \begin{pmatrix} \cos_j \\ \sin_j \end{pmatrix} (x - \phi) = \rho_j(\phi)^{\top} \begin{pmatrix} \cos_j \\ \sin_j \end{pmatrix} (x) \,,
$$

which shows that the basis functions $Y_j^1 = \cos_j$ and $Y_j^2 = \sin_j$ span an invariant subspace $V_j$ of $L^2_{\mathbb{R}}(S^1)$ under rotations. From a more abstract viewpoint, the Peter-Weyl theorem just states that $L^2_{\mathbb{R}}(S^1)$ splits into the orthogonal direct sum $\bigoplus_{j \in \widehat{\mathrm{SO}(2)}} V_j$.

**Tensor Products and Clebsch-Gordan Coefficients**  Finally, we need to investigate the tensor products of irreducible representations and their decomposition via Clebsch-Gordan coefficients. They will be used to correctly assemble harmonic basis functions to steerable kernels.

**Example A.5.** Remember the irreducible representations $\rho_l : \mathrm{SO}(2) \to \mathrm{O}(V_l)$ given in Example A.2. As we prove in Proposition E.4, including a description of the Clebsch-Gordan coefficients, the tensor products decompose as follows:

$$V_0 \otimes V_0 \cong V_0, \quad V_j \otimes V_0 \cong V_j, \quad V_0 \otimes V_l \cong V_l, \quad V_j \otimes V_l \cong V_{|j-l|} \oplus V_{j+l},$$

where the last isomorphism only holds if $j, l \geq 1$ and $j \neq l$. If $j, l \geq 1$ and $j = l$, then we obtain

$$V_l \otimes V_l \cong (V_0)^2 \oplus V_{2l},$$

i.e., $V_0$ here appears *twice* in the decomposition of a tensor product of irreducible representations. We therefore have multiplicities $[J(jl)]$ which are 1 for $[0(00)]$, $[j(j0)]$, $[l(0l)]$, $[|j - l|(jl)]$, $[j + l(jl)]$ and $[2l(ll)]$ while $[0(ll)] = 2$. Any other multiplicity is zero.

**Wigner-Eckart theorem for SO(2)-steerable kernels**     With these ingredients one can then determine all $\mathrm{SO}(2)$-steerable kernels. This is explained in Proposition E.6.

# B    REPRESENTATION THEORY OF COMPACT GROUPS

In this chapter, we outline the main ingredients of the representation theory of compact groups that we need for our applications to steerable CNNs. Usually, this theory is only developed for representations over the complex numbers. However, since we want to apply it also to steerable CNNs using real representations, we need to be a bit more careful. In particular, we need to make sure that the Peter-Weyl theorem is correctly stated and proven.

The outline is as follows: In Section B.1, we start by stating all the important definitions and concepts from group theory and representation theory of (unitary) representations that are needed for formulating the Peter-Weyl theorem. After defining Haar measures both for compact groups and their homogeneous spaces and shortly discussing their square-integrable functions, we formulate the Peter-Weyl Theorem B.22. In Section B.2, then, we give a proof of this version of the Peter-Weyl theorem, carefully making sure to not use properties that are only true over $\mathbb{C}$. In some essential steps, mainly the density of the matrix coefficients in the regular representation, we refer to the literature, since the proof clearly does not make use of $\mathbb{C}$ per se. While we initially only give the proof for the regular representation, i.e., the space of square-integrable functions on the group itself, we end this section with a discussion of general unitary representations and, in particular, the space of square-integrable functions for an arbitrary homogeneous space.

In the whole chapter, let $\mathbb{K}$ be the field of real or complex numbers.

## B.1   FOUNDATIONS OF REPRESENTATION THEORY AND THE PETER-WEYL THEOREM

### B.1.1   PRELIMINARIES OF TOPOLOGICAL GROUPS AND THEIR ACTIONS

In this section, we define preliminary concepts from topological groups and their actions. This material can, for example, be found in detail in Arkhangel'skii & Tkachenko (2008). For the topological concepts that we use, we refer to Chapter F.1.

**Definition B.1** (Group, Abelian Group). A *group* $G = (G, \cdot, (\cdot)^{-1}, e)$, most often simply written $G$, consists of the following data:

    1. A set $G$ of group elements $g \in G$.

    2. A multiplication $\cdot : G \times G \to G, (g, h) \mapsto g \cdot h$.

    3. An inversion $(\cdot)^{-1} : G \to G, g \mapsto g^{-1}$.

    4. A distinguished unit element $e \in G$. It is also called neutral element.

They are assumed to have the following properties for all $g, h, k \in G$:

    1. The multiplication is associative: $g \cdot (h \cdot k) = (g \cdot h) \cdot k$.

    2. The unit element is neutral with respect to multiplication: $e \cdot g = g = g \cdot e$.

3. The inversion of an element multiplied with itself is the neutral element: $g \cdot g^{-1} = g^{-1} \cdot g = e$.

A group is called *abelian* if, additionally, the multiplication is commutative: $g \cdot h = h \cdot g$ for all $g, h \in G$. If this is the case, a group is often written as $G = (G, +, -(\cdot), 0)$.

If we consider several groups at once, say $G$ and $H$, then we often do not distinguish their multiplication, inversion, and neutral elements in notation. It will be clear from the context which group the operation belongs to.

**Definition B.2** (Subgroup). Let $G$ be a group and $H \subseteq G$ a subset. $H$ is called a *subgroup* if:

1. For all $h, h' \in H$ we have $h \cdot h' \in H$.

2. For all $h \in H$ we have $h^{-1} \in H$.

3. The neutral element $e \in G$ is in $H$.

Consequently, $H$ is also a group with the restrictions of the multiplication and inversion of $G$ to $H$.

**Definition B.3** (Group Homomorphism). Let $G$ and $H$ be groups. A function $f : G \to H$ is called a *group homomorphism* if it respects the multiplication, inversion, and neutral element, i.e., for all $g, h \in G$:

1. $f(g \cdot h) = f(g) \cdot f(h)$.

2. $f(g^{-1}) = f(g)^{-1}$.

3. $f(e) = e$.

The second and third properties automatically follow from the first and so do not need to be verified in order to prove that a certain function is a group homomorphism.

**Definition B.4** (Topological Group, Compact Group). Let $G$ be a group and $\mathcal{T}$ be a topology of the underlying set of $G$. Then $G = (G, \mathcal{T})$ is called a *topological group* (Arkhangel'skii & Tkachenko, 2008) if both multiplication $G \times G \to G$, $(x, y) \mapsto x \cdot y$ and inversion $G \to G$, $x \mapsto x^{-1}$ are continuous maps. Additionally, we always assume the topology to be Hausdorff.

A topological group is called *compact* if the underlying topological space is compact.

From now on, all groups considered are compact topological groups. Furthermore, whenever $G$ is a finite group, we assume that it is a topological group with the *discrete topology*, i.e., the topology with respect to which all subsets of $G$ are open.

We will need the following definition in order to define homogeneous spaces:

**Definition B.5** (Group Action). Let $G$ be a compact group and $X$ a topological space. Then a *group action* of $G$ on $X$ is a continuous function $\cdot : G \times X \to X$ with the following properties:

1. $(g \cdot h) \cdot x = g \cdot (h \cdot x)$ for all $g, h, \in G$ and $x \in X$.

2. $e \cdot x = x$ for all $x \in X$.

We will often simply write $gx$ instead of $g \cdot x$. Also, note that the multiplication within $G$ is denoted by the same symbol as the group action on the space $X$.

**Definition B.6** (Orbit). Let $\cdot : G \times X \to X$ be a group action. Let $x \in X$. Then it's *orbit*, denoted $G \cdot x$, is given by the set

$$G \cdot x := \{g \cdot x \mid g \in G\} \subseteq X.$$

**Definition B.7** (Transitive Action, Homogeneous Space). Let $\cdot : G \times X \to X$ be a group action. This action is called *transitive* if for all $x, y \in X$ there exists $g \in G$ such that $gx = y$. Equivalently, each orbit is equal to $X$, that is: For all $x \in X$ we have $G \cdot x = X$.

$X$ is called a *homogeneous space* (with respect to the action) if the action is transitive, $X$ is Hausdorff and $X \neq \emptyset$.

The Hausdorff condition and non-emptiness in the definition of homogeneous spaces is needed for Lemma B.21, which is necessary to even define a normalized Haar measure on a homogeneous space. Some texts in the literature may define homogeneous spaces without these conditions.

**Definition B.8** (Stabilizer Subgroup). Let $\cdot : G \times X \to X$ be a group action. Let $x \in X$. The *stabilizer subgroup* $G_x$ is the subgroup of $G$ given by

$$G_x := \{g \in G \mid gx = x\} \subseteq G.$$

**Example B.9.** The multiplication of the group $G$ is a group action of $G$ on itself. $G$ is a homogeneous space with this action. Furthermore, for each $g \in G$ the stabilizers $G_g$ are the trivial subgroup $e$.

In general, homogeneous spaces with the property that all stabilizers are trivial are called *torsors* or *principal homogeneous spaces*. Principal homogeneous spaces are topologically indistinguishable from the group itself.

### B.1.2 LINEAR AND UNITARY REPRESENTATIONS

In this section, we define many of the foundational concepts about linear and unitary representations (Knapp, 2002; Kowalski, 2014).

Whenever we will consider linear or unitary representations of compact groups, we want those representations to be *continuous*. This requires that the vector spaces on which our groups act carry themselves a topology. Prototypical examples of such vector spaces are (pre-)Hilbert spaces. They are the main examples of vector spaces considered in this work. Foundational concepts about (pre-)Hilbert spaces can be found in Chapter F.3. The most important difference between how we view pre-Hilbert spaces and how it can often be found in the literature is that in this work, scalar products are antilinear in the first component and linear in the second. This is the convention usually chosen in physics.

For a vector space $V$ over $\mathbb{K}$ let $\mathrm{GL}(V)$ be the group of invertible linear functions from $V$ to $V$. Sometimes in the literature, this is also written $\mathrm{GL}(V, \mathbb{K})$. The multiplication is given by function composition and the neutral element by the identity function $\mathrm{id}_V$ on $V$.

**Definition B.10** (Linear Representation). Let $G$ be a compact group and $V$ be a $\mathbb{K}$-vector space carrying a topology, for example, a (pre)-Hilbert space. Then a *linear representation* of $G$ on $V$ is a group homomorphism $\rho : G \to \mathrm{GL}(V)$ which is continuous in the following sense: for all $v \in V$, the function

$$\rho^v : G \to V, \ \ g \mapsto \rho^v(g) := \rho(g)(v)$$

is continuous. From the definition we obtain $\rho(e) = \mathrm{id}_V$, $\rho(g \cdot h) = \rho(g) \circ \rho(h)$ and $\rho(g^{-1}) = \rho(g)^{-1}$ for all $g, h \in G$. For simplicity, we also just say *representation* or $G$-*representation* instead of *linear representation*. Instead of denoting the representation by $\rho$, we often denote it by $V$ if the function $\rho$ is clear from the context.

Note that in this definition, $V$ can be any abstract topological $\mathbb{K}$-vector space with a topology and does not need to be a space $\mathbb{K}^n$ or something similar. Consequently, we usually do not view the functions $\rho(g)$ as matrices, but as abstract linear automorphisms from $V$ to $V$.

**Definition B.11** (Intertwiner). Let $\rho : G \to \mathrm{GL}(V)$ and $\rho' : G \to \mathrm{GL}(V')$ be two representations over the same group $G$. An *intertwiner* between them is a linear function $f : V \to V'$ that is additionally equivariant with respect to $\rho$ and $\rho'$ and continuous. Equivariance means that for all $g \in G$ one has $f \circ \rho(g) = \rho'(g) \circ f$, which means the following diagram commutes:

$$
\begin{array}{ccc}
V & \xrightarrow{f} & V' \\
{\scriptstyle \rho(g)}\downarrow & & \downarrow{\scriptstyle \rho'(g)} \\
V & \xrightarrow{f} & V'
\end{array}
$$

**Definition B.12** (Equivalent Representations). Let $\rho : G \to \mathrm{GL}(V)$ and $\rho' : G \to \mathrm{GL}(V')$ be two representations. They are called *equivalent* if there is an intertwiner $f : V \to V'$ that has an inverse. That is, there exists an intertwiner $\tilde{f} : V' \to V$ such that $\tilde{f} \circ f = \mathrm{id}_V$ and $f \circ \tilde{f} = \mathrm{id}_{V'}$.

In categorical terms, equivalent representations are isomorphic in the category of linear representations. The reason we do not call them isomorphic is that there is a stronger notion of isomorphism between representations which we will later use, namely isomorphisms of *unitary* representations.

**Definition B.13** (Invariant Subspace, Subrepresentation, Closed Subrepresentation). Let $\rho : G \to \mathrm{GL}(V)$ be a representation. An *invariant subspace* $W \subseteq V$ is a linear subspace of $V$ such that $\rho(g)(w) \in W$ for all $g \in G$ and $w \in W$. Consequently, the restriction $\rho|_W : G \to \mathrm{GL}(W)$, $g \mapsto \rho(g)|_W : W \to W$ is a representation as well, called *subrepresentation* of $\rho$.

A subrepresentation is called *closed* if $W$ is closed in the topology of $V$.

**Definition B.14** (Irreducible Representation). A representation $\rho : G \to \mathrm{GL}(V)$ is called *irreducible* if $V \neq 0$ and if the only closed subrepresentations of $V$ are $0$ and $V$ itself. An irreducible representation is also shortly called *irrep*.

**Definition B.15** (Unitary Group). Let $V$ be a pre-Hilbert space. The *unitary group* $\mathrm{U}(V)$ of $V$ is defined as the group of all linear invertible maps $f : V \to V$ that respect the inner product, i.e., $\langle f(x)|f(y) \rangle = \langle x|y \rangle$ for all $x, y \in V$. It is a group with respect to the usual composition and inversion of invertible linear maps.

Note that if the field $\mathbb{K}$ is the real numbers, then what we call "unitary" is actually called *orthogonal*, and the group would be denoted $\mathrm{O}(V)$. However, the mathematical properties are essentially the same, and since the term "unitary" is more widely used (as normally, representations over the complex numbers are considered) we stick with "unitary".

More generally, we have the following:

**Definition B.16** (Unitary Transformation). Let $V, V'$ be two pre-Hilbert spaces. A *unitary transformation* $f : V \to V'$ is a bijective linear function such that $\langle f(x)|f(y) \rangle = \langle x|y \rangle$ for all $x, y \in V$. These can be regarded as isomorphisms between pre-Hilbert spaces.

Note that unitary transformations are in particular isometries, i.e., they keep the distances of vectors with respect to the metric defined by the scalar product. For the definition of this metric, see the discussion before and after Definition F.14.

**Definition B.17** (Unitary Representation). Let $V$ be a pre-Hilbert space and $G$ a group. Then a representation $\rho : G \to \mathrm{GL}(V)$ is called a *unitary representation* if $\rho(g) \in \mathrm{U}(V)$ for all $g \in G$. We then write $\rho : G \to \mathrm{U}(V)$.

In this whole chapter, the space $V$ of a unitary representation is supposed to be a *Hilbert space*, instead of just a pre-Hilbert space. Only in chapter D will we consider unitary representations on pre-Hilbert spaces. Note that all finite-dimensional pre-Hilbert spaces are already complete by Proposition F.47, so in these cases, there is no difference. The same proposition also shows that for finite-dimensional unitary representations, we can ignore the topological closedness condition in order to check whether it is irreducible. It will later turn out that all irreducible representations of a compact group are automatically finite-dimensional anyway, see Proposition B.31, so this further simplifies our considerations.

As before with the unitary group, a unitary representation is actually called "orthogonal representation" when the field is the real numbers $\mathbb{R}$. $\mathrm{U}(V)$ is then replaced by $\mathrm{O}(V)$. We again stick with $\mathrm{U}(V)$ whenever the field $\mathbb{K}$ is not specified.

**Definition B.18** (Isomorphism of Unitary Representations). Let $\rho : G \to \mathrm{U}(V)$, $\rho' : G \to \mathrm{U}(V')$ be unitary representations and $f : V \to V'$ an intertwiner. $f$ is called an *isomorphism* (of unitary representations) if, additionally, $f$ is a unitary transformation. The representations are then called *isomorphic*. For this, we write $\rho \cong \rho'$ or $V \cong V'$ depending on whether we want to emphasize the representations or the underlying vector spaces.

We note the following, which we will frequently use: due to the unitarity of $\rho(g)$ for a unitary representation $\rho$, we have $\rho(g)^* = \rho(g)^{-1}$, i.e., the adjoint is the inverse. Adjoints are defined in Definition F.42 and this statement is proven more generally in Proposition F.44. Overall, this means that $\langle \rho(g)(v)|w \rangle = \langle v|\rho(g)^{-1}(w) \rangle$ for all $v, w$ and $g$.

In the end, it will turn out that the Peter-Weyl theorem which we aim at is exclusively a statement about unitary representations. One may then wonder whether this is too restrictive. After all, the

representations that we consider for steerable CNNs (with precise definitions given in Section C.1) are not necessarily unitary, and so it is not immediately obvious how the Peter-Weyl theorem will be able to help for those. However, as it turns out, all linear representations on finite-dimensional spaces *can* be considered as unitary, and so the theory applies. We will discuss this in Proposition B.20 once we understand Haar measures on compact groups.

### B.1.3 THE HAAR MEASURE, THE REGULAR REPRESENTATION AND THE PETER-WEYL THEOREM

Now that we have introduced many notions in the representation theory of compact groups, we can formulate the most important result, the *Peter-Weyl theorem* that we will use throughout this work. In the next section, we will then go through a step-by-step proof of this theorem. The material in this section is based on Nachbin & Bechtolsheim (1965); Kowalski (2014) and Knapp (2002). We thank Stefan Dawydiak for a discussion about the Peter-Weyl theorem over the real numbers (Dawydiak, 2020).

We assume that the reader knows what a measure is (Tao, 2013). Let $G$ be a compact group. A standard result is that there exists a measure $\mu$ on $G$, called a *Haar measure* that, among other properties, fulfills the following:

1. $\mu(S)$ can be evaluated for all Borel sets $S \subseteq G$. Here, the Borel sets form the smallest so-called $\sigma$-algebra that contains all the open sets.

2. In particular, we can evaluate $\mu(S)$ for all open or closed sets $S \subseteq G$.

3. The Haar measure is normalized: $\mu(G) = 1$.

4. $\mu$ is left and right invariant: $\mu(gS) = \mu(S) = \mu(Sg)$ for all $g \in G$ and $S$ measurable.

5. $\mu$ is inversion invariant: $\mu(S^{-1}) = \mu(S)$ for all $S$ measurable.

These properties then translate into properties of the associated *Haar integral*: let $f : G \to \mathbb{K}$ be integrable with respect to $\mu$, then we obtain:

1. $\int_G 1 dg = 1$ for the constant function with value 1.

2. $\int_G f(hg)dg = \int_G f(g)dg = \int_G f(gh)dg$ for all $h \in G$.

3. $\int_G f(g^{-1})dg = \int_G f(g)dg$.

**Example B.19** (Finite Groups). If $G$ is a finite group with $n$ elements, then the Haar measure is just the normalized counting measure which assigns $\mu(g) = \frac{1}{n}$ for all $g \in G$. Each function $f : G \to \mathbb{K}$ is then integrable, and its integral is just given by

$$\int_G f(g)dg = \frac{1}{n}\sum_{g \in G} f(g).$$

In this special case, one can easily verify all properties of Haar measures and Haar integrals stated above.

With this measure defined, we can already understand why all linear representations on finite-dimensional spaces can be considered as unitary:

**Proposition B.20.** *Let $\rho : G \to \mathrm{GL}(V)$ be a linear representation on a finite-dimensional space $V$. Then there exists a scalar product $\langle \cdot | \cdot \rangle_\rho : V \times V \to \mathbb{K}$ that makes $(V, \langle \cdot | \cdot \rangle)$ a Hilbert space and such that $\rho$ becomes a unitary representation with respect to this scalar product.*

*Proof.* Since $V$ is finite-dimensional, there is an isomorphism of vector spaces to some $\mathbb{K}^n$. Consequently, there is *some* scalar product $\langle \cdot | \cdot \rangle : V \times V \to \mathbb{K}$ that makes $V$ a Hilbert space. However, this scalar product does not necessarily make $\rho$ a unitary representation. However, we can define $\langle \cdot | \cdot \rangle_\rho : V \times V \to \mathbb{K}$ by

$$\langle v | w \rangle_\rho := \int_G \langle \rho(g)(v) | \rho(g)(w) \rangle \, dg.$$

That this integral exists is due to the continuity of linear representations and since also the scalar product is continuous by Proposition F.38. It can easily be checked that this construction makes $V$ a

Hilbert space. And due to the right invariance of the Haar measure, we can check that $\rho$ is a unitary representation with respect to this scalar product. Namely, for arbitrary $g' \in G$ we have:

$$
\begin{aligned}
\langle \rho(g')(v) | \rho(g')(w) \rangle_\rho &= \int_G \langle \rho(g)\rho(g')v | \rho(g)\rho(g')w \rangle dg \\
&= \int_G \langle \rho(gg')(v) | \rho(gg')(w) \rangle dg \\
&= \int_G \langle \rho(g)(v) | \rho(g)(w) \rangle dg \\
&= \langle v | w \rangle_\rho.
\end{aligned}
$$

$\square$

Now, for a measure space $Y$ with corresponding measure $\mu$, we can consider the space of square-integrable functions on $Y$ with values in $\mathbb{K}$, denoted $L^2_{\mathbb{K}}(Y)$ (the measure is omitted in the notation since there is usually no ambiguity). In these spaces, functions are identified if they coincide on a set with measure 0. $L^2_{\mathbb{K}}(Y)$ is clearly a vector space over $\mathbb{K}$, but it turns out that it can even be considered to be a Hilbert space as follows:

$$
\langle f | g \rangle := \int_Y \overline{f(y)} g(y) dy.
$$

Here, the overline means complex conjugation. The Hilbert space properties are easily verified.

In particular, one can consider the space $L^2_{\mathbb{K}}(G)$ of square-integrable functions on the group $G$ itself. Now the claim is that $L^2_{\mathbb{K}}(G)$ can actually be equipped with a prototypical structure as a unitary representation over $G$ which makes this space, in some sense, "universal among unitary representations". This works with the following canonical representation, called the *regular representation*:

$$
\lambda : G \to U(L^2_{\mathbb{K}}(G)), \; [\lambda(g)(f)](g') := f(g^{-1}g').
$$

continuity of this map is non-trivial and is, for example, shown in Knapp (2002). However, the more algebraic properties of being a unitary representation are easy to appreciate. First of all, we clearly see that $\lambda$ is a group homomorphism mapping each group element to a linear automorphism. And finally, the unitarity of this representation can be understood as a direct consequence of the properties of the Haar measure, where we notably make only use of the left-invariance:

$$
\begin{aligned}
\langle \lambda(g)(f) | \lambda(g)(h) \rangle &= \int_G \overline{[\lambda(g)(f)](g')} \cdot [\lambda(g)(h)](g') dg' \\
&= \int_G \overline{f(g^{-1}g')} \cdot h(g^{-1}g') dg' \\
&= \int_G \overline{f(g')} h(g') dg' \\
&= \langle f | h \rangle.
\end{aligned}
$$

We saw in Example B.9 that $G$ is a homogeneous space with respect to the action on itself. We can now ask whether these constructions can also work if $X$ is an arbitrary homogeneous space of $G$. This requires us to define a suitable measure on $X$. This is indeed possible. For a fixed element $x^* \in X$, denote the stabilizer subgroup by $H = G_{x^*} \subseteq G$. Then the Hausdorff property of $X$ allows to write down a homeomorphism between $X$ and $G/H$, which in turn will allow us to use a canonical measure on $G/H$ that we study below. We denote cosets $gH \in G/H$ by $[g]$.

**Lemma B.21.** *Let $X$ be a homogeneous space of the compact group $G$ and $H$ the stabilizer subgroup of a fixed element $x^* \in X$. Then the map*

$$
\varphi : G/H \to X, \; [g] \mapsto gx^*
$$

*is a homeomorphism. Furthermore, $H$ is topologically closed.*

*Proof.* Let $\tilde{\varphi} : G \to X, g \mapsto gx^*$. This map is equal to the composition of the maps $G \to G \times X$, $g \mapsto (g, x^*)$ and $G \times X \to X, (g, x) \mapsto gx$. Both these are continuous, and thus $\tilde{\varphi}$ is continuous as well. Furthermore, note that if $g^{-1}g' \in H$, then there is $h \in H$ such that $g' = gh$, and thus

$$
\tilde{\varphi}(g') = \tilde{\varphi}(gh) = (gh)x^* = g(hx^*) = gx^* = \tilde{\varphi}(g)
$$

which means that by Proposition F.12, the map $\varphi : G/H \to X, [g] \mapsto gx^*$ is a well-defined continuous map. It is surjective since the action is transitive by definition of a homogeneous space. Furthermore, it is injective since if $gx^* = g'x^*$ then $x^* = (g^{-1}g')x^*$ and thus $g^{-1}g' \in H$, which means $[g] = [g']$.

Overall, $\varphi$ is a continuous bijective map from $G/H$ to $X$. Furthermore, $G/H$ is compact since it is the continuous image of the compact group $G$ under the projection $G \to G/H$, see Proposition F.8. Since $X$ is Hausdorff by definition of homogeneous spaces, $\varphi$ is a homeomorphism according to Proposition F.9.

Now, since $X$ is Hausdorff and $\varphi$ is a homeomorphism, it follows that $G/H$ is Hausdorff as well. Then, necessarily, $H$ is a topologically closed subgroup of $G$, see Bourbaki (1998), Chapter III, Section 2.5, Proposition 13. $\qquad\square$

Every space $G/H$ where $H$ is topologically closed allows a measure $\mu$ with similar properties to those of $G$ (Nachbin & Bechtolsheim, 1965). Since the stabilizer $H$ is closed and $X \cong G/H$ by Lemma B.21, we can do these constructions for $X$ as well, as we outline now. The only properties that we now miss are the right-invariance and inversion-invariance: We simply can't ask for them since $G$ does not naturally act on $X$ from the right and since we cannot invert elements in $X$. But left-invariance does hold and this means that

$$\lambda : G \to L^2_{\mathbb{K}}(X), \; [\lambda(g)(f)](x) := f(g^{-1}x)$$

makes $L^2_{\mathbb{K}}(X)$ a unitary representation over $G$, as can be shown in the exact same way as for $L^2_{\mathbb{K}}(G)$.

Let $\widehat{G}$ be the set of isomorphism classes of irreducible unitary representations over $G$. Furthermore, let $\rho_l : G \to V_l$ be a fixed representative of such an isomorphism class $l \in \widehat{G}$. We write isomorphism classes as "$l$" (and later also $j$ and $J$) in order to bring to mind quantum numbers used in quantum mechanics. Recall from linear algebra that a countable sum of subspaces of a vector space is called *direct* if no nontrivial subspace of any of the considered spaces is contained in the sum of all the other considered spaces.[9] Furthermore, recall that two subspaces $U, W \subseteq V$ of a Hilbert space $V$ are called perpendicular or orthogonal if $\langle u|w \rangle = 0$ for all $u \in U$ and $w \in W$. We then write $U \perp W$. We can now formulate the Peter-Weyl theorem. Intuitively, it says that $L^2_{\mathbb{K}}(X)$ splits into an orthogonal direct sum of the irreducible unitary representations, where each irreducible unitary representation appears maximally as often as its own dimension (and may not appear at all):

**Theorem B.22** (Peter-Weyl Theorem)**.** *Let $G$ be a compact group. Let $X$ be a homogeneous space. There are numbers $m_l \in \mathbb{N}_{\geq 0}$ for all $l \in \widehat{G}$ and closed-invariant subspaces $V_{li} \subseteq L^2_{\mathbb{K}}(X)$ for all $l \in \widehat{G}$ and $i \in \{1, \ldots, m_l\}$ such that the following hold:*

1. *$V_{li} \cong V_l$ as unitary representations for all $i$ and $l$.*

2. *$m_l \leq \dim(V_l) < \infty$ for all $l$.*

3. *$V_{li} \perp V_{l'j}$ whenever $l \neq l'$ or $i \neq j$.*

4. *$\bigoplus_{l \in \widehat{G}} \bigoplus_{i=1}^{m_l} V_{li}$ is topologically dense in $L^2_{\mathbb{K}}(X)$, written $L^2_{\mathbb{K}}(X) = \widehat{\bigoplus}_{l \in \widehat{G}} \bigoplus_{i=1}^{m_l} V_{li}$.*

*Now additionally consider $G$ as a homogeneous space of itself. Then the same holds for $L^2_{\mathbb{K}}(G)$ as well, with numbers $n_l \leq \dim(V_l) < \infty$. We additionally have the following:*

1. *$m_l \leq n_l$.*

2. *If $\mathbb{K} = \mathbb{C}$, then $n_l = \dim(V_l)$.*

Note that the representative $V_l$ is not assumed to be embedded in $L^2_{\mathbb{K}}(X)$. It is just isomorphic, as a unitary representation, to each of the $V_{li} \subseteq L^2_{\mathbb{K}}(X)$.

**Example B.23.** For $G = \mathrm{SO}(2)$ and $\mathbb{K} = \mathbb{C}$ we have $L^2_{\mathbb{C}}(\mathrm{SO}(2)) = \widehat{\bigoplus}_{l \in \mathbb{Z}} V_{l1}$ and all irreducible representations $V_l$ are 1-dimensional.

---

[9]For a vector space $V$ and subspaces $(U_i)_{i \in I}$, their sum $\sum_{i \in I} U_i$ is the set of sums $\sum_{i \in J} u_i$ with $J \subseteq I$ finite and $u_i \in U_i$ for all $i$. It is itself a subspace of $V$.

For $G = \mathrm{SO}(2)$ and $\mathbb{K} = \mathbb{R}$, we obtain $L^2_{\mathbb{R}}(\mathrm{SO}(2)) = \widehat{\bigoplus}_{l \geq 0} V_{l1}$, and all irreducible representations $V_l$ with $l \geq 1$ are two-dimensional, whereas $V_0$ is one-dimensional. Thus, here we see an example where the multiplicity of most irreducible representations in the regular representation is 1 and therefore smaller than their dimension, which cannot happen for representations over the complex numbers.

Both of these results are standard results in harmonic analysis. These examples are discussed in more detail, especially with respect to their applications in deep learning, in Section E.1 and E.2.

## B.2    A Proof of the Peter-Weyl Theorem

This section presents a proof of the Peter-Weyl theorem, as formulated in Theorem B.22. We mostly skip the *analytical* parts of the proof,[10] since they are well-presented in the literature and clearly work over both the real and complex numbers. However, the more *algebraic* parts of the proof usually make use of the property of the complex numbers to be algebraically closed, which does not hold for the real numbers. This is invoked usually both in the proof of a version of Schur's lemma, as well as in proving Schur's orthogonality. We therefore carefully adapt the proof of the Peter-Weyl theorem in the literature so that it also works over the real numbers, and formulate and prove versions of Schur's Lemma B.29 and Schur's orthogonality B.30 that work in general.

This section can be skipped if the interest is mainly in the applications of the Peter-Weyl theorem. In this case, the reader is advised to directly move on to Chapter C.

We note the following convention that applies to this section: for all unitary representations $\rho : G \to \mathrm{U}(V)$ that we consider here, $V$ is a Hilbert space (instead of just a pre-Hilbert space).

### B.2.1    Density of Matrix Coefficients

An important ingredient in the construction of the spaces $V_{li}$ that appear in the formulation of the Peter-Weyl Theorem B.22 are matrix coefficients, which together generate those spaces in case that one considers the regular representation on $L^2_{\mathbb{K}}(G)$.

**Definition B.24** (Matrix Coefficients)**.** Let $\rho : G \to \mathrm{U}(V)$ be a unitary representation. A *matrix coefficient* is any function of the form

$$\rho^{uv} : G \to \mathbb{K}, \ g \mapsto \overline{\langle u | \rho(g)(v) \rangle}$$

for arbitrary $u, v \in V$.

The term "matrix coefficient" comes from the analogy to matrix elements of linear maps between pre-Hilbert spaces of which orthonormal bases are fixed. Later, in Definition D.9 we will also define the notion of "matrix elements" separately. The term "matrix coefficient" only applies to unitary representations.

*Remark* B.25. By definition of linear representations, the function $g \mapsto \rho(g)(v)$ is continuous. Thus, since scalar products of Hilbert spaces are also continuous as functions on $V \times V$, see Proposition F.38, every matrix coefficient $\rho^{uv} : G \to \mathbb{K}$ is continuous. As a continuous function on a compact space, it is of course also square-integrable, i.e., $\rho^{uv} \in L^2_{\mathbb{K}}(G)$. The Peter-Weyl theorem basically asserts that these matrix coefficients can be considered as the building blocks of all square-integrable functions.

Furthermore, one may wonder why there is a complex conjugation in the definition. The reason for this is that, otherwise, the isomorphism that we will construct in Proposition B.35 is not linear but conjugate linear. The reason why this can nevertheless be called a matrix coefficient is that this actually *is* the matrix coefficient (without complex conjugation) on a conjugate Hilbert space, as explained in the next Proposition, which we took from Williams (1991).

**Proposition B.26.** *Let $\rho : G \to \mathrm{U}(V)$ be a unitary representation on a Hilbert space $V$ with scalar multiplication $\cdot_V$ and scalar product $\langle \cdot | \cdot \rangle_V$. We have the following:*

1. *$\tilde{V} := V$ (equality as abelian groups) with $\alpha \cdot_{\tilde{V}} v := \overline{\alpha} \cdot_V v$ and $\langle u | v \rangle_{\tilde{V}} := \overline{\langle u | v \rangle}$ is again a Hilbert space, the so-called* conjugate Hilbert space *of $V$.*

---

[10]I.e., those parts that deal with approximations of square-integrable functions by matrix elements.

2. $\tilde{\rho} : G \to \mathrm{U}(\tilde{V})$ with $\tilde{\rho}(g) := \rho(g)$ is again a unitary representation.

3. For the matrix coefficients, we have $\tilde{\rho}^{uv}(g) = \overline{\rho^{uv}(g)}$.

*Proof.* All these assertions are easy to check. As a demonstration, we do 3:

$$\tilde{\rho}^{uv}(g) = \overline{\langle u | \tilde{\rho}(g)(v) \rangle}_{\tilde{V}} = \overline{\overline{\langle u | \rho(g)(v) \rangle}}_V = \overline{\rho^{uv}(g)}.$$

That's what we wanted to show. □

As a consequence of this proposition, the matrix coefficient $\rho^{uv}(g)$ is equal to $\overline{\tilde{\rho}^{uv}(g)}$, thus being a "matrix coefficient without complex conjugation above the scalar product" of the conjugate unitary representation.

**Theorem B.27** (Density of Matrix Coefficients). *The linear span of the matrix-coefficients of finite-dimensional, unitary, irreducible representations of $G$ are dense in $L^2_{\mathbb{K}}(G)$ for all compact groups $G$.*

*Proof.* For $\mathbb{K} = \mathbb{C}$, this is shown in Knapp (2002). The same proof, without adaptions, also works for $\mathbb{K} = \mathbb{R}$. Note that the cited proof uses a definition of matrix coefficients without the complex conjugation. However, Proposition B.26 shows those span the same space, and thus we can apply it to our situation. □

### B.2.2 SCHUR'S LEMMA, SCHUR'S ORTHOGONALITY AND CONSEQUENCES

In this section, we state and prove versions of Schur's lemma and Schur's Orthogonality (Knapp, 2002) that are valid for both $\mathbb{K} = \mathbb{R}$ and $\mathbb{K} = \mathbb{C}$.

**Lemma B.28.** *Let $\rho : G \to \mathrm{U}(V)$ and $\rho' : G \to \mathrm{U}(V')$ be unitary representations. Furthermore, let $f : V \to V'$ be an intertwiner. Then the adjoint $f^* : V' \to V$ is also an intertwiner.*

*Proof.* The adjoint $f^* : V' \to V$ is the unique continuous linear function from $V'$ to $V$ such that, for all $v \in V$ and $v' \in V'$, we have

$$\langle f(v) | v' \rangle = \langle v | f^*(v') \rangle .$$

This always exists according to Definition F.42. Note that with $f$ being an intertwiner and using the unitarity of the representations, we obtain for all $g \in G, v \in V$ and $v' \in V'$:

$$\begin{aligned}
\langle v | \rho(g) f^*(v') \rangle &= \langle \rho(g^{-1})(v) | f^*(v') \rangle \\
&= \langle f \rho(g^{-1})(v) | v' \rangle \\
&= \langle \rho'(g^{-1}) f(v) | v' \rangle \\
&= \langle f(v) | \rho'(g)(v') \rangle \\
&= \langle v | f^* \rho'(g)(v') \rangle
\end{aligned}$$

from which we deduce $\rho(g) f^* = f^* \rho'(g)$ from Proposition F.45 for all $g \in G$, i.e., $f^*$ is an intertwiner. □

**Lemma B.29** (Schur's Lemma for unitary Representations). *Assume $\rho : G \to \mathrm{U}(V)$ and $\rho' : G \to \mathrm{U}(V')$ are irreducible unitary representations with $V$ finite-dimensional. Also assume that $f : V \to V'$ is an intertwiner. Then either $f = 0$ or there is $\mu \in \mathbb{R}_{>0}$ such that $\mu f$ is an isomorphism.*

*Proof.* For this proof, we follow the exposition of Tao (2011). We thank Terrence Tao for confirming in the discussion below his blogpost that this lemma can also be proven over the real numbers.

Let $f^* : V' \to V$ be the adjoint of $f$, which is also an intertwiner by Lemma B.28. Now, set $\varphi := f^* \circ f : V \to V$. As a composition of intertwiners, $\varphi$ is also an intertwiner. Furthermore, for arbitrary composable continuous linear functions between Hilbert spaces one always has $(g \circ$

$h)^* = h^* \circ g^*$ and $(g^*)^* = g$, which easily follows from the definition and uniqueness of adjoints. Consequently, we have

$$\varphi^* = (f^* \circ f)^* = f^* \circ (f^*)^* = f^* \circ f = \varphi,$$

and so $\varphi$ is self-adjoint. Thus, $\langle \varphi(v)|w \rangle = \langle v|\varphi(w) \rangle$ for all $v, w \in V$, from which we conclude that the matrix of $\varphi$ corresponding to any orthonormal basis of $V$ is Hermitian or, if $\mathbb{K} = \mathbb{R}$, even symmetric. Such an orthonormal basis exists by Proposition F.41. From the Spectral Theorem for Hermitian or symmetric matrices (Horn & Johnson, 2012) we conclude that $\varphi$ is unitarily (or for real matrices: orthogonally) diagonalizable with only real eigenvalues. Thus, there is an orthogonal decomposition of $V$ into eigenspaces: $V = \bigoplus_{\lambda \text{ eigenvalue}} E_\lambda(\varphi)$.

Let $E_\lambda(\varphi)$ be any eigenspace. We now claim that it is an invariant subspace of $\rho$. Indeed, for all $g \in G$ and $v \in E_\lambda(\varphi)$ we have since $\varphi$ is an intertwiner:

$$\varphi(\rho(g)(v)) = \rho(g)(\varphi(v)) = \rho(g)(\lambda v) = \lambda \rho(g)(v).$$

Since $V$ is finite-dimensional, $E_\lambda(\varphi)$ is topologically closed by Proposition F.47, and since $V$ is irreducible, we necessarily have $E_\lambda(\varphi) = 0$ or $E_\lambda(\varphi) = V$. Since not all eigenspaces can be zero, we conclude that there is an eigenvalue $\lambda$ with $E_\lambda(\varphi) = V$, meaning $\varphi = \lambda \operatorname{id}_V$.

Assume $f \neq 0$. We now claim that $\lambda > 0$. Indeed, note that for all $v \in V$ we have

$$
\begin{aligned}
\lambda \|v\|^2 &= \langle \varphi(v)|v \rangle \\
&= \langle f^* \circ f(v)|v \rangle \\
&= \langle f(v)|f(v) \rangle \\
&= \|f(v)\|^2.
\end{aligned}
$$

Thus, if $v \in V$ is any vector with $f(v) \neq 0$, then we obtain $\lambda = \left( \frac{\|f(v)\|}{\|v\|} \right)^2 > 0$.

Now define $g : V \to V'$ as $g = \lambda^{-\frac{1}{2}} f$. $g$ is clearly still an intertwiner. We can also show it is an isometry:

$$
\begin{aligned}
\langle g(v)|g(w) \rangle &= \lambda^{-1} \langle f(v)|f(w) \rangle \\
&= \lambda^{-1} \langle \varphi(v)|w \rangle \\
&= \lambda^{-1} \lambda \langle v|w \rangle \\
&= \langle v|w \rangle.
\end{aligned}
$$

Note that since $V'$ is irreducible and $f(V) \subseteq V'$ topologically closed due to $V$ being finite-dimensional, we necessarily have that $f$ is surjective. Thus, we have shown that $\mu f$ with $\mu := \lambda^{-\frac{1}{2}}$ is an isomorphism of unitary representations. $\square$

**Proposition B.30** (Schur's Orthogonality). *Let $\rho : G \to \mathrm{U}(V)$ and $\rho' : G \to \mathrm{U}(V')$ be non-isomorphic irreducible unitary representations of the compact group $G$, of which at least one is finite-dimensional. Let $\rho^{uv}$ and $\rho'^{u'v'}$ be matrix coefficients of them, which are functions in $L^2_{\mathbb{K}}(G)$ due to their continuity. Then they are orthogonal, i.e., $\left\langle \rho^{uv} \middle| \rho'^{u'v'} \right\rangle = 0$.*

*Proof.* Without loss of generality, we can assume $V'$ to be finite-dimensional. Assume that $l : V' \to V$ is *any* linear function. We can associate to it the function $f : V' \to V$ given by

$$f(w') := \int_G \rho(g) l \rho'(g)^{-1} w' dg.$$

For all $h \in G$ we have

$$
\begin{aligned}
\rho(h) f \rho'(h)^{-1} &= \int_G \rho(h)\rho(g) l \rho'(g)^{-1} \rho'(h)^{-1} dg \\
&= \int_G \rho(hg) l \rho'(hg)^{-1} dg \\
&= \int_G \rho(g) l \rho'(g)^{-1} dg \\
&= f,
\end{aligned}
$$

and thus $\rho(h)f = f\rho'(h)$, which means that $f$ is an intertwiner. In this derivation, $\rho(h)$ could be put insight the integral since $\rho(h)$ is continuous and an integral is a limit over finite sums, which commutes with the continuous $\rho(h)$. By Schur's Lemma B.29, we necessarily have $f = 0$. Now look at the specific linear function $l : V' \to V$ given by $l(w') := \langle v'|w'\rangle\, v$ with the fixed vectors $v, v'$ corresponding to the matrix coefficients. We obtain $f = 0$, for $f$ defined as before, and thus:

$$
\begin{aligned}
0 = \langle u|f(u')\rangle = \Big\langle u\Big| \int_G \rho(g)l\rho'(g)^{-1}(u')dg \Big\rangle \\
= \int_G \big\langle u\big|\rho(g)l\rho'(g)^{-1}(u')\big\rangle\, dg \\
= \int_G \Big\langle u\Big|\rho(g)\left[\langle v'|\rho'(g)^{-1}(u')\rangle\, v\right]\Big\rangle dg \\
= \int_G \langle u|\rho(g)(v)\rangle \cdot \langle v'|\rho'(g)^{-1}(u')\rangle\, dg \\
= \int_G \overline{\langle u|\rho(g)(v)\rangle} \cdot \overline{\langle u'|\rho'(g)(v')\rangle} dg \\
= \int_G \overline{\rho^{uv}(g)}\rho'^{u'v'}(g)dg \\
= \big\langle \rho^{uv}\big|\rho'^{u'v'}\big\rangle
\end{aligned}
$$

In this derivation, the integral could be put out of the scalar product since the scalar product is continuous, see Proposition F.38, and since integrals are certain limits over finite sums, with which the scalar product commutes. □

Note that there are more general Schur's orthogonality relations in the case that $\mathbb{K} = \mathbb{C}$, see Knapp (2002), Corollary 4.10. These then engage with the matrix coefficients of one and the same representation. This, together with a version of Schur's lemma that only holds over $\mathbb{C}$ leads to the strengthening of the Peter-Weyl theorem that shows that the multiplicities $n_l$ are given by $\dim(V_l)$.

**Proposition B.31.** *All irreducible unitary representations of a compact group $G$ are finite-dimensional.*

*Proof.* Assume $\rho : G \to \mathrm{U}(V)$ was an irreducible unitary representation on an infinite-dimensional space $V$. Let $\rho^{uv}$ be any of its matrix coefficients. By Proposition B.30, and since an infinite-dimensional representation can never be isomorphic to a finite-dimensional representation, $\rho^{uv}$ is perpendicular to all matrix coefficients of finite-dimensional irreducible unitary representations. Due to the linearity of the scalar product, $\rho^{uv}$ is perpendicular to the whole linear span of these matrix coefficients and thus to the topological closure of this span. The last step follows from the continuity of the scalar product, see Proposition F.38. By Theorem B.27 this closure is the whole space $L^2_{\mathbb{K}}(G)$. Therefore, $\rho^{uv}$ is even perpendicular to itself, and thus $\rho^{uv} = 0$.

Overall, for arbitrary $u, v \in V$ and $g \in G$ we obtain $0 = \rho^{uv}(g) = \overline{\langle u|\rho(g)(v)\rangle}$ and thus (by setting $u = \rho(g)(v)$) $\rho(g)(v) = 0$ and consequently $\rho(g) = 0$. We obtain $\rho = 0$, a contradiction. Thus infinite-dimensional irreducible unitary representations cannot exist. □

As a consequence, we mention that the finiteness conditions in Schur's lemma and Schur's Orthogonality were not necessary to state since all irreducible unitary representations are finite-dimensional anyway. We obtain from this and from Schur's Lemma B.29 that isomorphism classes and equivalence classes of irreducible unitary representations are one and the same.

### B.2.3 A PROOF OF THE PETER-WEYL THEOREM FOR THE REGULAR REPRESENTATION

In this section, we engage with the Peter-Weyl theorem for the regular representation on $L^2_{\mathbb{K}}(G)$. The case of $L^2_{\mathbb{K}}(X)$ for a homogeneous space $X$ will be dealt with in Section B.2.4. The core arguments in the proofs of this section are adapted from Williams (1991).

As before, let $\widehat{G}$ be the set of isomorphism classes of irreducible representations of $G$. For $l \in \widehat{G}$ let $\rho_l$ be a representative for the isomorphism class $l$. Furthermore, for each $\rho_l : G \to \mathrm{U}(V_l)$,

let $v_l^1, \ldots, v_l^{\dim(V_l)}$ be an arbitrary orthonormal basis, which exists due to Proposition F.41 (mostly written without the superscript, i.e., as $v^1, v^2, \ldots$, if the corresponding isomorphism class is clear). Denote $\rho_l^{ij} := \rho_l^{v^i v^j}$. Remember that matrix coefficients of unitary representations are continuous by Remark B.25, and thus functions in $L_\mathbb{K}^2(G)$. Then, let $\mathcal{E} \subseteq L_\mathbb{K}^2(G)$ be the linear span of the matrix coefficients of all irreducible unitary representations. In the next Lemma, we want to show that $\mathcal{E}$ is already spanned by the matrix coefficients corresponding to representatives of isomorphism classes and their orthonormal bases:

**Lemma B.32.** *We have*

$$\mathcal{E} = \mathrm{span}_\mathbb{K} \left\{ \rho_l^{ij} \mid l \in \widehat{G}, i, j \in \{1, \ldots, \dim(V_l)\} \right\}.$$

*Proof.* First, we show that isomorphic representations don't add distinct matrix coefficients. Thus, let $\rho \cong \rho_l$ and let $f : V \to V_l$ be the corresponding isomorphism. Then we have $\rho_l(g) \circ f = f \circ \rho(g)$ and thus, since $f$ is a unitary transformation, $\rho(g) = f^* \circ \rho_l(g) \circ f$, for all $g \in G$, see Proposition F.44. Now let $u, v \in V$ be arbitrary. We obtain

$$\begin{aligned}
\rho^{uv}(g) &= \overline{\langle u | \rho(g)(v) \rangle} \\
&= \overline{\langle u | f^* \rho_l(g) f(v) \rangle} \\
&= \overline{\langle f(u) | \rho_l(g)(f(v)) \rangle} \\
&= \rho_l^{f(u) f(v)}(g),
\end{aligned}$$

which proves the first claim. Now we want to show that we only need to consider the $\rho_l^{ij}$. Thus, let $u, v \in V_l$ be arbitrary. They allow for linear combinations

$$u = \sum_i \lambda^i v^i, \; v = \sum_i \mu^i v^i$$

with coefficients $\lambda^i, \mu^i \in \mathbb{K}$. We obtain:

$$\begin{aligned}
\rho_l^{uv}(g) &= \overline{\langle u | \rho_l(g)(v) \rangle} \\
&= \sum_i \sum_j \lambda^i \overline{\mu^j} \cdot \overline{\langle v^i | \rho_l(g)(v^j) \rangle} \\
&= \left( \sum_i \sum_j \lambda^i \overline{\mu^j} \rho_l^{ij} \right)(g),
\end{aligned}$$

thus showing that $\rho_l^{uv}$ is in the linear span of the matrix coefficients corresponding to the orthonormal basis. This concludes the proof. $\square$

For an isomorphism class $l \in \widehat{G}$, let $\mathcal{E}_l := \mathrm{span} \left\{ \rho_l^{ij} \mid i, j \in \{1, \ldots, \dim(V_l)\} \right\} \subseteq L_\mathbb{K}^2(G)$ be the linear subspace of $\mathcal{E}$ generated by matrix coefficients corresponding to $l$. Let furthermore for all $j$ the space $\mathcal{E}_l^j \subseteq \mathcal{E}_l$ be the subspace generated by all $\rho_l^{ij}$ for $i \in \{1, \ldots, \dim(V_l)\}$. In the next lemma, we prove that these are actually closed subrepresentations of the regular representation.

**Lemma B.33.** *For $j \in \{1, \ldots, \dim(V_l)\}$, $\mathcal{E}_l^j$ is a closed invariant subspace of $L_\mathbb{K}^2(G)$. In particular, $\mathcal{E}_l$ is a closed invariant subspace of $L_\mathbb{K}^2(G)$.*

*Proof.* Closedness follows immediately since this space is finite-dimensional and thus complete, see Proposition F.47. We need to show that $\lambda(g)\rho_l^{ij} \in \mathcal{E}_l^j$ for all $g \in G$ and all $i, j$. We can compute

this directly:

$$
\begin{aligned}
\left(\lambda(g)\rho_l^{ij}\right)(g') &= \rho_l^{ij}(g^{-1}g') \\
&= \overline{\langle v^i | \rho_l(g^{-1}g')(v^j)\rangle} \\
&= \overline{\langle \rho_l(g)(v^i) | \rho_l(g')(v^j)\rangle} \\
&= \overline{\left\langle \sum_{i'} \langle v^{i'} | \rho_l(g)(v^i)\rangle\, v^{i'} \,\middle|\, \rho_l(g')(v^j)\right\rangle} \\
&= \sum_{i'} \overline{\langle v^{i'} | \rho_l(g)(v^i)\rangle} \cdot \overline{\langle v^{i'} | \rho_l(g')(v^j)\rangle} \\
&= \sum_{i'} \overline{\langle v^i | \rho_l(g^{-1})(v^{i'})\rangle} \rho_l^{i'j}(g') \\
&= \left(\sum_{i'} \rho_l^{ii'}(g^{-1})\rho_l^{i'j}\right)(g')
\end{aligned}
$$

where the coefficients $\rho_l^{ii'}(g^{-1})$ do not depend on $g'$. Consequently, $\lambda(g)\rho_l^{ij} \in \mathcal{E}_l^j$. $\qquad\square$

**Lemma B.34.** *Let $\rho : G \to \mathrm{U}(V)$ and $\rho' : G \to \mathrm{U}(V')$ be unitary representations, $\rho$ being irreducible and $V' \neq 0$. Furthermore, assume that $f : V \to V'$ is a surjective intertwiner. Then $V'$ is also irreducible and $f$ an equivalence.*

*Proof.* Assume by contradiction that $V'$ is reducible. Thus, there is a nontrivial closed invariant subspace $0 \subsetneq W \subsetneq V'$. Now the following can easily be checked:

1. $0 \subsetneq f^{-1}(W) \subsetneq V$.

2. $f^{-1}(W)$ is an invariant subspace of $V$.

3. $f^{-1}(W)$ is a closed subset of $V$.

Once we have this, we have a contradiction to the fact that $V$ is irreducible.

1 and 2 can be checked by the reader, and 3 follows since $V$ is, as an irreducible representation, finite-dimensional by Proposition B.31 and thus every subspace is closed by Proposition F.47.

Therefore, we know that $V'$ is irreducible. Now use Schur's Lemma B.29 to conclude that $f$, being nonzero, necessarily is an equivalence. $\qquad\square$

**Proposition B.35.** *There is an equivalence of representations $f_l^j : V_l \to \mathcal{E}_l^j$ given on the orthonormal basis by $f_l^j(v^i) = \rho_l^{ij}$. Consequently, there is an isomorphism $V_l \cong \mathcal{E}_l^j$ of unitary representations.*

*Proof.* We need to show that $f_l^j$ is equivariant. Using the result of the derivation of Lemma B.33, we compute

$$
\begin{aligned}
f_l^j(\rho_l(g)(v^i)) &= f_l^j\left(\sum_{i'} \langle v^{i'} | \rho_l(g)(v^i)\rangle v^{i'}\right) \\
&= \sum_{i'} \langle v^{i'} | \rho_l(g)(v^i)\rangle f_l^j(v^{i'}) \\
&= \sum_{i'} \overline{\langle v^i | \rho_l(g^{-1})(v^{i'})\rangle} \rho_l^{i'j} \\
&= \sum_{i'} \rho_l^{ii'}(g^{-1})\rho_l^{i'j} \\
&= \lambda(g)\rho_l^{ij} \\
&= \lambda(g)\left(f_l^j(v^i)\right),
\end{aligned}
$$

so $f_l^j \circ \rho_l(g) = \lambda(g) \circ f_l^j$ for all $g \in G$, which is what we wanted to show. That $f$ is an intertwiner also requires it to be continuous: this follows since $V_l$ is finite-dimensional, and so all linear functions on it are continuous.

Now, that $f_l^j$ is even an equivalence follows from Lemma B.34 by noting that $\mathcal{E}_l^j \neq 0$. Indeed, if it was zero then we would have $\rho_l^{ij}(g) = 0$ for all $i$, and thus $\rho(g)$ would not be invertible, in contrast that it is a unitary automorphism.

Thus, there is even an isomorphism $V_l \cong \mathcal{E}_l^j$ by Schur's Lemma B.29. $\qquad\square$

**Lemma B.36.** *Let $\rho : G \to \mathrm{U}(V)$ be a unitary representation. Let $V_1 \subseteq V$ be a subrepresentation. Then the orthogonal complement $V_1^\perp$ is a subrepresentation as well.*

*Proof.* We have $\langle v | v_1 \rangle = 0$ for all $v \in V_1^\perp$ and all $v_1 \in V_1$. Now, let $g \in G$ be arbitrary. From the unitarity of $\rho$ we obtain

$$\langle \rho(g)(v) | v_1 \rangle = \langle v | \rho(g^{-1})(v_1) \rangle = 0.$$

The last step follows from $\rho(g^{-1})(v_1) \in V_1$, which holds since $V_1$ is a subrepresentation. Overall, this shows $\rho(g)(v) \in V_1^\perp$ as well, and so this is a subrepresentation. $\qquad\square$

**Lemma B.37.** *Let $\rho : G \to \mathrm{U}(V)$ be a finite-dimensional unitary representation. Furthermore, assume that $W_1, W_2$ are irreducible subrepresentations. If they are not isomorphic, then they are perpendicular, i.e., $\langle w_1 | w_2 \rangle = 0$ for all $w_1 \in W_1$ and $w_2 \in W_2$.*

*Proof.* Let $P : V \to W_1$ be the orthogonal projection from $V$ to $W_1$, defined as the adjoint of the canonical inclusion $i : W_1 \to V$, i.e., defined by the property

$$\langle w_1 | P(v) \rangle = \langle i(w_1) | v \rangle = \langle w_1 | v \rangle$$

for all $v \in V$ and $w_1 \in W_1$, see also Proposition F.46. We now show that $P$ is equivariant. For all $g \in G$, $v \in V$ and $w_1 \in W_1$ we have:

$$\begin{aligned}
\langle w_1 | P(\rho(g)(v)) \rangle &= \langle w_1 | \rho(g)(v) \rangle \\
&= \langle \rho(g^{-1})(w_1) | v \rangle \\
&= \langle \rho(g^{-1})(w_1) | P(v) \rangle \\
&= \langle w_1 | \rho(g)(P(v)) \rangle,
\end{aligned}$$

where we used in the third step that $W_1$ is a subrepresentation. Since this holds for all $w_1 \in W_1$, we obtain $P(\rho(g)(v)) = \rho(g)(P(v))$ by Proposition F.45 and overall that $P$ is equivariant.

In particular, also the restriction $P|_{W_2} : W_2 \to W_1$ is equivariant. Since $W_1$ and $W_2$ are not isomorphic, we obtain by Schur's Lemma B.29 that $P|_{W_2} = 0$, i.e., for all $w_1 \in W_1$ and $w_2 \in W_2$ we have $\langle w_1 | w_2 \rangle = \langle w_1 | P|_{W_2}(w_2) \rangle = \langle w_1 | 0 \rangle = 0$. Thus, $W_1$ and $W_2$ are perpendicular as claimed. $\qquad\square$

**Proposition B.38.** *Let $\rho : G \to \mathrm{U}(V)$ be any finite-dimensional unitary representation. Then $V$ decomposes into an orthogonal direct sum*

$$V = \bigoplus_{i=1}^n V_i$$

*such that $V_i \subseteq V$ are irreducible subrepresentations of $\rho$.*

*Proof.* Let $V_1$ be any irreducible subrepresentation of $V$: This can be obtained by noting that if $V$ is not already irreducible (in which case $V_1 = V$), then we find a nontrivial subrepresentation $0 \subsetneq W \subsetneq V$. By iteratively proceeding with $W$, we eventually need to reach an irreducible representation since $V$ is finite-dimensional.

Now, let $V_1^\perp$ be the orthogonal complement of $V_1$. From Lemma B.36 we know that this is a subrepresentation of $V$. By induction on the dimension of $V$, and since $V_1^\perp$ has strictly smaller dimension, we can assume that $V_1^\perp$ already splits into an orthogonal direct sum of irreducible subrepresentations $V_1^\perp = \bigoplus_{i=2}^n V_i$, and overall, $V = \bigoplus_{i=1}^n V_i$ is the decomposition we were looking for. $\qquad\square$

The following proposition will not be used now, but we make use of it later when showing that there are only finitely many basis kernels in a steerable CNN for a compact group:

**Proposition B.39** (Krull-Remak-Schmidt Theorem). *In the situation of Proposition B.38, the orthogonal direct sum decomposition is essentially unique. That is, the type and multiplicities of the irreducible direct summands is always the same.*

*Proof.* If one has one decomposition of $V$ in which an irreducible representation $U$ does not appear, then it cannot appear in *any* decomposition since $U$ would be perpendicular to all the irreps in the decomposition of $V$ by Lemma B.37 and thus zero. Therefore, the types of irreducible representations is always the same. That the multiplicities are always the same follows by the same argument and for dimension-reasons. $\qquad\square$

We can now finally prove The Peter-Weyl Theorem B.22 for the case that $X = G$:

*Proof.* By Proposition B.38 and Lemma B.33 there is some orthogonal decomposition $\mathcal{E}_l = \bigoplus_{i=1}^{n_l} V_{li}$ into irreducible invariant subspaces. Now assume that there is an $i$ such that $V_{li} \not\cong V_l$. By Proposition B.35 this means that $V_{li} \not\cong \mathcal{E}_l^j$ for all $j = 1, \ldots, \dim(V_l)$. By Lemma B.37 we obtain $V_{li} \perp \mathcal{E}_l^j$ for all $j$ and thus, since $\sum_j \mathcal{E}_l^j = \mathcal{E}_l$, we obtain $V_{li} \perp \mathcal{E}_l$ and overall $V_{li} = 0$, a contradiction.

Thus, the assumption was wrong and all $V_{li}$ in the orthogonal direct sum are isomorphic to $V_l$.

Now let $l \neq l'$ and $i, j$ be arbitrary. We have $\mathcal{E}_l \perp \mathcal{E}_{l'}$ by Proposition B.30, and thus in particular $V_{li} \perp V_{l'j}$. Furthermore, we have $n_l \leq \dim(V_l)$ since $\mathcal{E}_l = \sum_{j=1}^{\dim(V_l)} \mathcal{E}_l^j = \bigoplus_{i=1}^{n_l} V_{li}$, and $\dim(V_l) < \infty$ by Proposition B.31.

Moreover, we have $\bigoplus_{l \in \widehat{G}} \bigoplus_{i=1}^{n_l} V_{li} = \bigoplus_{l \in \widehat{G}} \mathcal{E}_l = \mathcal{E}$, which is topologically dense in $L_{\mathbb{K}}^2(G)$ by Theorem B.27.

Finally, that $n_l = \dim(V_l)$ if $\mathbb{K} = \mathbb{C}$ follows by invoking a stronger version of Schur's orthogonality than we have developed, and which works only over the complex numbers (Knapp, 2002). $\qquad\square$

### B.2.4 A PROOF OF THE PETER-WEYL THEOREM FOR GENERAL $L_{\mathbb{K}}^2(X)$

Now let $X$ be a homogeneous space of $G$. Then, as mentioned in Section B.1.3, there is a measure $\mu$ on $X$ which is left-$G$-invariant (Nachbin & Bechtolsheim, 1965) in the sense that we have for all $g \in G$ and all square-integrable functions $f \in L_{\mathbb{K}}^2(X)$:

$$\int_X f(g \cdot x) dx = \int_X f(x) dx.$$

Furthermore, let $\pi : G \to X$ be the projection given by $g \mapsto gx^*$ for a fixed element $x^* \in X$. One important result is that there is a Fubini-like theorem for evaluation of integrals on $G$ using the invariant measure on $X$. Namely, for arbitrary $x \in X$, let $g(x) \in G$ be any lift, i.e., any element in $G$ with $\pi(g(x)) = x$. This exists since the action is transitive. Let $H := G_{x^*} \subseteq G$ be the stabilizer subgroup. For a square-integrable function $f : G \to \mathbb{K}$, we can then construct the average $\mathrm{av}(f) : X \to \mathbb{K}$ by

$$\mathrm{av}(f)(x) := \int_H f(g(x)h) dh,$$

where we integrate using the Haar-measure on $H$.[11] If it is hard to understand why this is called an average, note that $X \cong G/H$, i.e., points in $X$ can be interpreted as cosets of $G$, and then the average just averages over cosets.[12]

This construction is well-defined, i.e., does not depend on the specific choice of the lift $g(x)$. Indeed, let $g(x)'$ be another lift of $x$. Then $g(x)' = g(x)h'$ for some $h' \in H$, since $H$ is the stabilizer

---

[11] Such a Haar measure exists since $H \subseteq G$ is a topologically closed subgroup of a compact group by Proposition B.21 and thus compact itself by standard topological results (Conway, 2014). Note that this measure fulfills $\mu(H) = 1$ and is thus *not* the same as the restriction of the measure on $G$ to $H$.

[12] Here, $G/H$ is the set of equivalence classes in $G$ with respect to the equivalence relation $g \sim g'$ if $g^{-1}g' \in H$, which has a quotient topology as explained in Definition F.11. The equivalence classes are given by the cosets $gH$ for $g \in G$.

subgroup. Consequently, using the invariance of the Haar measure, we see:

$$\int_H f(g(x)'h)dh = \int_H f(g(x)h'h)dh = \int_H f(g(x)h)dh,$$

and thus the well-definedness of the average $\mathrm{av}(f) : X \to \mathbb{K}$. Integration of $f$ on the whole of $G$ is a "complete" average, and thus we can hope that averaging $\mathrm{av}(f)$ leads to this complete integral. This is indeed the case, i.e., $\mathrm{av}(f)$ is square-integrable on $X$ and one has (Nachbin & Bechtolsheim, 1965)

$$\int_G f(g)dg = \int_X \mathrm{av}(f)(x)dx. \tag{9}$$

We will use this important result later in order to see that $L^2_{\mathbb{K}}(X)$ embeds with good properties into $L^2_{\mathbb{K}}(G)$.

We now want to prove the Peter-Weyl theorem for $L^2_{\mathbb{K}}(X)$. We first present a general argument showing an orthogonal decomposition of $L^2_{\mathbb{K}}(X)$ into irreducible subspaces, and then use a specific argument to deduce that the multiplicities of irreducible subrepresentations are necessarily bounded by the multiplicities in $L^2_{\mathbb{K}}(G)$.

**Proposition B.40.** *Let* $\rho : G \to \mathrm{U}(V)$ *be any* unitary representation. *Then there is a dense subrepresentation which splits as an orthogonal direct sum of irreducible subrepresentations.*

*Proof.* We sketch the proof in Kowalski (2014), Corollary 5.4.2. In this book, the proof is done only for the complex numbers $\mathbb{C}$, but it is obvious that each step carries over without any changes to arbitrary $\mathbb{K} \in \{\mathbb{R}, \mathbb{C}\}$. The rough steps are as follows:

1. From $\rho$ one builds a function $\overline{\rho} : L^2_{\mathbb{K}}(G) \to \mathrm{Hom}_{\mathbb{K}}(V, V)$, given by $\overline{\rho}(\varphi)(v) = \int_G \varphi(g)\rho(g)(v)dg$. This is analogous to our construction of kernel operators (special representation operators) from kernels, which we will handle in the next chapter, See Theorem C.7.

2. Given $v \in V$ fixed, one obtains the function $\overline{\rho}^v : L^2_{\mathbb{K}}(G) \to V$, $\varphi \mapsto \overline{\rho}(\varphi)(v)$. One can check easily that this is an intertwiner.

3. For each finite-dimensional subrepresentation $E \subseteq L^2_{\mathbb{K}}(G)$, the image $\overline{\rho}^v(E) \subseteq V$ is a finite-dimensional subrepresentation of $V$.

4. For $v \neq 0$, using analytical arguments and the Peter-Weyl theorem for $L^2_{\mathbb{K}}(G)$, one can prove that there is an $E$ such that $\overline{\rho}^v(E) \subseteq V$ is not zero.

Having that, one can use Proposition B.38 in order to deduce that $\overline{\rho}^v(E)$ contains an irreducible subrepresentation, and so does $V$.

With this at hand, one can proceed inductively as follows: Given an irreducible subrepresentation $V_1 \subseteq V$, one can consider the orthogonal complement $V_1^\perp$, which is by Lemma B.36 again a subrepresentation of $V$. Thus, this also has, by the same argument as above, an irreducible subrepresentation $V_2$ and so on. By induction (or better: using Zorn's Lemma), one can then "fill up" $V$ with orthogonal irreducible subrepresentations, deducing the result. $\square$

Consequently, since $L^2_{\mathbb{K}}(X)$ carries a unitary representation of $G$ by $[\lambda(g)(\varphi)](x) := \varphi(g^{-1}x)$, we can deduce that it contains a dense subrepresentation which splits as an orthogonal direct sum of irreducible subrepresentations. But we would like to know more details about this, in particular the multiplicities of the irreps. For this to work, we want to embed $L^2_{\mathbb{K}}(X)$ into $L^2_{\mathbb{K}}(G)$ and thus deduce a more specific result from the decomposition of $L^2_{\mathbb{K}}(G)$.

Let as before $x^* \in X$ be an arbitrary point and let $\pi : G \to X$ be the projection given by $\pi(g) := gx^*$. Consider the function $\pi^* : L^2_{\mathbb{K}}(X) \to L^2_{\mathbb{K}}(G)$ given by $\pi^*(\varphi) := \varphi \circ \pi$. It is unclear a priori whether this is well-defined: For example, it might be that an $f : X \to \mathbb{K}$ which is zero outside a measure 0 set gets lifted to $\pi^*(f) : G \to \mathbb{K}$ which does not have this property, and thus $\pi^*$ would not be an actual function.[13] Thus, we need some lemmas:

---

[13]Remember that functions in $L^2_{\mathbb{K}}(X)$ for any measurable space $X$ are identified if they agree outside a set of measure 0.

**Lemma B.41.** *Let $f : X \to \mathbb{K}$ be square-integrable. Then we have $\mathrm{av}(\pi^*(f)) = f$.*

*Proof.* Using Eq. (9) and that $H$ is the stabilizer subgroup we compute:

$$
\begin{aligned}
\mathrm{av}(\pi^*(f))(x) &= \int_H \pi^*(f)(g(x)h)dh \\
&= \int_H f(\pi(g(x)h))dh \\
&= \int_H f(\pi(g(x)))dh \\
&= \int_H f(x)dh \\
&= f(x)\int_H 1dh \\
&= f(x)\mu(H) \\
&= f(x).
\end{aligned}
$$

$\square$

**Lemma B.42.** *Let $A \subseteq X$ be any measurable set. Let $\mathbf{1}_A : X \to \{0,1\} \subseteq \mathbb{K}$ be its indicator function. Then $\pi^*(\mathbf{1}_A) = \mathbf{1}_{\pi^{-1}(A)}$.*

*Proof.* This can easily be checked. $\square$

**Lemma B.43.** *Let $\varphi : X \to \mathbb{K}$ be zero outside a measure zero set $A$. Then $\pi^*(\varphi)$ is zero outside $\pi^{-1}(A)$ which is also a measure zero set.*

*Proof.* If $g \notin \pi^{-1}(A)$ then $\pi(g) \notin A$ and thus:

$$
0 = \varphi(\pi(g)) = \pi^*(\varphi)(g)
$$

which proves the first statement. The second is shown as follows using both Lemmas B.41 and B.42 and Eq. (9):

$$
\begin{aligned}
\mu(\pi^{-1}(A)) &= \int_G \mathbf{1}_{\pi^{-1}(A)}(g)dg \\
&= \int_G \pi^*(\mathbf{1}_A)(g)dg \\
&= \int_X \mathrm{av}(\pi^*(\mathbf{1}_A))(x)dx \\
&= \int_X \mathbf{1}_A(x)dx \\
&= \mu(A) \\
&= 0,
\end{aligned}
$$

thus showing what was claimed. $\square$

Thus, our concern about well-definedness as a function is invalid and we can now prove an embedding result:

**Proposition B.44.** $\pi^* : L^2_{\mathbb{K}}(X) \to L^2_{\mathbb{K}}(G)$ *is a well-defined intertwiner and a unitary transformation, i.e., for all $\varphi, \psi \in L^2_{\mathbb{K}}(X)$ we have $\langle \pi^*(\varphi)|\pi^*(\psi)\rangle_{L^2_{\mathbb{K}}(G)} = \langle \varphi|\psi\rangle_{L^2_{\mathbb{K}}(X)}$.*

*Proof.* For well-definedness, we still need to show that $\pi^*(\varphi)$ is again square-integrable for square-integrable $\varphi : X \to \mathbb{K}$. This is indeed the case due to Eq. (9). Namely, let $|\pi^*(\varphi)|^2 : G \to \mathbb{K}$ and

consider its average $\mathrm{av}(|\pi^*(\varphi)|^2)$. Clearly, we have $|\pi^*(\varphi)|^2 = \pi^*(|\varphi|^2)$ and thus, using Lemma B.41, $\mathrm{av}(|\pi^*(\varphi)|^2) = |\varphi|^2$. We obtain:

$$
\begin{aligned}
\int_G |\pi^*(\varphi)|^2(g)dg &= \int_X \mathrm{av}(|\pi^*(\varphi)|^2)(x)dx \\
&= \int_X |\varphi(x)|^2 dx \\
&< \infty.
\end{aligned}
$$

Thus, $\pi^*$ is not only well-defined but even fulfills $\|\pi^*(\varphi)\|_{L^2_{\mathbb{K}}(G)} = \|\varphi\|_{L^2_{\mathbb{K}}(X)}$, which also shows the continuity of $\pi^*$. With similar arguments, we show that $\pi^*$ respects the whole scalar product, i.e., is a uniform transformation:

$$
\begin{aligned}
\langle \pi^*(\varphi)|\pi^*(\psi)\rangle_{L^2_{\mathbb{K}}(G)} &= \int_G \left( \overline{\pi^*(\varphi)} \cdot \pi^*(\psi) \right)(g)dg \\
&= \int_X \mathrm{av}(\overline{\pi^*(\varphi)} \cdot \pi^*(\psi))(x)dx \\
&= \int_X \overline{\varphi(x)}\psi(x)dx \\
&= \langle \varphi|\psi\rangle_{L^2_{\mathbb{K}}(X)}.
\end{aligned}
$$

The step from the second to the third line follows as before by noting that $\overline{\pi^*(\varphi)} \cdot \pi^*(\psi) = \pi^*(\overline{\varphi} \cdot \psi)$ and invoking Lemma B.41 again.

The linearity of $\pi^*$ is obvious, and the equivariance is done as follows: note that for arbitrary $g, g' \in G$ we have $\pi(g^{-1}g') = (g^{-1}g')x^* = g^{-1}(g'x^*) = g^{-1}\pi(g')$ and therefore:

$$
\begin{aligned}
\left[\pi^*(\lambda(g)\varphi)\right](g') &= (\lambda(g)\varphi)(\pi(g')) \\
&= \varphi(g^{-1}\pi(g')) \\
&= \varphi(\pi(g^{-1}g')) \\
&= \pi^*(\varphi)(g^{-1}g') \\
&= \left[\lambda(g)\pi^*(\varphi)\right](g').
\end{aligned}
$$

Thus, we shown everything which was to show. $\qquad\square$

Thus, $\pi^* : L^2_{\mathbb{K}}(X) \to L^2_{\mathbb{K}}(G)$ is an embedding which even preserves the scalar product. We can therefore view $L^2_{\mathbb{K}}(X)$ as a subspace: $L^2_{\mathbb{K}}(X) \subseteq L^2_{\mathbb{K}}(G)$.[14]

We can finally complete the proof of the Peter-Weyl Theorem B.22:

*Proof of Theorem B.22.* Assume that

$$
\bigoplus_{l\in\widehat{G}} \bigoplus_{i=1}^{m_l} V_{li} \subseteq L^2_{\mathbb{K}}(X) \subseteq L^2_{\mathbb{K}}(G)
$$

is a dense subspace such that the direct sum is orthogonal, where $V_{li} \cong V_l$ for all $l, i$. This exists by Proposition B.40.

Remember that $n_l$ denotes the multiplicity of $V_l$ as a subrepresentation in $L^2_{\mathbb{K}}(G)$. We now want to show that $m_l \leq n_l$. Since $V_{li}$ is perpendicular to all $\mathcal{E}_{l'}$ with $l' \neq l$ by Lemma B.37, $V_{li}$ must be contained in the orthogonal complement of $\bigoplus_{l'\neq l} \mathcal{E}_{l'}$. This is exactly $\mathcal{E}_l$, which we show in a final lemma after this proof. So $V_{li} \subseteq \mathcal{E}_l$ for all $i$. Thus, we obtain the result $m_l \leq n_l$ by dimension reasons. This was all there was left to show. $\qquad\square$

**Lemma B.45.** *We have* $\mathcal{E}_l = \left( \bigoplus_{l\neq l'\in\widehat{G}} \mathcal{E}_{l'} \right)^{\perp}$

---

[14]In this notation, we suppress that this embedding depends on the specific base point $x^*$ which was chosen. For another base point, the embedding differs by a unitary automorphism on $L^2_{\mathbb{K}}(G)$ as the reader may want to check.

*Proof.* We already know $\mathcal{E}_l \subseteq \left( \bigoplus_{l \neq l' \in \widehat{G}} \mathcal{E}_{l'} \right)^\perp$ from Proposition B.30. Now, assume this inclusion is not an equality. Then there is $v \notin \mathcal{E}_l$ such that $v \in \left( \bigoplus_{l \neq l' \in \widehat{G}} \mathcal{E}_{l'} \right)^\perp$. The space $\mathrm{span}_{\mathbb{K}}(v, \mathcal{E}_l)$ does contain an orthonormal basis by Proposition F.41, where the procedure of Gram-Schmidt orthonormalization allows starting with an orthonormal basis of $\mathcal{E}_l$ and to fill it up to one of the whole space $\mathrm{span}_{\mathbb{K}}(v, \mathcal{E}_l)$. Thus, we can assume $v \in \mathcal{E}_l^\perp$ as well. Overall, $v \in \left( \bigoplus_{l' \in \widehat{G}} \mathcal{E}_{l'} \right)^\perp$, and by taking topological closure and using that the scalar product is continuous by Proposition F.38, obtain $v \in \left( \widehat{\bigoplus}_{l' \in \widehat{G}} \mathcal{E}_{l'} \right)^\perp = (L^2_{\mathbb{K}}(G))^\perp$ by the Peter-Weyl theorem for the regular representation. This means $v = 0 \in \mathcal{E}_l$, a contradiction to $v \notin \mathcal{E}_l$.

Thus, our assumption is wrong and such a vector $v$ cannot exist. We obtain the equality as desired. $\square$

## C  THE CORRESPONDENCE BETWEEN STEERABLE KERNELS AND REPRESENTATION OPERATORS

In this chapter, we formulate and prove Theorem C.7, which gives a precise one-to-one correspondence between steerable kernels on the one hand, and certain representation operators which we call *kernel operators* on the other hand. Representation operators are a representation-theoretic abstraction of the scalar, vector and tensor operators from physics, that were explained in Section 2. The correspondence will allow us to prove a Wigner-Eckart theorem for steerable kernels in Chapter D and, ultimately, to obtain a complete description of steerable kernel bases. We formulate the correspondence in Section C.1, while Section C.2 gives a detailed and rigorous proof of it.

As in Chapter B, $\mathbb{K}$ is either of the two fields $\mathbb{R}$ or $\mathbb{C}$.

### C.1  FUNDAMENTALS OF THE CORRESPONDENCE

In Section C.1, we formulate the correspondence between steerable kernels and special representation operators that we name kernel operators. We do this by first studying steerable CNNs and the kernel constraint in Section C.1.1, which progressively leads us to consider steerable kernels on homogeneous spaces of general compact groups in Section C.1.2. This abstract formulation of steerable kernels will show apparent similarities to the concept of representation operators in Section C.1.3. We study them in purely representation-theoretic terms in Section C.1.4. However, they importantly differ in the fact that steerable kernels are not linear, whereas representation operators are – this is a difference that we need to bridge. Finally, after defining kernel operators as special representation operators, we give the formulation of the correspondence in Theorem C.7 in Section C.1.5 and shortly give some intuitions about why it is true.

### C.1.1  STEERABLE KERNELS AND THE RESTRICTION TO HOMOGENEOUS SPACES

The concept of steerable CNNs outlined here follows (Weiler et al., 2018a; Weiler & Cesa, 2019). In a nutshell, they work as follows:

The network is supposed to process feature fields $f : \mathbb{R}^d \to \mathbb{K}^c$ with $d \in \mathbb{N}$. $c$ is the dimension of the features themselves, i.e., the number of channels. For example, planar RGB-images correspond to the case $d = 2$ and $c = 3$.

Furthermore, a compact group $G$ (Definition B.4) is considered that acts on $\mathbb{R}^d$, for example, the special orthogonal group $\mathrm{SO}(d)$, the orthogonal group $\mathrm{O}(d)$ or the finite groups $\mathrm{C}_N$ or $\mathrm{D}_N$ if $d = 2$.[15] Then for each layer, the input and output features have a certain *type*, i.e., representation, which may differ from layer to layer. That is, the input (and output as well) consists of a function $f : \mathbb{R}^d \to \mathbb{K}^c$, and $G$ acts on $\mathbb{K}^c$ with a linear representation $\rho$, see Definition B.10. This action induces an action

---

[15]We will study some of these groups in the Examples in Chapter E.

of the semi-direct product $(\mathbb{R}^d, +) \rtimes G$ on the space of all signals,[16] where $t \in (\mathbb{R}^d, +)$ and $g \in G$:

$$\left( \left[ \text{Ind}_G^{\mathbb{R}^d \rtimes G} \rho \right] (tg) \cdot f \right)(x) := \rho(g) \cdot f(g^{-1}(x - t)).$$

Let the kernel that "maps" between the layers by convolution[17] be given by a function

$$K : \mathbb{R}^d \to \mathbb{K}^{c_{\text{out}} \times c_{\text{in}}}.$$

That is, for an input $f_{\text{in}} : \mathbb{R}^d \to \mathbb{K}^{c_{\text{in}}}$, the output $f_{\text{out}} : \mathbb{R}^d \to \mathbb{K}^{c_{\text{out}}}$ is given by

$$f_{\text{out}}(x) = [K \star f_{\text{in}}](x) = \int_{\mathbb{R}^d} K(y) f_{\text{in}}(x + y) dy,$$

where $K(y) \in \mathbb{K}^{c_{\text{out}} \times c_{\text{in}}}$ acts for any $y \in \mathbb{R}^d$ as a linear transformation from $\mathbb{K}^{c_{\text{in}}}$ to $\mathbb{K}^{c_{\text{out}}}$.

The goal is now to find kernels $K$ such that convolution with these kernels commutes with the induced actions on the input and output fields. That is, for all input fields $f_{\text{in}}$ and for all $t \in \mathbb{R}^d$ and $g \in G$ we want the following property:

$$K \star \left( \left[ \text{Ind}_G^{\mathbb{R}^d \rtimes G} \rho_{\text{in}} \right] (tg) \cdot f_{\text{in}} \right) = \left[ \text{Ind}_G^{\mathbb{R}^d \rtimes G} \rho_{\text{out}} \right] (tg) \cdot (K \star f_{\text{in}}).$$

It was shown in Weiler et al. (2018a) that a kernel $K$ has this equivariance property if and only if the kernel satisfies a certain constraint. We are rederiving it here for convenience.

Writing out both sides we obtain the following equality that needs to hold for all $f_{\text{in}}$ and all $x, t \in \mathbb{R}^d$:

$$\int_{\mathbb{R}^d} K(y) \rho_{\text{in}}(g) f_{\text{in}}(g^{-1}(x + y - t)) dy = \rho_{\text{out}}(g) \int_{\mathbb{R}^d} K(y) f_{\text{in}}(g^{-1}(x - t) + y) dy.$$

Substituting $y = g^{-1}y$ on the left side and using $|\det g| = 1$ due to the compactness of $G$, and putting $\rho_{\text{out}}(g)$ inside the integral on the right side, which is possible due to linearity, we obtain:

$$\int_{\mathbb{R}^d} \left[ K(gy) \rho_{\text{in}}(g) \right] f_{\text{in}}(g^{-1}x - g^{-1}t + y) dy = \int_{\mathbb{R}^d} \left[ \rho_{\text{out}}(g) K(y) \right] f_{\text{in}}(g^{-1}x - g^{-1}t + y) dy.$$

Since this needs to hold for all fields $f_{\text{in}}$, we necessarily have $K(gx) \rho_{\text{in}}(g) = \rho_{\text{out}}(g) K(x)$ for all $x \in \mathbb{R}^d$ and all $g \in G$ and obtain the kernel constraint

$$K(gx) = \rho_{\text{out}}(g) \circ K(x) \circ \rho_{\text{in}}(g)^{-1}. \tag{10}$$

This work will create a general theory for how to *solve* this kernel constraint, which means to find a parameterization for the space of all kernels that fulfill this constraint. We now explain how to make this problem more tractable: formally, the action of $G$ on $\mathbb{R}^d$ is a group action as in Definition B.5. However, it cannot be transitive as in Definition B.7 since $G$ is compact and $\mathbb{R}^d$ is not. Thus $\mathbb{R}^d$ splits into a disjoint union of orbits (Definition B.6), of the action:

$$\mathbb{R}^d = \bigsqcup_{k \in K} X_k.$$

That this is a *disjoint* union can be explained as follows: define the relation $\sim$ on $\mathbb{R}^d$ by $x \sim x'$ if $gx = x'$ for some $g \in G$. This is then an equivalence relation, and so $\mathbb{R}^d$ splits into a disjoint union of equivalence classes. One then can show that these equivalence classes are precisely the orbits of the group action. For example, such orbits take the form of spheres $S^{d-1}$ if $G = \text{SO}(d)$ or $G = \text{O}(d)$ and the form of a finite set of points if $G = C_N$ or $G = D_N$.

The idea is now that the kernel constraint 10 only constrains the behavior of the kernel at each orbit individually, and thus a solution on each orbit can be "patched together" to a solution on the whole

---

[16]The semidirect product $\mathbb{R}^d \rtimes G$ can be imagined as the smallest subgroup of the group of all isometries of $\mathbb{R}^d$ that contains both the translations $\mathbb{R}^d$ and the transformations $G$. It is not important to know the abstract definition of a semidirect product in our context.

[17]The operation is actually a so-called "correlation", but the term "convolution" is more widespread in the deep learning context and we follow this convention.

of $\mathbb{R}^d$. Indeed, assume that $K_k : X_k \to \mathbb{K}^{c_{\text{out}} \times c_{\text{in}}}$ individually fulfill the kernel constraint, which means that for all $x_k \in X_k$ and $g \in G$ we have

$$K_k(gx_k) = \rho_{\text{out}}(g) \circ K_k(x_k) \circ \rho_{\text{in}}(g)^{-1}.$$

Then, define the patch of these orbit-kernels by $K : \mathbb{R}^d \to \mathbb{K}^{c_{\text{out}} \times c_{\text{in}}}$ as $K(x) = K_k(x)$ if $x \in X_k$. This is well-defined since each $x$ is in precisely one orbit. Then clearly, $K$ satisfies the kernel constraint 10. Moreover, each kernel $K$ which fulfills the kernel constraint emerges from such a construction, since we can simply set $K_k := K|_{X_k}$. Overall, we see that we can restrict our attention to orbits. In Weiler et al. (2018b) and later Weiler et al. (2018a), a discretized implementation is done where the kernel is discretized into finitely many orbits with a smooth Gaussian radial profile. We will come back to these practical questions of parameterization in Remark D.19, once we have fully developed the theory of steerable CNNs.

### C.1.2   An Abstract Definition of Steerable Kernels

Motivated by the discussion in the last section, we now define steerable kernels in precise terms and will stick to that definition throughout this work. The definition will be more abstract than usual in the deep learning community, but we are rewarded since such an abstract definition makes it easier to apply representation-theoretic results.

Without loss of generality, we will in the rest of this work only consider kernels on orbits. Thus, let $X := G \cdot x$ be an arbitrary orbit. We consider steerable kernels $K : X \to \mathbb{K}^{c_{\text{out}} \times c_{\text{in}}}$. Note that the restriction of the action $G \times \mathbb{R}^d \to \mathbb{R}^d$ to $X$, written $G \times X \to X$, makes $X$ to a homogeneous space of $G$, see Definition B.7. Thus, instead of viewing $X$ as a subset of $\mathbb{R}^d$, we view $X$ as an arbitrary homogeneous space of an arbitrary compact group $G$. Notably, this framework is more general than usually studied in the context of steerable CNNs on $\mathbb{R}^d$, since we allow also groups that are not Lie groups and homogeneous spaces which are not naturally embedded in an $\mathbb{R}^d$, as well as finite homogeneous spaces of finite groups all at the same time.

Furthermore, we replace $\mathbb{K}^{c_{\text{in}}}$ and $\mathbb{K}^{c_{\text{out}}}$ by coordinate-independent $\mathbb{K}$-vector spaces $V_{\text{in}}$ and $V_{\text{out}}$, and therefore $\mathbb{K}^{c_{\text{out}} \times c_{\text{in}}}$ by the space of linear functions from $V_{\text{in}}$ to $V_{\text{out}}$, written $\text{Hom}_{\mathbb{K}}(V_{\text{in}}, V_{\text{out}})$. We assume there are linear representations $\rho_{\text{in}} : G \to \text{GL}(V_{\text{in}})$ and $\rho_{\text{out}} : G \to \text{GL}(V_{\text{out}})$.

Overall, this means that steerable kernels are certain maps $K : X \to \text{Hom}_{\mathbb{K}}(V_{\text{in}}, V_{\text{out}})$. The only property they need to fulfill is the kernel constraint $K(gx) = \rho_{\text{out}}(g) \circ K(x) \circ \rho_{\text{in}}(g)^{-1}$ for all $g \in G$ and $x \in X$. This can be viewed in representation-theoretic terms by defining the Hom-representation:

**Definition C.1** (Hom-Representation). Let $\rho_{\text{in}} : G \to \text{GL}(V_{\text{in}})$ and $\rho_{\text{out}} : G \to \text{GL}(V_{\text{out}})$ be two finite-dimensional $G$-representations over the field $\mathbb{K}$. The space $\text{Hom}_{\mathbb{K}}(V_{\text{in}}, V_{\text{out}})$ of $\mathbb{K}$-linear (not necessarily $G$-equivariant) functions from $V_{\text{in}}$ to $V_{\text{out}}$ also carries an induced $G$-representation, with action

$$[\rho_{\text{Hom}}(g)] (f) := \rho_{\text{out}}(g) \circ f \circ \rho_{\text{in}}(g)^{-1}.$$

We call this the *Hom-representation*.

*Remark* C.2. Of course, one needs to check that this is indeed a linear representation. Continuity follows from the continuity of $\rho_{\text{in}}$ and $\rho_{\text{out}}$ as follows: the topology on $\text{Hom}_{\mathbb{K}}(V_{\text{in}}, V_{\text{out}})$ is just the Euclidean topology of $\mathbb{K}^{c_{\text{out}} \times c_{\text{in}}}$ coming from a basis of $V_{\text{in}}$ and $V_{\text{out}}$. In these bases, $\rho_{\text{in}}(g)$ and $\rho_{\text{out}}(g)$ are given by matrices. All matrix coefficients are continuous by Remark B.25. Now, in order to show that $\rho_{\text{Hom}}$ is continuous, pick a fixed element $f \in \mathbb{K}^{c_{\text{in}} \times c_{\text{out}}}$. One needs to show that the map

$$\rho_{\text{Hom}}^f : G \to \mathbb{K}^{c_{\text{in}} \times c_{\text{out}}}, \; g \mapsto \rho_{\text{out}}(g) \circ f \circ \rho_{\text{in}}(g^{-1})$$

is continuous. Since all matrix coefficients are continuous and since also the inversion $G \to G$, $g \mapsto g^{-1}$ is continuous by the definition of a topological group, the map $\rho_{\text{Hom}}^f$ is basically just a stacked linear combination of continuous functions and thus continuous itself.

The linearity of each $\rho_{\mathrm{Hom}}(g)$ is also clear. So what needs to be checked is that $\rho_{\mathrm{Hom}}$ is a group homomorphism. And indeed, it is, exploiting the corresponding property of $\rho_{\mathrm{in}}$ and $\rho_{\mathrm{out}}$:

$$
\begin{aligned}
\left[\rho_{\mathrm{Hom}}(gg')\right](f) &= \rho_{\mathrm{out}}(gg') \circ f \circ \rho_{\mathrm{in}}(gg')^{-1} \\
&= \rho_{\mathrm{out}}(g) \circ \left(\rho_{\mathrm{out}}(g') \circ f \circ \rho_{\mathrm{in}}(g')^{-1}\right) \circ \rho_{\mathrm{in}}(g)^{-1} \\
&= \left[\rho_{\mathrm{Hom}}(g)\right]\left(\left[\rho_{\mathrm{Hom}}(g')\right](f)\right) \\
&= \left[\rho_{\mathrm{Hom}}(g) \circ \rho_{\mathrm{Hom}}(g')\right](f),
\end{aligned}
$$

and so the claim follows.

With this definition in mind, steerable kernels $K : X \to \mathrm{Hom}_{\mathbb{K}}(V_{\mathrm{in}}, V_{\mathrm{out}})$ are just functions with the property $K(gx) = \left[\rho_{\mathrm{Hom}}(g)\right](K(x))$. Summarizing, we have the following abstract definition of steerable kernels (different from Definition 3.2, we here allow also input- and output representations that are not irreducible and make explicit reference to the Hom-representation):

**Definition C.3** (Steerable Kernel). Let $G$ be any compact group and $X$ be any homogeneous space of $G$. Furthermore, let $\rho_{\mathrm{in}} : G \to \mathrm{GL}(V_{\mathrm{in}})$ and $\rho_{\mathrm{out}} : G \to \mathrm{GL}(V_{\mathrm{out}})$ be finite-dimensional representations of $G$. We assume that $\mathrm{Hom}_{\mathbb{K}}(V_{\mathrm{in}}, V_{\mathrm{out}})$ is equipped with the Hom-representation $\rho_{\mathrm{Hom}}$. A $G$-steerable kernel is an equivariant function $K : X \to \mathrm{Hom}_{\mathbb{K}}(V_{\mathrm{in}}, V_{\mathrm{out}})$, i.e., a function such that

$$
K(gx) = \left[\rho_{\mathrm{Hom}}(g)\right](K(x)) \tag{11}
$$

for all $g \in G$ and $x \in X$. We denote the vector-space of all these kernels by

$$
\mathrm{Hom}_G(X, \mathrm{Hom}_{\mathbb{K}}(V_{\mathrm{in}}, V_{\mathrm{out}})) = \{K : X \to \mathrm{Hom}_{\mathbb{K}}(V_{\mathrm{in}}, V_{\mathrm{out}}) \mid K \text{ is steerable}\}.
$$

Notably, steerable kernels are *not linear* in a meaningful sense with respect to their input.

That the space of steerable kernels forms a vector space, as claimed in this definition, can easily be checked.

### C.1.3 MORE DETAILS ON THE COMPARISON OF REPRESENTATION OPERATORS AND STEERABLE KERNELS

Steerable kernels satisfy the constraint

$$
K(gx) = \rho_{\mathrm{out}}(g) \circ K(x) \circ \rho_{\mathrm{in}}(g)^{-1}, \tag{12}
$$

whereas, as we saw in Section 2, representation operators are collections $(A_1, \ldots, A_N)$ of operators $A_i : \mathcal{H} \to \mathcal{H}$ that satisfy the constraint

$$
\sum_{j=1}^{N} \pi(g)_{ij} A_j = U(g)^{\dagger} A_i U(g) \qquad \forall\, g \in G.
$$

Hereby, $U : G \to \mathrm{U}(\mathcal{H})$ and $\pi : G \to \mathrm{U}(\mathbb{C}^N)$ are unitary representations. Unfortunately, these equations still look somewhat different from each other. We can make them more similar by inverting $g$ and using the unitarity of $\pi$ (note the swap of $j$ and $i$ and the complex conjugation):

$$
\sum_{j=1}^{N} \overline{\pi}(g)_{ji} A_j = U(g) A_i U(g)^{\dagger} \qquad \forall\, g \in G. \tag{13}
$$

In order to make the analogy to steerable kernels stronger, we would like to interpret a representation operator as *one object* $\boldsymbol{A}$ instead of separate operators $A_i$, in the same way as a kernel $K$ is one single object and not just a disjoint collection of linear functions in $\mathrm{Hom}_{\mathbb{K}}(V_{\mathrm{in}}, V_{\mathrm{out}})$. For this, we interpret $\boldsymbol{A}$ as a function that assigns to arbitrary vectors in $\mathbb{C}^N$ an operator. Namely, let $\{e_i\}$ be the standard basis of $\mathbb{C}^N$. We then define $\boldsymbol{A}$ as the *unique linear map* which is given on basis elements as follows:

$$
\boldsymbol{A} : e_i \mapsto A_i.
$$

We can then deduce the following, where we use the linearity of $\boldsymbol{A}$ in the second step, the definition of $A_j$ in the third and fifth step, and Eq. (13) in the fourth step:

$$
\begin{aligned}
\boldsymbol{A}\big(\overline{\pi}(g)(e_i)\big) &= \boldsymbol{A}\Big( \sum_j \overline{\pi}(g)_{ji} e_j \Big) \\
&= \sum_j \overline{\pi}(g)_{ji} \boldsymbol{A}\,(e_j) \\
&= \sum_j \overline{\pi}(g)_{ji} A_j \\
&= U(g) A_i U(g)^\dagger \\
&= U(g) \boldsymbol{A}(e_i) U(g)^\dagger .
\end{aligned}
\tag{14}
$$

If now $v = \sum_i \lambda_i e_i$ is an arbitrary vector in $\mathbb{C}^N$, not necessarily a standard basis vector, then from the linearity of $\boldsymbol{A}$ and Eq. (14) we obtain

$$
\boldsymbol{A}\big(\overline{\pi}(g)(v)\big) = U(g) \boldsymbol{A}(v) U(g)^{-1} .
\tag{15}
$$

This equation is essentially the starting point for the definition of a representation operator as it can be found in Jeevanjee [2011].

This, finally, really looks like Eq. (12). In this comparison, the action of the group $G$ on $\mathbb{R}^d$ in deep learning is replaced by the action of $G$ via $\overline{\pi}$ on the space $\mathbb{C}^N$. The main difference is that steerable kernels are not necessarily linear. This difference will be bridged in Theorem C.7.

### C.1.4 REPRESENTATION OPERATORS AND KERNEL OPERATORS

Now that we have a clear abstract idea of what steerable kernels are and saw strong analogies to representation operators, we can begin to formulate precise theoretical connections. In this section, we therefore begin with formulating a purely representation-theoretic and more abstract working definition of representation operators and will then formulate the main theorem of this chapter, Theorem C.7.

We come to the main definition, which is directly motivated from Eq. (15). It differs from (Jeevanjee, 2011) by allowing the input- and output representations to differ. We furthermore restrict to finite-dimensional input- and output representations due to our specific applications. As explained in Section C.1.3, this new definition furthermore somewhat differs from the one given in Section 2 since now we view representation operators as *one object* instead of viewing it as a collection of several linear operators.

**Definition C.4** (Representation Operator). Let $\rho_{\text{in}} : G \to \mathrm{GL}(V_{\text{in}})$ and $\rho_{\text{out}} : G \to \mathrm{GL}(V_{\text{out}})$ be finite-dimensional $G$-representations. Let $\lambda : G \to \mathrm{GL}(T)$ be a third $G$-representation, not necessarily finite-dimensional. Then a *representation operator* is an intertwiner $\mathcal{K} : T \to \mathrm{Hom}_{\mathbb{K}}(V_{\text{in}}, V_{\text{out}})$, where the right space is equipped with the $\mathrm{Hom}$-representation as in Definition C.1. We denote the vector space of all these representation operators by

$$
\mathrm{Hom}_{G,\mathbb{K}}(T, \mathrm{Hom}_{\mathbb{K}}(V_{\text{in}}, V_{\text{out}})) = \{\mathcal{K} : T \to \mathrm{Hom}_{\mathbb{K}}(V_{\text{in}}, V_{\text{out}}) \mid \mathcal{K} \text{ is an intertwiner}\} .
$$

Note that representation operators are by definition *linear*, which is a requirement that needs to be satisfied for the standard Wigner-Eckart theorem. We clearly see strong similarities between this definition and the formalization of steerable kernels in Definition C.3. The main difference is that we assume representation operators to be linear. This is in notation captured by the subscript $\mathbb{K}$ that we put in the corresponding $\mathrm{Hom}$-space. One may think that there is another difference, namely coming from the fact that intertwiners are by definition *continuous* with respect to the topologies involved. Two things need to be said about this:

1. First of all, one may wonder what continuity for representation operators actually means. This can be clarified as follows: By assumption, $G$-representations are always on vector spaces with topologies, and thus $T$ has a topology. Furthermore, in Remark C.2 we clarified the topology on $\mathrm{Hom}_{\mathbb{K}}(V_{\text{in}}, V_{\text{out}})$. Then, being continuous just means, as always, to be continuous with respect to the topologies of these two spaces.

2. The second remark is that this apparent difference in the requirement of continuity for steerable kernels and representation operators is actually non-existent. This is explained by the following Proposition which says that steerable kernels are automatically continuous. Note that this is not true for steerable kernels that are defined on the domain $\mathbb{R}^d$ – in that case, continuity is only guaranteed when restricting to orbits.

**Proposition C.5.** *Let $K : X \to \mathrm{Hom}_{\mathbb{K}}(V_{\mathrm{in}}, V_{\mathrm{out}})$ be a steerable kernel. Then $K$ is continuous.*

*Proof.* For brevity, denote $V := \mathrm{Hom}_{\mathbb{K}}(V_{\mathrm{in}}, V_{\mathrm{out}})$ and $\rho := \rho_{\mathrm{Hom}}$. Let $x^* \in X$ be any point and $G_{x^*}$ the stabilizer corresponding to the action of $G$ on $X$. Remember the homeomorphism $\varphi : G/H \to X$, $[g] \mapsto gx^*$ from Lemma B.21. Since this is a homeomorphism, the kernel $K$ is continuous if and only if the composition $K \circ \varphi$ is continuous, since then $K = (K \circ \varphi) \circ \varphi^{-1}$ is a composition of continuous functions. Thus, we evaluate $K \circ \varphi$:

$$(K \circ \varphi)([g]) = K(\varphi([g])) = K(gx^*) = \rho(g)(K(x^*)),$$

where in the last step we have used the equivariance of $K$. Thus, if we set $v^* := K(x^*) \in V$, then we obtain the simple relation $(K \circ \varphi)([g]) = \rho(g)(v^*)$. This is by definition just the unique map on the quotient, $G/H \to V$, coming from $\rho^{v^*} : G \to V$, $g \mapsto \rho(g)(v^*)$. This last map is continuous by definition of a linear representation. The universal property of quotients Proposition F.12 then shows that $K \circ \varphi$ is continuous as well, and so we are done. All of this is visualized in the following commutative diagram, where $q : G \to G/H$, $g \mapsto [g]$ is the canonical projection:

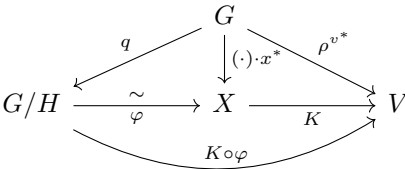

Thus, the only difference between steerable kernels and representation operators is indeed the linearity. We now look at special representation operators that play the main role in this work:

**Definition C.6** (Kernel Operator). Let $\rho_{\mathrm{in}} : G \to \mathrm{GL}(V_{\mathrm{in}})$ and $\rho_{\mathrm{out}} : G \to \mathrm{GL}(V_{\mathrm{out}})$ be finite-dimensional $G$-representations. Let $\lambda : G \to \mathrm{U}(L^2_{\mathbb{K}}(X))$ be the standard unitary representation on the space of square-integrable functions of a homogeneous space $X$, given, as in Section B.1.3, by

$$[\lambda(g)(\varphi)](g') = \varphi(g^{-1}g').$$

A *kernel operator* is a representation operator $\mathcal{K} : L^2_{\mathbb{K}}(X) \to \mathrm{Hom}_{\mathbb{K}}(V_{\mathrm{in}}, V_{\mathrm{out}})$. We denote the space of these by

$$\mathrm{Hom}_{G,\mathbb{K}}(L^2_{\mathbb{K}}(X), \mathrm{Hom}_{\mathbb{K}}(V_{\mathrm{in}}, V_{\mathrm{out}}))$$
$$= \left\{ \mathcal{K} : L^2_{\mathbb{K}}(X) \to \mathrm{Hom}_{\mathbb{K}}(V_{\mathrm{in}}, V_{\mathrm{out}}) \mid \mathcal{K} \text{ is an intertwiner} \right\}.$$

Notably, kernel operators are $\mathbb{K}$-linear in their input.

### C.1.5 FORMULATION OF THE CORRESPONDENCE BETWEEN STEERABLE KERNELS AND KERNEL OPERATORS

The following Theorem lies at the heart of our investigations and establishes that steerable kernels can be considered as kernel operators, which we defined as special representation operators. More precisely, we will give an explicit isomorphism between the space of steerable kernels and the space of kernel operators.

We shortly explain why the theorem is useful. First of all, using a Wigner-Eckart theorem for kernel operators that we prove in Theorem D.13, one can explicitly describe a basis $B$ of the space of kernel operators $\mathrm{Hom}_{G,\mathbb{K}}(L^2_{\mathbb{K}}(X), \mathrm{Hom}_{\mathbb{K}}(V_{\mathrm{in}}, V_{\mathrm{out}}))$. Then, since we have an isomorphism of vector spaces to the space of steerable kernels, one can "carry over" this basis to a basis for the space of steerable kernels, namely $\mathrm{Hom}_G(X, \mathrm{Hom}_{\mathbb{K}}(V_{\mathrm{in}}, V_{\mathrm{out}}))$. This basis will then have a convenient explicit form that we establish in Theorem D.16 and is exactly what we need in order to parameterize an equivariant neural network layer. We now come to a precise formulation of the theorem:

**Theorem C.7** (Kernel-Operator-Correspondence). *Let $\rho_{\mathrm{in}} : G \to \mathrm{GL}(V_{\mathrm{in}})$ and $\rho_{\mathrm{out}} : G \to \mathrm{GL}(V_{\mathrm{out}})$ be finite-dimensional $G$-representations and $X$ be a homogeneous space of $G$. Then there is an isomorphism*

$$\mathrm{Hom}_G(X, \mathrm{Hom}_{\mathbb{K}}(V_{\mathrm{in}}, V_{\mathrm{out}})) \xrightleftharpoons[(\cdot)|_X]{\widehat{(\cdot)}} \mathrm{Hom}_{G,\mathbb{K}}(L^2_{\mathbb{K}}(X), \mathrm{Hom}_{\mathbb{K}}(V_{\mathrm{in}}, V_{\mathrm{out}}))$$

*between the space of steerable kernels on the left and the space of kernel operators on the right. The two maps are defined as follows:*

1. *For a steerable kernel $K : X \to \mathrm{Hom}_{\mathbb{K}}(V_{\mathrm{in}}, V_{\mathrm{out}})$, the extension $\widehat{K} : L^2_{\mathbb{K}}(X) \to \mathrm{Hom}_{\mathbb{K}}(V_{\mathrm{in}}, V_{\mathrm{out}})$ is given by*

$$\widehat{K}(f) := \int_X f(x) K(x) dx.$$

2. *For a kernel operator $\mathcal{K} : L^2_{\mathbb{K}}(X) \to \mathrm{Hom}_{\mathbb{K}}(V_{\mathrm{in}}, V_{\mathrm{out}})$, the restriction $\mathcal{K}|_X : X \to \mathrm{Hom}_{\mathbb{K}}(V_{\mathrm{in}}, V_{\mathrm{out}})$ is given by*

$$\mathcal{K}|_X(x) := \lim_{U \in \mathcal{U}_x} \mathcal{K}(\delta_U).$$

*Hereby, $\mathcal{U}_x$ is the directed set of open neighborhoods of $x$, see Example F.27. $\delta_U : X \to \mathbb{K}$ is the approximated Dirac delta function with $\delta_U(y) = \frac{1}{\mu(U)}$ if $y \in U$ and $\delta_U(y) = 0$, else. The limit is a limit of nets as in Definition F.29.*

This theorem requires some explanation. First of all, $\widehat{K}$ is supposed to be a kernel operator, i.e., a map $L^2_{\mathbb{K}}(X) \to \mathrm{Hom}_{\mathbb{K}}(V_{\mathrm{in}}, V_{\mathrm{out}})$. Thus, $\widehat{K}(f)$ should be a linear function $V_{\mathrm{in}} \to V_{\mathrm{out}}$. The formal expression of it can indeed be considered as such:

$$\widehat{K}(f) = \int_X f(x) K(x) dx : v_{\mathrm{in}} \mapsto \int_X f(x) \left[K(x)\right](v_{\mathrm{in}}) dx \in V_{\mathrm{out}}. \tag{16}$$

Due to the continuity of $K$ proven in Proposition C.5[18] and the integrability of $f$, the function $X \to V_{\mathrm{out}}$, $x \mapsto f(x) \left[K(x)\right](v_{\mathrm{in}})$ is also integrable, meaning the expression in Eq. (16) can be evaluated. This explains the meaning of the map $\widehat{(\cdot)}$ in Theorem C.7.

For the map $(\cdot)|_X$ in the other direction, we want to shortly explain the intuitions in a more informal way. For this, we consider Dirac delta functions $\delta_x$ for $x \in X$. Such a "function" $\delta_x : X \to \mathbb{K}$ for a point $x \in X$ can be imagined as a function taking value infinity at $x$ and zero elsewhere. It is characterized by the property that $\int_X \delta_x(x') f(x') dx' = f(x)$ for any function $f \in L^2_{\mathbb{K}}(X)$. We think of $\delta_x$ as being a function in $L^2_{\mathbb{K}}(X)$, even though technically, it is not in this space. This is since $\infty \notin \mathbb{K}$.

Now, informally, we can think of the limit $\mathcal{K}|_X(x) = \lim_{U \in \mathcal{U}_x} \mathcal{K}(\delta_U)$ as being given by $\mathcal{K}(\delta_x)$, the value that $\mathcal{K}$ takes at the Dirac delta function $\delta_x$. This is since the limit of nets progressively "shrinks down" the open neighborhood $U$ of $x$. Of course, $\mathcal{K}(\delta_x)$ is not really well-defined since $\delta_x \notin L^2_{\mathbb{K}}(X)$, but we can pretend that it is for gaining intuitions.

Now that we have understood the formulation of the theorem, we might wonder, why should such a theorem be true? A first intuition comes from an analogy with linear algebra: namely, assume $B$ is a basis of a $\mathbb{K}$-vector space $V$ and $W$ any other vector space. Then linear maps $\hat{f} : V \to W$ are in one-to-one correspondence with (not assumed to be linear) functions $f : B \to W$, and this isomorphism is given by restriction and linear extension:

$$\mathrm{Hom}(B, W) \xrightleftharpoons[(\cdot)|_B]{\widehat{(\cdot)}} \mathrm{Hom}_{\mathbb{K}}(V, W).$$

---

[18]This means that all matrix elements of $K(x)$ for chosen bases of $V_{\mathrm{in}}$ and $V_{\mathrm{out}}$ are continuous.

Thus, we can think of the homogeneous space $X$ as a "continuous basis" of the space of square-integrable functions. Sums are then replaced by integrals, and evaluations at a basis element by evaluations at Dirac delta functions of elements in $X$.

For the actual proof of Theorem C.7, informally, one direction seems pretty clear from the properties of the Dirac delta:

$$\widehat{K}\big|_X(x) = \widehat{K}(\delta_x) = \int_X \delta_x(x')K(x')dx' = K(x).$$

But the other direction is less obvious: it seems like the space of kernel operators is considerably larger than the space of steerable kernels, since kernel operators are defined on a larger space. Therefore it is hard to believe that the construction is also inverse in the other direction. However, it pays off to ponder a bit more over what the Dirac delta construction does: Basically, we "embed" $X$ into $L^2_{\mathbb{K}}(X)$ by means of the Dirac delta functions, i.e., $x \mapsto \delta_x$ and, as such, view $X$ as a subset of $L^2_{\mathbb{K}}(X)$ (albeit a subset that is only in approximation in that space). Steerable kernels are then "partial" kernel operators in the sense that they are only defined on this subset $X \subseteq L^2_{\mathbb{K}}(X)$. What then needs to be understood is why there is only one unique extension of each steerable kernel $K$ to a kernel operator $\mathcal{K}$ on the whole of $L^2_{\mathbb{K}}(X)$: if this is understood, then the space of kernel operators cannot be larger than the space of steerable kernels. And indeed, if there is an extension of $K$ to $\mathcal{K}$ on $L^2_{\mathbb{K}}(X)$, it *has to* be unique: each $f \in L^2_{\mathbb{K}}(X)$ can be approximated by finite linear combinations of scaled indicator functions. Then by *linearity* of the kernel operator $\mathcal{K}$, we can evaluate $\mathcal{K}(f)$ by knowing $\mathcal{K}(\delta_U)$ for scaled indicator functions $\delta_U$ on small measurable sets $U$. And these approximate $K(x) = \mathcal{K}(\delta_x)$ for $x \in U$ arbitrarily well by construction. This determines the behavior of $\mathcal{K}$. The details of all of this can be found in the next section.

### C.2 A Proof of the Correspondence between Steerable Kernels and Kernel Operators

Here, we give a step-by-step proof of Theorem C.7. The details of this investigation will not be needed later, and so a reader who is mainly interested in the applications to steerable CNNs can safely skip reading this section and go on reading Chapter D.

#### C.2.1 A Reduction to Unitary Irreducible Representations

In this section, we make the proof more manageable by reducing $\mathrm{Hom}_{\mathbb{K}}(V_{\mathrm{in}}, V_{\mathrm{out}})$ to an irreducible representation. First, remember that Proposition B.20 shows that there is a scalar product on $\mathrm{Hom}_{\mathbb{K}}(V_{\mathrm{in}}, V_{\mathrm{out}})$ such that it's $\mathrm{Hom}$-representation becomes unitary. Since all norms on finite-dimensional spaces are equivalent, as is well known, this will not change the topology. Then, we can decompose $\mathrm{Hom}_{\mathbb{K}}(V_{\mathrm{in}}, V_{\mathrm{out}})$ into an orthogonal direct sum of irreducible unitary representations by Proposition B.38. Let $\mathrm{Hom}_{\mathbb{K}}(V_{\mathrm{in}}, V_{\mathrm{out}}) \cong \bigoplus_{i=1}^n V_i$ be such a decomposition. We get canonical[19] isomorphisms

$$\mathrm{Hom}_G(X, \mathrm{Hom}_{\mathbb{K}}(V_{\mathrm{in}}, V_{\mathrm{out}})) \cong \bigoplus_{i=1}^n \mathrm{Hom}_G(X, V_i)$$

and

$$\mathrm{Hom}_{G,\mathbb{K}}(L^2_{\mathbb{K}}(X), \mathrm{Hom}_{\mathbb{K}}(V_{\mathrm{in}}, V_{\mathrm{out}})) \cong \bigoplus_{i=1}^n \mathrm{Hom}_{G,\mathbb{K}}(L^2_{\mathbb{K}}(X), V_i).$$

Thus, we can show Theorem C.7 by showing it for irreducible unitary representations instead of $\mathrm{Hom}_{\mathbb{K}}(V_{\mathrm{in}}, V_{\mathrm{out}})$. Overall, we have reduced our Theorem to the following, simpler statement:

**Theorem C.8** (Kernel-Operator-Correspondence, Restated). *Let $\rho : G \to \mathrm{U}(V)$ be an irreducible unitary representation and $X$ a homogeneous space of $G$. Then there is an isomorphism*

$$\mathrm{Hom}_G(X, V) \xrightleftharpoons[(\cdot)|_X]{\widehat{(\cdot)}} \mathrm{Hom}_{G,\mathbb{K}}(L^2_{\mathbb{K}}(X), V)$$

---

[19]"Canonical" once the decompositions into irreducible representations is already chosen.

*which is given as follows: for $K \in \operatorname{Hom}_G(X, V)$ we set $\widehat{K}(f) = \int_X f(x)K(x)dx$ and for $\mathcal{K} \in \operatorname{Hom}_{G,\mathbb{K}}(L^2_{\mathbb{K}}(X), V)$ we set $\mathcal{K}|_X(x) = \lim_{U \in \mathcal{U}_x} \mathcal{K}(\delta_U)$, with $\delta_U$ being an approximated Dirac delta function as before.*

From now on, we assume that $X$ and $\rho : G \to \mathrm{U}(V)$ is fixed as in the formulation of Theorem C.8.

## C.2.2 WELL-DEFINEDNESS OF $\widehat{(\cdot)}$

**Lemma C.9.** *The function $\widehat{(\cdot)} : \operatorname{Hom}_G(X, V) \to \operatorname{Hom}_{G,\mathbb{K}}(L^2_{\mathbb{K}}(X), V)$ is well-defined, i.e.: for an equivariant function $K : X \to V$, the function $\widehat{K} : L^2_{\mathbb{K}}(X) \to V$ is linear, equivariant and continuous.*

*Proof.* Linearity of $\widehat{K}$ is clear. Equivariance can be proven using the equivariance of $K$ and the left invariance of the Haar measure on the homogeneous space $X$:

$$
\begin{aligned}
\widehat{K}(\lambda(g)f) &= \int_X (\lambda(g)f)(x)K(x)dx \\
&= \int_X f(g^{-1} \cdot x)K(x)dx \\
&= \int_X f(x)K(g \cdot x)dx \\
&= \int_X f(x)\left[\rho(g)\left(K(x)\right)\right]dx \\
&= \rho(g)\left[\int_X f(x)K(x)dx\right] \\
&= \rho(g)\left[\widehat{K}(f)\right].
\end{aligned}
$$

The action by $\rho(g)$ could be put out of the integral since $\rho(g)$ it is linear and continuous, and since integrals can be approximated by finite sums.

Now about continuity: By Proposition F.18, we only need to show continuity in $0$. Thus, let $(f_k)_k$ be a sequence of functions $f_k \in L^2_{\mathbb{K}}(X)$ with $\lim_{k \to \infty} \|f_k\|_{L^2} = 0$. Then we obtain

$$
\begin{aligned}
\|\widehat{K}(f_k)\|_V &= \left\|\int_X f_k(x)K(x)dx\right\|_V \\
&\leq \int_X |f_k(x)| \cdot \|K(x)\|_V dx \\
&\leq \max_{x'} \|K(x')\|_V \cdot \int_X |f_k(x)|dx,
\end{aligned}
$$

where the continuity of $K$ proven in Proposition C.5 was used.[20] For the right expression, using the Cauchy-Schwarz inequality Proposition F.34 we obtain

$$
\begin{aligned}
\int_X |f_k(x)|dx &= \int_X |f_k(x)| \cdot 1 dx \\
&= |\langle |f_k| \,|\, 1\rangle| \\
&\leq \|f_k\|_{L^2} \cdot \|1\|_{L^2} \\
&= \|f_k\|_{L^2}.
\end{aligned}
$$

So, overall, if $\lim_{k \to \infty} \|f_k\|_{L^2} = 0$, then $\lim_{k \to \infty} \|\widehat{K}(f_k)\|_V = 0$ as well, which proves continuity. $\square$

---

[20]since $\|K\|$ is continuous on $X$, which is compact by Proposition F.8 as an image of the compact group $G$, it has a maximum by Corollary F.25.

C.2.3 WELL-DEFINEDNESS OF $(\cdot)|_X$

While it is clear that the limit $\lim_{U \in \mathcal{U}_x} \mathcal{K}(\delta_U)$ from Theorem C.8 is unique if it exists (Conway, 2014), it is somewhat unclear why it exists in the first place. For this, we need to better understand the properties of the (approximated) Dirac delta. The most important one is the following, which we hinted at already in the intuitions we gave before this section: basically, Dirac deltas help for evaluating continuous functions at specific points:

**Lemma C.10.** *For each $x \in X$ and $Y : X \to \mathbb{K}$ continuous we have $\lim_{U \in \mathcal{U}_x} \langle \delta_U | Y \rangle = Y(x)$.*

*Proof.* We have

$$\left| \langle \delta_U | Y \rangle - Y(x) \right| = \left| \int_X \delta_U(x') Y(x') dx' - \mu(U) \cdot \frac{1}{\mu(U)} Y(x) \right|$$

$$= \left| \int_U \frac{1}{\mu(U)} Y(x') dx' - \int_U \frac{1}{\mu(U)} Y(x) dx' \right|$$

$$= \left| \int_U \frac{1}{\mu(U)} (Y(x') - Y(x)) dx' \right|$$

$$\leq \int_U \frac{1}{\mu(U)} |Y(x') - Y(x)| \, dx'.$$

Let $\epsilon > 0$. Since $Y$ is continuous in $x$, there is $U_\epsilon \in \mathcal{U}_x$ such that $Y(x') \in B_\epsilon(Y(x))$ for all $x' \in U_\epsilon$ or, equivalently, $|Y(x') - Y(x)| < \epsilon$. Thus, for all $U_\epsilon \supseteq U$, i.e., all $U_\epsilon \leq U$ in $\mathcal{U}_x$ we obtain

$$\left| \langle \delta_U | Y \rangle - Y(x) \right| \leq \int_U \frac{1}{\mu(U)} |Y(x') - Y(x)| dx'$$

$$\leq \int_U \frac{1}{\mu(U)} \epsilon dx'$$

$$= \epsilon \cdot \mu(U) \cdot \frac{1}{\mu(U)}$$

$$= \epsilon$$

and consequently $\lim_{U \in \mathcal{U}_x} \langle \delta_U | Y \rangle = Y(x)$. $\qquad \square$

Before we can show the well-definedness of $\mathcal{K}|_X$, we first want to get a better description of $\mathcal{K}$. For this, recall from the Peter-Weyl theorem that $L^2_{\mathbb{K}}(X) = \widehat{\bigoplus}_{l \in \widehat{G}} \bigoplus_{i=1}^{m_l} V_{li}$. With this at our disposal, we can formulate the following Lemma on the form of intertwiners on $L^2_{\mathbb{K}}(X)$:

**Lemma C.11.** *Let $\mathcal{K} : L^2_{\mathbb{K}}(X) \to V$ be an intertwiner. Let $l \in \widehat{G}$ be the unique index such that $V \cong V_{li}$ for all $i = 1, \ldots, m_l$. Let $Y_{li}^n$, $n = 1, \ldots, d_l$ be an orthonormal basis of $V_{li}$ where $d_l = \dim(V_l)$. Then*

$$\mathcal{K}(f) = \sum_{i=1}^{m_l} \sum_{n=1}^{d_l} \langle Y_{li}^n | f \rangle \, \mathcal{K}(Y_{li}^n)$$

*for all $f \in L^2_{\mathbb{K}}(X)$.*

*Proof.* We can write $f \in L^2_{\mathbb{K}}(X)$ according to the discussion after Definition F.40 as

$$f = \sum_{l' \in \widehat{G}} \sum_{i=1}^{m_{l'}} \sum_{n=1}^{[l']} \langle Y_{l'i}^n | f \rangle \, Y_{l'i}^n.$$

Note that $\mathcal{K}|_{V_{l'i}} : V_{l'i} \to V$ is an intertwiner as well, and so by Schur's Lemma B.29 it is necessarily zero unless $l' = l$ is the unique index such that $V_{li} \cong V$. Due to its continuity and linearity, $\mathcal{K}$ commutes with infinite sums and we obtain

$$\mathcal{K}(f) = \sum_{l' \in \widehat{G}} \sum_{i=1}^{m_{l'}} \sum_{n=1}^{[l']} \langle Y_{l'i}^n | f \rangle \, \mathcal{K}(Y_{l'i}^n)$$

$$= \sum_{l' \in \widehat{G}} \sum_{i=1}^{m_{l'}} \sum_{n=1}^{[l']} \langle Y_{l'i}^n | f \rangle \, \mathcal{K}|_{V_{l'i}}(Y_{l'i}^n)$$

$$= \sum_{i=1}^{m_l} \sum_{n=1}^{d_l} \langle Y_{li}^n | f \rangle \, \mathcal{K}(Y_{li}^n).$$

$\qquad \square$

**Corollary C.12.** *We have* $\mathcal{K}|_X(x) = \sum_{i=1}^{m_l} \sum_{n=1}^{d_l} \overline{Y_{li}^n(x)} \mathcal{K}(Y_{li}^n)$. *In particular, the defining limit exists.*

*Proof.* Since the $Y_{li}^n$ are by the proof of the Peter-Weyl theorem in the finite-dimensional space $\mathcal{E}_l$ spanned by matrix coefficients of the irreducible representation $\rho_l : G \to U(V_l)$ and since these matrix coefficients are continuous by Remark B.25, the $Y_{li}^n$ are as finite linear combinations of them also continuous functions. Thus, from Lemma C.10 and C.11 together we obtain:

$$\mathcal{K}|_X(x) = \lim_{U \in \mathcal{U}_x} \mathcal{K}(\delta_U)$$

$$= \lim_{U \in \mathcal{U}_x} \sum_{i=1}^{m_l} \sum_{n=1}^{d_l} \langle Y_{li}^n | \delta_U \rangle \mathcal{K}(Y_{li}^n)$$

$$= \sum_{i=1}^{m_l} \sum_{n=1}^{d_l} \left[ \lim_{U \in \mathcal{U}_x} \langle Y_{li}^n | \delta_U \rangle \right] \mathcal{K}(Y_{li}^n)$$

$$= \sum_{i=1}^{m_l} \sum_{n=1}^{d_l} \overline{Y_{li}^n(x)} \mathcal{K}(Y_{li}^n).$$

The complex conjugation came into play since the order in the scalar product is swapped compared to Lemma C.10. $\qquad\square$

Thus, since we now know that $\mathcal{K}|_X$ *as a function* makes sense, we can finally prove the well-definedness of $\mathcal{K} \mapsto \mathcal{K}|_X$,

**Lemma C.13.** *The function* $(\cdot)|_X : \mathrm{Hom}_{G,\mathbb{K}}(L^2_\mathbb{K}(X), V) \to \mathrm{Hom}_G(X, V)$ *is well-defined, that is: for a linear, equivariant and continuous function* $\mathcal{K} : L^2_\mathbb{K}(X) \to V$, *the restriction* $\mathcal{K}|_X : X \to V$ *is equivariant.*

*Proof.* We have

$$\mathcal{K}|_X(g \cdot x) = \lim_{U \in \mathcal{U}_{gx}} \mathcal{K}(\delta_U)$$

$$= \lim_{U \in \mathcal{U}_x} \mathcal{K}(\delta_{gU})$$

$$= \lim_{U \in \mathcal{U}_x} \mathcal{K}(\lambda(g)\delta_U)$$

$$= \lim_{U \in \mathcal{U}_x} \rho(g)[\mathcal{K}(\delta_U)]$$

$$= \rho(g)\left[ \lim_{U \in \mathcal{U}_x} \mathcal{K}(\delta_U) \right]$$

$$= \rho(g)[\mathcal{K}|_X(x)],$$

where the steps are justified as follows: The first step is just the definition of $\mathcal{K}|_X$. The second step uses that the open neighborhood of $gx$ are precisely the $g$-translated open neighborhoods of $x$ since $g : X \to X$ is a homeomorphism. The third step is easy to check. The fourth step uses the equivariance of $\mathcal{K}$. The fifth step uses the continuity of $\rho(g)$, which follows since $\rho(g)$ is a unitary transformation. The last step is again the definition of $\mathcal{K}|_X$. $\qquad\square$

### C.2.4 $\widehat{(\cdot)}$ AND $(\cdot)|_X$ ARE INVERSE TO EACH OTHER

We can now finish the proof of Theorem C.8 and consequently of Theorem C.7:

*Proof of Theorem C.8.* After all the preparation, we only need to still show that the maps $\widehat{(\cdot)}$ and $(\cdot)|_X$ are inverse to each other. For $\widehat{K}\big|_X = K$, i.e., the injectivity of the function $K \mapsto \widehat{K}$ and surjectivity of the function $\mathcal{K} \mapsto \mathcal{K}|_X$, we compute:

$$\widehat{K}\big|_X(x) = \lim_{U \in \mathcal{U}_x} \widehat{K}(\delta_U)$$

$$= \lim_{U \in \mathcal{U}_x} \int_X \delta_U(x') K(x') dx'$$

$$= K(x).$$

The last step follows from Lemma C.10 by identifying $V = V_l$ with $\mathbb{K}^{d_l}$ and viewing $K$ as consisting of continuous component functions $K^n : X \to \mathbb{K}$, $n \in \{1, \ldots, d_l\}$. The continuity of $K$ was shown in Proposition C.5.

For showing $\widehat{\mathcal{K}|_X} = \mathcal{K}$ we do a computation using the description of $\mathcal{K}$ from Lemma C.11 and the description of $\mathcal{K}|_X$ from Corollary C.12:

$$
\begin{aligned}
\widehat{\mathcal{K}|_X}(f) &= \int_X f(x)\mathcal{K}|_X(x)dx \\
&= \int_X f(x)\Big( \sum\nolimits_{i=1}^{m_l} \sum\nolimits_{n=1}^{d_l} \overline{Y_{li}^n(x)}\mathcal{K}(Y_{li}^n)\Big)dx \\
&= \sum\nolimits_{i=1}^{m_l} \sum\nolimits_{n=1}^{d_l} \Big( \int_X f(x)\overline{Y_{li}^n(x)}dx \Big)\mathcal{K}(Y_{li}^n) \\
&= \sum\nolimits_{i=1}^{m_l} \sum\nolimits_{n=1}^{d_l} \langle Y_{li}^n|f\rangle \, \mathcal{K}(Y_{li}^n) \\
&= \mathcal{K}(f).
\end{aligned}
$$

This finally finishes the proof. $\qquad\square$

# D  A WIGNER-ECKART THEOREM FOR STEERABLE KERNELS OF GENERAL COMPACT GROUPS

In Chapter C we have seen the most important theoretical insight of this work: steerable kernels on a homogeneous space $X$ correspond one-to-one to kernel operators (certain representation operators) on the space of square-integrable functions $L^2_{\mathbb{K}}(X)$. In this chapter, we will develop the most important consequence of this correspondence: a Wigner-Eckart theorem for steerable kernels and consequently a description of a basis for steerable kernels. This works for both fields $\mathbb{R}$ and $\mathbb{C}$, for an arbitrary compact group $G$, an arbitrary homogeneous space $X$ and arbitrary finite-dimensional input- and output fields. Additionally, it covers the general theory of equivariant CNNs on homogeneous spaces developed in (Cohen et al., 2019b).

In Section D.1 we will work towards formulating the most important theorems. Since these will involve tensor products, we will start with defining and studying tensor products of pre-Hilbert spaces and (unitary) representations. Afterward, we will define the Clebsch-Gordan coefficients, which relate a tensor product of irreducible representations to the irreducible subrepresentations of this tensor product. This will lead to a formulation of the original Wigner-Eckart theorem similar as it appears in quantum mechanics, including a proof. The original Wigner-Eckart theorem is a statement about representation operators on irreducible representations. However, we consider kernel operators on $L^2_{\mathbb{K}}(X)$ which is not irreducible. Also, different from the original Theorem, we also consider representations over the real numbers, which leads to a replacement of reduced matrix elements by *endomorphisms*. Therefore we then formulate a generalization of the original theorem. Then, using the correspondence between kernel operators and steerable kernels from Theorem C.7, we can transform this into a Wigner-Eckart theorem for steerable kernels and ultimately a statement about a basis of the space of steerable kernels. We conclude with some remarks about how to use the basis kernels in practice.

Afterward, in Section D.2, we give the remaining proof of the Wigner-Eckart theorem for kernel operators, which we omit in the section before. First, we reduce the statement to the dense subspace of $L^2_{\mathbb{K}}(X)$ which is a direct sum of all irreducible subrepresentations. We then describe a correspondence between representation operators and intertwiners on a certain tensor product, the so-called hom-tensor adjunction. Finally, we finish with the full proof of the Wigner-Eckart theorem.

As always, let $\mathbb{K}$ be either of the two fields $\mathbb{R}$ and $\mathbb{C}$ and $G$ be a compact topological group. $X$ is any homogeneous space of $G$.

## D.1 A WIGNER-ECKART THEOREM FOR STEERABLE KERNELS AND THEIR KERNEL BASES

### D.1.1 TENSOR PRODUCTS OF PRE-HILBERT SPACES AND UNITARY REPRESENTATIONS

In order to state the Wigner-Eckart theorem, we need the notion of representations on tensor products. This is defined similarly to Hom-representations, see Definition C.1. For this, we first need to discuss the notion of a tensor product of vector spaces:

**Definition D.1** (Tensor Product). Let $V$ and $V'$ be two vector spaces over $\mathbb{K}$. Then $V \otimes V'$, the *tensor product* of $V$ and $V'$, is a vector space over $\mathbb{K}$ with the following properties:

1. There is a bilinear function $\otimes : V \times V' \to V \otimes V'$, $(v, v') \mapsto v \otimes v'$. $V \otimes V'$ is generated by elements of the form $v \otimes v'$.

2. It has the following universal property: for any *bilinear* function $\beta : V \times V' \to P$ into a vector space $P$, there is a *unique linear* function $\overline{\beta} : V \otimes V' \to P$ given on elements of the form $v \otimes v'$ by $\overline{\beta}(v \otimes v') = \beta(v, v')$. In other words, the following diagram commutes:

$$
\begin{array}{ccc}
V \times V' & \xrightarrow{\beta} & P \\
\otimes \downarrow & \nearrow_{\overline{\beta}} & \\
V \otimes V' & &
\end{array}
$$

3. If $V$ and $V'$ are finite-dimensional with bases $\{v_1, \ldots, v_n\} \subseteq V$ and $\{v'_1, \ldots, v'_m\} \subseteq V'$, then $\{v_i \otimes v'_j\}_{i,j} \subseteq V \otimes V'$ is a basis of $V \otimes V'$. In particular, the dimension of $V \otimes V'$ is $n \cdot m$.

Property 3 follows from 1 and 2 and would therefore not necessarily be needed in the definition. The explicit construction of tensor products shall not matter for our purposes since the properties above characterize it up to isomorphism. The second property stated in the definition is of large importance since it tells us how we can define linear functions on $V \otimes V'$: if we have a guess for such a function $\varphi : V \otimes V' \to P$ (of which we don't yet know whether its "assignment rule" is well-defined), then we just need to test whether the function $\tilde{\varphi} : V \times V' \to P$ given by $\tilde{\varphi}(v, v') := \varphi(v \otimes v')$ is bilinear. If it is, then $\varphi$ is a well-defined linear function. We will use this soon in the following context: Assume $f : V \to V$ and $g : V' \to V'$ are linear functions. Then we would like to define a function $f \otimes g : V \otimes V' \to V \otimes V'$ by $(f \otimes g)(v \otimes v') = f(v) \otimes g(v')$. For this to work, we need to test whether the assignment $(v, v') \mapsto f(v) \otimes g(v')$ is a bilinear function $V \times V' \to V \otimes V'$. Clearly, it is, and so $f \otimes g$ is a well-defined linear function! We use this in Definition D.3 in order to define the tensor product of representations.

Since we actually deal with Hilbert spaces most of the time, we would like to build tensor products of Hilbert spaces. However, their definition is not completely straightforward since one cannot just take the tensor product of the underlying vector spaces but needs to additionally build the *completion* of the resulting space (Kadison & Ringrose, 1997). Since this complicates the considerations related to a correspondence we later formulate in Proposition D.23, we go a slightly different route. Instead of describing the tensor product of Hilbert spaces, we describe the tensor product of *pre-Hilbert spaces*, which does not require a completion step. Recall from Definition F.3 that a pre-Hilbert space is basically a Hilbert space that is not necessarily complete.

**Definition D.2** (Tensor Product of pre-Hilbert spaces). Let $V, V'$ be two pre-Hilbert spaces with scalar products $\langle \cdot | \cdot \rangle$ and $\langle \cdot | \cdot \rangle'$. Then the tensor product of vector spaces $V \otimes V'$ can be made into a pre-Hilbert space using the scalar product which is given on generators by

$$
\langle v \otimes v' | w \otimes w' \rangle_{\otimes} := \langle v | w \rangle \cdot \langle v' | w' \rangle' .
$$

This is then anti-linearly extended in the first (i.e., "Bra"), and linearly extended in the second (i.e., "Ket") component.

One can show that this makes $V \otimes V'$ a pre-Hilbert space. For simplicity, we will from now on not notationally distinguish the different scalar products involved. With this preparation, we can come to the notion of tensor product representations:

**Definition D.3** (Tensor Product Representation). Let $\rho : G \to \mathrm{GL}(V)$ and $\rho' : G \to \mathrm{GL}(V')$ be two linear representations, where $V$ and $V'$ are pre-Hilbert spaces. Then on the tensor product $V \otimes V'$ of pre-Hilbert spaces, we can define the *tensor product representation $\rho \otimes \rho'$* by

$$\rho \otimes \rho' : G \to \mathrm{GL}(V \otimes V'), \ g \mapsto \rho(g) \otimes \rho'(g),$$

where $\rho(g) \otimes \rho'(g) : V \otimes V' \to V \otimes V'$ is given on generators by

$$(\rho(g) \otimes \rho'(g))(v \otimes v') := \rho(g)(v) \otimes \rho'(g)(v').$$

**Lemma D.4.** *The map $\rho \otimes \rho' : G \to \mathrm{GL}(V \otimes V')$ defined above is a linear representation.*

*Proof.* Clearly, each $(\rho \otimes \rho')(g)$ is linear and we have $(\rho \otimes \rho')(gg') = (\rho \otimes \rho')(g) \circ (\rho \otimes \rho')(g')$. Thus, for showing that it is a linear representation, we need to show it is continuous. Assume we already knew continuity of all maps $(\rho \otimes \rho')^{v \otimes v'} : G \to V \otimes V', g \mapsto \left[ (\rho \otimes \rho')(g) \right](v \otimes v')$. Then for linear combinations $\xi = \sum_{i=1}^{n} \lambda_i (v_i \otimes v_i')$ we obtain using the linearity of $(\rho \otimes \rho')(g)$:

$$\begin{aligned}
(\rho \otimes \rho')^\xi(g) &= \left[ (\rho \otimes \rho')(g) \right](\xi) \\
&= \left[ (\rho \otimes \rho')(g) \right]\left( \sum_{i=1}^{n} \lambda_i (v_i \otimes v_i') \right) \\
&= \sum_{i=1}^{n} \lambda_i \left[ (\rho \otimes \rho')(g) \right](v_i \otimes v_i') \\
&= \left( \sum_{i=1}^{n} \lambda_i (\rho \otimes \rho')^{v_i \otimes v_i'} \right)(g).
\end{aligned}$$

Now, since scalar multiplication and addition in topological vector spaces is continuous, and since pre-Hilbert spaces are special topological vector spaces, the continuity of $(\rho \otimes \rho')^\xi$ follows from that of all $(\rho \otimes \rho')^{v \otimes v'}$.

What's left is proving the continuity of functions of the form $(\rho \otimes \rho')^{v \otimes v'}$. For notational simplicity, write $f = \rho^v : G \to V$ and $f' : \rho'^{v'}$, which are both continuous since $\rho$ and $\rho'$ are linear representations. We want to show that also $f \otimes f' : G \to V \otimes V'$ is continuous. We can test continuity in each point $g_0 \in G$ separately by Definition F.6. For each $g \in G$ we then obtain, with $\mathrm{Re}$ being the real part of a complex number:

$$\begin{aligned}
\| (f &\otimes f')(g) - (f \otimes f')(g_0) \|^2 \\
&= \left\| \left[ f(g) \otimes f'(g) - f(g) \otimes f'(g_0) \right] + \left[ f(g) \otimes f'(g_0) - f(g_0) \otimes f'(g_0) \right] \right\|^2 \\
&= \left\| f(g) \otimes \left[ f'(g) - f'(g_0) \right] + \left[ f(g) - f(g_0) \right] \otimes f'(g_0) \right\|^2 \\
&= \left\| f(g) \otimes \left[ f'(g) - f'(g_0) \right] \right\|^2 + \left\| \left[ f(g) - f(g_0) \right] \otimes f'(g_0) \right\|^2 \\
&\quad + 2 \, \mathrm{Re} \left\langle f(g) \otimes \left[ f'(g) - f'(g_0) \right] \, \middle| \, \left[ f(g) - f(g_0) \right] \otimes f'(g_0) \right\rangle \\
&= \| f(g) \|^2 \cdot \| f'(g) - f'(g_0) \|^2 + \| f(g) - f(g_0) \|^2 \cdot \| f'(g_0) \|^2 \\
&\quad + 2 \, \mathrm{Re} \left( \langle f(g) | f(g) - f(g_0) \rangle \cdot \langle f'(g) - f'(g_0) | f'(g_0) \rangle \right).
\end{aligned}$$

All in all we see the following: If $g$ is sufficiently close to $g_0$, then due to the continuity of $f$, $f'$, the scalar product, multiplication in $\mathbb{K}$ and the real part, $\| (f \otimes f')(g) - (f \otimes f')(g_0) \|^2$ gets arbitrarily close to 0. This shows the continuity of $f \otimes f'$ and we are done. $\square$

**Lemma D.5.** *Let $\rho : G \to \mathrm{U}(V)$ and $\rho' : G \to \mathrm{U}(V')$ be* unitary *representations on pre-Hilbert spaces. Then also $\rho \otimes \rho' : G \to \mathrm{U}(V \otimes V')$ is a well-defined unitary representation.*

*Proof.* According to Lemma D.4 we only need to check whether all $\rho(g) \otimes \rho'(g)$ are unitary transformations. This follows immediately from the unitarity of $\rho(g)$ and $\rho'(g)$. $\square$

### D.1.2 THE CLEBSCH-GORDAN COEFFICIENTS AND THE ORIGINAL WIGNER-ECKART THEOREM

In this section, we describe the Clebsch-Gordan coefficients and the original Wigner-Eckart theorem. Except for the proof, we roughly follow Jeevanjee (2011). For the proof, we follow the more general treatment in Agrawala (1980).[21]

For our aims, let $\rho_j : G \rightarrow \mathrm{U}(V_j)$ and $\rho_l : G \rightarrow \mathrm{U}(V_l)$ be representatives of isomorphism classes of irreducible unitary representations.[22] Then consider their tensor product representation

$$\rho_j \otimes \rho_l : G \rightarrow \mathrm{U}(V_j \otimes V_l)$$

which is again a unitary representation according to Lemma D.5. If $V_j$ and $V_l$ are of dimension $d_j$ and $d_l$, respectively, then $V_j \otimes V_l$ is of dimension $d_j \cdot d_l$. Since it is a finite-dimensional unitary representation, it is itself an orthogonal direct sum of finitely many irreducible unitary representations by Proposition B.38:

$$V_j \otimes V_l \cong \bigoplus_{J \in \widehat{G}} \bigoplus_{s=1}^{[J(jl)]} V_J.$$

Here $\widehat{G}$ is, as before, the set of isomorphism classes of irreducible unitary representations and $[J(jl)]$ is the number of times that $\rho_J : G \rightarrow \mathrm{U}(V_J)$ appears in the direct sum decomposition of $V_j \otimes V_l$. Note that for most $J$ we have $[J(jl)] = 0$, and for some $J$ we may have $[J(jl)] > 1$, see Section E.2, where it turns out that $\rho_0$ is contained twice in $\rho_m \otimes \rho_m$.

Now, choose – once and for all – orthonormal bases of all involved irreps, which exists according to Proposition F.41:

$$\left\{ Y_j^m \mid m = 1, \ldots, d_j \right\} \subseteq V_j,$$
$$\left\{ Y_l^n \mid n = 1, \ldots, d_l \right\} \subseteq V_l,$$
$$\left\{ Y_J^M \mid M = 1, \ldots, d_J \right\} \subseteq V_J.$$

This notation is supposed to remind about spherical harmonics since they form a basis for irreducible representations of the group $\mathrm{SO}(3)$. But as mentioned in the footnote, we do not consider these basis elements to be functions here.

Furthermore, let $l_s : V_J \rightarrow V_j \otimes V_l$ be the linear, equivariant and isometric (i.e., scalar product preserving) embeddings that correspond to the direct sum decomposition of $V_j \otimes V_l$ into irreps, where $s$ ranges in $\{1, \ldots, [J(jl)]\}$. With this in mind, we can define the Clebsch-Gordan coefficients:

**Definition D.6** (Clebsch-Gordan Coefficients). The *Clebsch-Gordan Coefficients* are given by

$$\langle s, JM | jm; ln \rangle := \left\langle l_s(Y_J^M) \middle| Y_j^m \otimes Y_l^n \right\rangle.$$

Note that in the literature, people usually only consider Clebsch-Gordan coefficients of the specific groups $\mathrm{SO}(3)$, $\mathrm{SU}(2)$, $\mathrm{SU}(3)$ or similar groups appearing in physics. Also note that in the physics context, there is only one linear, equivariant, isometric embedding $l_s$, which follows directly from Schur's Lemma D.8. Therefore, it is sensible that the embedding is usually not part of the notation of these coefficients. In our case, however, when considering real representations, there can be several such embeddings $l_s$. This happens if the endomorphism space of $V_J$ is nontrivial. An example is given by the two-dimensional irreducible representations of $\mathrm{SO}(2)$ over the real numbers which we discuss in Section E.2. Since, however, we do not want to depart too much from the notation usually considered in physics, we also omit the embedding from the notation. The index $s$ however needs to be present in order to index the possibly different appearances of $V_J$ in $V_j \otimes V_l$.

With this preparation, we can explain the Wigner-Eckart theorem the way it is usually considered in physics, as a prelude for the generalization that we consider in the next section.

---

[21]It is more general in that it considers arbitrary groups and the situation that the considered irreducible representation appears several times in a tensor product representation instead of just once.

[22]Those are a priori *not* assumed to be embedded in a space of square-integrable functions. For such embedded representations, we write $V_{ji}$ instead.

In this (and only this!) section, we assume that our field is $\mathbb{C}$, since this is the case considered in physics. The Wigner-Eckart theorem aims to obtain a description for all possible representation operators $\mathcal{K} : V_j \to \mathrm{Hom}_{\mathbb{C}}(V_l, V_J)$. This is, for example, useful for describing state transitions in the electrons of hydrogen atoms. To motivate the generalization in the next section, we shortly explain the derivation: we can consider the equivalent function $\tilde{\mathcal{K}} : V_j \otimes V_l \to V_J$ given by $\tilde{\mathcal{K}}(v_j \otimes v_l) := [\mathcal{K}(v_j)](v_l)$ on the tensor product. As one can compute, and as we will see in more generality in Proposition D.23, $\tilde{\mathcal{K}} : V_j \otimes V_l \to V_J$ is an intertwiner, where on the left we consider the tensor product representation. We assume, as is the case for $G = \mathrm{SO}(3)$ or $G = \mathrm{SU}(2)$ for usual applications in physics, that $V_J$ is exactly once a direct summand of $V_j \otimes V_l$. Then, since by Schur's Lemma B.29 there cannot be nontrivial equivariant linear maps between nonisomorphic irreps, $\tilde{\mathcal{K}}$ restricted to each direct summand of $V_j \otimes V_l$ vanishes, except the one isomorphic to $V_J$. More precisely, assume that

$$V_j \otimes V_l \cong V_J \oplus \bigoplus_{l'} V_{l'}$$

is a decomposition of $V_j \otimes V_l$ into copies of irreducible representations, where each $V_{l'}$ is nonisomorphic to $V_J$. Then the information contained in $\tilde{\mathcal{K}}$ is equal to the information contained in the restriction $\tilde{\mathcal{K}}|_{V_J} : V_J \to V_J$. Since it is an intertwiner from a representation to itself, it deserves a special name. We state the following definition for arbitrary $\mathbb{K} \in \{\mathbb{R}, \mathbb{C}\}$, since it will be of crucial importance in our generalization of the Wigner-Eckart theorem:

**Definition D.7** (Endomorphism). Let $\rho : G \to \mathrm{GL}(V)$ be a linear representation. An intertwiner from $V$ to $V$ is called *endomorphism*. The vector space of endomorphisms is written as

$$\mathrm{End}_{G,\mathbb{K}}(V) := \mathrm{Hom}_{G,\mathbb{K}}(V, V).$$

A version of Schur's lemma gives a simple description for endomorphisms of irreducible representations in the case that the underlying field is the complex numbers $\mathbb{C}$. It makes use of the property of the complex numbers to be algebraically closed:

**Lemma D.8** (Schur's Lemma). *Let $\rho : G \to \mathrm{GL}(V)$ be an irreducible representation. If the underlying field is the complex numbers $\mathbb{C}$, then the set of endomorphisms, i.e., intertwiners from $V$ to $V$, only consists of the complex multiples of the identity:*

$$\mathrm{End}_{G,\mathbb{C}}(V) = \{c \cdot \mathrm{id}_V \mid c \in \mathbb{C}\} \cong \mathbb{C}.$$

*Proof.* See Jeevanjee (2011). □

This means that $\tilde{\mathcal{K}}|_{V_J} = c \cdot \mathrm{id}_{V_J}$ for some complex number $c \in \mathbb{C}$. Now if we let $p : V_j \otimes V_l \to V_J$ be the projection corresponding to the direct sum decomposition of $V_j \otimes V_l$, then we obtain

$$\tilde{\mathcal{K}} = \tilde{\mathcal{K}}|_{V_J} \circ p = (c \cdot \mathrm{id}_{V_J}) \circ p = c \cdot p.$$

That is, we have just found out that one complex number, $c$, is able to completely characterize $\tilde{\mathcal{K}}$ and consequently $\mathcal{K}$! This is *basically* already the Wigner-Eckart theorem. However, it is useful to find a formulation that describes $\mathcal{K}$ with respect to bases of the different irreducible representations. For this, we define matrix elements of representation operators. Before we come to the definition, we introduce some notation: If $f : V \to V'$ is a linear continuous map between Hilbert spaces, we set

$$\langle y|f|x \rangle := \langle y|f(x) \rangle$$

for each $x \in V$ and $y \in V'$. The symmetry in this notation is supposed to remind about the fact that $f$ has an adjoint, see Definition F.42, and thus can be applied to $y$ just as well as to $x$, but we will not make use of this fact.

**Definition D.9** (Matrix Element). Let $T$, $V_l$ and $V_J$ be unitary representations with orthonormal bases $\{Y_j^m\} \subseteq T$ (with $j$ possibly also varying), $\{Y_l^n\} \subseteq V_l$ and $\{Y_J^M\} \subseteq V_J$, respectively. Let $\mathcal{K} : T \to \mathrm{Hom}_{\mathbb{K}}(V_l, V_J)$ be a representation operator. Then it's *matrix elements* are given by the scalars

$$\langle JM|\mathcal{K}_j^m|ln \rangle := \langle Y_J^M|\mathcal{K}(Y_j^m)|Y_l^n \rangle.$$

In the same way, if $f : V_l \to V_J$ is any linear (not necessarily equivariant) map, then its matrix elements are given by the scalars

$$\langle JM|f|ln \rangle := \langle Y_J^M|f|Y_l^n \rangle.$$

*Remark* D.10. We shortly explain this term. Usually, in linear algebra, one has to do with linear functions $f : V \rightarrow V'$ between vector spaces carrying bases $\{v_j\} \subseteq V$ and $\{v_i'\} \subseteq V'$. For each basis element $v_j \in V$ one can then find coefficients $A_{ij} \in \mathbb{K}$ such that

$$f(v_j) = \sum_i A_{ij} v_i'.$$

The $A_{ij}$ are called the *matrix elements* of $f$ and characterize $f$ completely. Now if the bases are orthonormal bases as in Definition F.40, then the coefficients are given by

$$A_{ij} = \langle v_i' | f(v_j) \rangle = \langle v_i' | f | v_j \rangle.$$

In a similar way we can understand the matrix elements of a representation operator, only that the linear function itself depends on a chosen basis vector of $V_j$. As for linear functions, the matrix elements of a representation operator completely characterize it.

One last remark: since in this section, $V_J$ appears only once as a direct summand in $V_j \otimes V_l$, we omit the additional "quantum number" $s$ in the notation for the Clebsch-Gordan coefficients. With this preparation, we can formulate and prove the original version of the Wigner-Eckart theorem. Remember that there is a unique complex number $c$ such that $\tilde{\mathcal{K}}$ is given by $\tilde{\mathcal{K}} = c \cdot p$ for a projection $p : V_j \otimes V_l \rightarrow V_J$. We now denote this by $\langle J \| \mathcal{K} \| l \rangle := c$.

**Theorem D.11** (Wigner-Eckart Theorem). *The matrix elements of the representation operator $\mathcal{K}$ : $V_j \rightarrow \mathrm{Hom}_{\mathbb{C}}(V_l, V_J)$ are given by*

$$\langle JM | \mathcal{K}_j^m | ln \rangle = \langle J \| \mathcal{K} \| l \rangle \cdot \langle JM | jm; ln \rangle,$$

*with the $\langle JM | jm; ln \rangle$ being the Clebsch-Gordan coefficients (which are independent from the representation operator $\mathcal{K}$).*

*Proof.* Let $i : V_J \rightarrow V_j \otimes V_l$ be the embedding corresponding to the direct sum decomposition of $V_j \otimes V_l$. It is an adjoint of the projection $p : V_j \otimes V_l \rightarrow V_J$ according to the proof of Proposition F.46. By what we've argued above, there exists some $c \in \mathbb{C}$ such that:

$$\begin{aligned}
\langle JM | \mathcal{K}_j^m | ln \rangle &= \langle Y_J^M | \mathcal{K}(Y_j^m) | Y_l^n \rangle \\
&= \langle Y_J^M | \tilde{\mathcal{K}}(Y_j^m \otimes Y_l^n) \rangle \\
&= \langle Y_J^M | c \cdot p(Y_j^m \otimes Y_l^n) \rangle \\
&= c \cdot \langle Y_J^M | p(Y_j^m \otimes Y_l^n) \rangle \\
&= c \cdot \langle i(Y_J^M) | Y_j^m \otimes Y_l^n \rangle \\
&= \langle J \| \mathcal{K} \| l \rangle \cdot \langle JM | jm; ln \rangle.
\end{aligned}$$

As a short explanation: in the fifth step it was used that $i$ and $p$ are adjoint to each other, and consequently, we move from considering the tensor product in $V_J$ to that one in $V_j \otimes V_l$. In the last step, the definition of the Clebsch-Gordan coefficients was used, and additionally, the notation $\langle J \| \mathcal{K} \| l \rangle := c$ that we mentioned before the theorem. The index $s$ is everywhere missing since $V_J$ appears only once in $V_j \otimes V_l$. This finishes the proof. $\square$

**Definition D.12** (Reduced Matrix Element). The unique number $c = \langle J \| \mathcal{K} \| l \rangle \in \mathbb{C}$ in this theorem is called the *reduced matrix element*. To reiterate, it characterizes the representation operator completely.

### D.1.3 REDUCTION TO IRREDUCIBLE UNITARY REPRESENTATIONS

Let $G$ be any compact group and $X$ any homogeneous space of $G$. Before we state the Wigner-Eckart Theorem for steerable kernels in the next section, we first want to explain why we can restrict to the case of irreducible unitary input- and output representations. Our explanations are adapted from Weiler & Cesa (2019).

Thus, let $\rho_{\mathrm{in}} : G \rightarrow \mathrm{GL}(V_{\mathrm{in}})$ and $\rho_{\mathrm{out}} : G \rightarrow \mathrm{GL}(V_{\mathrm{out}})$ be general finite-dimensional input- and output representations. We consider the task of finding a basis for the space of steerable kernels

$\mathrm{Hom}_G(X, \mathrm{Hom}_{\mathbb{K}}(V_{\mathrm{in}}, V_{\mathrm{out}}))$. By Theorem B.20 and Proposition B.38, there are equivalences of representations (i.e., linear isomorphisms that intertwine between the representations)

$$Q_{\mathrm{in}} : V_{\mathrm{in}} \to \bigoplus_{\mu \in I_{\mathrm{in}}} V_\mu, \qquad Q_{\mathrm{out}} : V_{\mathrm{out}} \to \bigoplus_{\nu \in I_{\mathrm{out}}} V_\nu,$$

where $\rho_\mu : G \to \mathrm{U}(V_\mu)$ and $\rho_\nu : G \to \mathrm{U}(V_\nu)$ are irreducible unitary representations. Both for the input- and the output representation, the same irrep can appear several times, e.g., there can be $\mu \neq \mu'$ such that $\rho_\mu \cong \rho_{\mu'}$. Now, notice that the map

$$\Phi_{Q_{\mathrm{out}}, Q_{\mathrm{in}}} : \mathrm{Hom}_G \left( X, \mathrm{Hom}_{\mathbb{K}} \left( \bigoplus_{\mu \in I_{\mathrm{in}}} V_\mu, \bigoplus_{\nu \in I_{\mathrm{out}}} V_\nu \right) \right) \to \mathrm{Hom}_G(X, \mathrm{Hom}_{\mathbb{K}}(V_{\mathrm{in}}, V_{\mathrm{out}}))$$

given for all $x \in X$ by

$$\left[ \Phi_{Q_{\mathrm{out}}, Q_{\mathrm{in}}}(K) \right](x) := Q_{\mathrm{out}}^{-1} \circ K(x) \circ Q_{\mathrm{in}}$$

is clearly an isomorphism. Thus, once a basis for the first kernel space is known, we just need to postcompose and precompose each basis kernel with $Q_{\mathrm{out}}^{-1}$ and $Q_{\mathrm{in}}$, respectively, in order to get a basis for the space we actually care about. Furthermore, the map

$$\Psi : \bigoplus_{\nu \in I_{\mathrm{out}}} \bigoplus_{\mu \in I_{\mathrm{in}}} \mathrm{Hom}_G(X, \mathrm{Hom}_{\mathbb{K}}(V_\mu, V_\nu)) \to \mathrm{Hom}_G \left( X, \mathrm{Hom}_{\mathbb{K}} \left( \bigoplus_{\mu \in I_{\mathrm{in}}} V_\mu, \bigoplus_{\nu \in I_{\mathrm{out}}} V_\nu \right) \right)$$

given by

$$\left[ \Psi((K^{\nu\mu})_{\nu,\mu})(x) \right] ((v_\mu)_\mu) := \left( \sum_{\mu \in I_{\mathrm{in}}} K^{\nu\mu}(x)(v_\mu) \right)_\nu \in \bigoplus_{\nu \in I_{\mathrm{out}}} V_\nu,$$

where $x \in X$ and $(v_\mu)_\mu \in \bigoplus_{\mu \in I_{\mathrm{in}}} V_\mu$ are arbitrary, is also clearly an isomorphism. It expresses that we can take a collection of steerable kernels $(K^{\nu\mu})_{\nu,\mu}$ and build with it a block-matrix, which is steerable again, as can easily be checked. Accordingly, if we have basis kernels for a space $\mathrm{Hom}_G(X, \mathrm{Hom}_{\mathbb{K}}(V_\mu, V_\nu))$ for some $\mu, \nu$, then we can, by applying $\Psi$, map it to block basis kernels which are zero outside the block with indices $\nu$ and $\mu$. Overall, by doing this for all $\mu, \nu$, we thus recover a full basis for the space $\mathrm{Hom}_G \left( X, \mathrm{Hom}_{\mathbb{K}} \left( \bigoplus_{\mu \in I_{\mathrm{in}}} V_\mu, \bigoplus_{\nu \in I_{\mathrm{out}}} V_\nu \right) \right)$. By applying the base change $\Phi_{Q_{\mathrm{out}}, Q_{\mathrm{in}}}$ from above, we thus get a basis for $\mathrm{Hom}_G(X, \mathrm{Hom}_{\mathbb{K}}(V_{\mathrm{in}}, V_{\mathrm{out}}))$. In summary, knowing a basis of steerable kernels for irreducible unitary input- and output representations gives us one for all finite-dimensional input- and output representations. Finally, note that the transformation of basis kernels using $\Phi_{Q_{\mathrm{out}}, Q_{\mathrm{in}}}$ and $\Psi$ can be done in the network initialization process and does not need to be performed in each forward pass.

### D.1.4 The Wigner-Eckart Theorem for Steerable Kernels

Now that we have seen the Wigner-Eckart theorem in a version similar to how it usually appears in physics, it is time to state the version which we will need in this work for applications in deep learning. The treatment is similar to the formulation in Agrawala (1980), which presents a generalization of the Wigner-Eckart theorem to the case that $V_J$ may appear several times as a direct summand in the direct sum decomposition of the tensor product. However, this paper still only considers the Wigner-Eckart theorem for the case of the complex numbers $\mathbb{C}$. If we allow the real numbers as well, we cannot be sure that endomorphisms of irreducible representations are just given by one number. This is a complication we will deal with below by allowing matrix elements of general endomorphisms. Furthermore, we will deal with topological considerations that did not play a role in Agrawala (1980). And lastly, we transport the theorem over into the nonlinear realm of steerable kernels.

As discussed in the last section, we can restrict the considerations to (representatives of isomorphism classes of) irreducible unitary input- and output representations. Thus, assume the input-representation to be the irrep $\rho_l : G \to \mathrm{U}(V_l)$ and the output-representation to be the irrep $\rho_J : G \to \mathrm{U}(V_J)$. The idea is now that kernel operators $\mathcal{K} : L^2_{\mathbb{K}}(X) \to \mathrm{Hom}_{\mathbb{K}}(V_l, V_J)$ can be described on each direct summand of the domain individually, and that on each of these summands, arguments similar to those for the original Wigner-Eckart theorem apply.

According to the Peter-Weyl Theorem B.22 the space $L^2_{\mathbb{K}}(X)$ has a dense subset which is a direct sum of irreducible unitary representations:

$$L^2_{\mathbb{K}}(X) = \widehat{\bigoplus_{j \in \widehat{G}}} \bigoplus_{i=1}^{m_j} V_{ji}.$$

Each $V_{ji}$ is, as a subrepresentation of $L^2_{\mathbb{K}}(X)$, isomorphic to $V_j$. $V_j$ is itself not assumed to be embedded in $L^2_{\mathbb{K}}(X)$.

For arbitrary $j \in \widehat{G}$, fix once and for all orthonormal bases $\{Y^m_{ji}\} \subseteq V_{ji}$ corresponding to the basis $\{Y^m_j\}$ of $V_j$.[23] Furthermore, assume that for all $s = 1, \ldots, [J(jl)]$, $p_{jis} : V_{ji} \otimes V_l \to V_J$ is a projection which is an adjoint of the linear equivariant isometric embedding $l_{jis} : V_J \to V_{ji} \otimes V_l$. This is assumed to be aligned with the embeddings $V_J \to V_j \otimes V_l$ with respect to the isomorphisms $V_j \cong V_{ji}$ that underlie the correspondence of basis elements $Y^m_j \sim Y^m_{ji}$. What this means is that the Clebsch-Gordan coefficients with respect to all of these embeddings, for all $i$, are equal:

$$\left\langle l_{jis}(Y^M_J) \middle| Y^m_{ji} \otimes Y^n_l \right\rangle = \langle s, JM | jm; ln \rangle .$$

Now we state and prove the Wigner-Eckart theorem, which gives an explicit description of representation operators $\mathcal{K} : L^2_{\mathbb{K}}(X) \to \mathrm{Hom}_{\mathbb{K}}(V_l, V_J)$ in terms of endomorphisms of $V_J$ and then transfers this statement over to a statement about steerable kernels $K : X \to \mathrm{Hom}_{\mathbb{K}}(V_l, V_J)$. Before we state the theorem, we want to shortly explain what to *expect*: in the derivation of the original Wigner-Eckart theorem in Section D.1.2, we saw that a kernel operator could be expressed as $\tilde{\mathcal{K}} : V_j \otimes V_l \to V_J$. This was in turn equal to $\tilde{\mathcal{K}} = c \circ p$ for an endomorphism $c : V_J \to V_J$ and the projection $p$ corresponding to the appearance of $V_J$ in the direct sum decomposition of $V_j \otimes V_l$. This time, however, $V_J$ can be found often in $L^2_{\mathbb{K}}(X) \otimes V_l$, namely:

1. For each isomorphism class of irreps $j \in \widehat{G}$,
2. For each appearance $i = 1, \ldots, m_j$ of the irrep $V_j$ in $L^2_{\mathbb{K}}(X)$ and
3. For each appearance $s = 1, \ldots, [J(jl)]$ of the irrep $V_J$ in the tensor product representation $V_j \otimes V_l$. $[J(jl)]$ can be zero, which means that $j$ does not contribute.

We therefore expect $\tilde{\mathcal{K}}$ to be a whole sum of compositions of endomorphisms with projections, for each combination of valid $j$, $i$ and $s$. Furthermore, the specific structure of $L^2_{\mathbb{K}}(X)$ will be exploited as well by using orthogonal projections from $L^2_{\mathbb{K}}(X)$ to summands $V_{ji}$. Overall, we hope this sufficiently motivates the theorem:

**Theorem D.13** (Wigner-Eckart Theorem for Steerable Kernels). *We state the theorem in three parts:*

1. *(Basis-independent Wigner-Eckart for Kernel Operators) There is an isomorphism of vector spaces*

$$\mathrm{Rep} : \bigoplus_{j \in \widehat{G}} \bigoplus_{i=1}^{m_j} \bigoplus_{s=1}^{[J(jl)]} \mathrm{End}_{G,\mathbb{K}}(V_J) \to \mathrm{Hom}_{G,\mathbb{K}}(L^2_{\mathbb{K}}(X), \mathrm{Hom}_{\mathbb{K}}(V_l, V_J))$$

*which is given by*

$$\left[\mathrm{Rep}((c_{jis})_{jis})(\varphi)\right](v_l) := \sum_{j \in \widehat{G}} \sum_{i=1}^{m_j} \sum_{s=1}^{[J(jl)]} \sum_{m=1}^{d_j} \left\langle Y^m_{ji} \middle| \varphi \right\rangle \cdot c_{jis} \left( p_{jis}(Y^m_{ji} \otimes v_l) \right) \quad (17)$$

*where $(c_{jis})_{jis}$ is a tuple of endomorphisms, $\varphi : X \to \mathbb{K}$ is any square-integrable function and $v_l \in V_l$ is any element.*

2. *(Basis-independent Wigner-Eckart for Steerable Kernels) There is an isomorphism of vector spaces*

$$\mathrm{GKer} : \bigoplus_{j \in \widehat{G}} \bigoplus_{i=1}^{m_j} \bigoplus_{s=1}^{[J(jl)]} \mathrm{End}_{G,\mathbb{K}}(V_J) \to \mathrm{Hom}_G(X, \mathrm{Hom}_{\mathbb{K}}(V_l, V_J))$$

---

[23]$i$ is like an additional quantum number in physics.

*which is given by*

$$[\text{GKer}((c_{jis})_{jis})(x)](v_l) := \sum_{j \in \widehat{G}} \sum_{i=1}^{m_j} \sum_{s=1}^{[J(jl)]} \sum_{m=1}^{d_j} \langle i, jm | x \rangle \cdot c_{jis} \left( p_{jis}(Y_{ji}^m \otimes v_l) \right)$$

*where $(c_{jis})_{jis}$ is a tuple of endomorphisms, $x \in X$ is any point and $v_l \in V_l$ is any element. Here, $\langle i, jm | x \rangle := \lim_{U \in \mathcal{U}_x} \langle Y_{ji}^m | \delta_U \rangle$, which is according to Proposition C.10 equal to $\overline{Y_{ji}^m(x)}$.*

3. *(Basis-dependent Wigner-Eckart for Steerable Kernels) Let $K = \text{GKer}((c_{jis})_{jis})$ be the steerable kernel corresponding to the tuple of endomorphisms $(c_{jis})_{jis}$ according to the isomorphism above. Then the matrix elements of $K(x) \in \text{Hom}_{\mathbb{K}}(V_l, V_J)$ are explicitly given by*

$$\langle JM | K(x) | ln \rangle =$$
$$\sum_{j \in \widehat{G}} \sum_{i=1}^{m_j} \sum_{s=1}^{[J(jl)]} \sum_{m=1}^{d_j} \sum_{M'=1}^{d_J} \langle JM | c_{jis} | JM' \rangle \cdot \langle s, JM' | jm; ln \rangle \cdot \langle i, jm | x \rangle. \tag{18}$$

*Remark* D.14. Before we come to the proof, we have some remarks to make about this theorem:

1. In line with the usual convention, we call the $\langle JM | c_{jis} | JM' \rangle$ the *generalized reduced matrix elements* of the representation operator $\mathcal{K}$. Different from the situation in physics, these can depend nontrivially on the specific basis indices $M$ and $M'$. If the space of endomorphisms is 1-dimensional, as is the case when considering representations over $\mathbb{C}$, then each $c_{jis}$ is a diagonal matrix, meaning that it is characterized by only one complex number, for simplicity with the same name $c_{jis}$. Then one has $\langle JM | c_{jis} | JM' \rangle = \delta_{MM'} \cdot c_{jis}$ and the sum over $M'$ disappears. What this means for the matrix form of basis kernels of steerable CNNs will be discussed in Corollary D.17.

2. The coefficients $\langle s, JM' | jm; ln \rangle$ are as before the *Clebsch-Gordan coefficients*. Note that the input $x$ of $K$ appears only in $\langle i, jm | x \rangle$. Those two parts of the right-hand side of the formula are always the same, independent of the kernel $K$.

3. The Clebsch-Gordan coefficients are traditionally defined with respect to isometric embeddings $l_{jis} : V_J \to V_j \otimes V_l$ since this makes them less ambiguous. However, we mention that the property of being isometric is no requirement for the construction of Clebsch-Gordan coefficients or the proof of the Wigner-Eckart theorem, being equivariant and linear is sufficient. This then means that the copies $l_s(Y_J^M)$ do not anymore form an orthonormal basis. We will use this relaxation in the example in Section E.2, where we do not want to be bothered with obtaining *isometric* embeddings.

4. The names for the isomorphisms in the theorem are meant as follows: $\text{Rep}$ is the map that maps a tuple of endomorphisms to a kernel operator, which is a special **rep**resentation operator. $\text{GKer}$ maps a tuple of endomorphisms to a **G**-steerable **ker**nel. It is *not* meant as a notation for a kernel in the sense of a nullspace in linear algebra.

5. Furthermore, a reader with a background in abstract algebra may wonder why we build the direct sum of spaces of endomorphisms instead of the direct product. The reason is that a posteriori, it turns out that only finitely many $j$ contribute nontrivially, and so the direct sum is equal to the direct product. For a proof of the finiteness, see Remark D.18 below.

6. As a last remark, we want to mention that part 1 of the theorem is not the most general version we could do. We chose to formulate the Wigner-Eckart theorem for $L_{\mathbb{K}}^2(X)$ specifically since this is the space we use it for. However, an appropriate isomorphism can probably be formulated for any unitary representation instead of $L_{\mathbb{K}}^2(X)$, only that we then need to take care that we replace direct sums by direct products if the index sets on the left side are infinite. Additionally, $V_l$ and $V_J$ could be replaced by arbitrary finite-dimensional representations, and an appropriate adaptation of the theorem would apply. Whether $V_l$ and $V_J$ could also be replaced by infinite-dimensional unitary representations would need to be explored, but an extension to such a case seems possible.

*Proof of Theorem D.13.* The proof of 1 will be done in Section D.2 since it requires some work. However, the proofs of 2 and 3 are relatively straightforward once we believe 1 and so we do them here:

From 1 we know that $\mathrm{Rep}$ is an isomorphism. Furthermore, from Theorem C.7 we know that

$$(\cdot)|_X : \mathrm{Hom}_{G,\mathbb{K}}(L^2_{\mathbb{K}}(X), \mathrm{Hom}_{\mathbb{K}}(V_l, V_J)) \to \mathrm{Hom}_{G,\mathbb{K}}(X, \mathrm{Hom}_{\mathbb{K}}(V_l, V_J))$$

is an isomorphism as well, and this is given by $\mathcal{K}|_X(x) := \lim_{U \in \mathcal{U}_x} \mathcal{K}(\delta_U)$, where we take the limit over the directed set of open neighborhoods of $x$. We define the isomorphism $\mathrm{GKer}$ now simply as the composition, i.e., $\mathrm{GKer} := (\cdot)|_X \circ \mathrm{Rep}$. This isomorphism is then explicitly given by:

$$
\begin{aligned}
[\mathrm{GKer}((c_{jis})_{jis})(x)](v_l) &= [\mathrm{Rep}((c_{jis})_{jis})|_X(x)](v_l) \\
&= \lim_{U \in \mathcal{U}_x} [\mathrm{Rep}((c_{jis})_{jis})(\delta_U)](v_l) \\
&= \lim_{U \in \mathcal{U}_x} \sum_{j \in \widehat{G}} \sum_{i=1}^{m_j} \sum_{s=1}^{[J(jl)]} \sum_{m=1}^{d_j} \langle Y_{ji}^m | \delta_U \rangle \cdot c_{jis}\left(p_{jis}(Y_{ji}^m \otimes v_l)\right) \\
&= \sum_{j \in \widehat{G}} \sum_{i=1}^{m_j} \sum_{s=1}^{[J(jl)]} \sum_{m=1}^{d_j} \left[\lim_{U \in \mathcal{U}_x} \langle Y_{ji}^m | \delta_U \rangle\right] \cdot c_{jis}\left(p_{jis}(Y_{ji}^m \otimes v_l)\right) \\
&= \sum_{j \in \widehat{G}} \sum_{i=1}^{m_j} \sum_{s=1}^{[J(jl)]} \sum_{m=1}^{d_j} \langle i, jm | x \rangle \cdot c_{jis}\left(p_{jis}(Y_{ji}^m \otimes v_l)\right).
\end{aligned}
$$

This already proves 2. Now, in the following computation, we will use that $c_{jis} \circ p_{jis} = c_{jis} \circ \mathrm{id}_{V_J} \circ p_{jis}$ and that, inspired by notation in physics, we can write the identity on $V_J$ as $\mathrm{id}_{V_J} = \sum_{M'=1}^{d_J} |Y_J^{M'}\rangle \cdot \langle Y_J^{M'}|$. For 3, we then compute

$$
\begin{aligned}
&\langle JM | K(x) | ln \rangle \\
&= \langle Y_J^M | K(x) | Y_l^n \rangle \\
&= \langle Y_J^M | [\mathrm{GKer}((c_{jis})_{jis})(x)](Y_l^n) \rangle \\
&= \sum_{j \in \widehat{G}} \sum_{i=1}^{m_j} \sum_{s=1}^{[J(jl)]} \sum_{m=1}^{d_j} \langle i, jm | x \rangle \cdot \langle Y_J^M | c_{jis} \circ p_{jis} | Y_{ji}^m \otimes Y_l^n \rangle \\
&= \sum_{j \in \widehat{G}} \sum_{i=1}^{m_j} \sum_{s=1}^{[J(jl)]} \sum_{m=1}^{d_j} \sum_{M'=1}^{d_J} \langle i, jm | x \rangle \cdot \langle Y_J^M | c_{jis} | Y_J^{M'} \rangle \cdot \langle Y_J^{M'} | p_{jis} | Y_{ji}^m \otimes Y_l^n \rangle \\
&= \sum_{j \in \widehat{G}} \sum_{i=1}^{m_j} \sum_{s=1}^{[J(jl)]} \sum_{m=1}^{d_j} \sum_{M'=1}^{d_J} \langle JM | c_{jis} | JM' \rangle \cdot \langle s, JM' | jm; ln \rangle \cdot \langle i, jm | x \rangle.
\end{aligned}
$$

In the last step, we used the Clebsch-Gordan coefficients, see Definition D.6 and, as mentioned before, that $p_{jis}$ is adjoint to the embedding $l_{jis} : V_J \to V_{ji} \otimes V_l$. $\qquad\square$

*Remark* D.15. Here, we want to argue that our kernel space solution also covers that of general equivariant CNNs on homogeneous spaces (Cohen et al., 2019b). One definition of the kernel space in that setting is

$$
\begin{aligned}
&\mathrm{Hom}_{G_{\mathrm{in}} \times G_{\mathrm{out}}}(H, \mathrm{Hom}_{\mathbb{K}}(V_{\mathrm{in}}, V_{\mathrm{out}})) \\
&= \left\{ K : H \to \mathrm{Hom}_{\mathbb{K}}(V_{\mathrm{in}}, V_{\mathrm{out}}) \mid K(g_{\mathrm{out}} h g_{\mathrm{in}}) = \rho_{\mathrm{out}}(g_{\mathrm{out}}) \circ K(h) \circ \rho_{\mathrm{in}}(g_{\mathrm{in}}) \right\},
\end{aligned}
\tag{19}
$$

where $H$ is a locally compact group and $G_{\mathrm{in}}, G_{\mathrm{out}} \subseteq H$ are subgroups with input- and output representations $\rho_{\mathrm{in}} : G_{\mathrm{in}} \to \mathrm{GL}(V_{\mathrm{in}})$ and $\rho_{\mathrm{out}} : G_{\mathrm{out}} \to \mathrm{GL}(V_{\mathrm{out}})$. For compact groups $G_{\mathrm{in}}$ and $G_{\mathrm{out}}$, this is covered by our setting as follows: we define $G := G_{\mathrm{out}} \times G_{\mathrm{in}}$ and $g := (g_{\mathrm{out}}, g_{\mathrm{in}})$. We can define the left action of $G$ on $H$ by $g \cdot h := g_{\mathrm{out}} h g_{\mathrm{in}}^{-1}$. Furthermore, we can reformulate the representations of $G_{\mathrm{in}}$ and $G_{\mathrm{out}}$ to representations of the group $G$ by setting $\rho_{\mathrm{in}} : G \to \mathrm{GL}(V_{\mathrm{in}})$

with $\boldsymbol{\rho_{\mathbf{in}}}(\boldsymbol{g}) := \rho_{\mathrm{in}}(g_{\mathrm{in}})$, and similarly for $\boldsymbol{\rho_{\mathbf{out}}}$. We furthermore notice that in Eq. (19) we could also have inverted $g_{\mathrm{in}}$ since that constraint needs to apply to all elements of $G_{\mathrm{in}}$. Thus, we then see that the kernel space can be equivalently defined by

$$
\begin{aligned}
&\mathrm{Hom}_{G_{\mathrm{in}} \times G_{\mathrm{out}}}(H, \mathrm{Hom}_{\mathbb{K}}(V_{\mathrm{in}}, V_{\mathrm{out}})) \\
&= \left\{ K : H \to \mathrm{Hom}_{\mathbb{K}}(V_{\mathrm{in}}, V_{\mathrm{out}}) \mid K(\boldsymbol{g} \cdot h) = \boldsymbol{\rho_{\mathbf{out}}}(\boldsymbol{g}) \circ K(h) \circ \boldsymbol{\rho_{\mathbf{in}}}(\boldsymbol{g})^{-1} \right\},
\end{aligned}
\tag{20}
$$

which precisely is the kernel constraint of steerable CNNs in Eq. (2). Thus, if we restrict to a homogeneous space of the action of $\boldsymbol{G}$ on $H$, we recover steerable kernels as in Definition 3.2 and can apply Theorem 4.1.

### D.1.5 GENERAL STEERABLE KERNEL BASES

Now that we have a Wigner-Eckart theorem for steerable kernels, which gives a one-to-one correspondence between steerable kernels and tuples of endomorphisms, we can finally describe what a *basis* of the space of steerable kernels looks like. For this, additionally to the notation in the last section, we assume that $\{c_r \mid r = 1, \ldots, E_J\}$ is a basis of $\mathrm{End}_{G,\mathbb{K}}(V_J)$.

**Theorem D.16** (Steerable Kernel Bases). *A basis of the space of steerable kernels* $\mathrm{Hom}_G(X, \mathrm{Hom}_{\mathbb{K}}(V_l, V_J))$ *is given by*

$$
\{K_{jisr} : X \to \mathrm{Hom}_{\mathbb{K}}(V_l, V_J) \mid j \in \widehat{G}, i \in \{1, \ldots, m_j\}, s \in \{1, \ldots, [J(jl)]\}, r = 1, \ldots, E_J\},
$$

*where the basis kernels $K_{jisr}$ have matrix elements*

$$
\langle JM | K_{jisr}(x) | ln \rangle = \sum_{m=1}^{d_j} \sum_{M'=1}^{d_J} \langle JM | c_r | JM' \rangle \cdot \langle s, JM' | jm; ln \rangle \cdot \langle i, jm | x \rangle.
\tag{21}
$$

*Now, for each $M' \in \{1, \ldots, d_J\}$, let $\mathrm{CG}_{J(jl)s}^{M'}$ be the $d_j \times d_l$-matrix of Clebsch-Gordan coefficients $\langle s, JM' | jm; ln \rangle$, with only $m$ and $n$ varying. Furthermore, let $\langle i, j | x \rangle$ be the row vector with entries $\langle i, jm | x \rangle$ for $m = 1, \ldots, d_j$. In matrix-notation with respect to the bases $\{Y_J^M\} \subseteq V_J$ and $\{Y_l^n\} \subseteq V_l$, we can then express the basis kernel $K_{jisr}(x) : V_l \to V_J$ as follows:*

$$
K_{jisr}(x) = c_r \cdot \begin{pmatrix} \langle i, j | x \rangle \cdot \mathrm{CG}_{J(jl)s}^1 \\ \vdots \\ \langle i, j | x \rangle \cdot \mathrm{CG}_{J(jl)s}^{d_J} \end{pmatrix}.
\tag{22}
$$

*In this formula, all "dots" mean conventional matrix multiplication and $c_r$ is by abuse of notation the matrix of the endomorphism $c_r$.*

*Proof.* For the first statement, note that a basis for $\bigoplus_{j \in \widehat{G}} \bigoplus_{i=1}^{m_j} \bigoplus_{s=1}^{[J(jl)]} \mathrm{End}_{G,\mathbb{K}}(V_J)$ is given by all the tuples $t_{jisr} := (0, \ldots, c_r, \ldots, 0)$ that have $c_r$ at position $jis$, for all combinations of $j, i, s$ and $r$. Thus, from the isomorphism GKer in the second part of Theorem D.13 we obtain that all $K_{jisr} := \mathrm{GKer}(t_{jisr})$ together form a basis for the space of steerable kernels $\mathrm{Hom}_G(X, \mathrm{Hom}_{\mathbb{K}}(V_l, V_J))$. When applying the basis-dependent form in part 3 of that theorem to $K_{jisr}$, the first three sums in Eq. (18) just disappear since $t_{jisr}$ is zero almost everywhere. Furthermore, $c_{jis}$ is replaced by the basis endomorphism $c_r$. We obtain the claimed result.

For the final statement on the matrix representation, note that

$$
\begin{aligned}
\langle JM | K_{jisr}(x) | ln \rangle &= \sum_{m=1}^{d_j} \sum_{M'=1}^{d_J} \langle JM | c_r | JM' \rangle \cdot \langle s, JM' | jm; ln \rangle \cdot \langle i, jm | x \rangle \\
&= \sum_{M'=1}^{d_J} \langle JM | c_r | JM' \rangle \sum_{m=1}^{d_j} \langle i, jm | x \rangle \cdot \langle s, JM' | jm; ln \rangle \\
&= c_r^M \cdot \left( \sum_{m=1}^{d_j} \langle i, jm | x \rangle \cdot \langle s, JM' | jm; ln \rangle \right)_{M'=1}^{d_J} \\
&= c_r^M \cdot \left( \langle i, j | x \rangle \cdot \mathrm{CG}_{J(jl)s}^{M'-n} \right)_{M'=1}^{d_J}.
\end{aligned}
$$

Here, $c_r^M$ is the $M$'th row of the matrix $c_r$. The result follows by dropping the indices $M$ and $n$. □

The next corollary means that endomorphisms can be ignored if the space of endomorphisms is 1-dimensional, which is in particular the case if $\mathbb{K} = \mathbb{C}$.

**Corollary D.17.** *Assume that* $\dim(\mathrm{End}_{G,\mathbb{K}}(V_J)) = 1$. *Then a basis of steerable kernels* $K : X \to \mathrm{Hom}_{\mathbb{K}}(V_l, V_J)$ *is given by all* $K_{jis}$ *with matrices*

$$K_{jis}(x) = \begin{pmatrix} \langle i, j | x \rangle \cdot \mathrm{CG}^1_{J(jl)s} \\ \vdots \\ \langle i, j | x \rangle \cdot \mathrm{CG}^{d_J}_{J(jl)s} \end{pmatrix}. \tag{23}$$

*In particular, this is the case if* $\mathbb{K} = \mathbb{C}$.

*Proof.* In this case, a basis for the space of endomorphisms is given by the single endomorphism $c = \mathrm{id}_{V_J}$. Postcomposition with the identity does not change the matrix, and so the result follows.

For $\mathbb{K} = \mathbb{C}$ we have $\dim(\mathrm{End}_{G,\mathbb{C}}(V_J)) = 1$ by Schur's Lemma D.8, and thus the result follows. □

We end with two remarks regarding the parameterization of steerable CNNs. The first remark considers the case of steerable CNNs of the form $K : X \to \mathrm{Hom}_{\mathbb{K}}(V_l, V_J)$ on a homogeneous space $X$. The second remark connects this back to the case that $X$ is an orbit embedded in $\mathbb{R}^d$.

*Remark* D.18 (Parameterization in the abstract). First of all, we want to understand that there are only finitely many basis kernels $K_{jisr}$. To this end, note that the index sets for $i$, $s$, and $r$ are necessarily finite for all $j$, and thus we need to understand the finite range of $j$. A priori, $j$ can run over the whole set $\widehat{G}$, which can be infinite. But, as we argue now, for only finitely many $j \in \widehat{G}$ we can have $V_J$ in a direct sum decomposition of $V_j \otimes V_l$, which rescues the finiteness:

Namely, $V_J$ is in the direct sum decomposition of $V_j \otimes V_l$ if and only if the vector space $\mathrm{Hom}_{G,\mathbb{K}}(V_j \otimes V_l, V_J)$ is nonzero by Schur's Lemma B.29. By the hom-tensor adjunction that we will show in Proposition D.23 in more generality, this is the case if an only if $\mathrm{Hom}_{G,\mathbb{K}}(V_j, \mathrm{Hom}_{\mathbb{K}}(V_l, V_J))$ is nonzero. And finally, this is the case if and only if $V_j$ is in a direct sum decomposition of the representation $\mathrm{Hom}_{\mathbb{K}}(V_l, V_J)$, again by Schur's lemma. Now, since $\mathrm{Hom}_{\mathbb{K}}(V_l, V_J)$ is finite-dimensional, this can only be the case for finitely many $j$, and so we are done.[24]

Overall, this means the following: To parameterize an equivariant neural network, one needs arbitrary parameters $w_{jisr} \in \mathbb{K}$ for all combinations of $j \in \widehat{G}$, $i \in \{1, \ldots, m_j\}$, $s \in \{1, \ldots, [J(jl)]\}$ and $r = 1, \ldots, E_J$. A general steerable Kernel $K : X \to \mathrm{Hom}_{\mathbb{K}}(V_l, V_J)$ then takes the form

$$K = \sum\nolimits_{j \in \widehat{G}} \sum\nolimits_{i=1}^{m_j} \sum\nolimits_{s=1}^{[J(jl)]} \sum\nolimits_{r=1}^{E_J} w_{jisr} K_{jisr},$$

with the basis kernels $K_{jisr}$ as in Theorem D.16.

*Remark* D.19 (Parameterization in practice). Remember that our original motivation for the use of homogeneous spaces in Section C.1.1 was that $\mathbb{R}^d$ splits as a disjoint union of homogeneous spaces, on which the kernel constraint acts separately. For simplicity, we assume that the compact group acting on $\mathbb{R}^d$ is either $G = \mathrm{SO}(d)$ or $G = \mathrm{O}(d)$, but the general ideas hold also for the finite transformation groups in $\mathbb{R}^d$ – the only difference is that in these finite cases, the set of representatives of orbits becomes larger.

Thus, $\mathbb{R}^d$ splits into orbits $\mathbb{R}^d = \bigsqcup_{r \geq 0} S^{n-1}(r)$, where $S^{n-1}(r)$ is the sphere of radius $r$ (with $S(0) = \{0\}$ being a single point).

We'll discuss the orbit $X_0 = \{0\}$, the origin, separately below. But note that all other orbits are necessarily homeomorphic to each other and thus can be treated on equal footing. Therefore, let $S^{n-1}$ be the standard sphere with radius 1 and $K_{jisr} : S^{n-1} \to \mathrm{Hom}_{\mathbb{K}}(V_l, V_J)$ be basis kernels for this choice. Then for a general steerable kernel $K : \mathbb{R}^d \to \mathrm{Hom}_{\mathbb{K}}(V_l, V_J)$ there are *arbitrary* functions $w_{jisr} : \mathbb{R}_{>0} \to \mathbb{K}$ such that, for all $x \in \mathbb{R}^d \setminus \{0\}$, we have:

$$K(x) = \sum\nolimits_{j \in \widehat{G}} \sum\nolimits_{i=1}^{m_j} \sum\nolimits_{s=1}^{[J(jl)]} \sum\nolimits_{r=1}^{E_J} w_{jisr}(\|x\|) \cdot K_{jisr}\left(\frac{x}{\|x\|}\right).$$

---

[24]Of course, for this argument, we need the uniqueness of direct sum decompositions. But this follows if we assume the $\mathrm{Hom}$-representation to be unitary, which works by Proposition B.20 and then using the Krull-Remak-Schmidt Theorem, Proposition B.39.

For $x = 0$, we might use our heavy theory to solve the kernel constraint, but it is more illuminating to do it from scratch since this case is so simple: we have $K(0) : V_l \to V_J$, and the kernel constraint takes the form

$$K(0) = K(g \cdot 0) = \rho_J(g) \circ K(0) \circ \rho_l(g)^{-1}$$

for all $g \in G$, which is equivalent to $K(0) \circ \rho_l(g) = \rho_J(g) \circ K(0)$ for all $g \in G$. This just means that $K(0) : V_l \to V_J$ is an intertwiner, and by Schur's Lemma B.29 it is either 0 if $l \neq J$ or an arbitrary endomorphism $V_J \to V_J$ if $l = J$. Thus, assuming $l = J$ and choosing basis-endomorphisms $c_r : V_J \to V_J$, there are coefficients $w_r \in \mathbb{K}$ such that

$$K(0) = \sum_{r=1}^{E_J} w_r \cdot c_r.$$

The reader may find it interesting to check that this solution is precisely what is also predicted by our theory using that $L^2_{\mathbb{K}}(\{0\}) \cong \mathbb{K}$ is just isomorphic to the trivial representation of $G$.

All in all, we now know what the most general steerable kernels look like. In practice, one needs to choose the functions $w_{jisr} : \mathbb{R}_{>0} \to \mathbb{K}$. For representations over the real numbers, i.e., with $\mathbb{K} = \mathbb{R}$, one choice is to only consider finitely many radii and Gaussian radial profiles around them. Then instead of learning the whole function $w_{jisr}$, one learns finitely many real parameters that choose "how activated" a basis kernel $K_{jisr}$ is for a certain radius. This is, for example, the route taken in Weiler et al. (2018b;a); Weiler & Cesa (2019). If one deals with complex representations, one usually goes the same route, only that the parameters that choose how "activated" the basis kernels are will then be *complex numbers*. One can either parameterize them as $a + ib$ with a real part $a$ and a complex part $b$. This intuitively means that $a$ activates the standard version of the kernel $K_{jisr}$, whereas $b$ activates the kernel $iK_{jisr}$, which can be imagined as a version of the kernel turned by 90°. One other possibility is to parameterize a complex number as $\alpha \cdot e^{i\beta}$ with a scaling factor $\alpha > 0$ and a phase shift $\beta$. This is the route chosen in Worrall et al. (2016).

In Chapter E we will look at examples of determining the basis kernels $K_{jisr}$, which will hopefully further illuminate the theorem. In the next section, we go back to the theory and prove the remaining parts of the Wigner-Eckart theorem.

## D.2    Proof of the Wigner-Eckart Theorem for Kernel Operators

In this section, we prove the first part of Theorem D.13, the Wigner-Eckart theorem for Kernel Operators, since we have skipped this in the last section. It is not necessary to read this section and the reader may wish to directly go to the chapter on examples E. We will make frequent use of topological concepts from Chapter F.1 in this section.

The strategy is the following: in Section D.2.1, we show that

$$\mathrm{Hom}_{G,\mathbb{K}}(L^2_{\mathbb{K}}(X), \mathrm{Hom}_{\mathbb{K}}(V_l, V_J)) \cong \mathrm{Hom}_{G,\mathbb{K}}\left( \bigoplus_{j \in \widehat{G}} \bigoplus_{i=1}^{m_j} V_{ji}, \mathrm{Hom}_{\mathbb{K}}(V_l, V_J) \right),$$

which basically means that we can ignore the "topological closure" of the direct sum which is dense in $L^2_{\mathbb{K}}(X)$. This works, intuitively, since kernel operators are continuous, and so they are determined by what they do on a dense subset. Then, in section D.2.2, we show that

$$\mathrm{Hom}_{G,\mathbb{K}}\left( \bigoplus_{j \in \widehat{G}} \bigoplus_{i=1}^{m_j} V_{ji}, \mathrm{Hom}_{\mathbb{K}}(V_l, V_J) \right) \cong \mathrm{Hom}_{G,\mathbb{K}}\left( \bigoplus_{j \in \widehat{G}} \bigoplus_{i=1}^{m_j} V_{ji} \otimes V_l, V_J \right),$$

which is the main step that we need in order to be able to make use of the Clebsch-Gordan coefficients, namely when we decompose the tensor product. Finally, in Section D.2.3, we finish the proof of Theorem D.13.

### D.2.1    Reduction to a Dense Subspace of $L^2_{\mathbb{K}}(X)$

In this section, we reduce the statement to representation operators on $\bigoplus_{j \in \widehat{G}} \bigoplus_{i=1}^{m_j} V_{ji}$. For simplicity, we write the double direct sum from now on as $\bigoplus_{ji}$.

Furthermore, remember that $V_l$ and $V_J$ are finite-dimensional, and thus $\mathrm{Hom}_{\mathbb{K}}(V_l, V_J)$ can be identified with matrices in $\mathbb{K}^{d_J \times d_l}$. This space is a Euclidean space and thus has a scalar product and consequently also a norm, see Chapter F.1. Consequently, each kernel operator is a continuous map between *normed* vector spaces, which we'll use in the following.

A short terminological note: kernel operators are just representation operators on $L^2_{\mathbb{K}}(X)$ and only have their name due to the relation to steerable kernels. Thus, the terminological difference to representation operators in the following reduction result has no further meaning:

**Lemma D.20.** *The restriction map*

$$\mathrm{Hom}_{G,\mathbb{K}}(L^2_{\mathbb{K}}(X), \mathrm{Hom}_{\mathbb{K}}(V_l, V_J)) \to \mathrm{Hom}_{G,\mathbb{K}}\left(\bigoplus\nolimits_{ji} V_{ji}, \mathrm{Hom}_{\mathbb{K}}(V_l, V_J)\right)$$

*given by $\mathcal{K} \mapsto \mathcal{K}|_{\bigoplus_{ji} V_{ji}}$, between kernel operators on the left and representation operators on the right is an isomorphism.*

*Proof.* First of all, the kernel operators on the left are actually uniformly continuous by Proposition F.18. Thus, by Lemma F.22, the restriction map is an injection into *uniformly continuous* representation operators on $\bigoplus_{ji} V_{ji}$. The set of all these maps is equal to the set of all representation operators by Proposition F.18 again.

Thus, in order to be finished, we only need to see that the unique extension of a representation operator $\mathcal{K} : \bigoplus_{ji} V_{ji} \to \mathrm{Hom}_{\mathbb{K}}(V_l, V_J)$ to a continuous function $\overline{\mathcal{K}} : L^2_{\mathbb{K}}(X) \to \mathrm{Hom}_K(V_l, V_J)$ is a kernel operator, which means it is linear and equivariant.

For linearity, let $a \in \mathbb{K}$ and $f \in L^2_{\mathbb{K}}(X)$. Let $(f_k)_k$ be a sequence in $\bigoplus_{ji} V_{ji}$ that converges to $f$. Using the continuity of $\overline{\mathcal{K}}$ and the linearity of $\mathcal{K}$ we obtain:

$$\begin{aligned}
\overline{\mathcal{K}}(a \cdot f) &= \overline{\mathcal{K}}\big(\lim_{k \to \infty}(a \cdot f_k)\big) \\
&= \lim_{k \to \infty} \overline{\mathcal{K}}(a \cdot f_k) \\
&= \lim_{k \to \infty} \mathcal{K}(a \cdot f_k) \\
&= \lim_{k \to \infty} a \cdot \mathcal{K}(f_k) \\
&= a \cdot \lim_{k \to \infty} \overline{\mathcal{K}}(f_k) \\
&= a \cdot \overline{\mathcal{K}}\big(\lim_{k \to \infty} f_k\big) \\
&= a \cdot \overline{\mathcal{K}}(f).
\end{aligned}$$

Linearity with respect to addition can be shown similarly. For the equivariance we can argue in the same way, only that we additionally need to use the continuity of the representations $\lambda : G \to \mathrm{U}(L^2_{\mathbb{K}}(X))$ and $\rho_{\mathrm{Hom}} : G \to \mathrm{GL}(\mathrm{Hom}_{\mathbb{K}}(V_l, V_J))$. $\qquad\square$

### D.2.2 THE HOM-TENSOR ADJUNCTION

**Lemma D.21.** *Let $\mathcal{K} : \bigoplus_{li} V_{li} \to V$ be linear and equivariant, where $V$ is an irrep. Then $\mathcal{K}$ is continuous.*

*Proof.* By Schur's Lemma D.8,[25] we know that $\mathcal{K}$ factors through the irreducible representations that are isomorphic to $V$. That is, let $V_j$ be that irrep and $p_{ji} : \bigoplus_{li} V_{li} \to V_{ji}$ be the canonical projections. Then there are intertwiners $c_i : V_{ji} \to V$ such that $\mathcal{K} = \sum_i c_i \circ p_{ji}$. Each $c_i$ is continuous since it is a linear function between *finite-dimensional* normed vector spaces. Since also summation on normed vector spaces is continuous, we only need to show that the projections $p_{ji}$ are continuous.

---

[25]Schur's lemma applies since it is a statement about irreducible representations which are necessarily finite-dimensional. This means that the continuity condition in the definition of intertwiners is vacuous and thus we don't need to worry about $\mathcal{K}$ not being continuous *a priori*.

This follows from the following fact on how the norm on $\bigoplus_{li} V_{li}$ is composed from the norms on each $V_{li}$: For an element $f = \sum_{li} f_{li} \in \bigoplus_{li} V_{li}$ with $f_{li} \in V_{li}$, we have:

$$\|f\|^2 = \sum_{li} \|f_{li}\|^2.$$

The reason for this is that the $V_{li}$ are perpendicular to each other. Consequently, if $(f^k)_k$ with $f^k \in \bigoplus_{li} V_{li}$ converges to 0, then also $(p_{ji}(f^k))_k = (f_{ji}^k)_k$ converges to 0, which shows the continuity of $p_{ji}$ in 0 and thus general continuity by Proposition F.18. □

*Remark* D.22. Note the curious fact that we cannot get rid of the equivariance condition in the preceding Lemma. I.e., if we have a linear function $\mathcal{K} : \bigoplus_l V_l \to V$, then we cannot deduce that $\mathcal{K}$ is continuous. We omit the index $i$ for simplicity. If equivariance is no requirement, then we only deal with vector spaces, which are in general isomorphic to spaces of (maybe infinite) tuples of elements in $\mathbb{K}$. Thus, let the function $\mathcal{K} : \bigoplus_{l \in \mathbb{N}} \mathbb{K} \to \mathbb{K}$ given by

$$(a_l)_l \mapsto \sum_l l \cdot a_l.$$

This is linear but not continuous in 0. The latter can be seen by considering the sequence $(a^k)_k$ with $a^k = (0, \ldots, 0, \frac{1}{k}, 0, \ldots)$ that has value $\frac{1}{k}$ on position $k$ and otherwise only zeros. This sequence converges to the 0-sequence in norm. However, we have $\mathcal{K}(a^k) = 1$ for all $k$, thus the images do not converge to $0 = \mathcal{K}(0)$. □

From the preceding lemma, we are able to obtain the following alternative description of representation operators:

**Proposition D.23** (Hom-tensor Adjunction). *The map*

$$\tilde{(\cdot)} : \mathrm{Hom}_{G,\mathbb{K}}\left(\bigoplus_{ji} V_{ji}, \mathrm{Hom}_{\mathbb{K}}(V_l, V_J)\right) \to \mathrm{Hom}_{G,\mathbb{K}}\left(\left(\bigoplus_{ji} V_{ji}\right) \otimes V_l, V_J\right)$$

*given by*

$$\tilde{\mathcal{K}}(v_j \otimes v_l) \coloneqq [\mathcal{K}(v_j)](v_l)$$

*is an isomorphism.*

*Proof.* For continuity, note the following: by straightforward extensions of Lemma D.21, all linear and equivariant maps $\bigoplus_{ji} V_{ji} \to \mathrm{Hom}_{\mathbb{K}}(V_l, V_J)$ and $\left(\bigoplus_{ji} V_{ji}\right) \otimes V_l \to V_J$ are necessarily continuous, and thus we can ignore continuity altogether. The rest of the proof can be done as in Agrawala (1980). For illustrating the most important part, we show that $\tilde{\mathcal{K}}$ is actually equivariant:

$$\begin{aligned}
\tilde{\mathcal{K}}\big([(\rho_j \otimes \rho_l)(g)](v_j \otimes v_l)\big) &= \tilde{\mathcal{K}}\big([\rho_j(g)](v_j) \otimes [\rho_l(g)](v_l)\big) \\
&= [\mathcal{K}(\rho_j(g)(v_j))](\rho_l(g)(v_l)) \\
&= \big[\rho_{\mathrm{Hom}}(g)(\mathcal{K}(v_j))\big](\rho_l(g)(v_l)) \\
&= \big(\rho_J(g) \circ \mathcal{K}(v_j) \circ \rho_l(g)^{-1}\big)(\rho_l(g)(v_l)) \\
&= \rho_J(g)(\mathcal{K}(v_j)(v_l)) \\
&= \rho_J(g)(\tilde{\mathcal{K}}(v_j \otimes v_l)).
\end{aligned}$$

□

*Remark* D.24. Some readers may wonder why this is called an *adjunction*. With removing some of the notation in the Proposition, one has

$$\mathrm{Hom}_{G,\mathbb{K}}(T, \mathrm{Hom}_{\mathbb{K}}(U, V)) \cong \mathrm{Hom}_{G,\mathbb{K}}(T \otimes U, V).$$

Now, for notational clarity, set $F \coloneqq \mathrm{Hom}_{\mathbb{K}}(U, \cdot)$ and $H \coloneqq (\cdot) \otimes U$ and remove the subscripts. Then the formula can be written as

$$\mathrm{Hom}(T, F(V)) \cong \mathrm{Hom}(H(T), V).$$

With replacing the notation if the $\mathrm{Hom}$-spaces with a scalar product, and the isomorphism sign with equality, this reads as follows:

$$\langle T | F(V) \rangle = \langle H(T) | V \rangle.$$

Similar to adjoints in Hilbert spaces, we can then view $F$ and $H$ as adjoint to each other. In categorical terms, they are a pair of adjoint functors, see Lane et al. (1998).

### D.2.3 PROOF OF THEOREM D.13

After the work done in the prior sections, we are ready to complete the proof of Theorem D.13!

*Proof of Theorem D.13.* Only the first part of that theorem still needs to be proven. We have the following string of isomorphisms, which we will explain below:

$$\operatorname{Hom}_{G,\mathbb{K}}(L_{\mathbb{K}}^2(X), \operatorname{Hom}_{\mathbb{K}}(V_l, V_J)) \cong \operatorname{Hom}_{G,\mathbb{K}}\Big(\bigoplus_{ji} V_{ji}, \operatorname{Hom}_{\mathbb{K}}(V_l, V_J)\Big)$$

$$\cong \operatorname{Hom}_{G,\mathbb{K}}\Big(\Big(\bigoplus_{ji} V_{ji}\Big) \otimes V_l, V_J\Big)$$

$$\cong \operatorname{Hom}_{G,\mathbb{K}}\Big(\bigoplus_{ji}(V_{ji} \otimes V_l), V_J\Big)$$

$$\cong \bigoplus_{ji} \operatorname{Hom}_{G,\mathbb{K}}(V_{ji} \otimes V_l, V_J)$$

$$\cong \bigoplus_{ji} \bigoplus_{s=1}^{[J(jl)]} \operatorname{Hom}_{G,\mathbb{K}}(V_J, V_J)$$

$$= \bigoplus_{j \in \widehat{G}} \bigoplus_{i=1}^{m_j} \bigoplus_{s=1}^{[J(jl)]} \operatorname{End}_{G,\mathbb{K}}(V_J).$$

The steps are justified as follows:

1. For the first step, use Lemma D.20.

2. For the second step, use Proposition D.23.

3. For the third step, use that there is a natural isomorphism $\Big(\bigoplus_{ji} V_{ji}\Big) \otimes V_l \cong \bigoplus_{ji}(V_{ji} \otimes V_l)$.

4. For the fourth step, use that linear equivariant maps can be described on each direct summand individually (and that we do not need to worry about continuity due to Lemma D.21).

5. For the fifth step, precompose with the linear equivariant isometric embeddings $l_{jis} : V_J \to V_{ji} \otimes V_l$ and use, again, that linear equivariant maps can be described on each direct summand individually. Furthermore, use Schur's Lemma B.29 in order to see that the other summands disappear.

6. The last step is just a reformulation.

Now, we call the string of isomorphisms from right to left

$$\operatorname{Rep} : \bigoplus_{j \in \widehat{G}} \bigoplus_{i=1}^{m_j} \bigoplus_{s=1}^{[J(jl)]} \operatorname{End}_{G,\mathbb{K}}(V_J) \to \operatorname{Hom}_{G,\mathbb{K}}(L_{\mathbb{K}}^2(X), \operatorname{Hom}_{\mathbb{K}}(V_l, V_J))$$

and are only left with understanding that it is actually given by Eq. (17). For this, we take a tuple $(c_{jis})_{jis}$ of endomorphisms and explicitly trace back "where it comes from". As in Lemma D.21, let $p_{ji} : \bigoplus_{j'i'} V_{j'i'} \to V_{ji}$ be the canonical projection, which is by Proposition F.46 explicitly given by $p_{ji}(\varphi) = \sum_{m=1}^{d_j} \langle Y_{ji}^m | \varphi \rangle Y_{ji}^m$. Furthermore, let $p_{jis} : V_{ji} \otimes V_l \to V_J$ be the projections corresponding to the embeddings $l_{jis}$. Then from bottom to top, $(c_{jis})_{jis}$ gets transformed as follows:

$$(c_{jis})_{jis} \mapsto \Big( \sum_{s=1}^{[J(jl)]} c_{jis} \circ p_{jis} \Big)_{ji}$$

$$\mapsto \sum_{j \in \widehat{G}} \sum_{i=1}^{m_j} \sum_{s=1}^{[J(jl)]} c_{jis} \circ p_{jis} \circ (p_{ji} \otimes \operatorname{id}_{V_l})$$

$$\mapsto \operatorname{Rep}((c_{jis})_{jis})$$

In the last step, the hom-tensor adjunction Proposition D.23 is used, but in the other direction. As an illustration, the composition of functions over which we sum can be shown in the following commutative diagram:

$$
\bigoplus_{i'j'} V_{j'i'} \otimes V_l \xrightarrow{p_{ji} \otimes \mathrm{id}_{V_l}} V_{ji} \otimes V_l \xrightarrow{p_{jis}} V_J \xrightarrow{c_{jis}} V_J
$$

$$
c_{jis} \circ p_{jis} \circ (p_{ji} \otimes \mathrm{id}_{V_l})
$$

We obtain:

$$
\left[ \mathrm{Rep}((c_{jis})_{jis})(\varphi) \right](v_l) = \sum_{j \in \widehat{G}} \sum_{i=1}^{m_j} \sum_{s=1}^{[J(jl)]} \left[ c_{jis} \circ p_{jis} \circ (p_{ji} \otimes \mathrm{id}_{V_l}) \right](\varphi \otimes v_l)
$$

$$
= \sum_{j \in \widehat{G}} \sum_{i=1}^{m_j} \sum_{s=1}^{[J(jl)]} (c_{jis} \circ p_{jis})(p_{ji}(\varphi) \otimes v_l)
$$

$$
= \sum_{j \in \widehat{G}} \sum_{i=1}^{m_j} \sum_{s=1}^{[J(jl)]} (c_{jis} \circ p_{jis}) \left( \sum_{m=1}^{d_j} \langle Y_{ji}^m | \varphi \rangle \, Y_{ji}^m \otimes v_l \right)
$$

$$
= \sum_{j \in \widehat{G}} \sum_{i=1}^{m_j} \sum_{s=1}^{[J(jl)]} \sum_{m=1}^{d_j} \langle Y_{ji}^m | \varphi \rangle \cdot c_{jis} \left( p_{jis} \left( Y_{ji}^m \otimes v_l \right) \right).
$$

That, finally, finishes the proof. □

# E    EXAMPLE APPLICATIONS

In this chapter, we develop some relevant examples of the theory outlined in prior chapters. All of these examples are applications of Theorem D.16 and Corollary D.17. These examples are concerned with the following question: Given a *specific* field $\mathbb{K} \in \{\mathbb{R}, \mathbb{C}\}$, compact transformation group $G$ and homogeneous space $X$ of $G$, how can a basis of steerable kernels $K : X \to \mathrm{Hom}_{\mathbb{K}}(V_l, V_J)$ for given irreducible representations $\rho_l : G \to \mathrm{U}(V_l)$ and $\rho_J : G \to \mathrm{U}(V_J)$ be determined? The theorems give an outline for what needs to be done in order to succeed in this task, and the steps are always as follows:

1. For each $l \in \widehat{G}$, a representative for the isomorphism class of irreducible representations $l$ needs to be determined. That is, one needs to determine $\rho_l : G \to \mathrm{U}(V_l)$ and an orthonormal basis $\{Y_l^n \mid n \in \{1, \ldots, d_l\}\}$. We omit the index $n$ if there is only one basis element. Usually, we have $V_l = \mathbb{K}^{d_l}$ and the orthonormal basis is just the standard basis.

2. The Peter-Weyl Theorem B.22 gives the existence-statement for a decomposition of $L_{\mathbb{K}}^2(X)$ into irreducible subrepresentations. We need an *explicit* such decomposition, i.e.: we need to find multiplicities $m_j$, irreducible subrepresentations $V_{ji} \cong V_j$ for $i \in \{1, \ldots, m_j\}$ and basis functions $Y_{ji}^m \in V_{ji} \subseteq L_{\mathbb{K}}^2(X)$ corresponding to the $Y_j^m$ such that $L_{\mathbb{K}}^2(X) = \widehat{\bigoplus}_{j \in \widehat{G}} \bigoplus_{i=1}^{m_j} V_{ji}$.

3. For each combination of $j, l$ and $J$ in $\widehat{G}$, one needs to find the number of times $[J(jl)]$ that $V_J$ appears in a direct sum decomposition of $V_j \otimes V_l$. Then, for each $s \in \{1, \ldots, [J(jl)]\}$, and for all basis-indices $M, m$ and $n$, one needs to determine the Clebsch-Gordan coefficients $\langle s, JM | jm; ln \rangle$. We omit the index $s$ if $V_J$ appears only once in the direct sum decomposition of $V_j \otimes V_l$.

4. For each $J$ one needs to determine a basis $\{c_r \mid r = 1, \ldots, E_J\}$ of the space of endomorphisms of $V_J$, namely $\mathrm{End}_{G,\mathbb{K}}(V_J)$.

Once all of this is done, one can then simply write down the basis kernels according to Eq. (22) or, in case that the space of endomorphisms is 1-dimensional, Eq. (23). The ingredients determined

above are purely representation-theoretic information about the situation at hand, which hopefully makes the reader appreciate the results even more: we do not simply determine basis kernels; we understand in detail, along the way, the representation theory of the group and homogeneous space.

Note that we are not concerned with practical considerations related to how fine-grained to do this in practice (for example, if the space on which the kernels operate splits into infinitely many orbits). For such questions, we refer back to Remark D.19.

In the following sections, we discuss harmonic networks (SO(2)-equivariant CNNs with complex representations), SO(2)-equivariant CNNs with real representations, reflection-equivariant networks, SO(3)-equivariant CNNs with both complex and real representations, and O(3)-equivariant CNNs with both complex and real representations. For each of these examples, we go through the four steps outlined above. We recommend looking at the first example in detail: we conduct it in the greatest detail and it is the easiest to understand and thus serves as a nice introduction.

### E.1    SO(2)-STEERABLE KERNELS FOR COMPLEX REPRESENTATIONS – HARMONIC NETWORKS

Here, we explain how the kernel constraint for harmonic networks (Worrall et al., 2016) can be solved using our theory. In the case of harmonic networks, we have $\mathbb{K} = \mathbb{C}$, $G = \mathrm{SO}(2)$, $X = S^1$. As in most examples that follow, we ignore the solution of the kernel constraint in the origin, since it is usually easy to solve. For simplifying the formulas, we employ the isomorphism

$$\mathrm{SO}(2) \xrightarrow{\sim} \mathrm{U}(1), \qquad \begin{pmatrix} a & -b \\ b & a \end{pmatrix} \mapsto a + ib$$

and always write $\mathrm{U}(1)$ instead of $\mathrm{SO}(2)$. Here, $\mathrm{U}(1)$ is the group of rotations of $\mathbb{C}$, i.e., the group of elements in $\mathbb{C}$ with absolute value 1. It is also called the *circle group* since the group elements lie on a circle in the complex plane. Note that the change from $\mathrm{SO}(2)$ to the isomorphic group $\mathrm{U}(1)$ is done purely for convenience reasons, and $\mathrm{SO}(2)$ could be used just as well.

We now go through the four steps outlined above. Our statements about the representation theory of the circle group can be found in Kowalski (2014), chapter 5.

#### E.1.1    CONSTRUCTION OF THE IRREDUCIBLE REPRESENTATIONS OF U(1)

We have $\widehat{\mathrm{U}(1)} = \mathbb{Z}$, and for $l \in \mathbb{Z}$ we can construct a representative $\rho_l : \mathrm{U}(1) \to \mathrm{U}(V_l)$ as follows: $V_l = \mathbb{C}$ is just the canonical 1-dimensional $\mathbb{C}$-vector space, and $\rho_l$ is given by

$$[\rho_l(g)](z) := g^l \cdot z,$$

where $g$ is regarded as an element in $\mathbb{C}$. One can easily check that this is an irreducible representation. The orthonormal basis element for each such representation is just given by $1 \in \mathbb{C} = V_l$. This already answers step 1 of the outline above.

#### E.1.2    THE PETER-WEYL THEOREM FOR $L^2_{\mathbb{C}}(S^1)$

For step 2, we need to determine the Peter-Weyl decomposition of $L^2_{\mathbb{C}}(S^1)$, where we regard $S^1$ as a subset of $\mathbb{C}$. Let $Y_{l1} : S^1 \to \mathbb{C}$ be given by $Y_{l1}(z) = z^{-l}$. Let $V_{l1} \subseteq L^2_{\mathbb{C}}(S^1)$ just be given by its span: $V_{l1} = \mathrm{span}_{\mathbb{C}}(Y_{l1})$. We want to see that this is a subrepresentation of $L^2_{\mathbb{C}}(S^1)$. To see this, remember that the unitary representation on $L^2_{\mathbb{C}}(X)$ is given by $\lambda : \mathrm{U}(1) \to \mathrm{U}(L^2_{\mathbb{C}}(S^1))$ with $[\lambda(g)\varphi](z) = \varphi(g^{-1}z)$. We have

$$[\lambda(g)Y_{l1}](z) = Y_{l1}(g^{-1}z) = (g^{-1}z)^{-l} = g^l \cdot z^{-l} = (g^l \cdot Y_{l1})(z) \tag{24}$$

and thus $\lambda(g)Y_{l1} = g^l Y_{l1} \in V_{l1}$, which is what we claimed. Since the $V_{l1}$ are 1-dimensional, they are necessarily irreducible for dimension reasons. Now, an important result from Fourier analysis is that the $Y_{l1}$ for $l \in \mathbb{Z}$ actually form an orthonormal basis of $L^2_{\mathbb{C}}(S^1)$ and that, consequently, the Peter-Weyl decomposition of $L^2_{\mathbb{C}}(S^1)$ looks as follows:

$$L^2_{\mathbb{C}}(S^1) = \widehat{\bigoplus_{l \in \mathbb{Z}}} V_{l1}.$$

From this we see that the multiplicities $m_l$ are all given by 1. What is missing is the connection to the irreps $\rho_l : \mathrm{U}(1) \to \mathrm{U}(V_l)$, but we have already indicated this in the notation. Namely, the map $f_l : V_l \to V_{l1}$ given by $z \mapsto z \cdot Y_{l1}$ is clearly an isomorphism of vector spaces, and due to Eq. (24) even an isomorphism of representations:

$$
\begin{aligned}
f_l\big(\rho_l(g)(z)\big) &= f_l\big(g^l \cdot z\big) \\
&= (g^l \cdot z) \cdot Y_{l1} \\
&= z \cdot (g^l \cdot Y_{l1}) \\
&= z \cdot (\lambda(g)(Y_{l1})) \\
&= \lambda(g)\big(z \cdot Y_{l1}\big) \\
&= \lambda(g)\big(f_l(z)\big).
\end{aligned}
$$

Thus, $f_l \circ \rho_l(g) = \lambda(g) \circ f_l$ for all $g \in \mathrm{U}(1)$ and, as claimed, $f_l$ turns out to be an isomorphism. This finishes step 2 of the outline above.

### E.1.3   THE CLEBSCH-GORDAN DECOMPOSITION

For step 3, we proceed as follows: The map

$$
f : V_j \otimes V_l \to V_{j+l}, \ z_j \otimes z_l \mapsto z_j \cdot z_l
$$

is clearly well-defined and linear by the universal property of tensor products, see Definition D.1. Furthermore, it is an isometry: namely, since the scalar product in $\mathbb{C}$ is just the usual multiplication (with the left entry being complex conjugated), we obtain

$$
\begin{aligned}
\big\langle f(z_j \otimes z_l)\big|f(z_j' \otimes z_l')\big\rangle &= \big\langle z_j z_l \big| z_j' z_l' \big\rangle \\
&= \overline{z_j z_l} \cdot z_j' z_l' \\
&= \overline{z_j} z_j' \cdot \overline{z_l} z_l' \\
&= \big\langle z_j \big| z_j' \big\rangle \cdot \big\langle z_l \big| z_l' \big\rangle \\
&= \big\langle z_j \otimes z_l \big| z_j' \otimes z_l' \big\rangle .
\end{aligned}
$$

In the last step, we have used the definition of the scalar product on the tensor product, Definition D.2. Thus, $f$ is an isomorphism of Hilbert spaces. Finally, it also respects the representations since

$$
\begin{aligned}
f\big(\,[(\rho_j \otimes \rho_l)(g)]\,(z_j \otimes z_l)\big) &= f\big(\,[\rho_j(g)]\,(z_j) \otimes [\rho_l(g)]\,(z_l)\big) \\
&= f\big(g^j z_j \otimes g^l z_l\big) \\
&= g^j z_j \cdot g^l z_l \\
&= g^{j+l} \cdot (z_j z_l) \\
&= [\rho_{j+l}(g)]\,(f(z_j \otimes z_l))
\end{aligned}
$$

and thus $f \circ (\rho_j \otimes \rho_l)(g) = \rho_{j+l}(g) \circ f$ for all $g \in \mathrm{U}(1)$. Finally, the basis vectors correspond in the simplest possible way since $f(1 \otimes 1) = 1$.

Overall, what we've shown is the following: $V_J$ is a direct summand of $V_j \otimes V_l$ if and only if $J = j + l$. If this is the case, we have $[J(jl)] = 1$ and can thus omit the index $s$. The only Clebsch-Gordan coefficient is then given by $\langle J1|j1l1 \rangle = 1$ since the basis elements directly correspond.

### E.1.4   ENDOMORPHISMS OF $V_J$

This is the simplest part: Since we are considering representations over $\mathbb{C}$, Schur's Lemma D.8 tells us that $\mathrm{End}_{\mathrm{U}(1),\mathbb{C}}(V_J)$ is 1-dimensional for each irrep $J$, and thus we can ignore the endomorphisms altogether.

### E.1.5   BRINGING EVERYTHING TOGETHER

We now show that a basis of steerable kernels $K : S^1 \to \mathrm{Hom}_{\mathbb{C}}(V_l, V_J)$ of the group $\mathrm{U}(1)$ is given, when expressed as $1 \times 1$-matrix parameterized by $S^1$, by the basis function $Y_{l-J} : S^1 \to \mathbb{C}$. We

remove the index "1" at the basis function to remove clutter. How can we see this result, using Eq. (23)?

Note that $V_J$ can only appear as a direct summand of $V_j \otimes V_l$ if $j = J - l$ by what we've shown above. The "matrix" of Clebsch-Gordan coefficients $\mathrm{CG}_{J((J-l)l)}$ is then just the number 1. We can omit the vacuous indices $i$ and $s$ and obtain that the only basis kernel is given by

$$
\begin{aligned}
K_{J-l}(x) = \langle Y_{J-l} | x \rangle &= \overline{Y_{J-l}(x)} \\
&= \overline{x^{-(J-l)}} \\
&= x^{-(l-J)} \\
&= Y_{l-J}(x).
\end{aligned}
$$

This result is precisely equal to the one obtained in the original paper (Worrall et al., 2016). This concludes our investigations of harmonic networks.

### E.2    SO(2)-STEERABLE KERNELS FOR REAL REPRESENTATIONS

In this section, we look at the case $\mathbb{K} = \mathbb{R}$, $G = \mathrm{SO}(2)$ and $X = S^1$. In the following sections, we again step by step determine the representation-theoretic ingredients that we need for the application of our theorem. Compared to Chapter A, which focuses more on the components themselves and how they relate to the general situation, this section has a stronger focus on actually determining the final kernels, which also involves the task of determining the Clebsch-Gordan coefficients explicitly. We remark that the resulting kernels are not new, since Weiler & Cesa (2019) have solved for this kernel basis already. However, we want to emphasize again that with our method, we learn more about the representation theory of $\mathrm{SO}(2)$ and thus get an overall better conceptual understanding of how the kernels arise.

Since it will help the presentation of our results, we set $\mathrm{SO}(2) = \mathbb{R}/2\pi\mathbb{Z}$, i.e., we view $\mathrm{SO}(2)$ as a group of angles. We also set $S^1 = \mathbb{R}/2\pi\mathbb{Z}$, i.e., we take the interval $[0, 2\pi]$ as the space where our functions are defined. Consequently, since we want our Haar measure to be normalized, we have to put the fraction $\frac{1}{2\pi}$ before all of our integrals, different from what we did in our treatment of $\mathrm{SO}(2)$ over $\mathbb{C}$.

Note that since we now consider representations over the real numbers, unitary representations become *orthogonal* and we write $\mathrm{O}(V)$ instead of $\mathrm{U}(V)$.

#### E.2.1    CONSTRUCTION OF THE IRREDUCIBLE REPRESENTATIONS OF SO(2)

The irreps of $\mathrm{SO}(2)$ over $\mathbb{R}$ are given by $\rho_l : \mathrm{SO}(2) \to \mathrm{O}(V_l)$, $l \in \mathbb{N}_{\geq 0}$. For $l = 0$, we have $V_0 = \mathbb{R}$ and the action is trivial. For $l \geq 1$, $V_l = \mathbb{R}^2$ as a vector space. The action is given by

$$
\big[\rho_l(\phi)\big](v) = \begin{pmatrix} \cos(l\phi) & -\sin(l\phi) \\ \sin(l\phi) & \cos(l\phi) \end{pmatrix} \cdot v
$$

for $\phi \in \mathrm{SO}(2) = \mathbb{R}/2\pi\mathbb{Z}$. The orthonormal basis is in both cases just given by standard basis vectors.

#### E.2.2    THE PETER-WEYL THEOREM FOR $L^2_{\mathbb{R}}(S^1)$

Now we look at square-integrable functions $L^2_{\mathbb{R}}(S^1)$ that we now assume to take *real values*. As before, $\mathrm{SO}(2)$ acts on this space by $(\lambda(\phi)f)(x) = f(x - \phi)$.[26] For notational simplicity, we write $\cos_l$ for the function that maps $x$ to $\cos(lx)$, and analogously for $\sin_l$. One then can show the following, which is a standard result in Fourier analysis:

**Proposition E.1.** *The functions* $\cos_l$, $\sin_l$, $l \geq 1$ *span an irreducible invariant subspace of* $L^2_{\mathbb{R}}(S^1)$ *of dimension* 2*, explicitly given by*

$$
\mathrm{span}_{\mathbb{R}}(\cos_l, \sin_l) = \big\{ \alpha \cos_l + \beta \sin_l \mid \alpha, \beta \in \mathbb{R} \big\}
$$

---

[26]Note that we have a subtraction now instead of a multiplicative inversion. This is because we view our group as additive.

*which is isomorphic as an orthogonal representation to* $V_l$ *by* $\sqrt{2}\cos_l \mapsto \begin{pmatrix} 1 \\ 0 \end{pmatrix}$ *and* $\sqrt{2}\sin_l \mapsto$ $\begin{pmatrix} 0 \\ 1 \end{pmatrix}$.[27] *Furthermore,* $\sin_0 = 0$ *and* $\cos_0 = 1$ *are constant functions and their span is 1-dimensional and equivariantly isomorphic to* $V_0$ *by* $\cos_0 \mapsto 1$.

*Finally, the functions* $\sqrt{2} \cdot \cos_l$, $\sqrt{2} \cdot \sin_l$ *form an orthonormal basis of* $L^2_{\mathbb{R}}(S^1)$, *i.e., every function can be written uniquely as a (possibly infinite) linear combination of these basis functions.*

When setting $V_{l1} = \mathrm{span}_{\mathbb{R}}(\cos_l, \sin_l)$, we thus obtain a decomposition

$$L^2_{\mathbb{R}}(S^1) = \widehat{\bigoplus_{l \geq 0}} V_{l1}.$$

Thus, we have $m_l = 1$ for all $l \in \mathbb{N}$. All in all, we know everything there is to know about the Peter-Weyl theorem in our situation.

### E.2.3 THE CLEBSCH-GORDAN DECOMPOSITION

We now do the explicit decomposition of $V_j \otimes V_l$ into irreps, which will give us the Clebsch-Gordan coefficients that we need. Instead of doing the decomposition in terms of $V_j$ and $V_l$ themselves, in the proofs we actually use the isomorphic images $V_{j1}$ and $V_{l1}$ in $L^2_{\mathbb{R}}(S^1)$. For doing so, we first need some trigonometric formulas in our disposal:

**Lemma E.2.** *The sine and cosine functions fulfill the following rules:*

1. $\cos_{j+l} = \cos_j \cos_l - \sin_j \sin_l$.

2. $\sin_{j+l} = \sin_j \cos_l + \cos_j \sin_l$.

3. $\cos_{j-l} = \cos_j \cos_l + \sin_j \sin_l$.

4. $\sin_{j-l} = \sin_j \cos_l - \cos_j \sin_l$.

*Proof.* The first two are well-known and the last two follow directly from the first two using $\sin_{-j} = -\sin_j$ and $\cos_{-j} = \cos_j$. $\qquad\square$

We will need the following general lemma:

**Lemma E.3.** *Let* $f : V \to V'$ *be an intertwiner between representations* $\rho : G \to \mathrm{GL}(V)$ *and* $\rho' : G \to \mathrm{GL}(V')$. *Then* $\mathrm{null}(f) = \{v \in V \mid f(v) = 0\}$ *is an invariant linear subspace of V.*

*Proof.* This can easily be checked by the reader. $\qquad\square$

As a remark on notation for the following proposition: We write the Clebsch-Gordan coefficients $\mathrm{CG}_{J(jl)s}$ of irreps $V_J$, $V_j$ and $V_l$ with dimensions $d_J$, $d_j$ and $d_l$ as a $d_J \times (d_j \times d_l)$-tensor. That is, it consists of $d_J$ "rows", each of which is a $d_j \times d_l$-matrix. If $V_J$ appears only once in the tensor product, we omit the index $s$ as before.

**Proposition E.4.** *We have the following decomposition results:*

1. *For* $j = l = 0$ *we have* $V_0 \otimes V_0 \cong V_0$ *and Clebsch-Gordan coefficients* $\mathrm{CG}_{0(00)} = ([1])$.

2. *For* $j = 0$, $l > 0$ *we have* $V_0 \otimes V_l \cong V_l$ *and Clebsch-Gordan coefficients* $\mathrm{CG}_{l(0l)} = \left(\begin{bmatrix} 1 & 0 \\ 0 & 1 \end{bmatrix}\right)$.

3. *For* $j > 0$, $l = 0$, *we get* $V_j \otimes V_0 \cong V_j$ *and Clebsch-Gordan coefficients* $\mathrm{CG}_{j(j0)} = \left(\begin{bmatrix} 1 \\ 0 \end{bmatrix} \\ \begin{bmatrix} 0 \\ 1 \end{bmatrix}\right)$.

---

[27]$\sqrt{2}$ acts as a normalization.

4. *For $j > l > 0$ we get $V_j \otimes V_l \cong V_{j-l} \oplus V_{j+l}$. The Clebsch-Gordan coefficients are given*

$$\text{by } \mathrm{CG}_{j-l,(jl)} = \left( \begin{bmatrix} 1 & 0 \\ 0 & 1 \\ 0 & -1 \\ 1 & 0 \end{bmatrix} \right) \text{ and } \mathrm{CG}_{j+l,(jl)} = \left( \begin{bmatrix} 1 & 0 \\ 0 & -1 \\ 0 & 1 \\ 1 & 0 \end{bmatrix} \right).$$

5. *For $l > j > 0$ we get $V_j \otimes V_l \cong V_{l-j} \oplus V_{j+l}$. The Clebsch-Gordan coefficients are given*

$$\text{by } \mathrm{CG}_{(l-j)(jl)} = \left( \begin{bmatrix} 1 & 0 \\ 0 & 1 \\ 0 & 1 \\ -1 & 0 \end{bmatrix} \right) \text{ and } \mathrm{CG}_{j+l,(jl)} = \left( \begin{bmatrix} 1 & 0 \\ 0 & -1 \\ 0 & 1 \\ 1 & 0 \end{bmatrix} \right).$$

6. *For $j = l > 0$, we get an isomorphism $V_l \otimes V_l \cong V_0^2 \oplus V_{2l}$. We obtain the Clebsch-Gordan coefficients* $\mathrm{CG}_{0(ll)1} = \left( \begin{bmatrix} 1 & 0 \\ 0 & 1 \end{bmatrix} \right)$, $\mathrm{CG}_{0(ll)2} = \left( \begin{bmatrix} 0 & \mp 1 \\ \pm 1 & 0 \end{bmatrix} \right)$, *and*

$$\mathrm{CG}_{2l,(ll)} = \left( \begin{bmatrix} 1 & 0 \\ 0 & -1 \\ 0 & 1 \\ 1 & 0 \end{bmatrix} \right), \text{ the last one being the same as the Clebsch-Gordan coefficients}$$

$\mathrm{CG}_{j+l,(jl)}$ *from above. In* $\mathrm{CG}_{0(ll)1}$ *and* $\mathrm{CG}_{0(ll)2}$, *a fourth index is present, namely $1$ and $2$, respectively. This is the index "s" that was missing in all the prior examples, since this is the first time an irrep appears more than once in a tensor product decomposition. Note that for* $\mathrm{CG}_{0(ll)2}$, *we have exactly one positive and one negative entry present, but both are equally valid and mirror the lower halves in* $\mathrm{CG}_{j-l,(jl)}$ *from part $4$ and* $\mathrm{CG}_{l-j,(jl)}$ *from part $5$.*

*Proof.* In the proof, instead of working directly with the irreps $\rho_j : \mathrm{SO}(2) \to \mathrm{O}(V_j)$, we use the isomorphic copies $V_{j1}$ in $L^2_{\mathbb{R}}(S^1)$ given in Proposition E.1. Since we think that it does not help understanding to carry the index "1" in all computations, we omit this index.

The proof of 1, 2, and 3 is clear.

For 4 and 5, consider the (unnormalized) basis $\{\cos_j \otimes \cos_l, \cos_j \otimes \sin_l, \sin_j \otimes \cos_l, \sin_j \otimes \sin_l\}$ of $V_j \otimes V_l$. Our goal is to express these basis elements with respect to basis elements of invariant subspaces. We do this by explicitly constructing an isomorphism to a decomposition of irreps. To that end, let $p : V_j \otimes V_l \to L^2_{\mathbb{R}}(S^1)$ be given by $f \otimes g \mapsto f \cdot g$, which is clearly a well-defined intertwiner. We get as image of $p$ the set

$$\mathrm{im}(p) = \mathrm{span}_{\mathbb{R}} \left( p(\cos_j \otimes \cos_l), \ p(\cos_j \otimes \sin_l), \ p(\sin_j \otimes \cos_l), \ p(\sin_j \otimes \sin_l) \right)$$
$$= \mathrm{span}_{\mathbb{R}} \left( \cos_j \cdot \cos_l, \ \cos_j \cdot \sin_l, \ \sin_j \cdot \cos_l, \ \sin_j \cdot \sin_l \right).$$

From Lemma E.2 we obtain:

$$\begin{aligned} (a) \quad & p(\cos_j \otimes \cos_l) \ - \ p(\sin_j \otimes \sin_l) \ = \ \cos_{j+l}, \\ (b) \quad & p(\cos_j \otimes \sin_l) \ + \ p(\sin_j \otimes \cos_l) \ = \ \sin_{j+l}, \\ (c) \quad & p(\cos_j \otimes \cos_l) \ + \ p(\sin_j \otimes \sin_l) \ = \ \cos_{j-l}, \\ (d) \quad & p(\sin_j \otimes \cos_l) \ - \ p(\cos_j \otimes \sin_l) \ = \ \sin_{j-l}. \end{aligned} \tag{25}$$

Since the right hand sides are linearly independent basis functions of $L^2_{\mathbb{R}}(S^1)$, we obtain:

$$\mathrm{im}(p) = \mathrm{span}_{\mathbb{R}} \left( \cos_{j+l}, \ \sin_{j+l}, \ \cos_{j-l}, \ \sin_{j-l} \right) = V_{|j-l|} \oplus V_{j+l}.$$

Note for the last step that due to symmetry, $\cos_{j-l} = \cos_{l-j}$ and $\sin_{j-l} = -\sin_{l-j}$.

We now specialize to the case of 4, i.e., $j > l > 0$. In this case, $V_{|j-l|} = V_{j-l}$, and the basis is given by $\cos_{j-l}$ and $\sin_{j-l}$, as in the right hand sides of Eq. (25) (c) and (d). Consequently, Eq. (25) is already the expansion of the new basis elements with the old, and the coefficients are consequently the Clebsch-Gordan coefficients.[28] More precisely, if we want to compute, for example, $\mathrm{CG}_{j-l,(jl)}$,

---

[28] Note that for two orthonormal bases in a Hilbert space, when expressing one basis $\{b_i\}$ with respect to another basis $\{c_i\}$, then the expansion coefficients are given by the scalar products $\langle c_j | b_i \rangle$. Since we work over

then we observe from (c) that

$$\cos_{j-l} = \begin{array}{ll} +1 \cdot p(\cos_j \otimes \cos_l) & +0 \cdot p(\cos_j \otimes \sin_l) \\ +0 \cdot p(\sin_j \otimes \cos_l) & +1 \cdot p(\sin_j \otimes \sin_l) \end{array}$$

from which we can already read the upper half of $CG_{j-l,(jl)}$ as the coefficients in this equation (which we conveniently visually arranged in the right way). For the lower half, we do proceed the same for $\sin_{j-l}$, using (d). Then, for $CG_{j+l,(jl)}$, we proceed exactly the same, using parts (a) and (b). That proves 4.

For 5, we have $l > j > 0$. In this case, $V_{|j-l|} = V_{l-j}$, i.e., the basis is given by $\cos_{l-j} = \cos_{j-l}$ and $\sin_{l-j} = -\sin_{j-l}$. The latter means that in part (d) of Eq. (25), we need to replace $\sin_{j-l}$ by $\sin_{l-j}$ and thus change the signs on the left hand side. This change means that $CG_{j+l,(jl)}$ will remain the same as in 4, the upper half of $CG_{l-j,(jl)}$ will remain the same as the upper half of $CG_{j-l,(jl)}$ from part 4 since the cosine in part (c) of Eq. (25) is symmetric, and the lower part will flip the signs. This fully proves 5.

Finally, we prove 6. We have $j = l$ and still consider the same function $p$. Note that $p(\cos_j \otimes \cos_l) + p(\sin_j \otimes \sin_l) = 1$ and $p(\sin_j \otimes \cos_l) - p(\cos_j \otimes \sin_l) = 0$ are constant functions that span the 1-dimensional trivial representation. Thus, we see that $p$ is a surjection

$$p : V_l \otimes V_l \to V_0 \oplus V_{2l}$$

with null space spanned by $\sin_j \otimes \cos_l - \cos_j \otimes \sin_l$. Such a null space is automatically an invariant subspace as well, and since it is one-dimensional, it also must be isomorphic to the trivial representation. Overall, this gives us an isomorphism

$$V_l \otimes V_l \cong V_0^2 \oplus V_{2l}.$$

From this, we can as before read off the Clebsch-Gordan coefficients. The only thing that changes is that parts (c) and (d) of Eq. (25) now correspond to two different copies of $V_0$, which means that the Clebsch-Gordan coefficients $CG_{0(ll)}$ now split up in two parts $CG_{0(ll)1}$ and $CG_{0(ll)2}$. Note that in the trivial representation, the isomorphism that sends the basis vector to its negative is clearly equivariant, which means that both combinations of signs that we give in the final formula for $CG_{0(ll)2}$ are valid. □

### E.2.4 Endomorphisms of $V_J$

We now describe the endomorphisms of the irreducible representations, our last ingredient:

**Proposition E.5.** *We have* $\mathrm{End}_{SO(2),\mathbb{R}}(V_0) \cong \mathbb{R}$*, i.e., multiplications with all real numbers are valid endomorphisms of* $V_0$*. For* $l \geq 1$*, we get*

$$\mathrm{End}_{SO(2),\mathbb{R}}(V_l) = \left\{ \begin{pmatrix} a & -b \\ b & a \end{pmatrix} \Big| a, b \in \mathbb{R} \right\},$$

*which is the set of all scaled rotations of* $\mathbb{R}^2$*. When identifying* $\mathbb{R}^2 \cong \mathbb{C}$*, we can also view these transformations as arbitrary multiplications with a complex number.*

*As a consequence,* $\mathrm{id}_{\mathbb{R}}$ *is a basis for* $\mathrm{End}_{SO(2),\mathbb{R}}(V_0)$ *and* $\left\{ \begin{pmatrix} 1 & 0 \\ 0 & 1 \end{pmatrix}, \begin{pmatrix} 0 & -1 \\ 1 & 0 \end{pmatrix} \right\}$ *a basis for* $\mathrm{End}_{SO(2),\mathbb{R}}(V_l)$ *for* $l \geq 1$*.*

*Proof Sketch.* For $l \geq 1$ and an arbitrary matrix $E = \begin{pmatrix} a & b \\ c & d \end{pmatrix}$ that commutes with all rotation matrices $\rho_l(\phi)$, i.e., $E \circ \rho_l(\phi) = \rho_l(\phi) \circ E$, one can easily show the constraints $a = d$ and $b = -c$, from which the result follows. □

the real numbers, the scalar product is symmetric, and these coefficients are thus also the expansion coefficients when expressing $\{c_i\}$ with $\{b_j\}$. This is why we do not have to rearrange the expressions in Eq. (25), it simply doesn't matter which of the two bases is expanded. Note, however, that our bases are not normalized, and so the Clebsch-Gordan coefficients differ by a constant if the equation is rearranged. This constant does not matter for us since we are only interested in a basis of the space of steerable kernels, and constant multiples of bases are still bases.

### E.2.5 BRINGING EVERYTHING TOGETHER

Now we have done all needed preparation and can solve the kernel constraint explicitly, using the matrix-form of the Wigner-Eckart theorem for steerable kernels, Theorem D.16. This is, as mentioned before, a new derivation of the results in Weiler & Cesa (2019). One can compare with table 8 in their appendix which only differs by (irrelevant) constants.

**Proposition E.6.** *We consider steerable kernels* $K : S^1 \to \mathrm{Hom}_\mathbb{R}(V_l, V_J)$, *where* $V_l$ *and* $V_J$ *are irreducible representations of* $\mathrm{SO}(2)$. *Then the following holds:*

1. *For* $l = J = 0$, *we get* $K(x) = a \cdot (1)$ *for every* $x \in S^1$ *and an arbitrary real number* $a \in \mathbb{R}$ *independent of* $x$.

2. *For* $l = 0$, $J > 0$, *a basis for steerable kernels is given by* $\begin{pmatrix} \cos_J \\ \sin_J \end{pmatrix}$ *and* $\begin{pmatrix} -\sin_J \\ \cos_J \end{pmatrix}$.

3. *For* $l > 0$ *and* $J = 0$, *a basis for steerable kernels is given by* $(\cos_l \quad \sin_l)$, $(\sin_l \quad -\cos_l)$.

4. *For* $l, J > 0$, *a basis for steerable kernels is given by* $\begin{pmatrix} \cos_{J-l} & -\sin_{J-l} \\ \sin_{J-l} & \cos_{J-l} \end{pmatrix}$, $\begin{pmatrix} -\sin_{J-l} & -\cos_{J-l} \\ \cos_{J-l} & -\sin_{J-l} \end{pmatrix}$, $\begin{pmatrix} \cos_{J+l} & \sin_{J+l} \\ \sin_{J+l} & -\cos_{J+l} \end{pmatrix}$, *and* $\begin{pmatrix} -\sin_{J+l} & \cos_{J+l} \\ \cos_{J+l} & \sin_{J+l} \end{pmatrix}$.

*Proof.* The proof of 1 is clear.

For 2, note that $V_J$ can only appear in $V_j \otimes V_0$ if $j = J$. The relevant Clebsch-Gordan coefficients are by Proposition E.4 therefore $\mathrm{CG}_{J(J0)} = \begin{pmatrix} \begin{bmatrix} 1 \\ 0 \\ 0 \\ 1 \end{bmatrix} \end{pmatrix}$. Furthermore, the orthonormal basis of $V_{j1} = V_{J1}$ is given by Proposition E.1 up to constants by $\{\cos_J, \sin_J\}$, which we have to write as a row-vector according to Theorem D.16. Thereby, we can ignore the complex conjugation since we work over the real numbers. Our final ingredient is the endomorphism basis of $V_J$, which is by Proposition E.5 given by $c_1 = \mathrm{id}_{\mathbb{R}^2}$ and $c_2 = \begin{pmatrix} 0 & -1 \\ 1 & 0 \end{pmatrix}$. Overall, the basis kernels are given by

$$ c_i \cdot \begin{pmatrix} [\cos_J \quad \sin_J] \cdot \begin{bmatrix} 1 \\ 0 \end{bmatrix} \\ [\cos_J \quad \sin_J] \cdot \begin{bmatrix} 0 \\ 1 \end{bmatrix} \end{pmatrix} = c_i \cdot \begin{pmatrix} \cos_J \\ \sin_J \end{pmatrix}. $$

The result follows.

For 3, we find $V_0$ only in $V_j \otimes V_l$ if $j = l$, and even twice so. The relevant Clebsch-Gordan coefficients are therefore by Proposition E.4 given by $\mathrm{CG}_{0(ll)1} = \begin{pmatrix} \begin{bmatrix} 1 & 0 \\ 0 & 1 \end{bmatrix} \end{pmatrix}$ and $\mathrm{CG}_{0(ll)2} = \begin{pmatrix} \begin{bmatrix} 0 & -1 \\ 1 & 0 \end{bmatrix} \end{pmatrix}$. The basis-functions in $V_{j1} = V_{l1}$ are by Proposition E.1 up to constants $\{\cos_l, \sin_l\}$, again written as a row-vector. Finally, $V_J = V_0$ has only $\mathrm{id}_\mathbb{R}$ as a basis-endomorphism by Proposition E.5, so this can be ignored altogether by Corollary D.17. We obtain the following basis for steerable kernels:

$$ \left( [\cos_l \quad \sin_l] \begin{bmatrix} 1 & 0 \\ 0 & 1 \end{bmatrix} \right) = (1 \cos_l \quad 1 \sin_l) $$

$$ \left( [\cos_l \quad \sin_l] \begin{bmatrix} 0 & -1 \\ 1 & 0 \end{bmatrix} \right) = (1 \sin_l \quad -1 \cos_l). $$

For 4, we consider only the case $l < J$. By Proposition E.4 we have

$$ V_{J-l} \otimes V_l \cong V_{|2l-J|} \oplus V_J, \quad V_{l+J} \otimes V_l \cong V_J \oplus V_{2l+J}, $$

i.e., $j = J - l$ and $j = l + J$ leads to a tensor product decomposition containing $V_J$, but no other $j$ does. Thus, the relevant Clebsch-Gordan coefficients are by Proposition E.4 the matrices $\mathrm{CG}_{J,(J-l,l)}$ and $\mathrm{CG}_{J,(l+J,l)}$.

We now consider the first case, i.e., $j = J - l$. The Clebsch-Gordan coefficients are $\mathrm{CG}_{J,(J-l,l)} = \mathrm{CG}_{l+j,(jl)} = \left( \begin{bmatrix} 1 & 0 \\ 0 & -1 \\ 0 & 1 \\ 1 & 0 \end{bmatrix} \right)$. The basis functions of $V_{(J-l)1}$ are by Proposition E.1 furthermore given by $\{\cos_{J-l}, \sin_{J-l}\}$. Finally, $V_J$ has again the two basis endomorphisms $c_1 = \mathrm{id}_{\mathbb{R}^2}$ and $c_2$ from above. Thus, we obtain the following basis kernel for $c_1$:

$$c_1 \cdot \left( \begin{bmatrix} [\cos_{J-l} & \sin_{J-l}] \cdot \begin{bmatrix} 1 & 0 \\ 0 & -1 \end{bmatrix} \\ [\cos_{J-l} & \sin_{J-l}] \cdot \begin{bmatrix} 0 & 1 \\ 1 & 0 \end{bmatrix} \end{bmatrix} \right) = \begin{pmatrix} \cos_{J-l} & -\sin_{J-l} \\ \sin_{J-l} & \cos_{J-l} \end{pmatrix}. \tag{26}$$

Consequently, for $c_2$ as the basis endomorphism we need to postcompose with $c_2$ and get:

$$\begin{pmatrix} 0 & -1 \\ 1 & 0 \end{pmatrix} \cdot \begin{pmatrix} \cos_{J-l} & -\sin_{J-l} \\ \sin_{J-l} & \cos_{J-l} \end{pmatrix} = \begin{pmatrix} -\sin_{J-l} & -\cos_{J-l} \\ \cos_{J-l} & -\sin_{J-l} \end{pmatrix}. \tag{27}$$

These are half of the basis kernels. For the other half, we need to look at the case $j = l + J$. The Clebsch-Gordan coefficients are by part 4 of Proposition E.4 given by $\mathrm{CG}_{J,(l+J,l)} = \mathrm{CG}_{j-l,(jl)} = \left( \begin{bmatrix} 1 & 0 \\ 0 & 1 \\ 0 & -1 \\ 1 & 0 \end{bmatrix} \right)$. The basis functions of $V_{(l+J)1}$ are by Proposition E.1 furthermore given by $\{\cos_{J+l}, \sin_{J+l}\}$. For the basis endomorphism $c_1$ we thus get the basis kernel

$$c_1 \cdot \left( \begin{bmatrix} [\cos_{J+l} & \sin_{J+l}] \cdot \begin{bmatrix} 1 & 0 \\ 0 & 1 \end{bmatrix} \\ [\cos_{J+l} & \sin_{J+l}] \cdot \begin{bmatrix} 0 & -1 \\ 1 & 0 \end{bmatrix} \end{bmatrix} \right) = \begin{pmatrix} \cos_{J+l} & \sin_{J+l} \\ \sin_{J+l} & -\cos_{J+l} \end{pmatrix}. \tag{28}$$

Consequently, for $c_2$ as the basis endomorphism we need to postcompose with $c_2$ and get:

$$\begin{pmatrix} 0 & -1 \\ 1 & 0 \end{pmatrix} \cdot \begin{pmatrix} \cos_{J+l} & \sin_{J+l} \\ \sin_{J+l} & -\cos_{J+l} \end{pmatrix} = \begin{pmatrix} -\sin_{J+l} & \cos_{J+l} \\ \cos_{J+l} & \sin_{J+l} \end{pmatrix}. \tag{29}$$

Overall, for the case $l < J$ we have determined all four basis kernels in Eqs. (26), (27), (28), and (29). The cases $l = J$ and $l > J$ can be considered analogously, and in every case the correct Clebsch-Gordan coefficients have to be picked. By using $\cos_{l-J} = \cos_{J-l}$ and $\sin_{l-J} = -\sin_{J-l}$, this will, in the end, always lead to the same final formulas. This result is consistent with Table 8 in Weiler & Cesa (2019). □

### E.3 $\mathbb{Z}_2$-STEERABLE KERNELS FOR REAL REPRESENTATIONS

In this section, we discuss steerable CNNs that use the finite group $\mathbb{Z}_2$, which we identify with $(\{-1, +1\}, \cdot)$, for their symmetries. We let this group act on the plane $\mathbb{R}^2$ by vertical reflections, though other choices are possible as well:

$$x \cdot \begin{pmatrix} a \\ b \end{pmatrix} = \begin{pmatrix} xa \\ b \end{pmatrix}.$$

This example is simple and one may see it as contrived to apply our relatively heavy theory to it. We include it mainly as a demonstration that our results can also be applied to non-smooth finite groups as instances of compact groups. Furthermore, we will fully recover the relationship to the original group convolutional CNNs from Cohen & Welling (2016a) and thereby demonstrate that all the different developed theories are consistent with each other.

### E.3.1 THE IRREDUCIBLE REPRESENTATIONS OF $\mathbb{Z}_2$ OVER THE REAL NUMBERS

Let $\rho : \mathbb{Z}_2 \to \mathrm{GL}(V)$ be an irreducible real representation. Note that

$$\rho(-1) \circ \rho(-1) = \rho((-1) \cdot (-1)) = \rho(1) = \mathrm{id}_V,$$

and thus $\rho(-1)$ is an involution satisfying the equation $\rho(-1)^2 - \mathrm{id}_V = 0$. It is well-known from linear algebra that involutions are diagonalizable, and thus $\rho(-1)$ leaves 1-dimensional subspaces invariant. By irreducibility of $\rho$ this means that $V$ itself needs to be 1-dimensional. Consequently, we can assume $V = \mathbb{R}$ without loss of generality. Note that the computations above mean that we have

$$\big(\rho(-1) - \mathrm{id}_\mathbb{R}\big) \circ \big(\rho(-1) + \mathrm{id}_\mathbb{R}\big) = 0$$

and thus we need to have $\rho(-1) - \mathrm{id}_\mathbb{R} = 0$ or $\rho(-1) + \mathrm{id}_\mathbb{R} = 0$. It follows $\rho(-1) = \mathrm{id}_\mathbb{R}$ or $\rho(-1) = -\mathrm{id}_\mathbb{R}$. Overall, all these investigations mean that we have precisely two irreducible representations of $\mathbb{Z}_2$ up to equivalence. We call them $\rho_+ : \mathbb{Z}_2 \to \mathrm{O}(V_+)$ and $\rho_- : \mathbb{Z}_2 \to \mathrm{O}(V_-)$, where $\rho_+(-1) = \mathrm{id}_\mathbb{R}$ and $\rho_-(-1) = -\mathrm{id}_\mathbb{R}$ and $V_+ = V_- = \mathbb{R}$.

### E.3.2 THE PETER-WEYL THEOREM FOR $L^2_\mathbb{R}(X)$

Here we do the Peter-Weyl decomposition for $L^2_\mathbb{R}(X)$, where $X$ is one of the two homogeneous spaces $X = \{-1, 1\}$ and $X = \{0\}$ with the obvious actions coming from the groups $\mathbb{Z}_2$. This time, we also discuss orbits with only one point since we later want to get a description of kernels on the whole of $\mathbb{R}^2$ for comparisons with group convolutional CNNs.

We start with $X = \{-1, 1\}$. Note that the measure on $X$ is just the normalized counting measure, and thus *all* functions $f : X \to \mathbb{R}$ are square-integrable. We define the two functions

$$f_+ : X \to \mathbb{R}, \ f_+(x) = 1 \text{ for all } x \in X = \{-1, 1\},$$
$$f_- : X \to \mathbb{R}, \ f_-(x) = x \text{ for all } x \in X = \{-1, 1\}.$$

We then define $V_{+1} = \mathrm{span}_\mathbb{R}(f_+)$ and $V_{-1} = \mathrm{span}_\mathbb{R}(f_-)$. This gives a decomposition

$$L^2_\mathbb{R}(X) = V_{+1} \oplus V_{-1}$$

since we have for all $f \in L^2_\mathbb{R}(X)$

$$f = \frac{f(1) + f(-1)}{2} \cdot f_+ + \frac{f(1) - f(-1)}{2} \cdot f_-.$$

Furthermore, the maps $1 \mapsto f_+$ and $1 \mapsto f_-$ give isomorphisms of representations $V_+ \cong V_{+1}$ and $V_- \cong V_{-1}$, respectively.

Now, assume that $X = \{0\}$ with the trivial action coming from $\mathbb{Z}_2$. Then $L^2_\mathbb{R}(X) = V_{+1}$ generated from the function $f_+ : X \to \mathbb{R}$, $f_+(0) = 1$. As before, $1 \mapsto f_+$ gives an isomorphism $V_+ \cong V_{+1}$. This concludes the investigations of the Peter-Weyl theorem.

### E.3.3 THE CLEBSCH-GORDAN DECOMPOSITION

We have the following four isomorphisms of representations:

$$V_+ \otimes V_+ \cong V_+, \quad V_+ \otimes V_- \cong V_-,$$
$$V_- \otimes V_+ \cong V_-, \quad V_- \otimes V_- \cong V_+,$$

each time simply given by $a \otimes b \mapsto ab$. It can easily be checked that these are isomorphisms. In Section E.6.3 the reader can find a proof for similar, sign-dependent isomorphisms for the case that the group is $\mathrm{O}(3)$. For each such isomorphism, there is precisely one Clebsch-Gordan coefficient and it is just given by 1. Thus, as in the case of harmonic networks in Section E.1.5, we can just ignore the Clebsch-Gordan coefficients altogether in the final formulas for our basis kernels.

### E.3.4 ENDOMORPHISMS OF $V_+$ AND $V_-$

Since $V_+$ and $V_-$ are themselves only 1-dimensional, the endomorphism spaces are necessarily 1-dimensional as well and just given by arbitrary $1 \times 1$-matrices, i.e., arbitrary stretchings. As in the example of harmonic networks, we can therefore ignore the endomorphisms as well.

### E.3.5 Bringing Everything Together

Different from the other examples, we will in this section not only engage with the final steerable kernels on homogeneous spaces but also discuss how these assemble to kernels defined on the whole plane $\mathbb{R}^2$. In the end, we will then also discuss how kernels for the regular representation would look like.

But first, we engage with the homogeneous spaces. We start with $X = \{-1, 1\}$ and consider steerable kernels $K : X \to \mathrm{Hom}_{\mathbb{R}}(V_{\mathrm{in}}, V_{\mathrm{out}})$ for irreducible $V_{\mathrm{in}}$ and $V_{\mathrm{out}}$. There are four possibilities for the input and output representations:

**Steerable Kernels $K : X \to \mathrm{Hom}_{\mathbb{R}}(V_+, V_+)$:**

$V_+$ can only be in a tensor product $V \otimes V_+$ if the sign of $V$ is positive as well. Such a space appears precisely once in $L^2_{\mathbb{R}}(X)$ according to Section E.3.2. Since endomorphisms and Clebsch-Gordan coefficients do not appear by what we've shown before, and since complex conjugation doesn't do anything over the real numbers, a basis for steerable kernels is just given by the one kernel $K_+ = f_+$ itself. Here, we identify $\mathrm{Hom}_{\mathbb{R}}(V_+, V_+)$ with $\mathbb{R}$ since it only consists of $1 \times 1$-matrices.

**Steerable Kernels $K : X \to \mathrm{Hom}_{\mathbb{R}}(V_+, V_-)$:**

By the same arguments, a basis is given by the one kernel $K_- = f_-$.

**Steerable Kernels $K : X \to \mathrm{Hom}_{\mathbb{R}}(V_-, V_+)$:**

Again, a basis for steerable kernels is given by $K_- = f_-$.

**Steerable Kernels $K : X \to \mathrm{Hom}_{\mathbb{R}}(V_-, V_-)$:**

A basis is given by $K_+ = f_+$.

Finally, we also need to engage with the case that $X = \{0\}$ consists only of a single point. Similarly to above, in the "even" case that the signs of input- and output representations agree, a basis is given by $K_+ = f_+$ with $f_+(0) = 1$. If, however, the signs do not agree, then only $K = 0$ fulfills the constrained and the basis is empty.

Now, we assemble this to kernels on the whole of $\mathbb{R}^2$. We saw above that we only need to distinguish two cases, namely (a) the case that the signs of input and output representation agree and (b) that they do not.

For case (a), let $K : \mathbb{R}^2 \to \mathbb{R}$ be a steerable kernel, where $\mathbb{R}$ is isomorphic to the $\mathrm{Hom}$-space between equal-sign representations. $\mathbb{R}^2$ splits disjointly into orbits, namely $\left\{ \begin{pmatrix} a \\ b \end{pmatrix}, \begin{pmatrix} -a \\ b \end{pmatrix} \right\}$ for all $a \in \mathbb{R}_{\geq 0}$ and $b \in \mathbb{R}$. If $a = 0$, then the orbit is just a single point, which means that we have a vertical line of single-point orbits. The solution above showed that on each orbit, the kernel needs to be constant (since $f_+$ is constant) and overall this just translates to

$$K \begin{pmatrix} a \\ b \end{pmatrix} = K \begin{pmatrix} -a \\ b \end{pmatrix}$$

for all $a \geq 0$ and $b \in \mathbb{R}$. Consequently, $K$ is just an arbitrary left-right symmetric kernel.

In the case that the input- and output representations do not share their sign, by the same arguments we see that $K : \mathbb{R}^2 \to \mathbb{R}$ is an arbitrary left-right *anti*-symmetric kernel which is zero on the vertical line $\begin{pmatrix} 0 \\ b \end{pmatrix}$ for arbitrary $b \in \mathbb{R}$.

Other than these left-right restrictions, the kernel can be freely learned. Overall, this means that we learn one "half" of the kernel and can recover the other half by the symmetry property derived above.

### E.3.6 GROUP CONVOLUTIONAL CNNS FOR $\mathbb{Z}_2$

We now investigate what all this means if we consider regular representations instead of irreducible representations, thus corresponding to group convolutional kernels as in (Cohen & Welling, 2016a). In this case, we will see an interesting "twist" in the kernel, which makes this example more interesting than one might initially think. The twist emerges as follows: For regular representations, we consider steerable kernels

$$K : \mathbb{R}^2 \to \mathrm{Hom}_{\mathbb{R}}(L^2_{\mathbb{R}}(\mathbb{Z}_2), L^2_{\mathbb{R}}(\mathbb{Z}_2))$$

Now, there are two relatively canonical bases we can choose in the left and the right space. We already know from above that $\{f_+, f_-\}$ is the basis to choose if we want to express steerable kernels corresponding to irreducible representations. However, for vanilla group convolutional CNNs, the basis usually chosen is $\{e_{+1}, e_{-1}\}$ where $e_{+1}(x) = \delta_{+1,x}$ and $e_{-1}(x) = \delta_{-1,x}$. We then obtain the following four base change relations:

$$f_+ = e_{+1} + e_{-1}, \quad f_- = e_{+1} - e_{-1},$$
$$e_{+1} = \frac{1}{2}f_+ + \frac{1}{2}f_-, \quad e_{-1} = \frac{1}{2}f_+ - \frac{1}{2}f_-.$$

Thus, the base change matrices are given by

$$B = \begin{pmatrix} 1 & 1 \\ 1 & -1 \end{pmatrix}, \quad B^{-1} = \begin{pmatrix} \frac{1}{2} & \frac{1}{2} \\ \frac{1}{2} & -\frac{1}{2} \end{pmatrix}.$$

Now, assume that $K : \mathbb{R}^2 \to \mathrm{Hom}_{\mathbb{R}}(L^2_{\mathbb{R}}(\mathbb{Z}_2), L^2_{\mathbb{R}}(\mathbb{Z}_2)) \cong \mathbb{R}^{2\times 2}$ is expressed with respect to the basis $\{f_+, f_-\}$. If we write $K$ as a matrix

$$K = \begin{pmatrix} K_{11} & K_{12} \\ K_{21} & K_{22} \end{pmatrix}$$

then we know that $K_{11}$ and $K_{22}$ map between equal-sign representations and $K_{12}$ and $K_{21}$ between unequal-sign representations. Consequently, from what we've found above, $K_{11}$ and $K_{22}$ are symmetric, whereas $K_{12}$ and $K_{21}$ are antisymmetric. What we now want to figure out is how exactly this translates to a property of the kernel expressed in the basis $\{e_+, e_-\}$.

Thus, let $K'$ be this corresponding kernel. Then using the base change matrices above we obtain

$$\begin{pmatrix} K'_{11} & K'_{12} \\ K'_{21} & K'_{22} \end{pmatrix} = K'$$
$$= B \cdot K \cdot B^{-1}$$
$$= \begin{pmatrix} 1 & 1 \\ 1 & -1 \end{pmatrix} \cdot \begin{pmatrix} K_{11} & K_{12} \\ K_{21} & K_{22} \end{pmatrix} \cdot \begin{pmatrix} \frac{1}{2} & \frac{1}{2} \\ \frac{1}{2} & -\frac{1}{2} \end{pmatrix}$$
$$= \begin{pmatrix} \frac{1}{2}[K_{11} + K_{12} + K_{21} + K_{22}] & \frac{1}{2}[K_{11} - K_{12} + K_{21} - K_{22}] \\ \frac{1}{2}[K_{11} + K_{12} - K_{21} - K_{22}] & \frac{1}{2}[K_{11} - K_{12} - K_{21} + K_{22}] \end{pmatrix}.$$

What symmetry properties does this kernel obey? In order to understand this, we use the following convention: for $y \in \mathbb{R}^2$ we set $-y = \begin{pmatrix} -y_1 \\ y_2 \end{pmatrix}$, i.e., the vertically flipped image of $y$. Then we have, using the symmetry and anti-symmetry of the entries of the original kernel $K$:

$$K'_{22}(-y) = \frac{1}{2}[K_{11}(-y) - K_{12}(-y) - K_{21}(-y) + K_{22}(-y)]$$
$$= \frac{1}{2}[K_{11}(y) + K_{12}(y) + K_{21}(y) + K_{22}(y)]$$
$$= K'_{11}(y),$$
$$K'_{21}(-y) = \frac{1}{2}[K_{11}(-y) + K_{12}(-y) - K_{21}(-y) - K_{22}(-y)]$$
$$= \frac{1}{2}[K_{11}(y) - K_{12}(y) + K_{21}(y) - K_{22}(y)]$$
$$= K'_{12}(y).$$

Thus the second row of $K'$ is basically the same as the first, only that the kernels swap with each other and are internally flipped. This is a special case of the outcome in Cohen & Welling (2016a), which is also described clearly in Weiler et al. (2018b): in group convolutional kernels which are steerable with respect to finite groups, the kernels get copied and applied in all orientations demanded by the group.

What we would still like to understand is if we can also reverse the direction: That is, assume that we start with a group convolutional kernel $K'$ of which we know that $K_{22'}(-y) = K'_{11}(y)$ and $K'_{21}(-y) = K'_{12}(y)$ for all $y \in \mathbb{R}^2$. If we then do a base change, we would like to know if the resulting kernel consists of symmetric and antisymmetric entries. Namely, set

$$
\begin{aligned}
\begin{pmatrix} K_{11} & K_{12} \\ K_{21} & K_{22} \end{pmatrix} &= K \\
&= B^{-1} \cdot K' \cdot B \\
&= \begin{pmatrix} \frac{1}{2} & \frac{1}{2} \\ \frac{1}{2} & -\frac{1}{2} \end{pmatrix} \cdot \begin{pmatrix} K'_{11} & K'_{12} \\ K'_{21} & K'_{22} \end{pmatrix} \cdot \begin{pmatrix} 1 & 1 \\ 1 & -1 \end{pmatrix} \\
&= \begin{pmatrix} \frac{1}{2}\left[K'_{11} + K'_{12} + K'_{21} + K'_{22}\right] & \frac{1}{2}\left[K'_{11} - K'_{12} + K'_{21} - K'_{22}\right] \\ \frac{1}{2}\left[K'_{11} + K'_{12} - K'_{21} - K'_{22}\right] & \frac{1}{2}\left[K'_{11} - K'_{12} - K'_{21} + K'_{22}\right] \end{pmatrix}.
\end{aligned}
$$

The reader can easily check that we can deduce that $K_{11}$ and $K_{22}$ are symmetric and that $K_{12}$ and $K_{21}$ are anti-symmetric. We have thus fully shown the equivalence of the kernel solutions in the setting of steerable CNNs compared to the setting of group convolutional CNNs for the specific group $\mathbb{Z}_2$.

### E.4 SO(3)-Steerable Kernels for Complex Representations.

In the first two sections, we have discussed SO(2)-equivariant kernels (i.e., SE(2)-equivariant neural networks) both over $\mathbb{C}$ and $\mathbb{R}$. The situation over $\mathbb{R}$ was considerably more complicated and required new arguments. In this section, we will discuss SO(3)-equivariant kernels (i.e., SE(3)-equivariant neural networks) for complex representations. In Section E.5 we will then look at the real case, which will essentially give the exact same results, thus differing somewhat from the considerations about SO(2). Different from the earlier sections, we will from now on be less explicit and care more about the general properties of the different functions and coefficients we consider. SO(3)-equivariant networks with real representations have before been implemented in Weiler et al. (2018a) and Thomas et al. (2018), among others.

#### E.4.1 The Irreducible Representations of SO(3) over the Complex Numbers

In this section, we state the complex irreducible representations of SO(3). We will not state the matrices explicitly since the matrix elements are considerably more complicated than in the earlier examples that we saw. For each $l \in \mathbb{N}_{\geq 0}$, there is one irreducible unitary representation

$$
D_l : \mathrm{SO}(3) \to \mathrm{U}(V_l), \text{ where } V_l = \mathbb{C}^{2l+1}.
$$

The matrices $D_l(g)$ for $g \in \mathrm{SO}(3)$ are called the *Wigner D-matrices*.[29] There are, up to equivalence, no other irreducible representations of SO(3) over $\mathbb{C}$. A reference for all this is the original work Wigner (1944).

We note that the indices for the dimensions in $\mathbb{C}^{2l+1}$ are $-l, -l+1, \ldots, l-1, l$ by general convention.

#### E.4.2 The Peter-Weyl Theorem for $L^2_{\mathbb{C}}(S^2)$ as a Representation of SO(3)

Here, we describe how $L^2_{\mathbb{C}}(S^2)$, considered as a unitary representation via $\lambda : \mathrm{SO}(3) \to \mathrm{U}(L^2_{\mathbb{C}}(S^2))$, with $[\lambda(g)\varphi](x) = \varphi(g^{-1}x)$, contains densely a direct sum of irreducible representations. For doing so, we proceed by first describing spherical harmonics without formulas and stating their orthonormality properties, and then stating how they transform under rotation. This will then yield the result. Note that we do not need to describe explicit formulas for the spherical harmonics, which are again somewhat complicated since we are more interested in their properties

---

[29] Here, the letter "D" stands for "Darstellung" which is the German term for "representation".

in relation to Hilbert space theory and representation theory. A reference for all this is MacRobert (1947).

The spherical harmonics are continuous functions $Y_l^n : S^2 \to \mathbb{C}$ for $l \in \mathbb{N}_{\geq 0}$ and $n = -l, \dots, l$. Thus, they are elements of $L_{\mathbb{C}}^2(S^2)$. They have the following properties:

1. $\left\langle Y_l^n \middle| Y_{l'}^{n'} \right\rangle = \delta_{ll'} \delta_{nn'}$ for all $l, l', n, n'$.

2. The linear span of the spherical harmonics is dense in $L_{\mathbb{C}}^2(S^2)$.

3. They transform as follows under rotation: $\lambda(g)(Y_l^n) = \sum_{n'=-l}^{l} D_l^{n'n}(g) Y_l^{n'}$, where $D_l^{n'n}(g)$ are the matrix elements of the Wigner D-matrices defined in Section E.4.1.

Properties 1 and 2 together imply that the spherical harmonics form an orthonormal basis of $L_{\mathbb{C}}^2(S^2)$, see Definition F.40. Let
$$V_{l1} := \mathrm{span}_{\mathbb{C}}(Y_l^n \mid n = -l, \dots, l).$$
Then we already obtain $L_{\mathbb{C}}^2(S^2) = \widehat{\bigoplus}_{l \geq 0} V_{l1}$. Now, let $e^n \in \mathbb{C}^{2l+1}$ be the $n$'th standard basis vector, for $n = -l, \dots, l$. Then property 3 means that the linear map given on basis vectors by
$$f : V_l \to V_{l1}, \ e^n \mapsto Y_l^n$$
is an isomorphism of unitary representations. More precisely, $f$ is clearly a unitary transformation and a linear isomorphism, and it is furthermore equivariant on basis vectors since

$$
\begin{aligned}
f\left(D_l(g)(e^n)\right) &= f\left(\sum_{n'=-l}^{l} D_l^{n'n}(g) e^{n'}\right) \\
&= \sum_{n'=-l}^{l} D_l^{n'n}(g) f(e^{n'}) \\
&= \sum_{n'=-l}^{l} D_l^{n'n}(g) Y_l^{n'} \\
&= \lambda(g)(Y_l^n) \\
&= \lambda(g)(f(e^n)).
\end{aligned}
\tag{30}
$$

General equivariance then follows from equivariance on basis vectors. This concludes this section.

### E.4.3 THE CLEBSCH-GORDAN DECOMPOSITION

Explicit formulas for the Clebsch-Gordan coefficients of $\mathrm{SO}(3)$ are given in Bohm & Löwe (1993). The most important fact is the following: There is a decomposition

$$V_j \otimes V_l \cong \bigoplus_{J=|l-j|}^{l+j} V_J$$

of representations. Furthermore, the Clebsch-Gordan coefficients $\langle JM|jm; ln \rangle$ are all real numbers, a fact that we will use in Section E.5.

### E.4.4 ENDOMORPHISMS OF $V_J$

As in the case of harmonic networks, this is again simple: we are considering representations over $\mathbb{C}$, and so Schur's Lemma D.8 tells us that $\mathrm{End}_{\mathrm{SO}(3)}(V_J)$ is 1-dimensional for each irrep $J$. We can therefore ignore the endomorphisms once again.

### E.4.5 BRINGING EVERYTHING TOGETHER

Now, with all this prior work, let us determine the equivariant kernels $K : S^2 \to \mathrm{Hom}_{\mathbb{C}}(V_l, V_J)$ for the irreducible representations $D_l : \mathrm{SO}(3) \to \mathrm{U}(V_l)$ and $D_J : \mathrm{SO}(3) \to \mathrm{U}(V_J)$. For this, we use Eq. (23). Since each $V_j$ appears only once in the direct sum decomposition of $L_{\mathbb{C}}^2(S^2)$ according to Section E.4.2 and since $V_J$ can only appear once in the direct sum decomposition of a tensor product $V_j \otimes V_l$ according to Section E.4.3 , we do not need the indices $i$ and $s$. Furthermore, as

mentioned in the last section, the endomorphisms are trivial, which is why we also do not need the index $r$. Overall, we see that we simply have basis kernels $K_j : S^2 \to \mathrm{Hom}_{\mathbb{C}}(V_l, V_J)$ for all $j$ with $|l - J| \leq j \leq l + J$.[30] They are explicitly given by

$$
K_j(x) = \begin{pmatrix} \langle j|x \rangle \cdot \mathrm{CG}^1_{J(jl)} \\ \vdots \\ \langle j|x \rangle \cdot \mathrm{CG}^{d_J}_{J(jl)} \end{pmatrix}
$$

for all $x \in S^2$. Remembering that $\langle jm|x \rangle = \overline{Y_j^m(x)}$, the individual matrix elements of $K_j(x)$ are then given by

$$
\langle JM|K_j(x)|ln \rangle = \sum_{m=-j}^{j} \langle JM|jm; ln \rangle \cdot \overline{Y_j^m}(x).
$$

This ends the discussion.

### E.5   SO(3)-Steerable Kernels for Real Representations

In this section, we want to argue why the results in the last section transfer over to the real case as well. Most of the investigations in this section are probably well-known. However, we were not able to find sources that explicitly explain the representation theory of $SO(3)$ over the real numbers, and so we develop lots of it here from scratch. We thereby make use of the theory over $\mathbb{C}$, some results about real spherical harmonics, and the general theory of real and quaternionic representations outlined in Bröcker & Dieck (2003). We need to somewhat turn the order around in this section in order to develop the results. Therefore we first investigate the Peter-Weyl theorem, then look at the endomorphism spaces of the appearing irreducible representations and afterward, as a consequence, show that the representations appearing in the decomposition of $L^2_{\mathbb{R}}(S^2)$ are already exhaustive.

### E.5.1   The Peter-Weyl Theorem for $L^2_{\mathbb{R}}(S^2)$ as a Representation of $SO(3)$

The most important finding is the following, which is taken from Gallier & Quaintance (2020): One can do a base change for the spherical harmonics as follows to obtain real versions of them. Namely, let

$$
{}^rY_l^n = \begin{cases} \dfrac{i}{\sqrt{2}} \left( Y_l^n - (-1)^n Y_l^{-n} \right) & \text{if } n < 0, \\ Y_l^0 & \text{if } n = 0, \\ \dfrac{1}{\sqrt{2}} \left( Y_l^{-n} + (-1)^n Y_l^n \right) & \text{if } n > 0. \end{cases} \tag{31}
$$

One can then show that these functions are real-valued continuous functions and therefore ${}^rY_l^n \in L^2_{\mathbb{R}}(S^2)$. Furthermore, they are an orthonormal basis of this space. We can then, as before, set ${}^rV_{l1}$ as the span of the ${}^rY_l^n \in L^2_{\mathbb{R}}(S^2)$ and obtain a decomposition

$$
L^2_{\mathbb{R}}(S^2) = \widehat{\bigoplus}_{l \geq 0} {}^rV_{l1}.
$$

We need to understand the transformation properties of these real-valued spherical harmonics under rotation. To understand this explicitly, we set $B_l \in \mathbb{C}^{(2l+1) \times (2l+1)}$ as the (complex) base change matrix between the complex and real spherical harmonics. Its entries are given according to Eq. (31) such that the following relation holds for all $n = -l, \ldots, l$:

$$
{}^rY_l^n = \sum_{n'=-l}^{l} B_l^{n'n} \cdot Y_l^{n'}.
$$

Since for a given $l$, both the complex and real spherical harmonics are linearly independent, the matrix $B_l$ is invertible. Let $B_l^{-1}$ be its inverse. Then it is generally known from linear algebra that

---

[30]We saw that $V_J$ is a direct summand of $V_j \otimes V_l$ if and only if $|l - j| \leq J \leq l + j$. By doing case distinctions, one can show that this is the case if and only if $|l - J| \leq j \leq l + J$.

we also obtain the inverse relation:

$$Y_l^n = \sum_{n'=-l}^{l} (B_l^{-1})^{n'n} \cdot {}^r Y_l^{n'}.$$

Using both these relations and the rotation properties of the complex spherical harmonics from Section E.4.2 we obtain the following rotation property for the real spherical harmonics:

$$\lambda(g)({}^r Y_l^n) = \sum_{n_1=-l}^{l} B_l^{n_1 n} \cdot \lambda(g)(Y_l^{n_1})$$

$$= \sum_{n_1=-l}^{l} B_l^{n_1 n} \sum_{n_2=-l}^{l} D_l^{n_2 n_1}(g) \cdot Y_l^{n_2}$$

$$= \sum_{n_1=-l}^{l} B_l^{n_1 n} \sum_{n_2=-l}^{l} D_l^{n_2 n_1}(g) \cdot \sum_{n'=-l}^{l} (B_l^{-1})^{n' n_2} \cdot {}^r Y_l^{n'}$$

$$= \sum_{n'=-l}^{l} \left( \sum_{n_1=-l}^{l} \sum_{n_2=-l}^{l} (B_l^{-1})^{n' n_2} \cdot D_l^{n_2 n_1}(g) \cdot B_l^{n_1 n} \right) {}^r Y_l^{n'}$$

$$= \sum_{n'=-l}^{l} \left( B_l^{-1} \cdot D_l(g) \cdot B_l \right)^{n' n} \cdot {}^r Y_l^{n'}.$$

Now if we set ${}^r D_l(g) := B_l^{-1} \cdot D_l(g) \cdot B_l$, then we obtain the transformation property

$$\lambda(g)({}^r Y_l^n) = \sum_{n'=-l}^{l} {}^r D_l(g)^{n' n} \cdot {}^r Y_l^{n'} \qquad (32)$$

which is analogous to the one in Section E.4.2.

**Lemma E.7.** ${}^r D_l(g)^{n' n} \in \mathbb{R}$ *for all* $l \geq 0$, $n', n = -l, \ldots, l$ *and* $g \in \mathrm{SO}(3)$.

*Proof.* Note that since ${}^r Y_l^n$ is a real-valued function, the rotation $\lambda(g)({}^r Y_l^n)$ is real-valued as well. Thus, it is in the space $L_{\mathbb{R}}^2(S^2)$. The real spherical harmonics are a basis of this space, which means that the coefficients when expanding $\lambda(g)({}^r Y_l^n)$ in this basis are necessarily real as well. These coefficients are precisely given by the ${}^r D_l(g)^{n' n}$ according to Eq. (32). $\qquad \square$

Now, we have the choice to view ${}^r D_l$ as either a real or a complex representation, but first we take the complex viewpoint and see it as a function ${}^r D_l : \mathrm{SO}(3) \to \mathrm{GL}(\mathbb{C}^{2l+1})$. Notationwise, the following is important: the "$r$" in ${}^r D_l$ indicates that the elements in this matrix are real but does not tell us on which space it acts. This will always be clarified by the context. We have the following:

**Lemma E.8.** ${}^r D_l : \mathrm{SO}(3) \to \mathrm{U}(\mathbb{C}^{2l+1})$ *is an irreducible unitary representation and isomorphic to* $D_l$.

*Proof.* First of all, it is an actual linear representation since

$${}^r D_l(gg') = B_l^{-1} D_l(gg') B_l = B_l^{-1} D_l(g) B_l B_l^{-1} D_l(g') B_l = {}^r D_l(g) \cdot {}^r D_l(g')$$

where we used that $D_l$ is a linear representation. Now since $Y_l^n$ and ${}^r Y_l^n$ are both orthonormal bases of $L_{\mathbb{C}}^2(S^2)$, the base change matrix $B_l$ needs to be a unitary matrix. Consequently, ${}^r D_l(g) = B_l^{-1} D_l(g) B_l$ is as a product of unitary transformations itself unitary, which means that ${}^r D_l$ is a unitary representation. Furthermore, we obtain $B_l \cdot {}^r D_l(g) = D_l(g) \cdot B_l$, which means that $B_l$ gives an isomorphism ${}^r D_l \cong D_l$ of unitary representations. From the fact that $D_l$ is irreducible, we obtain that ${}^r D_l$ is irreducible as well. $\qquad \square$

Now we take the real viewpoint. Let ${}^r V_l = \mathbb{R}^{2l+1}$.

**Lemma E.9.** ${}^r D_l : \mathrm{SO}(3) \to \mathrm{O}({}^r V_l)$ *is an irreducible orthogonal representation.*

*Proof.* ${}^r D_l(g)$ is a unitary matrix for each $g \in \mathrm{SO}(3)$ by Lemma E.8, and since its matrix elements are real by Lemma E.7, it automatically is an orthogonal matrix. If it was reducible, then there would be a *real* base change matrix that brings ${}^r D_l$ in a nontrivial block-diagonal shape. However, this base change would in particular be complex, meaning that we would conclude that the complex version of the representation ${}^r D_l$ is reducible. But it is not, due to Lemma E.8. $\qquad\square$

Now, remember that $L^2_{\mathbb{R}}(S^2) = \widehat{\bigoplus}_{l \geq 0} {}^r V_{l1}$ and that ${}^r V_{l1}$ is generated from the real spherical harmonics. Also, remember that the real spherical harmonics transform as in Eq. (32). Thus, with the same arguments as in Eq. (30) we obtain ${}^r V_{l1} \cong {}^r V_l$, which is from the preceding lemmas an irreducible orthogonal representation. Thus, we have found the Peter-Weyl decomposition of $L^2_{\mathbb{R}}(S^2)$.

### E.5.2    Endomorphisms of ${}^r V_J$

In the next section, we will show that the ${}^r D_J : \mathrm{SO}(3) \to \mathrm{O}({}^r V_J)$ already given an exhaustive list of the irreducible representations of $\mathrm{SO}(3)$ over the real numbers. In this section, we first describe their endomorphism spaces since this will help in showing that there cannot be any other irreducible representations. Fortunately, the situation is again simple:

**Proposition E.10.** $\mathrm{End}_{\mathrm{SO}(3),\mathbb{R}}({}^r V_J)$ *is one-dimensional for each $J \geq 0$.*

*Proof.* Let $f : {}^r V_J \to {}^r V_J$ be an endomorphism. Since ${}^r V_J = \mathbb{R}^{2J+1}$ we can view $f$ as a matrix in $\mathbb{R}^{(2J+1) \times (2J+1)}$. That $f$ is an endomorphism then means

$$f \cdot {}^r D_J(g) = {}^r D_J(g) \cdot f$$

for all $g \in \mathrm{SO}(3)$. Now note that as a real matrix, f is in particular a complex matrix, i.e., $f \in \mathbb{C}^{(2J+1) \times (2J+1)}$. Also, remember that we can view ${}^r D_J$ also as a complex irreducible representation ${}^r D_J : \mathrm{SO}(3) \to \mathrm{U}(\mathbb{C}^{2J+1})$ by Lemma E.8. What this means is that $f \in \mathrm{End}_{\mathrm{SO}(3),\mathbb{C}}(\mathbb{C}^{2J+1})$, which is isomorphic to $\mathbb{C}$ by Schur's Lemma D.8. Thus, $f$ is a complex multiple of the identity. Since $f$ is a real matrix, it is thus a real multiple of the identity. The result follows. $\qquad\square$

### E.5.3    General Notes on the Relation between Real and Complex Representations

In the next section we show that there can, up to isomorphism, not be *other* irreducible representations than the ${}^r D_l : \mathrm{SO}(3) \to \mathrm{O}({}^r V_l)$. In order to do so, we first need to better understand the relationship between real and complex representations of compact groups. These investigations will carry over to the investigations for $\mathrm{O}(3)$ that we do in Section E.6 as well.

The following definition of a classification of real irreducible representations of a compact group $G$ can be found in Bröcker & Dieck (2003), Theorem II.6.7. In this book, it is a theorem, since the authors give an independent but equivalent definition of these notions.

**Definition E.11** (Real, Complex, and Quaternionic Type Irreducible Representations)**.** Let $\rho : G \to \mathrm{O}(V)$ be a *real* irreducible representation of a compact group $G$. Then $\rho$ is said to be of

1. *real type* if $\mathrm{End}_{G,\mathbb{R}}(V) \cong \mathbb{R}$,

2. *complex type* if $\mathrm{End}_{G,\mathbb{R}}(V) \cong \mathbb{C}$ and

3. *quaternionic type* if $\mathrm{End}_{G,\mathbb{R}}(V) \cong \mathbb{H}$, where $\mathbb{H}$ are the quaternions.

Here, these isomorphisms respect both addition and multiplication. The multiplication in the endomorphism spaces is thereby given by composition of functions.

Furthermore, Bröcker & Dieck (2003) shows in Theorem II.6.3 that there is no other possibility for an irreducible real representation, i.e., they can be completely categorized by being of real, complex or quaternionic type. Additionally, since $\mathbb{R}$, $\mathbb{C}$ and $\mathbb{H}$ already differ in their $\mathbb{R}$-dimension, it is enough to check whether the $\mathbb{R}$-dimension of an endomorphism space is $1$, $2$ or $4$ in order to do the classification.

In order to compare real and complex representations we need to define two functors between those:[31]

**Definition E.12** (Restriction and Extension). Let $^c\rho : G \to \mathrm{GL}(^cV)$ be a complex representation. Furthermore, let $^r\rho : G \to \mathrm{GL}(^rV)$ be a real representation. Then we define their *restriction* and *extension* as follows:

1. Set $r(^cV)$ as the $\mathbb{R}$-vector space that has the same underlying abelian group as $^cV$ and the scalar multiplication from $\mathbb{R}$ which is the restriction of the multiplication from $\mathbb{C}$. The *restriction* $r(^c\rho) : G \to \mathrm{GL}(r(^cV))$ is defined as the exact same map as $^c\rho$, only that $r(^c\rho)(g) : r(^cV) \to r(^cV)$ is now viewed as an automorphism of *real* vector spaces.

2. We define the *extension* by $e(^rV) := \mathbb{C} \otimes_{\mathbb{R}} {}^rV$, where $\mathbb{C}$ is regarded as an $\mathbb{R}$-vector space. This construction becomes a $\mathbb{C}$-vector space by scalar multiplication $z \cdot (z' \otimes v) := (zz') \otimes v$. We can then define $e(^r\rho) : G \to \mathrm{GL}(e(^rV))$ by setting $e(^r\rho)(g) := \mathrm{id}_{\mathbb{C}} \otimes (^r\rho(g))$.

Note that the extension operation doubles the $\mathbb{R}$-dimension, whereas for the restriction it stays equal. Therefore, we can not hope that these operations are inverse to each other. However, we have the following, almost as nice statement:

**Proposition E.13.** *For each real representation $\rho : G \to \mathrm{GL}(V)$ there is a natural isomorphism $r(e(V)) \cong V \oplus V$ of $\mathbb{R}$-representations.*

*Proof.* This is the first statement in Bröcker & Dieck (2003), Proposition II.6.1. $\square$

The following definition is actually not the definition that Bröcker & Dieck (2003) formulate. However, it is an equivalent characterization that follows from their Proposition II.6.6 (vii), (viii) and (ix) and is more convenient for our needs:

**Definition E.14** (Real Type *Complex* Representation). Let $\rho : G \to \mathrm{GL}(V)$ be a *complex* irreducible representation. Then $\rho$ is called *of real type* if there is an isomorphism of real representations $r(V) \cong U \oplus U$ where

1. $\rho_U : G \to \mathrm{GL}(U)$ is an irreducible real representation and

2. $r(\rho) : G \to \mathrm{GL}(r(V))$ is the restriction of $\rho$, as defined in Definition E.12.

**Proposition E.15.** *Assume $G$ is a compact group such that all complex irreducible representations are of real type. Then also all real irreducible representations are of real type.*

*Proof.* This follows from Bröcker & Dieck (2003), Proposition II.6.6 (ii) and (iii). $\square$

**Proposition E.16.** *Let $\rho : G \to \mathrm{GL}(V)$ be an irreducible real representation of real type. Then its extension $e(\rho) : G \to \mathrm{GL}(e(V))$ given as in Definition E.12 is an irreducible complex representation (also of real type).*

*Proof.* This is precisely Bröcker & Dieck (2003), Proposition II.6.6(i). $\square$

### E.5.4 THE IRREDUCIBLE REPRESENTATIONS OF $\mathrm{SO}(3)$ OVER THE REAL NUMBERS

The rough strategy is to use the fact that the $^rD_l$, viewed as complex irreducible representations, are an exhaustive list of all the complex irreps. Then, using the restriction and extension operators $r$ and $e$ between real and complex representations, we can show that in the specific case of $\mathrm{SO}(3)$, there can not be any other real irreducible representations than the $^rD_l$, viewed as *real* representations.

**Lemma E.17.** *All complex irreducible representations of $\mathrm{SO}(3)$ are of real type.*

---

[31]We only define these functors on objects and not on morphisms. The reason is that we will never explicitly use their definitions on morphisms. More details on this can be found in Bröcker & Dieck (2003), including other functors which are needed in the general theory. The reader should not worry if he or she does not know what a functor is.

*Proof.* From Section E.4.1 and Lemma E.8 we know that the $^rD_l : \mathrm{SO}(3) \to \mathrm{U}(\mathbb{C}^{2l+1})$ give us, up to equivalence, all the complex irreducible representations of $\mathrm{SO}(3)$. According to Definition E.14 we now need to understand that its restriction splits into the direct sum of twice the same irreducible *real* representation. We do this as follows:

We can write $r(\mathbb{C}^{2l+1}) = \mathbb{R}^{2l+1} \oplus (i\mathbb{R})^{2l+1} = {}^rV_l \oplus i{}^rV_l$, which is a decomposition of $\mathbb{C}^{2l+1}$ when viewed as an $\mathbb{R}$-vector space. Then, we can note that both

$$^rD_l : \mathrm{SO}(3) \to \mathrm{O}(^rV_l) \text{ and}$$
$$^rD_l : \mathrm{SO}(3) \to \mathrm{O}(i^rV_l)$$

are well-defined $\mathbb{R}$-representations, which follows from the fact that the matrix elements are all real. Furthermore, the first map is actually an irreducible real representation by Lemma E.9. The second one is isomorphic to the first since one can show that

$$i : {}^rV_l \to i^rV_l, \ a \mapsto i \cdot a$$

is an isomorphism of real $\mathrm{SO}(3)$-representations. This gives us precisely the splitting of $r(\mathbb{C}^{2l+1})$ as a representation that we were looking for. $\qquad \square$

**Corollary E.18.** *All irreducible real representations of* $\mathrm{SO}(3)$ *are of real type.*

*Proof.* This follows directly from Lemma E.17 and Proposition E.15. $\qquad \square$

**Proposition E.19.** *The* $^rD_l : \mathrm{SO}(3) \to \mathrm{O}(^rV_l)$ *are, up to equivalence, all real irreducible representations of* $\mathrm{SO}(3)$.

*Proof.* Assume that $\rho : \mathrm{SO}(3) \to \mathrm{GL}(V)$ is an irreducible real representation of $\mathrm{SO}(3)$. It is of real type by Corollary E.18. By Proposition E.16, the extension $e(\rho) : G \to \mathrm{GL}(e(V))$ is an irreducible complex representation. Since the $^rD_l$ give us all complex irreducible representations up to equivalence by Section E.4.1 and Lemma E.8, there is an equivalence of complex $\mathrm{SO}(3)$-representations $e(V) \cong \mathbb{C}^{2l+1}$ for some $l$. Since functors respect isomorphisms (and equivalences are isomorphisms in the categories of $G$-representations) and the restriction operation is a functor,[32] and using Proposition E.13 as well as the proof of Lemma E.17 we obtain:

$$V \oplus V \cong r(e(V)) \cong r(\mathbb{C}^{2l+1}) \cong {}^rV_l \oplus i^rV_l = {}^rV_l \oplus {}^rV_l.$$

Using the Krull-Remak-Schmidt Theorem B.39, we see that there is an isomorphism of $\mathrm{SO}(3)$-representations $V \cong {}^rV_l$. This finishes the proof. $\qquad \square$

### E.5.5 THE CLEBSCH-GORDAN DECOMPOSITION

We are almost there. The only thing left to understand is the Clebsch-Gordan decomposition. Remember the following from Section E.4.3: For the *complex* irreducible representations there are decompositions

$$V_j \otimes V_l \cong \bigoplus_{J=|l-j|}^{l+j} V_J$$

where on each space, the representations $D_j$, $D_l$ and $D_J$ are given by the Wigner D-matrices. Furthermore, the Clebsch-Gordan coefficients are all real. Now, we know that $^rD_l$ is, as a complex representation, isomorphic to $D_l$ by Lemma E.8, and such a representation then acts on $\mathbb{C}^{2l+1}$ as well. Consequently, we also get the decomposition

$$\mathbb{C}^{2j+1} \otimes \mathbb{C}^{2l+1} \cong \bigoplus_{J=|l-j|}^{l+j} \mathbb{C}^{2J+1}$$

of the complex representations $^rD_j$ and $^rD_l$. Obviously, the Clebsch-Gordan coefficients can be chosen to be exactly the same as before, and thus they are again real.

---

[32]The reader does not need to know what a functor is if he or she believes these statements.

Let the above isomorphism be called $f$. Now, we can view all involved vector spaces as $\mathbb{R}$-vector spaces as well. Furthermore, we have subspaces ${}^r V_j = \mathbb{R}^{2j+1}$, ${}^r V_l = \mathbb{R}^{2l+1}$ and ${}^r V_J = \mathbb{R}^{2J+1}$ which are also invariant under the representations ${}^r D_j$, ${}^r D_l$ and ${}^r D_J$. Consequently, we can just restrict the isomorphism above to a map

$$f| : {}^r V_j \otimes {}^r V_l \to \bigoplus_{J=|l-j|}^{l+j} {}^r V_J.$$

which is well-defined since the Clebsch-Gordan coefficients are real. It needs to be injective, since it is a restriction of an isomorphism. For dimension reasons, the restriction then needs to be an isomorphism, and obviously, it has the exact same Clebsch-Gordan coefficients as the original map $f$.[33]

### E.5.6 BRINGING EVERYTHING TOGETHER

By what we've shown in the last sections, we see that the situation is basically the same as in Section E.4.5. The only thing that changes is that we now use the *real* spherical harmonics, and therefore the complex conjugation disappears. What this overall means is the following: let ${}^r D_l : \mathrm{SO}(3) \to \mathrm{O}({}^r V_l)$ and ${}^r D_J : \mathrm{SO}(3) \to \mathrm{O}({}^r V_J)$ be the representations determining the input and output fields. Then a basis for steerable kernels $K : S^2 \to \mathrm{Hom}_{\mathbb{R}}({}^r V_l, {}^r V_J)$ is given by kernels $K_j : S^2 \to \mathrm{Hom}_{\mathbb{R}}({}^r V_l, {}^r V_J)$ for all $|l - J| \le j \le l + J$. The matrix elements are given by

$$\langle JM|K_j(x)|ln\rangle = \sum_{m=-j}^{j} \langle JM|jm;ln\rangle \cdot {}^r Y_j^m(x). \tag{33}$$

### E.6 O(3)-STEERABLE KERNELS FOR COMPLEX REPRESENTATIONS

In this section, we deal with O(3)-equivariant kernels for complex representations and then, in the next section, will transport the results over to real representations. In the earlier examples, we saw that the Peter-Weyl decomposition of $L^2_{\mathbb{K}}(X)$ always contained each irreducible representation of the symmetry group exactly once. The example of O(3) is the first in which this is not the case: *parity* will play a role in determining which irreducible representations make their way in the space of square-integrable functions and which do not. Overall, we hope that the example of O(3) is a sufficient justification for our use of the multiplicities $m_j$ of irreducible representations that we considered in all our theorems. O(3)-equivariant networks are to the best of our knowledge not described in any published work yet.

### E.6.1 THE IRREDUCIBLE REPRESENTATIONS OF O(3)

The most important observation is the following, after which we can deduce the irreducible representations of O(3) from those of SO(3):

**Lemma E.20.** *Let* $\mathbb{Z}_2 := (\{-1, +1\}, \cdot)$ *be the group with two elements. Then the map*

$$\cdot : \mathbb{Z}_2 \times \mathrm{SO}(3) \to \mathrm{O}(3), \quad (s, g) \mapsto sg$$

*is an isomorphism of groups.*

*Proof.* It is a group homomorphism since $s \in \{-1, +1\}$ can be represented by a multiple of the identity matrix, and as such it commutes with every matrix $g$. That $\cdot$ is an isomorphism follows since all matrices in O(3) either have determinant $1$ or $-1$. The matrices with determinant $1$ form SO(3) and are the image of $\{+1\} \times \mathrm{SO}(3)$. The matrices with determinant $-1$ are the image of $\{-1\} \times \mathrm{SO}(3)$. $\square$

Note the fact that for $g \in \mathrm{SO}(3)$, $-g$ has determinant $-1$, which we used in the proof. This does only hold for $g \in \mathrm{SO}(d)$ with $d$ being *odd*. Therefore, the above lemma is not true for $d$ even. In the even case, we obtain a *semidirect* product and the story complicates somewhat.

---

[33]The reason for this is that the standard basis vectors in $\mathbb{C}^k$ which are used for the Clebsch-Gordan coefficients are exactly the standard basis vectors in $\mathbb{R}^k \subseteq \mathbb{C}^k$ by definition of this embedding.

Earlier, we already considered tensor product representations of one and the same group. A related notion is that of tensor product representations of two different groups:[34]

**Definition E.21** (Tensor Product Representation)**.** Let $G$ and $H$ be two compact groups. Let $\rho_G : G \to \mathrm{GL}(V_G)$ and $\rho_H : H \to \mathrm{GL}(V_H)$ be representations of the two groups $G$ and $H$. Then the *tensor product representation* is given by

$$\rho_G \otimes \rho_H : G \times H \to \mathrm{GL}(V_G \otimes V_H),$$
$$\big[(\rho_G \otimes \rho_H)(g,h)\big](v_G \otimes v_H) := \rho_G(g)(v_G) \otimes \rho_H(h)(v_H).$$

This is again a linear representation.

**Proposition E.22.** *Representatives of isomorphism classes of irreducible representations of $G \times H$ are given precisely by all the $\rho_G \otimes \rho_H$, where $\rho_G$ and $\rho_H$ run through representatives of isomorphism classes of irreducible representations of $G$ and $H$, respectively.*

*Proof.* This is proven in chapter II, Proposition $4.14$ and $4.15$ of Bröcker & Dieck (2003). □

It is important to note that the proof of the above proposition uses the property of the complex numbers to be algebraically closed in crucial steps, and therefore it is unclear how exactly a generalization to representations over the real numbers looks like. Therefore, we will not use the above proposition in our later considerations for real representations of $\mathrm{O}(3)$.

However, in our current situation, we can apply it without problems. This proposition, together with Lemma E.20, suggests that we should understand the irreducible representations of $\mathbb{Z}_2$. We already saw this for real representations before and essentially obtain the same result:

**Lemma E.23.** *The irreducible representations of $\mathbb{Z}_2$ are up to equivalence precisely the following two, which we state for simplicity only on the generator:*

$$\rho_+ : \mathbb{Z}_2 \to \mathrm{GL}(\mathbb{C}), \quad \rho_+(-1) = \mathrm{id}_\mathbb{C}$$
$$\rho_- : \mathbb{Z}_2 \to \mathrm{GL}(\mathbb{C}), \quad \rho_-(-1) = -\mathrm{id}_\mathbb{C}.$$

*Proof.* This can be shown in exactly the same way as in Section E.3.1. □

Thus we are ready to state our result about the irreducible representations of $\mathrm{O}(3)$:

**Proposition E.24.** *The irreducible representations of $\mathrm{O}(3)$ are up to equivalence given as follows: for each $l \in \mathbb{N}_{\geq 0}$ there are precisely two representations $D_{l+} : \mathrm{O}(3) \to \mathrm{U}(V_{l+})$ and $D_{l-} : \mathrm{O}(3) \to \mathrm{U}(V_{l-})$ with $V_{l+} = \mathbb{C}^{2l+1} = V_{l-}$, given as follows:*

$$D_{l+}(sg) = D_l(g) \quad \text{for all } s \in \mathbb{Z}_2, \ g \in \mathrm{SO}(3).$$
$$D_{l-}(sg) = sD_l(g) \quad \text{for all } s \in \mathbb{Z}_2, \ g \in \mathrm{SO}(3).$$

*Proof.* Remember from Section E.4.1 that the irreducible representations of $\mathrm{SO}(3)$ are given by the Wigner D-matrices $D_l$. From Lemma E.23 we know that the irreducible representations of $\mathbb{Z}_2$ are given by $\rho_+$ and $\rho_-$. From the isomorphism $\mathrm{O}(3) \cong \mathbb{Z}_2 \times \mathrm{SO}(3)$ from Lemma E.20 and from Proposition E.22 we thus obtain that the irreducible representations of $\mathrm{O}(3)$ are precisely given by all $\rho_+ \otimes D_l$ and $\rho_- \otimes D_l$. We now show that $\rho_- \otimes D_l$ is equivalent to $D_{l-}$: We have

$$\rho_- \otimes D_l : \mathrm{O}(3) \to \mathrm{GL}(\mathbb{C} \otimes V_l), \quad \big[(\rho_- \otimes D_l)(sg)\big](z \otimes v) = sz \otimes [D_l(g)](v).$$

Now, consider the linear isomorphism $f : \mathbb{C} \otimes V_l \to V_{l+}, z \otimes v \mapsto zv$. We only need to check that it is equivariant and are then done:

$$\begin{aligned}
f\big(\big[(\rho_- \otimes D_l)(sg)\big](z \otimes v)\big) &= f\big(sz \otimes [D_l(g)](v)\big) \\
&= sz \cdot [D_l(g)](v) \\
&= [sD_l(g)](zv) \\
&= [D_{l-}(sg)](f(z \otimes v)).
\end{aligned}$$

The statement about $D_{l+}$ can be shown using the exact same map $f$. □

---

[34]It is not a direct generalization due to the presence of two different group elements being applied.

### E.6.2 THE PETER-WEYL THEOREM FOR $L^2_{\mathbb{C}}(S^2)$ AS REPRESENTATION OF O(3)

The considerations in this section follow almost entirely from Section E.4.2. There we saw that, as a representation over SO(3), we have a decomposition

$$L^2_{\mathbb{C}}(S^2) = \widehat{\bigoplus_{l \geq 0}} V_{l1}$$

with the spaces $V_{l1}$ being spanned by the spherical harmonics $Y_l^n$, $n = -l, \ldots, l$. We immediately see that in $L^2_{\mathbb{C}}(S^2)$, viewed as a representation over O(3), there is *not enough space* for all the irreducible representations, since they appear in pairs as shown in Proposition E.24.[35] Thus, we need to figure out which irreducible representations are present and which are not. The core of this question is answered by the following proposition:

**Lemma E.25** (Parity in spherical harmonics)**.** *The spherical harmonics obey the following parity rules:*

$$Y_l^n(sx) = s^l \cdot Y_l^n(x)$$

*for all $l \geq 0$, $n = -l, \ldots, l$, $s \in \mathbb{Z}_2$ and $x \in S^2$.*

*Proof.* This is a well-known property of the spherical harmonics. $\qquad\square$

Thus, together with Section E.4.2 we get the following transformation behavior of spherical harmonics under the group O(3), where $s \in \mathbb{Z}_2$ and $g \in SO(3)$:

$$\lambda(sg)(Y_l^n) = s^l \lambda(g)(Y_l^n)$$
$$= s^l \sum_{n'=-l}^{l} D_l^{n'n}(g) Y_l^{n'}$$
$$= \sum_{n'=-l}^{l} \left( s^l D_l^{n'n}(g) \right) Y_l^{n'}$$
$$= \begin{cases} \sum_{n'=-l}^{l} D_{l+}^{n'n}(sg) Y_l^{n'}, & l \text{ even} \\ \sum_{n'=-l}^{l} D_{l-}^{n'n}(sg) Y_l^{n'}, & l \text{ odd}. \end{cases}$$

Thus, we obtain the following decomposition of $L^2_{\mathbb{C}}(S^2)$:

$$L^2_{\mathbb{C}}(S^2) = \widehat{\bigoplus_{\substack{l \geq 0 \\ l \text{ even}}}} V_{l1+} \oplus \widehat{\bigoplus_{\substack{l \geq 0 \\ l \text{ odd}}}} V_{l1-}.$$

Here, $V_{l1+}$ and $V_{l1-}$ are generated from the spherical harmonics of order $l$ and we have $V_{l1+} \cong V_{l+}$ and $V_{l1-} \cong V_{l-}$ as representations according to the transformation behavior we saw above.

### E.6.3 THE CLEBSCH-GORDAN DECOMPOSITION

Remember from Section E.4.3 that we have a decomposition of SO(3)-representations

$$V_j \otimes V_l \cong \bigoplus_{J=|l-j|}^{l+j} V_J$$

given by real Clebsch-Gordan coefficients. Now for O(3), remember that as vector spaces we have for all $j$ (and equally for $l$ and $J$) equalities $V_j = V_{j-} = V_{j+}$, and so we guess that in the isomorphism above, we just need to figure out the correct signs in order to be compatible with the

---

[35]With this, we mean the following: the irreducible representations of SO(3) already cover $L^2_{\mathbb{C}}(S^2)$. O(3) has *even more* irreducible representations than SO(3), so it is a priori clear that they cannot all fit into $L^2_{\mathbb{C}}(S^2)$.

corresponding representations. The idea is that "multiplying the signs at the left" should lead to the "sign at the right", and this paradigm leads us to believe that there are the following isomorphisms:

$$V_{j+} \otimes V_{l+} \cong \bigoplus_{J=|l-j|}^{l+j} V_{J+}, \quad V_{j+} \otimes V_{l-} \cong \bigoplus_{J=|l-j|}^{l+j} V_{J-},$$

$$V_{j-} \otimes V_{l+} \cong \bigoplus_{J=|l-j|}^{l+j} V_{J-}, \quad V_{j-} \otimes V_{l-} \cong \bigoplus_{J=|l-j|}^{l+j} V_{J+}.$$

We just show the lower-left isomorphism since the arguments are always the same. So, assume that $f : V_j \otimes V_l \to \bigoplus_{J=|l-j|}^{l+j} V_J$ is an isomorphism and thus in particular intertwines the given representations. Now, we take the exact same map $f : V_{j-} \otimes V_{l+} \to \bigoplus_{J=|l-j|}^{l+j} V_{J-}$ and only need to figure out that it is equivariant with respect to the given representations, using the same property for the original isomorphism we started with:

$$f \circ \left[ D_{j-}(sg) \otimes D_{l+}(sg) \right] = f \circ \left[ s(D_j(g) \otimes D_l(g)) \right]$$

$$= s \bigoplus_{J=|l-j|}^{l+j} D_J(g) \circ f$$

$$= \bigoplus_{J=|l-j|}^{l+j} D_{J-}(sg) \circ f.$$

This shows the claim. From these considerations, it also follows that the Clebsch-Gordan coefficients do not in any way depend on the signs of the spaces $V_j$, $V_l$, $V_J$. Thus, we write them generically as $\langle JM|jm;ln\rangle$.

### E.6.4 ENDOMORPHISMS OF $V_J$

As always over $\mathbb{C}$, Schur's Lemma D.8 shows that the endomorphism spaces are 1-dimensional, and thus we can ignore endomorphisms.

### E.6.5 BRINGING EVERYTHING TOGETHER

Now we can finally compute the basis for steerable kernels. The section on the Clebsch-Gordan decomposition suggests that we need to do a case distinction for this. Namely, the possible kernels depend on the signs of $V_l$ and $V_J$. The results basically follow analogously to the results in Section E.4.5.

STEERABLE KERNELS $K : S^2 \to \mathrm{Hom}_{\mathbb{C}}(V_{l+}, V_{J+})$:

$V_{J+}$ can only be in a tensor product $V_j \otimes V_{l+}$ if the sign of $j$ is positive. Spaces $V_{j1+}$ appear in the tensor product decomposition of $L_{\mathbb{C}}^2(S^2)$ precisely for even $j$, according to Section E.6.2. Thus, a basis for steerable kernels is given by all $K_j$ with *even* $j \in \{|l - J|, \ldots, l + J\}$ . It has matrix elements

$$\langle JM|K_j(x)|ln\rangle = \sum_{m=-j}^{j} \langle JM|jm;ln\rangle \cdot \overline{Y_j^m}(x),$$

exactly as in Section E.4.5.

STEERABLE KERNELS $K : S^2 \to \mathrm{Hom}_{\mathbb{C}}(V_{l+}, V_{J-})$:

Analogously, a basis for steerable kernels is given by all $K_j$, with *odd* $j \in \{|l - J|, \ldots, l + J\}$.

STEERABLE KERNELS $K : S^2 \to \mathrm{Hom}_{\mathbb{C}}(V_{l-}, V_{J+})$:

Again, a basis for steerable kernels is given by all $K_j$ with *odd* $j \in \{|l - J|, \ldots, l + J\}$.

STEERABLE KERNELS $K : S^2 \to \mathrm{Hom}_{\mathbb{C}}(V_{l-}, V_{J-})$:

As in the first case, a basis for steerable kernels is given by all $K_j$ with *even* $j \in \{|l - J|, \dots, l + J\}$.

Thus, we have determined all kernel bases for the group $\mathrm{O}(3)$ over the complex numbers. Compared to $\mathrm{SO}(3)$, we see that the kernel spaces get roughly halved. The reason for this is that with a bigger symmetry group, the kernel needs to obey more rules, which means that the kernel constraint has fewer solutions.

### E.7 O(3)-STEERABLE KERNELS FOR REAL REPRESENTATIONS

Basically, we can argue exactly as in Section E.5.4 in order to transport the results for complex representations over to the real world. We shortly sketch the procedure and outcome. As we know from Section E.3.1, $\rho_- : \mathbb{Z}_2 \to \mathrm{O}(\mathbb{R})$ and $\rho_+ : \mathbb{Z}_2 \to \mathrm{O}(\mathbb{R})$ are the only irreducible real representations of $\mathbb{Z}_2$. Thus, for each $l \geq 0$ we obtain two irreducible real representations ${}^r D_{l+} : \mathrm{O}(3) \to \mathrm{O}({}^r V_{l+})$ and ${}^r D_{l-} : \mathrm{O}(3) \to \mathrm{O}({}^r V_{l-})$. As before, they also act on complex vector spaces and are as such isomorphic to the complex irreducible representations of $\mathrm{O}(3)$. One can then show as in Lemma E.17 that all complex irreducible representations are of real type since they split into two copies of the real version of this representation. Thus, by Corollary E.18, all real irreducible representations are of real type, and this means that we can proceed exactly as in Proposition E.19 in order to show that the ${}^r D_{l+}$ and ${}^r D_{l-}$ are already all the irreducible real representations of $\mathrm{O}(3)$ up to equivalence.

For the Peter-Weyl decomposition of $L_{\mathbb{R}}^2(S^2)$, we only need to note that the real spherical harmonics emerge with a base change from the complex ones, as seen in Eq. (31), and thus fulfill the same parity rules as the complex spherical harmonics. This gives us a decomposition

$$L_{\mathbb{R}}^2(S^2) = \widehat{\bigoplus_{\substack{l \geq 0 \\ l \text{ even}}}} \left({}^r V_{l1+}\right) \oplus \widehat{\bigoplus_{\substack{l \geq 0 \\ l \text{ odd}}}} \left({}^r V_{l1-}\right).$$

For the Clebsch-Gordan coefficients, we again get decompositions

$$ {}^r V_j \otimes {}^r V_l \cong \bigoplus_{J=|l-j|}^{l+j} {}^r V_J $$

where the signs on the left must "multiply to" the signs on the right, as in Section E.6.3. Finally, the endomorphism spaces must be 1-dimensional since the endomorphism spaces of the complex versions are 1-dimensional.

Overall, we obtain the same kernels as in Section E.6.5, only that we need to use the real spherical harmonics as our steerable filters and can get rid of the complex conjugation.

## F MATHEMATICAL PRELIMINARIES

In this chapter, we state mathematical preliminaries that we use throughout the earlier chapters. In this whole chapter, $\mathbb{K}$ is one of the two fields $\mathbb{R}$ or $\mathbb{C}$.

### F.1 TOPOLOGICAL SPACES, NORMED SPACES, AND METRIC SPACES

Since in this work, we want to develop the theory of representations over *compact* groups, and since this is a topological property, we need to formulate some topological concepts (Conway, 2014). Additionally, the vector spaces on which our compact groups act also carry a topology, mostly coming from their Hilbert space structure.

**Definition F.1** (Topological Space, Open Sets, Closed Sets). A *topological space* $(X, \mathcal{T})$ consists of a set $X$ and a set $\mathcal{T}$ of subsets of $X$, called the *open* sets, such that arbitrary unions and finite intersections of open sets are open. In particular, $X$ and the empty set $\emptyset$ are open. *Closed* sets are the complements of open sets and fulfill dual axioms: arbitrary intersections and finite unions of closed sets are closed.

Let in the following $X$ and $Y$ be topological spaces.

**Definition F.2** (Open Neighborhood). Let $x \in X$. An open set $U \subseteq X$ is called *open neighborhood* of $x$ if $x \in U$.

**Definition F.3** (Hausdorff Space). $X$ is called a *Hausdorff space* if two distinct points can always be separated by open sets, i.e., for all $x, y \in X$ there exist $U_x, U_y$ open such that $x \in U_x, y \in U_y$, and $U_x \cap U_y = \emptyset$.

In this work, all topological spaces are Hausdorff.

**Definition F.4** (Subspace). Assume $A \subseteq X$ is a subset. Then the set $\mathcal{T}_A := \{U \cap A \mid U \in \mathcal{T}\}$ is a topology for $A$ and thus makes $A$ a topological space as well. It is called a *subspace* of $X$.

Whenever we consider a subset of a topological space, it is viewed as a topological space with this construction.

**Definition F.5** (Closure, Density). For $A \subseteq X$, its *closure* $\overline{A}$ is defined as the smallest closed subset of $X$ that contains $A$. Equivalently, it is the intersection of all closed subsets of $X$ containing $A$, which is closed by the axioms of a topology. $A$ is called *dense in $X$* if $\overline{A} = X$.

**Definition F.6** (Continuous Function, Homeomorphism). A function $f : X \to Y$ is called *continuous* if preimages of open sets are always open. Equivalently, for each point $x_0 \in X$ and each open neighborhood $V$ of $f(x_0)$ there is an open neighborhood $U$ of $x_0$ such that $f(U) \subseteq V$.

A *homeomorphism* is a continuous bijective function with a continuous inverse.

Note that compositions of continuous functions are continuous as well.

**Definition F.7** (Open Cover, Compact Space). An *open cover* of $X$ is a family of open sets $\{U_i\}_{i \in I}$ that cover $X$, i.e., $X = \bigcup_{i \in I} U_i$. $X$ is called *compact* if all open covers have a finite subcover, that is: For all open covers $\{U_i\}_{i \in I}$ there exists a finite subset $J \subseteq I$ such that $\{U_i\}_{i \in J}$ is still an open cover of $X$.

**Proposition F.8.** *If $X$ is compact and $f : X \to Y$ is continuous, then $f(X) \subseteq Y$ is compact as well. In particular, if $f$ surjective, then $Y$ is compact.*

*Proof.* See Sutherland (1975), Proposition 13.15. $\qquad\square$

**Proposition F.9.** *Let $f : X \to Y$ be a continuous bijection and assume that $X$ is compact and that $Y$ is Hausdorff. Then the inverse $f^{-1}$ is continuous as well and thus $f$ is a homeomorphism.*

*Proof.* See Sutherland (1975), Proposition 13.26. $\qquad\square$

**Definition F.10** (Product Topology). The *product topology* on $X \times Y$ is the coarsest (i.e., smallest in terms of inclusion) topology that makes both projections $p_X : X \times Y \to X$ and $p_Y : X \times Y \to Y$ continuous.

If $Z$ is a third topological space and we have two continuous functions $f_X : Z \to X$ and $f_Y : Z \to Y$, then the function $f_X \times f_Y : Z \to X \times Y, z \mapsto (f_X(z), f_Y(z))$ is continuous as well.

**Definition F.11** (Quotient Map, Quotient Space). A continuous function $f : X \to Y$ is called a *quotient map* if $f$ is surjective and if $U \subseteq Y$ is open if and only if $f^{-1}(U) \subseteq X$ is open.

Let $\sim$ be any equivalence relation on $X$ and $X/\sim$ be the quotient set formed by identifying equivalent elements. Let $q : X \to X/\sim$ be the canonical function sending each element to its equivalence class. We define $U \subseteq X/\sim$ to be open if $q^{-1}(U) \subseteq X$ is open. Then $q$ is a quotient map and $X/\sim$ is called a *quotient space*.

**Proposition F.12** (Universal property of Quotient Maps). *Let $q : X \to X/\sim$ be a standard quotient map and $f : X \to Y$ be any continuous function such that $f(x) = f(x')$ whenever $x \sim x'$. Then there is a unique continuous function $\overline{f} : X/\sim \to Y$ such that the following diagram commutes:*

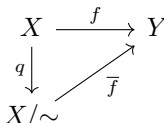

$\overline{f}$ *is given on equivalence classes by* $\overline{f}([x]) = f(x)$.

*Proof.* See Conway (2014), Proposition 2.8.7. $\qquad\qquad\qquad\qquad\qquad\qquad\qquad$ □

It can be shown that all quotient maps are equivalent to a construction of the form $q : X \to X/\sim$. Namely, for a quotient map $f : X \to Y$, define $\sim$ by $x \sim x'$ if $f(x) = f(x')$. Then the map $\overline{f} : X/\sim \to Y$, $[x] \mapsto f(x)$ is a well-defined continuous map by the universal property of quotient maps Proposition F.12. One can show that this is a homeomorphism. Thus for a quotient map $f : X \to Y$ we also call $Y$ a quotient space.

Our route for defining concrete topologies is in most cases through the existence of inner products on Hilbert spaces, which will be defined in detail in Definition F.32. Namely, inner products define norms, which define metrics (Kaplansky, 2001), which in turn define topologies. For this, we need some definitions:

**Definition F.13** (Norm). Let $V$ be a $\mathbb{K}$-vector space, A *norm* on $V$ is a map $\| \cdot \| : V \to \mathbb{R}_{\geq 0}$ with the following properties for all $\lambda \in \mathbb{K}$ and $v, w \in V$:

1. $\|v\| = 0$ if and only if $v = 0$.

2. $\|\lambda v\| = |\lambda| \cdot \|v\|$.

3. Triangle inequality: $\|v + w\| \leq \|v\| + \|w\|$.

If $\langle \cdot | \cdot \rangle : V \times V \to \mathbb{K}$ is an inner product on a Hilbert space, then it defines a norm $\| \cdot \| : V \to \mathbb{R}_{\geq 0}$ by $\|x\| := \sqrt{\langle x | x \rangle}$.

**Definition F.14** (Metric). Let $Y$ be a set. A *metric* on $Y$ is a function $d : Y \times Y \to \mathbb{R}_{\geq 0}$ with the following properties for all $x, y, z \in Y$:

1. $d(x, y) = 0$ if and only if $x = y$.

2. Symmetry: $d(x, y) = d(y, x)$.

3. Triangle inequality: $d(x, z) \leq d(x, y) + d(y, z)$.

A norm $\| \cdot \| : V \to \mathbb{R}_{\geq 0}$ defines a metric $d : V \times V \to \mathbb{R}$ by setting $d(x, y) := \|x - y\|$. In turn, a metric defines a topology as follows: open balls are given by all sets of the form $\mathrm{B}_\epsilon(x) := \{y \in V \mid d(x, y) < \epsilon\}$ for all $x \in V$ and $\epsilon > 0$. Open sets are then defined as arbitrary unions of arbitrary open balls.

Additionally, we need notions about convergence in this work. Since we will deal with them mostly in the context of metric spaces (with normed vector spaces and Hilbert spaces being special cases, as explained above), we focus on these notions for metric spaces.

**Definition F.15** (Convergent Sequence). Let $Y$ be a metric space. Then a sequence $(y_k)_k$ in $Y$ is said to *converge to* $y$ if for all $\epsilon > 0$ there is a $k_\epsilon \in \mathbb{N}$ such that $y_k \in \mathrm{B}_\epsilon(y)$ for all $k \geq k_\epsilon$.

With this in mind, one can give an equivalent definition of continuity that applies to metric spaces:

**Definition F.16** (Continuity in Metric Spaces). A function $f : Y \to Z$ between metric spaces is *continuous in* $y \in Y$ if for each sequence $(y_k)_k$ of points $y_k \in Y$ converging to a point $y \in Y$, we also have that the sequence $f(y_k)$ converges to $f(y)$. This can be understood in terms of the function "commuting with limits":

$$\lim_{k \to \infty} f(y_k) = f\left( \lim_{k \to \infty} y_k \right).$$

Furthermore, $f : Y \to Z$ is called *continuous* if it is continuous in all points $y \in Y$.

Equivalently, the following holds: $f : Y \to Z$ is continuous in $y \in Y$ if and only of for all $\epsilon > 0$ there is a $\delta > 0$ such that $f(\mathrm{B}_\delta(y)) \subseteq \mathrm{B}_\epsilon(f(y))$.

**Definition F.17** (Uniform Continuity). A function $f : Y \to Z$ between metric spaces is called *uniformly continuous* if for each $\epsilon > 0$ there is a $\delta > 0$ such that for all $y, y' \in Y$ with $d_Y(y, y') < \delta$ we obtain $d_Y(f(y), f(y')) < \epsilon$.

The following is a result we use several times in the main text:

**Proposition F.18.** *Let $f : V \to V'$ be a linear function between normed vector spaces. Then the following are equivalent:*

1. *$f$ is uniformly continuous.*

2. *$f$ is continuous.*

3. *$f$ is continuous in $0$.*

*Proof.* Trivially, 1 implies 2, which in turn implies 3. Now assume 3, i.e., $f$ is continuous in 0. Let $\epsilon > 0$. Then by continuity in 0, there exists $\delta > 0$ such that for all $v \in V$ with $\|v\| = \|v - 0\| < \delta$ we obtain $\|f(v)\| = \|f(v) - f(0)\| < \epsilon$. Now let $v, v' \in V$ be arbitrary with $\|v - v'\| < \delta$. Then by the linearity of $f$ we obtain:

$$\|f(v) - f(v')\| = \|f(v - v')\| < \epsilon,$$

which is exactly what we wanted to show. □

Sometimes, sequences *look like* they converge since their elements get ever closer to each other. However, not all such sequences need to converge. Therefore, there is the following notion:

**Definition F.19** (Cauchy Sequence). Let $Y$ be a metric space. A sequence $(y_k)_k$ in $Y$ is a *Cauchy Sequence* if for all $\epsilon > 0$ there is $k_\epsilon \in \mathbb{N}$ such that for all $k, k' > k_\epsilon$ we have $d(y_k, y_{k'}) < \epsilon$.

For example, one can consider the metric space $\mathbb{R} \setminus \{0\}$ together with the usual metric. Then the sequence $\left(\frac{1}{k}\right)_k$ is a Cauchy sequence but does not converge since the limit (in $\mathbb{R}$!), which would be 0, is not in $\mathbb{R} \setminus \{0\}$. Thus, the following notion is useful:

**Definition F.20** (Complete Metric Space). A metric space $Y$ is called *complete* if every Cauchy sequence converges.

**Definition F.21** (Completion). Let $Y$ be a metric space. A *completion of $Y$* is a metric space $Y'$ which contains $Y$ as a dense subspace and such that $Y'$ is complete.

**Proposition F.22** (Universal Property of Completions). *Assume that $Y \subseteq Y'$ is a pair of metric spaces, where $Y'$ is a completion of $Y$. Then the following universal property holds:*

*Let $Z$ be any complete metric space and $f : Y \to Z$ be any uniformly continuous function. Then there is a unique continuous function $f' : Y' \to Z$ that extends $f$, i.e., such that $f'|_Y = f$. $f'$ furthermore is also uniformly continuous. This can be expressed by the following commutative diagram, where $i : Y \to Y'$ is the canonical inclusion:*

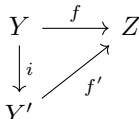

*Proof.* See, for example, Kaplansky (2001). □

**Definition F.23** (Boundedness). Let $Y$ be a metric space. A subset $A \subseteq Y$ is called *bounded* if there is a constant $C > 0$ such that $d(a, b) \leq C$ for all $a, b \in A$.

**Theorem F.24** (Heine-Borel Theorem). *A subset $A \subseteq \mathbb{K}^d$ is compact if and only if it is closed and bounded.*

*Proof.* See Conway (2014), Theorem 1.4.8. □

**Corollary F.25** (Extreme Value Theorem). *Let $f : X \to \mathbb{R}$ be continuous, where $X$ is any nonempty compact topological space. Then $f$ has a maximum and a minimum.*

*Proof.* By Proposition F.8, $f(X) \subseteq \mathbb{R}$ is compact. By Theorem F.24 this means that $f(X)$ is closed and bounded. Boundedness means that the supremum is finite and closedness means that the supremum must lie in $f(X)$, and consequently it is a maximum. For the minimum, the same arguments apply. □

## F.2 LIMITS OF NETS AND APPROXIMATED DIRAC DELTA FUNCTIONS

In this section, we discuss "limits of nets", where a net can be imagined as a sequence over an index set which may be "too big to be handled as a sequence over the natural numbers". They appear in the formulation of Theorem C.7. This material can, for example, be found in (Conway, 2014).

**Definition F.26** (Partially Ordered Set, Directed Set). Let $I$ be an index set and $\leq$ a relation on it. $I = (I, \leq)$ is a *partially ordered set* if:

1. $\leq$ is *reflexive*, i.e., $i \leq i$ for all $i \in I$.

2. $\leq$ is *antisymmetric*, that is: $i \leq j$ and $j \leq i$ together imply $i = j$.

3. $\leq$ is *transitive*, that is: $i \leq j$ and $j \leq k$ together imply $i \leq k$.

A partially ordered set $I$ is called *directed* if for all $i, j \in I$ there exists $k \in I$ such that $i \leq k$ and $j \leq k$.

**Example F.27.** Clearly, the natural numbers together with the standard order relation form a directed set.

An important example for our purposes is the following: let $Z$ be any topological space (for example, a homogeneous space $X$ of a compact group $G$) and $x \in Z$ be any point. Furthermore, define $\mathcal{U}_x$ as the set of open neighborhoods of $x$, i.e., open sets $U \subseteq Z$ such that $x \in U$. On this set, we define $U \leq V$ if $U \supseteq V$, i.e., by *reversed* inclusion. Then $(\mathcal{U}_x, \leq)$ is a directed set:

1. Reflexivity is clear since $V \supseteq V$ for all $V$.

2. Antisymmetry is clear since $U \supseteq V$ and $V \supseteq U$ together clearly imply $U = V$.

3. Transitivity is clear since $U \supseteq V$ and $V \supseteq W$ together clearly imply $U \supseteq W$.

4. For directedness, let $U, V \in \mathcal{U}_x$. Define $W = U \cap V$. Then $W \in \mathcal{U}_x$ and clearly $U \supseteq W$ and $V \supseteq W$, which is what was to show.

Note that $\mathcal{U}_x$ is usually *not totally ordered*, i.e., there are usually $U, V \in \mathcal{U}_x$ such that neither $U \supseteq V$ nor $V \supseteq U$.

**Definition F.28** (Net). Let $Z$ be any topological space and $I$ a directed set. Then a *net* in $Z$ is a function $x : I \to Z$. We write a net as $(x_i)_{i \in I}$, in analogy to sequences.

**Definition F.29** (Convergence of Nets). Let $(x_i)_{i \in I}$ be a net in a topological space $Z$. Let $x \in Z$. We say that $(x_i)_{i \in I}$ *converges to* $x$, written $\lim_{i \in I} x_i = x$, if the following holds: for all open neighborhoods $U$ of $x$ there is an $i_0 \in I$ such that for all $i \geq i_0$ we have $x_i \in U$.

Now we define the approximated Dirac delta for the special case that $X$ is a homogeneous space of a compact group $G$. Remember that there is a Haar measure $\mu$ on $X$.

**Definition F.30** (Approximated Dirac Delta). For $\emptyset \neq U \subseteq X$ open, we define the approximated Dirac delta by $\delta_U : X \to \mathbb{K}$ with

$$\delta_U(x) = \frac{1}{\mu(U)} \cdot \mathbf{1}_U(x) = \begin{cases} \frac{1}{\mu(U)}, & x \in U \\ 0, & \text{else.} \end{cases}$$

We have $\delta_U \in L^2_{\mathbb{K}}(X)$.

A priori, it is unclear that open sets have positive measure, which is needed for the well-definedness of this construction, since otherwise we divide by zero. Thus, we need the following lemma:

**Lemma F.31.** *Let $\emptyset \neq U \subseteq X$ be an open set. Then $\mu(U) > 0$.*

*Proof.* Consider the family of open sets $(gU)_{g \in G}$. That all of these sets are necessarily open follows since the action $G \times X \to X$ is continuous, and thus by the definition of a group action, each $g \in G$ induces a homeomorphism $X \to X, x \mapsto gx$. Now, since the action is transitive, $(gU)_{g \in G}$ is an open cover of $X$, and since $X$ is compact, see Definition F.7, it has an open subcover $(g_i U)_{i=1}^n$ with

$g_i \in G$. Note that $\mu(g_i U) = \mu(U)$ for all $i$ since the measure $\mu$ on $X$ is by definition left invariant under the action of $G$. Overall, we obtain

$$1 = \mu(X) = \mu\left(\bigcup_{i=1}^{n} g_i U\right) \leq \sum_{i=1}^{n} \mu(g_i U) = \sum_{i=1}^{n} \mu(U) = n \cdot \mu(U)$$

and thus $\mu(U) \geq \frac{1}{n} > 0$. $\qquad\square$

### F.3 PRE-HILBERT SPACES AND HILBERT SPACES

Here, we state foundational concepts in the theory of Hilbert spaces (Debnath & Mikusinski, 2005).

**Definition F.32** (pre-Hilbert Space, Hilbert space). A *pre-Hilbert space* $V = (V, \langle \cdot | \cdot \rangle)$ consists of the following data:

1. A vector space $V$ over $\mathbb{K}$.

2. An inner product $\langle \cdot | \cdot \rangle : V \times V \to \mathbb{K}$, $(x, y) \mapsto \langle x | y \rangle$.

It has the following properties that hold for all $x, x', y, y' \in V$, $\lambda \in \mathbb{K}$:

1. The inner product is conjugate linear in the first component: $\langle x + x' | y \rangle = \langle x | y \rangle + \langle x' | y \rangle$ and $\langle \lambda x | y \rangle = \overline{\lambda} \langle x | y \rangle$, where $\overline{\lambda}$ is the complex conjugate of $\lambda$.

2. The inner product is linear in the second component: $\langle x | y + y' \rangle = \langle x | y \rangle + \langle x | y' \rangle$ and $\langle x | \lambda y \rangle = \lambda \langle x | y \rangle$.

3. The inner product is conjugate symmetric: $\langle y | x \rangle = \overline{\langle x | y \rangle}$

4. The inner product is positive definite: $\langle x | x \rangle > 0$ unless $x = 0$.

If additionally, the following statement holds, then $V$ is called a *Hilbert Space*:

5. $V$, together with the norm $\|\cdot\| : V \to V$ induced from the inner product by $\|x\| := \sqrt{\langle x | x \rangle}$, and consequently the metric defined by $d(x, y) := \|x - y\|$, is a complete metric space as in Definition F.20.

*Remark F.33.* Of course, all Hilbert Spaces are pre-Hilbert spaces, and so all Propositions about pre-Hilbert spaces in the following apply to Hilbert spaces just as well.

Note that the first property follows from the second and third. We also mention that usually, inner products on Hilbert spaces are assumed to be linear in the first and conjugate linear in the second component, in contrast to how we view it. The reason for our choice is that our work is inspired by connections to physics where our convention is more common. It is basically the bra-ket convention. Furthermore, note that if $\mathbb{K} = \mathbb{R}$, then conjugate linear maps are linear and thus the inner product will be linear in both components. Additionally, it will be symmetric instead of only conjugate symmetric.

**Proposition F.34** (Cauchy-Schwartz Inequality). *For any two elements $v, w$ in a pre-Hilbert space $V$, we have*
$$|\langle v | w \rangle| \leq \|v\| \cdot \|w\|.$$
*We have equality if and only if $v$ and $w$ are linearly dependent.*

*Proof.* See Debnath & Mikusinski (2005), Theorem 3.2.9. $\qquad\square$

**Definition F.35** (Orthogonality). Two vectors $v, w$ in a pre-Hilbert space $V$ are called *orthogonal*, written $v \perp w$, if $\langle v | w \rangle = 0$.

Obviously, being orthogonal is a symmetric relation.

**Definition F.36** (Orthogonal Complement). Let $V$ be a pre-Hilbert space and $W \subseteq V$ a subset. $v \in V$ is *orthogonal to $W$* if $\langle v \mid w \rangle = 0$ for all $w \in W$.

The *orthogonal complement* of $W$, denoted $W^\perp$, is the set of all vectors in $V$ that are orthogonal to $W$.

**Proposition F.37** (Closedness of Complements). *Let $W \subseteq V$ be a subset of a pre-Hilbert space $V$. Then $W^\perp$ is a topologically closed linear subspace of $V$.*

*Proof.* See Debnath & Mikusinski (2005), Theorem 3.6.2. □

**Proposition F.38** (Continuity of Scalar Product). *For any pre-Hilbert space $V$, the scalar product $\langle \cdot | \cdot \rangle : V \times V \to \mathbb{K}$ is continuous.*

*Proof.* See Debnath & Mikusinski (2005), Theorem 3.3.12. □

**Definition F.39** (Orthonormal System). A family $(v_i)_{i \in I}$ of elements in a pre-Hilbert space is called *orthonormal system* if $\|v_i\| = 1$ for all $i \in I$ and $v_i \perp v_j$ for all $i \neq j$.

**Definition F.40** (Orthonormal Basis). An orthonormal system $(v_i)_{i \in I}$ in a Hilbert space $V$ is called *orthonormal basis* if the linear span of all $\{v_i\}_{i \in I}$ is dense in $V$. If this is the case, then each $v \in V$ can be uniquely written as

$$v = \sum_{i \in I} \alpha_i v_i$$

with only countably many $\alpha_i \in \mathbb{K}$ being nonzero. The coefficients are given by $\alpha_i = \langle v_i | v \rangle$.

We stress that while the index set $I$ can be uncountably infinite, the sequence expansions of each element in $V$ only have countably many entries. It is obvious from the Peter-Weyl Theorem B.22 and this definition that the functions

$$\left\{ Y_{li}^m \mid l \in \widehat{G}, i \in \{1, \ldots, m_l\}, m \in \{1, \ldots, d_l\} \right\}$$

form an orthonormal basis of $L_{\mathbb{K}}^2(X)$.

**Proposition F.41** (Gram-Schmidt Orthonormalization). *For every linearly independent sequence $(y_k)_k$ in a pre-Hilbert space $V$ with $N \in \mathbb{N} \cup \{\infty\}$ elements, one can find an orthonormal sequence $(v_k)_k$ in $V$ such that the following holds: for all $n \in \mathbb{N}$, $n \leq N$, the progressive linear span stays the same:*

$$\mathrm{span}_{\mathbb{K}}(v_1, \ldots, v_n) = \mathrm{span}_{\mathbb{K}}(y_1, \ldots, y_n).$$

*In particular, since every finite-dimensional Hilbert space has a vector space basis, it necessarily also has an orthonormal basis.*

*Proof.* See Debnath & Mikusinski (2005), page 110. □

**Definition F.42** (Adjoint of an Operator). Let $f : V \to V'$ be a continuous linear function between Hilbert spaces. Then there is a unique continuous linear function $f^* : V' \to V$ such that for all $v \in V$ and $v' \in V'$ one has:

$$\langle f(v) | v' \rangle_{V'} = \langle v | f^*(v') \rangle_V .$$

$f^*$ is called the *adjoint* of $f$.

The existence of adjoints is, for example, discussed in Debnath & Mikusinski (2005), page 158. This book only considers the case of operators on a Hilbert space to itself, but these considerations generalize to the setting with two different Hilbert spaces. One has the following:

**Proposition F.43.** *Let $f : V \to V'$ and $g : V' \to V''$ be continuous linear functions between Hilbert spaces. Then:*

1. *$(f^*)^* = f$.*

2. *$\mathrm{id}_V^* = \mathrm{id}_V$.*

3. *$(g \circ f)^* = f^* \circ g^*$.*

*Proof.* All of these properties follow directly from the uniqueness of adjoints. □

**Proposition F.44.** *Let $f : V \to V'$ be a unitary transformation between Hilbert spaces, i.e., an invertible linear function such that $\langle f(v) | f(w) \rangle = \langle v | w \rangle$ for all $v, w \in V$. Then the adjoint is the inverse, i.e., $f^* = f^{-1}$.*

*Proof.* First of all, the inverse $f^{-1}$ is again continuous due to the unitarity of $f$. Furthermore, due to the unitarity, we obtain

$$\langle f(v)|v'\rangle = \langle f(v)\big|f(f^{-1}(v'))\rangle$$
$$= \langle v\big|f^{-1}(v')\rangle$$

for all $v \in V$ and $v' \in V'$. Due to the uniqueness of adjoints, we obtain $f^{-1} = f^*$. □

The following proposition is sometimes used in the main text:

**Proposition F.45.** *Let $v, w \in V$ be two elements in a pre-Hilbert space such that $\langle v|u\rangle = \langle w|u\rangle$ for all $u \in V$. Then $v = w$.*

*Proof.* We have

$$\langle v - w|u\rangle = \langle v|u\rangle - \langle w|u\rangle = 0$$

for all $u \in V$. In particular, when setting $u = v - w$ we obtain

$$\langle v - w|v - w\rangle = 0$$

and thus $v - w = 0$, i.e., $v = w$. □

**Proposition F.46** (Orthogonal Projection Operators). *Let $W \subseteq V$ be a topologically closed sub-space of a Hilbert space. Then there is a continuous linear function $P : V \to W$ such that for all $v \in V$ and $w \in W$ we have*

$$\langle P(v)|w\rangle = \langle v|w\rangle.$$

*Furthermore, if $W$ is finite-dimensional and $w_1, \ldots, w_n$ and orthonormal basis, then $P$ is given explicitly by*

$$P(v) = \sum_{i=1}^{n} \langle w_i|v\rangle \, w_i.$$

*Proof.* That $W$ is topologically closed means that $W$, with the scalar product inherited from $V$, is a complete metric space. Thus, $W$ is a Hilbert space as well. Therefore, the continuous linear embedding $i : W \to V$ given by $w \mapsto w$ has an adjoint $i^* : V \to W$ by Definition F.42. Set $P := i^*$. For arbitrary $v \in V$ and $w \in W$ we obtain:

$$\langle P(v)|w\rangle = \langle i^*(v)|w\rangle$$
$$= \langle v|i(w)\rangle$$
$$= \langle v|w\rangle.$$

For the second statement, note that for all $j \in \{1, \ldots, n\}$ we have, using that the $w_i$ are orthonormal:

$$\left\langle \sum_{i=1}^{n} \langle w_i|v\rangle \, w_i \Big| w_j \right\rangle = \sum_{i=1}^{n} \overline{\langle w_i|v\rangle} \, \langle w_i|w_j\rangle$$
$$= \langle v|w_j\rangle$$
$$= \langle P(v)|w_j\rangle.$$

By Proposition F.45 and since the $w_j$ generate $W$ we obtain $\sum_{i=1}^{n} \langle w_i|v\rangle \, w_i = P(v)$ as claimed. □

**Proposition F.47.** *Let $(V, \langle \cdot|\cdot \rangle)$ be a finite-dimensional pre-Hilbert space. Then this space is already complete and thus a Hilbert space.*

*In particular, all finite-dimensional subspaces of Hilbert spaces are topologically closed.*

*Proof.* The proof of the Gram-Schmidt orthonormalization in Proposition F.41 does not make use of the completeness of the Hilbert space, and thus it holds for pre-Hilbert spaces as well. Consequently, $V$, being finite-dimensional, has an orthonormal basis. It is thus isomorphic to $\mathbb{K}^n$ together with the standard scalar product, which is well-known to be complete. Thus, $V$ is a Hilbert space.

Now, let $W \subseteq V$ be a finite-dimensional subspace of a Hilbert space $V$ which may be infinite-dimensional. Then $W$ is a pre-Hilbert space and by what was just shown a Hilbert space. Consequently, all sequences in $W$ which have a limit in $V$ need, by completeness, to have that limit already in $W$. This shows that $W$ is topologically closed. □

