# OpenReview forum: "A Wigner-Eckart Theorem for Group Equivariant Convolution Kernels"
_ICLR.cc/2021/Conference — ICLR 2021 Poster_

### Official Review · AnonReviewer2 · 2020-10-27
**Physics techniques for group equivariant convolution**

**Rating:** 6
**Confidence:** 3

**Review:**

For a physicist familiar with machine learning, the title of this work says it all.  It is a lengthy explanation of how to use well known techniques from physics in constructing convolutional neural networks with a group symmetry.  The paper also spells out what are likely to be the most used cases of U(1) and SU(2) and their quotients by discrete groups.

I give the paper high marks on the value of the topic it addresses, but middling marks on presentation.  The results I spot checked were correct, but the paper reads like a compilation of material from the many math and physics textbooks on the topic, and lacks the coherence of a good textbook.  If I had a student who needed this material I would instead give him or her the original Cohen Welling paper and a representation theory textbook such as Hall 2015 or one of the several textbooks they cite.  In addition the explicit results of appendix E would be rather more useful if implemented in a ML package, and I hope that is on the author's to-do list.

To summarize, a serviceable reference work which will probably be made obsolete by the appearance of a proper textbook on group theory in ML before long.

---

> ### Author Response · Authors · 2020-11-16
> **Response to Reviewer 2**
>
> We thank you for your review and are glad you find the topic that we address valuable.
>
> Q: the paper reads like a compilation of material from the many math and physics textbooks on the topic, and lacks the coherence of a good textbook. If I had a student who needed this material I would instead give him or her the original Cohen Welling paper and a representation theory textbook such as Hall 2015 or one of the several textbooks they cite.
>
> A: We understand the impression that our work looks similar to existing results like the Wigner-Eckart Theorem for spherical tensor operators and representation operators. Nevertheless, we disagree with the implied sentiment that our results are well-known. The main theoretical contributions which are not found in existing research are as follows:
>
> - Representation operators can be described as linear maps on a representation space, as we explain in Section $C.1.3$. In contrast, steerable kernels on a homogeneous space $X$ are not linear (indeed, $X$ does not even have a vector space structure), and so we needed to bridge this gap. This is accomplished by our Theorem $C.7$, which relates steerable kernels on $X$ to certain representation operators which we call "kernel operators" on $L^2\_{\mathbb{K}}(X)$.
> - Note that also for these kernel operators, the Wigner-Eckart Theorem seems to not have been established in the literature before, in particular considering that kernel operators on $L^2\_{\mathbb{K}}(X)$ are by definition also continuous, thus requiring topological arguments.
> - We acknowledge (here and in the paper), however, that earlier work like Thomas et al. (TFN) and Weiler and Cesa (E2CNNs) have kernel space solutions for specific groups that match our Wigner-Eckart Theorem. If we consider the following, we think it is appropriate to say that our results are new:
>    - These papers do not mention the Wigner-Eckart Theorem explicitly.
>    - They do not prove the completeness of the kernel space solutions (Thomas et al.) or use proofs different from ours (Weiler and Cesa).
>    - They both do not mention endomorphisms.
>    - They are less general, working for specific groups instead of all compact groups.
> - Different from considerations in physics, we furthermore allow for both the fields $\mathbb{R}$ and $\mathbb{C}$, and thus our treatment crucially highlights the importance of endomorphism spaces.
> - Finally, for the Peter-Weyl Theorem (in our case formulated for a homogeneous space $X$ of a compact group $G$) we could not find literature that describes it for real representations, and we show how the version for real representations can be proven.
>
> If you can point us to related work in physics and representation theory which we accidentally omitted, then we are happy to include such work.
>
> Q: In addition the explicit results of appendix E would be rather more useful if implemented in a ML package, and I hope that is on the author's to-do list.
>
> A: Note that for the two-dimensional case, i.e. compact subgroups of O(2), such a library was already developed by Weiler and Cesa. We indeed consider to implement a library for other groups. Nevertheless, given the theoretical nature and the scope of our current submission, we believe that these implementations deserve another publication that focuses more on practical considerations.

---

### Official Review · AnonReviewer4 · 2020-10-28
**Theorem 4.1 unifies previous efforts on Steerable CNN and provides a useful method for approaching further G.  This paper is a significant contribution to the field.**

**Rating:** 8
**Confidence:** 4

**Review:**


The authors prove a theorem (thm 4.1) which describes a basis for the space of kernels in a G-steerable CNN for any compact group G.  Steerable CNNs are similar to CNNs but replace channels with G-reps and enforce an equivariance constraint on the kernels.  Though Cohen et al 2019 state the constraint, and Cohen et al 2019, Cohen & Welling 2016, and Weiler & Cesa 2019 and several other papers solve this constraint for different groups and representations, there has not been a general formulation which applies to all compact groups.  Here, solving the constraint means to construct a basis of the space of steerable kernels.  Any steerable kernel is then a linear combination of this basis and the network can then be trained by learning the coefficients.

The problem of finding a basis for the space of steerable kernels is non-trivial and critical for constructing G-steerable CNNs.  Up until now this has done group by group.  The theorem proved in this work unifies such previous efforts and provides a useful method for approaching further G.  This paper is a significant contribution to the field.

The appendix provides a complete and approachable background in the area as well as detailed and precise proofs.  Moreover, the effort by Cohen & Welling to frame equivariant deep learning in the proper context of representation theory is continued and extended here to good profit.  The appendix is quite verbose and the language is more casual than I am accustomed to in a mathematical text or research paper, but it serves the goal of being didactic and approachable.

A practical consideration remains. Though theorem 4.1 reduces construction of a basis of steerable kernels to 1) finding Clebsch-Gordon decomposition of tensor products, 2) describing endomorphisms of irreps, and 3) describing harmonic functions, none of these problems is trivial (or even necessarily solved) for a general compact group G.  That said, Appendix E does a good job providing evidence that this can be done for many individual groups.  However, in that case, we are still back to solving the problem on a group by group basis.

Specific Additional Points:
1.My opinion is that the language of physics does not add to the paper.  While Clebsch-Gordan and harmonic functions first arose in physics, they can be described in terms of representation theory.  In this way, both steerable CNN and quantum mechanics are applications of rep. theory and so it is not necessary to use physics here to describe steerable CNN.
2.Page 5, the notation $[j] = dim(V_j)$ seems unusual to me. It would be better to use something more standard.  In particular, in Defn 3.5, brackets are used as parenthesis, making this more confusing.
3.Page 5, the fact that input and output representations $V_{in}$ and $V_{out}$ decompose into irreps does not immediately explain how to construct a steerable kernel basis for $V_{in} \to V_{out}$ given ones between irreps $V_i \to V_j$.  Though it is not complicated, I would include this.
4.Page 5, I was confused by the inclusion of $End_{G,K}(V_j)$ at first since it does not appear when working over $\mathbb{C}$ due to Shurr’s lemma.  It is explained in the paper and more so in the appendix that it is necessary over $\mathbb{R}$, but it could be a bit clearer and earlier in the paper.  Namely, you could note that over $\mathbb{C}$ $End_{G,K}(V_j) = \mathbb{C}$ and give the possibilities over $\mathbb{R}$.
5.Page 5, Thm 3.4 and Page 43, Thm C.7.  Given that the purpose of this paper is partially to formulate Steerable CNN in precise terms, the fact that $\delta_x$ is informally considered as in $L^2(X)$ is very imprecise.  Not only does this make the proof informal, it means the maps in the theorem are not even defined.  Can you replace $L^2(X)$ with an appropriate space of distributions in order to make the statement precise?
6.Page 6, “which is zero for almost all J” should be “which is zero for all but finitely many J”
7.Appendix E, $U(1)$ is isomorphic to $SO(2)$, so it is strange to use both notations.  The difference between these subsections is whether the representations are real or complex.

**Updates from Author Feedback**
While I would have liked to see a draft with the changes, I feel reasonably sure the authors will improve Appendix C to make the statements mathematically precise. I am confident in the statements and proofs. While the presentation can be verbose and casual, I think it is justified to increase accessibility, so long as the proofs and statements are formal and precise. Based on Author responses I have increased my confidence.

---

> ### Author Response · Authors · 2020-11-16
> **Response to Reviewer 4, Part 1**
>
> We thank you for the detailed and constructive review and for highlighting that our work unifies previous efforts to construct kernel bases of equivariant CNNs. We are happy to see that you found the appendix didactic and approachable, as this was indeed our intention. We now answer each of your further comments:
>
> Q: None of these problems [i.e., finding harmonics, CG-coefficients, or endomorphisms] is trivial (or even necessarily solved) for a general compact group G.
>
> A: We agree with this comment and will update our submission accordingly to reflect this better. Our contribution is to fully clarify the general _structure_ of the kernel space solutions in terms of the natural ingredients of harmonics, CG-coefficients, and endomorphisms. The ingredients then have to be determined for each new symmetry group, and we hope that the examples in Appendix E convince the reader that this is usually doable in practice.
>
> 1: In this way, both steerable CNN and quantum mechanics are applications of rep. theory and so it is not necessary to use physics here to describe steerable CNN.
>
> A: We fully agree that representation theory is the basis of our work, as is especially reflected in our appendix. We still think it is useful to highlight similarities to physics: the structural similarities of steerable kernels to spherical tensor operators were what sparked our initial ideas in the first place. Additionally, the Wigner-Eckart Theorem is best known in its context in physics, and so we think it is useful for many readers to be pointed to this connection (but also to important differences, most importantly that steerable kernels are nonlinear) in order to believe and appreciate our results. And finally, we hope to inspire new principled research in deep learning for physics problems.
>
> 2: Page 5, the notation $[j] = \text{dim}(V_j)$ seems unusual to me. It would be better to use something more standard. In particular, in Defn 3.5, brackets are used as parenthesis, making this more confusing.
>
> A: We will change the notation from $[j]$ to $d_j$ to avoid confusion.
>
> 3: Page 5, the fact that input and output representations $V_{\text{in}}$ and $V_{\text{out}}$ decompose into irreps does not immediately explain how to construct a steerable kernel basis for $V_{\text{in}} \to V_{\text{out}}$ given ones between irreps $V_{l} \to V_{J}$. Though it is not complicated, I would include this.
>
> A: Originally, we omitted this explanation and pointed to Weiler and Cesa, who explain the reduction to irreducible input- and output representations. But we agree that it would be more convenient for the reader to find this discussion, so we will include it in the final version of the paper.
>
> 4: Namely, you could note that over $\mathbb{C}$ that $\text{End}_{G,\mathbb{K}}(V_J) = \mathbb{C}$ and give the possibilities over $\mathbb{R}$.
>
> A: We will include this discussion two paragraphs after Definition $3.3$. In particular, we will mention that over $\mathbb{R}$, we have the possibilities $\text{End}_{G,\mathbb{R}}(V_J) \cong \mathbb{R}$, $\text{End}_{G,\mathbb{R}}(V_J) \cong \mathbb{C}$, and $\text{End}_{G,\mathbb{R}}(V_J) \cong \mathbb{H}$, leading to dimension $1$, $2$, and $4$, respectively. The representation $V_J$ is in these cases then denoted as being of real type, complex type, and quaternionic type. This is shown, for example, in "Representations of Compact Lie Groups" by Bröcker and Dieck.

---

> > ### Author Response · Authors · 2020-11-16
> > **Response to Reviewer 4, Part 2**
> >
> > 5: Page 5, Thm 3.4, and Page 43, Thm C.7. Given that the purpose of this paper is partially to formulate Steerable CNN in precise terms, the fact that $\delta_x$ is informally considered as in $L^2_{\mathbb{K}}(X)$ is very imprecise. Not only does this make the proof informal, it means the maps in the theorem are not even defined. Can you replace $L^2_{\mathbb{K}}(X)$ with an appropriate space of distributions in order to make the statement precise?
> >
> > A: We agree that Thm. C.7 was formulated in an imprecise way. However, it is in essence correct if $\mathcal{K}(\delta_x)$ is understood as a limit. More detailed, we note:
> >
> > - That the map $\widehat{(\cdot)}: \text{Hom}\_{G}(X, \text{Hom}\_{\mathbb{K}}(V_{\text{in}}, V_{\text{out}})) \to \text{Hom}\_{G,\mathbb{K}}(L^2\_{\mathbb{K}}(X), \text{Hom}\_{\mathbb{K}}(V_{\text{in}}, V_{\text{out}}))$ is well-defined and an isomorphism is true and a precise statement, and the informal Dirac delta $\delta\_x$ does not appear in this part of the statement.
> > - We agree that the map $(\cdot)|\_X: \text{Hom}\_{G,\mathbb{K}}(L^2\_{\mathbb{K}}(X), \text{Hom}\_{\mathbb{K}}(V\_{\text{in}}, V\_{\text{out}})) \to \text{Hom}\_{G}(X, \text{Hom}\_{\mathbb{K}}(V\_{\text{in}}, V\_{\text{out}}))$, $\mathcal{K} \mapsto \mathcal{K}|\_X$ with $\mathcal{K}|\_X(x) := \mathcal{K}(\delta\_x)$ is, at it stands, informal. However, in Definition C.16 we make this formal and precise by defining $\mathcal{K}(\delta\_x) := \lim_{U \ni \{x\}} \mathcal{K}(\delta_U)$, with scaled indicator functions $\delta_U$ for $U \ni \{x\}$ open. These scaled indicator functions are indeed in $L^2\_{\mathbb{K}}(X)$, and the limit (of nets) is shown to exist in Corollary C.19. This means that while the original formulation of the theorem was imprecise, the proof is actually precise and formal.
> >
> > We have now decided to make the formulation of Theorem $C.7$ immediately precise by working with the limit over the net of open sets $U \ni \{x\}$ right away. We believe that it would be possible to use a space of distributions, as you suggest, however, it is not necessary.
> >
> >
> > 6: Page 6, “which is zero for almost all J” should be “which is zero for all but finitely many J”
> >
> > A: We agree and thank you for this hint.
> >
> > 7.Appendix E, $\text{U}(1)$ is isomorphic to $\text{SO}(2)$, so it is strange to use both notations. The difference between these subsections is whether the representations are real or complex.
> >
> > A: We will update our paper accordingly and will for clarity write about $\text{SO}(2)$ with real and complex representations.

---

### Official Review · AnonReviewer3 · 2020-10-29
**Important result characterizing group symmetries in learning problems**

**Rating:** 8
**Confidence:** 4

**Review:**

The paper considers Group Equivariant Convulation Neural Networks (GCNNs) which are convolutional neural networks that are equivariant wrt group symmetries of the underlying space. The equivariance requirement places constraints on the parameterization of the corresponding CNN. This work extends previous results for particular symmetries and outlines a method for obtaining these parameterizations for any compact group symmetry.

Technically, the paper establishes a Wigner-Eckert Theorem for G-steerable kernels, which in turn allows any admissible kernel to be expressed using a basis of kernels thereby establishing a natural parameterization. This procedure is carried out for U(1), SO(2), SO(3) among others.

The paper highlights important ideas from representation theory that can be used to obtain paramaterizations for symmetry-constrained learning models, and the mathematical methods could be of independent interest. The results obtained here are significant for any learning problem where there are inherent natural symmetries, as the authors point out this could be especially beneficial for data arising from physical processes.

---

> ### Author Response · Authors · 2020-11-16
> **Response to Reviewer 3**
>
> We thank you for the positive assessment of our work! We agree that the parameterization is quite natural, in particular considering the meaningful decomposition of basis kernels into harmonic basis functions, Clebsch-Gordan coefficients, and endomorphisms. Indeed, we also think that some of the mathematical results can be of independent interest. We especially think about our Wigner-Eckart Theorem for "kernel operators" (i.e. representation operators on $L^2_{\mathbb{K}}(X)$), but also about Theorem C.7 which establishes the bridge from kernel operators to steerable kernels and thus from a linear domain to a nonlinear one.

---

### Official Review · AnonReviewer1 · 2020-10-31
**Review of "A Wigner-Eckart Theorem for Group Equivariant Convolution Kernels"**

**Rating:** 6
**Confidence:** 1

**Review:**

The paper under review is a very technical contribution to the study of group-equivariance of convolution kernels. The problem of group-equivariance is studied in the most general setup, thus encompassing the previous achievements of Cohen and Welling, Cohen, Geiger, and Weiler, Weiler and Cesa, etc.

The most general tools from classical harmonic analysis and Lie group representations are put to work in order to provide the most general framework for the analysis of equivariance.

I learnt a lot about the mathematics of equivariance, a very interesting topic. However, I wonder wether the ICLR conference is the most appropriate venue for such thorough study. The application section is in particular way too sketchy to convince the novice that this impressive work will be useful to the machine learning community and an effort in this direction should be made to clarify the expected impact.

---

> ### Author Response · Authors · 2020-11-16
> **Response to Reviewer 1**
>
> We thank you for your comments and are glad you found our work interesting! We agree that this work is quite mathematical compared to most other work at ICLR, but considering the relevance of equivariance in deep learning, we still think it is the appropriate venue. More applied research can use our results for implementing equivariant CNNs for new symmetry groups.
>
> Q: The most general tools from classical harmonic analysis and Lie group representations are put to work in order to provide the most general framework for the analysis of equivariance.
>
> A: We agree that we use quite general tools. For clarity, note that we not only work with Lie groups, but in the setting of compact groups, which encompasses compact Lie groups, but additionally also includes finite groups or any other compact group which is not a Lie group (i.e., not a manifold).
>
> Q: The application section is in particular way too sketchy to convince the novice that this impressive work will be useful to the machine learning community and an effort in this direction should be made to clarify the expected impact.
>
> A: In the final version of the paper, we will update the application section in order to clarify the expected impact: as shown by Cohen, every group equivariant convolutional network is based on G-steerable kernels, and we describe for general compact group G how to parameterize the G-steerable kernel space. Thus, any practitioner who wants to use convolutions with symmetries described by a compact group can find in our theorem the answer of how to parameterize the neural network. In some instances, like Weiler and Cesa (E(2)-equivariant networks) such implementations already exist, and in other instances, e.g., O(3) or SU(N), such steerable kernel spaces are yet to be implemented. Our paper describes precisely how to make this work.

---

### Comment · ~Anatoliy_Malyarenko1 · 2021-03-19
**A suggestion**

Good sources for real representations of O(3) and the corresponding Clebsch--Gordan coefficients:

Gordienko, V. M. Matrix elements of real representations of the groups {${\rm O}(3)$} and {${\rm SO}(3)$}, Sibirsk. Mat. Zh., 43, no. 1 (2002), 51--63, MR1888117

Godunov, S. K. and Gordienko, V. M. Clebsch--{G}ordan coefficients in the case of various choices of bases of unitary and orthogonal representations of the groups {${\rm SU}(2)$} and {${\rm SO}(3)$}}, Sibirsk. Mat. Zh., 45, no. 3 (2004), 51--63, MR2078714

---

> ### Author Response · Authors · 2021-06-09
> **Thank you!**
>
> Thank you for this suggestion!

---

### Decision · Program_Chairs · 2021-01-07
**Final Decision**

**Decision:**

Accept (Poster)

**Comment:**

Reviewers generally agree that the main result of the paper, which generalizes the classical Wigner-Eckart Theorem and provides a  basis for the space of G-steerable kernels for any compact group G, is a significant result. There are also several concerns
that need to be addressed. R4 notes that the use of the Dirac delta function (e.g. Theorem C.7) is informal and mathematically imprecise and needs to be fixed. R1 notes that it would be helpful to at least describe how this general formulation can be applied in machine learning.

Presentation and accessibility: the current version of the paper will be accessible to only  a small part of the machine learning audience, i.e. those already with advanced knowledge in mathematics and/or theoretical physics, in particular in representation theory. If the authors aim to make it more accessible, the writing would need to be substantially improved.